# Seasonal changes of mélange thickness coincide with Greenland calving dynamics

Yue Meng [1] ✉, Ching-Yao Lai [1], Riley Culberg [2], Michael G. Shahin [3], Leigh A. Stearns [3,4], Justin C. Burton [5] & Kavinda Nissanka[5]

Iceberg calving is a major contributor to Greenland's ice mass loss. Ice mélange, tightly packed sea ice and icebergs, has been hypothesized to buttress the calving fronts. However, quantifying the mélange buttressing force from field observations remains a challenge. Here we show that such quantification can be achieved with a single field measurement: thickness of mélange at the glacier terminus. We develop a three-dimensional discrete element model of mélange along with a simple analytical model to quantify the mélange buttressing using mélange thickness data from ArcticDEM over 32 Greenland glacier termini. We observed a strong seasonality in mélange thickness: thin mélange (averaged thickness $34^{+17}_{-15}$ m) in summertime when terminus retreats, and thick mélange (averaged thickness $119^{+31}_{-37}$ m) in wintertime when terminus advances. The observed seasonal changes of mélange thickness strongly coincide with observed Greenland calving dynamics and the modeled buttressing effects.

The Greenland Ice Sheet (GrIS), holding 7.2 m of sea level equivalent, has become the largest single source of barystatic sea-level rise in the cryosphere[1,2]. Under high carbon emission scenario, the GrIS is projected to contribute about 79–167 mm of sea-level rise by 2100, 30% to 60% of which comes from iceberg calving at marine-terminating glaciers[3,4]. Projections of sea level rise by 2100 can vary by 374 mm depending on the rate of iceberg calving at Greenland ice sheet margins[5]. Calving laws used in current ice-sheet models predict calving rates using empirically tuned strain rate or stress criteria, which is inadequate to capture the complex external interactions that modulate calving and are strongly coupled with the warming climate[3,6–8]. In particular, how calving depends on ice-ocean interactions is poorly understood.

Recent large calving front retreats at some Greenland outlet glaciers have been correlated with rapid breakup of mélange, a collection of sea ice and icebergs tightly packed in tidewater glacier fjords adjacent to glacier termini[9–15]. Remote observations reveal that seasonal advance and retreat of glacier termini coincides with the formation and disappearance of mélange[16], and variations in the mélange rigidity induced by sea ice that grows in winter and decays in summer[17–21].

Rigid, immobile mélange has been observed from either persistently low surface temperatures[18] or a coherent and uniform velocity field at the terminus measured by feature tracking[17,19–21]. These observations suggest that the presence of rigid mélange can mitigate iceberg calving by providing a back force at the glacier terminus[17–19,22–29]. The force exerted by the mélange to support the glacier terminus is called the mélange buttressing force[28]. Prescribing a periodic change in the magnitude of the mélange buttressing force in ice-sheet models successfully reproduces observed seasonal calving dynamics[4,25,26,30–34]. In a warming climate, a complete loss of mélange buttressing may prevent terminus advances in winter and spring while exacerbating summer retreats, shifting the seasonal range in the terminus position inland[4].

To capture physical processes that dictate the buttressing force magnitude, recent studies have taken a granular mechanics approach to quantify the flow and stress within ice mélange[27,28]. Discrete element models[27] successfully reproduce the observed jamming wave propagation during calving events[35]. These discrete element models are two-dimensional and assume a constant thickness of ice mélange and disk-shaped grains for simplicity. However, field observations show that mélange thickness can be non-uniform and decays with distance from

[1]Department of Geophysics, Stanford University, Stanford, CA, USA. [2]Department of Earth and Atmospheric Sciences, Cornell University, Ithaca, NY, USA. [3]Department of Geology, University of Kansas, Lawrence, KS, USA. [4]Department of Earth and Environmental Sciences, University of Pennsylvania, Philadelphia, PA, USA. [5]Department of Physics, Emory University, Atlanta, GA, USA. ✉e-mail: olivmeng@stanford.edu

the terminus[23,29]. In early summer 2016 for Jakobshavn Isbræ, an unusually thick mélange wedge at the glacier front coincided with a one-month glacier quiescence period, implying that thick mélange can inhibit calving[23]. Continuum theories state that assuming mélange of a constant thickness, the mélange buttressing force per unit width linearly scales with mélange thickness ($F/W \sim H$)[28], whereas in three dimensions with along-flow mélange thickness variations, it scales with the square of the mélange thickness ($F/W \sim H^2$) built up at the terminus[29]. Therefore, it is crucial to consider the three-dimensional nature of mélange. To quantify the mélange buttressing force, previous two-dimensional models assuming mélange of uniform thickness require estimates of many parameters, including fjord/mélange friction/cohesion properties, and the mélange width/length[28]. Motivated by recent observations of weekly to seasonal variations in mélange thickness, we hypothesize that mélange thickness dictates its buttressing force per unit width. To test this hypothesis, we use terrestrial laser scanning and ICESat-2 data to demonstrate that mélange thickness is well-correlated with terminus position and calving dynamics at four major Greenland outlet glaciers. Motivated by these

observations, we develop a continuum mélange model, validated with a three-dimensional discrete element model, which confirms that mélange thickness and packing density at the terminus are the only observations required to estimate mélange buttressing force per unit width. We then apply this continuum model to estimate mélange buttressing forces around Greenland from ArcticDEM observations, and show that the inferred buttressing force is highly consistent with observed patterns of terminus advance and retreat.

## Results

### Mélange thickness associated with calving dynamics at Helheim Glacier in 2019–2020

To investigate the correlation between the mélange thickness and calving dynamics, we derive time series of mélange elevation at the terminus, calving events, and terminus position inferred from TLS data and satellite images (Sentinel-1, Sentinel-2, and Landsat 8) from 1 Sep 2019 to 1 Sep 2020 at Helheim Glacier (Fig. 1b). Throughout 2019, a REIGL VZ-6000 terrestrial laser scanner (TLS), located at the triangle marked in Fig. 1a, scanned the terminus and ice mélange of Helheim

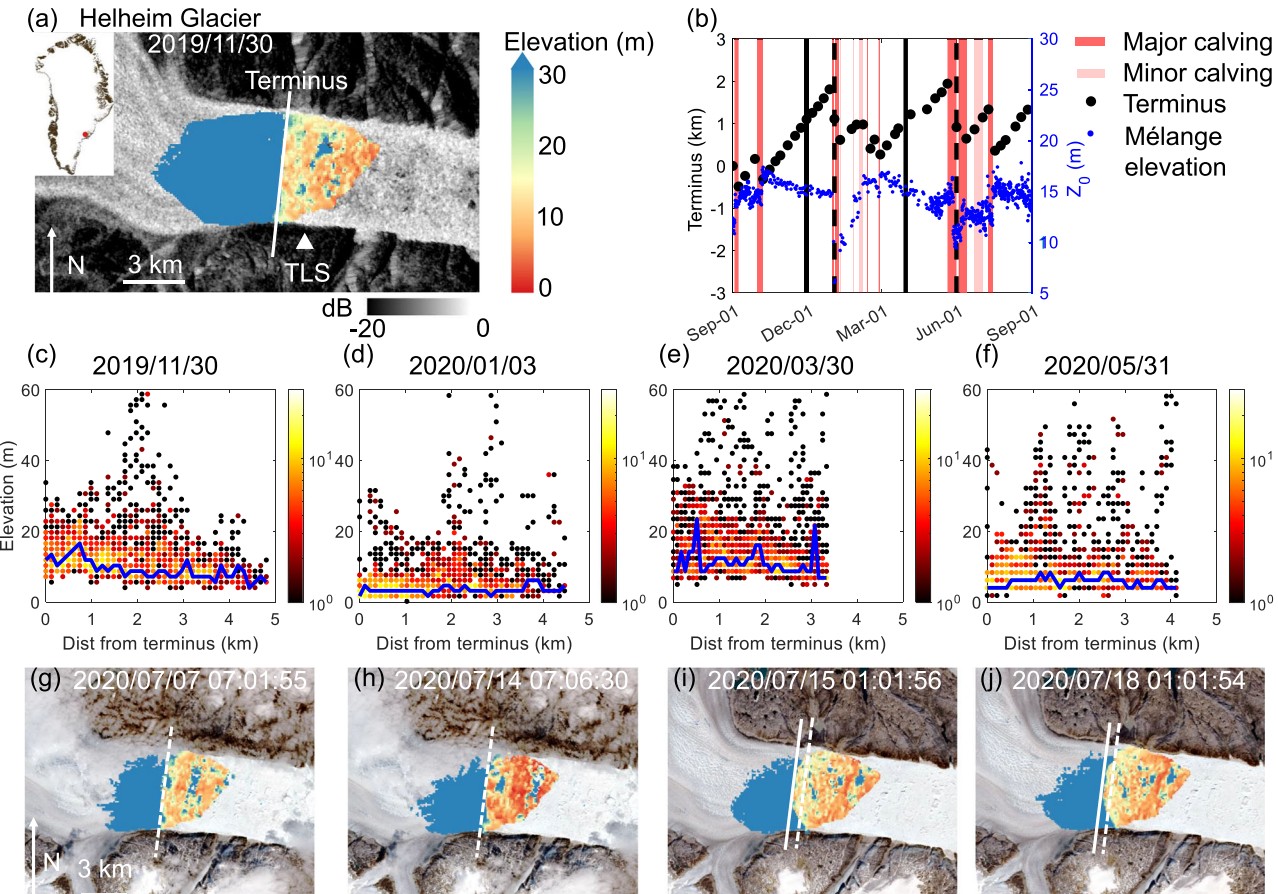

**Fig. 1 | Helheim Glacier and ice mélange. a** Terrestrial laser scanner (TLS) measured elevation map overlain on a Sentinel-1 HV image (both acquired on 30 Nov 2019). The white line across the fjord indicates the glacier front location. The white triangle indicates the TLS location. The upper left inset shows the location of Helheim Glacier in Greenland. The image is in polar stereographic projection (EPSG: 3413). **b** Terminus position relative to 1 Sep 2019, where the positive sign indicates terminus advance. Blue dots denote the averaged mélange elevation within 1 km of the terminus, $Z_0$. We mark the time period during which a calving event occurs by a red-shade rectangle. Four vertical black lines mark the dates for the TLS-measured elevation data presented in (**c**–**f**), which corresponds to 30 Nov 2019, 3 Jan 2020, 30 Mar 2020, and 31 May 2020, respectively. Solid black lines mark the dates with terminus advances, and dashed black lines mark the dates with terminus retreats. **c**–**f** Surface elevation profiles for the mélange displayed as density plots (1510–1859

data points in total); the color bar denotes the number of data points that have the same elevation and distance from terminus values. For any specific distance from terminus, we find the elevation value that has the maximum number of data points. Solid blue lines connect these elevation values along the distance from terminus as the representative mélange elevation profiles. Major calving occurred on 2 Jan 2020 and 28 May 2020 led to noticeable mélange thinning on 3 Jan (**d**) and 31 May (**f**). **g**–**j** The TLS-measured mélange surface elevation map from 7 Jul to 18 Jul 2020. Dashed white lines indicate positions of the terminus before calving. Solid white lines indicate positions of the terminus after calving. Dates on the images show the acquisition dates for TLS data and times are in UTC. TLS scans are overlain on Sentinel-2 images acquired around the same dates. Mélange thinning was observed from 7 Jul to 14 Jul 2020, followed by a major calving event occurred between 14 Jul and 15 Jul 2020.

Glacier. Figure 1a shows the TLS-measured surface elevation field for ice mélange on 30 Nov 2019 (see "Methods" for the TLS processing steps). Mélange surface elevation was derived from TLS every 24 h in the winter months, and every 6 h in non-winter months. To display the spatial profile of the mélange elevation, we calculate distances from terminus for all data points in the ice mélange and plot them as density maps in Fig. 1c–f. For any specific distance from the terminus, there is a spread of mélange elevation that exhibits a long-tail distribution due to the presence of large icebergs. To reflect the mélange elevation that is piled up from small icebergs and sea ice, we connect elevation values that have maximum numbers of data points along the distance from terminus as the representative mélange elevation profiles (solid blue lines in Fig. 1c–f, see Methods and Fig. 2 in SI for a sensitivity study of this metric). In doing so, we prevent large icebergs from disrupting the mélange elevation profiles without cutting off the data by an arbitrary elevation threshold. The resulting representative mélange elevation profiles are always below 30 m, indicating that it is a reasonable elevation threshold to exclude large icebergs from consideration. To estimate mélange elevation near the glacier terminus ($Z_0$), we take an average of all data points below the 30 m threshold within 1 km (i.e., 1/5 of the fjord width) of the terminus. We infer thickness of the mélange based on TLS-derived surface elevations and assuming hydrostatic equilibrium.

We digitize the terminus position on each TLS scan with a straight line that intersects and is tangent to the furthest upstream point of the calving front (the white line in Fig. 1a). The terminus advance/retreat refers to the changes in the terminus position, and is quantified by distances between these straight lines. This can indicate whether the glaciers are advancing (growing) or retreating (shrinking). With reference to previous classification of calving events[23], here, we define "major calving" events as those with block size >0.25 km² (observed from TLS scans), and causing significant mélange motion and an overall terminus retreat (observed from satellite images); "minor calving" events are those in which visible blocks calved, but the mélange or terminus position remained largely unchanged. We observed two episodes of calving cessation, from 8 Oct 2019–31 Dec 2019 and 1 Mar 2020–20 May 2020, when no major or minor calving occurred and the terminus advanced steadily for 2 km and 1.7 km, respectively. We found that mélange elevation at the terminus averaged 15 m during these periods. We include TLS scans during these two episodes of calving cessation in Supplementary material (Fig. 3 in SI). We identified two periods during which noticeable mélange thinning occurred after calving. The mélange elevation at the terminus decreased by 5.3 m to 9.4 ± 0.05 m from 31 Dec 2019 to 3 Jan 2020 (Fig. 1d), and by 4 m to 10 ± 0.05 m from 26 May 2020 to 31 May 2020 (Fig. 1f). Major calving occurred on 2 Jan 2020 and 28 May 2020 with corresponding retreats at the terminus of 1.2 km and 1.3 km, respectively, accompanied by large increases in fraction of mélange area composed of large icebergs (see SI Figs. 4 and 5 for associated TLS scans and temporal evolution of iceberg fraction). We attribute the noticeable mélange thinning to calving-induced divergent motion within the mélange that helped to advect ice away[23]. This was followed by 1 to 2 months of active calving events from Jan to Mar, and Jun to Jul in 2020 (Fig. 1b). In addition, we identified two periods during which noticeable mélange thinning occurred prior to calving. The mélange elevation at the terminus decreased by 1.6 m to 10.8 ± 0.05 m from 1 Sep to 3 Sep 2019, and by 2.3 m to 11.6 ± 0.05 m from 7 Jul to 14 Jul 2020 (Fig. 1g, h). Major calving occurred on 4 Sep 2019 and 15 Jul 2020 (Fig. 1i) with corresponding retreats at the terminus of 0.5 km and 1.0 km, respectively.

Apart from the weekly terminus variability of Helheim Glacier presented in this section, satellite remote sensing observations on many Greenland glacier termini have shown significant terminus-position seasonality, with advance from winter to spring and retreat from summer to fall through enhanced calving[36–38]. Previous studies

have attributed seasonal calving dynamics to buttressing from ice mélange[15,18,39], which motivates us to further explore seasonal changes of mélange thickness and calving dynamics on other Greenland glaciers.

## Seasonal changes of mélange thickness and calving dynamics

To investigate whether there are correlations between ice mélange thickness and calving dynamics on other glaciers, we use ICESat-2 observations of mélange surface elevation. While this dataset does not provide the temporal resolution to study individual calving events, we can leverage the observed seasonality in terminus advance and retreat at many Greenland glaciers to assess whether mélange thickness is correlated with periods of quiescence versus vigorous calving.

There are very few ICESat-2 tracks passing through the fronts of termini in different seasons, because positions of termini vary seasonally but ICESat-2 tracks are generally fixed in space. We identify ICESat-2 tracks passing over glacier termini in different seasons for Jakobshavn Isbræ (Fig. 2a), Kangerlussuaq Glacier (Fig. 2b), and Store Glacier (Fig. 2c), which are large discharging glaciers contributing to Greenland's mass losses[40]. We use the ATL06 data set from ICESat-2 that provides geolocated, land-ice surface heights above the WGS 84 ellipsoid[41]. After subtracting the geoid heights at the glacier termini[42], the mélange surface elevations are approximately heights above mean sea level[23]. Next, we correct for the tidal influence on the mélange elevations using the Greenland 1-km tide model[43], ensuring that the adjusted surface elevations represent heights above sea level at the time of data acquisition. Surface elevation data is acquired along the ICESat-2 track and displayed as a function of the distance from terminus. We found that mélange was continuously present at Jakobshavn Isbræ from 2021 to 2022 and at Kangerlussuaq Glacier in 2020, while it was seasonally present at Store Glacier in 2019. Where mélange persisted, we calculated seasonally distinct freeboard heights from winter to early spring (solid black lines) and in summer (dashed black lines) (Fig. 2d, e). Near the termini, mélange for the two glaciers both exhibit different ranges of freeboard heights during the two seasons: 20–35 m in winter or spring, and below 5 m in summer. The seasonal changes in mélange thickness at the terminus may explain the observed calving dynamics and terminus motion: zero or minor calving with an advancing terminus from winter to spring, and vigorous calving with a retreating terminus from summer to fall (Fig. 2g, h). At Store Glacier, the mélange was present from 1 Jan to 14 June, after which calving resumed and the terminus kept retreating (Fig. 2i). The mélange elevation profile on 22 March 2019 exhibits a thickness gradient with a freeboard height of around 30 m near the terminus (Fig. 2f). ICESat-2 data provide a snapshot of the mélange elevation profile, which may vary significantly within days or across different profile lines in the fjord, as evidenced by results from as evidenced by results from the previous section. Although ICESat-2 data cannot fully capture the spatiotemporal variability in mélange thickness distributions, it still offers valuable insights into the correlations between mélange thickness and terminus seasonality.

## A three-dimensional continuum model of ice mélange

Remote sensing observations reveal a strong correlation between mélange thickness and calving dynamics. As a result, quantifying the buttressing force of mélange in terms of its thickness is the first step to better representing ice-ocean interactions and developing process-based calving models. Building on the one-dimensional model of ice flow[29,44], we derive a three-dimensional continuum model for ice mélange, and then validate it by discrete element modeling. Figure 3 shows a schematic of the glacier-ocean-mélange system. We use Cartesian coordinate system, with $x$ starting from the terminus and in the direction along the fjord, $y$ in the direction across the fjord, and $z$ in the vertical direction with $z = 0$ at sea level. We begin by defining a number

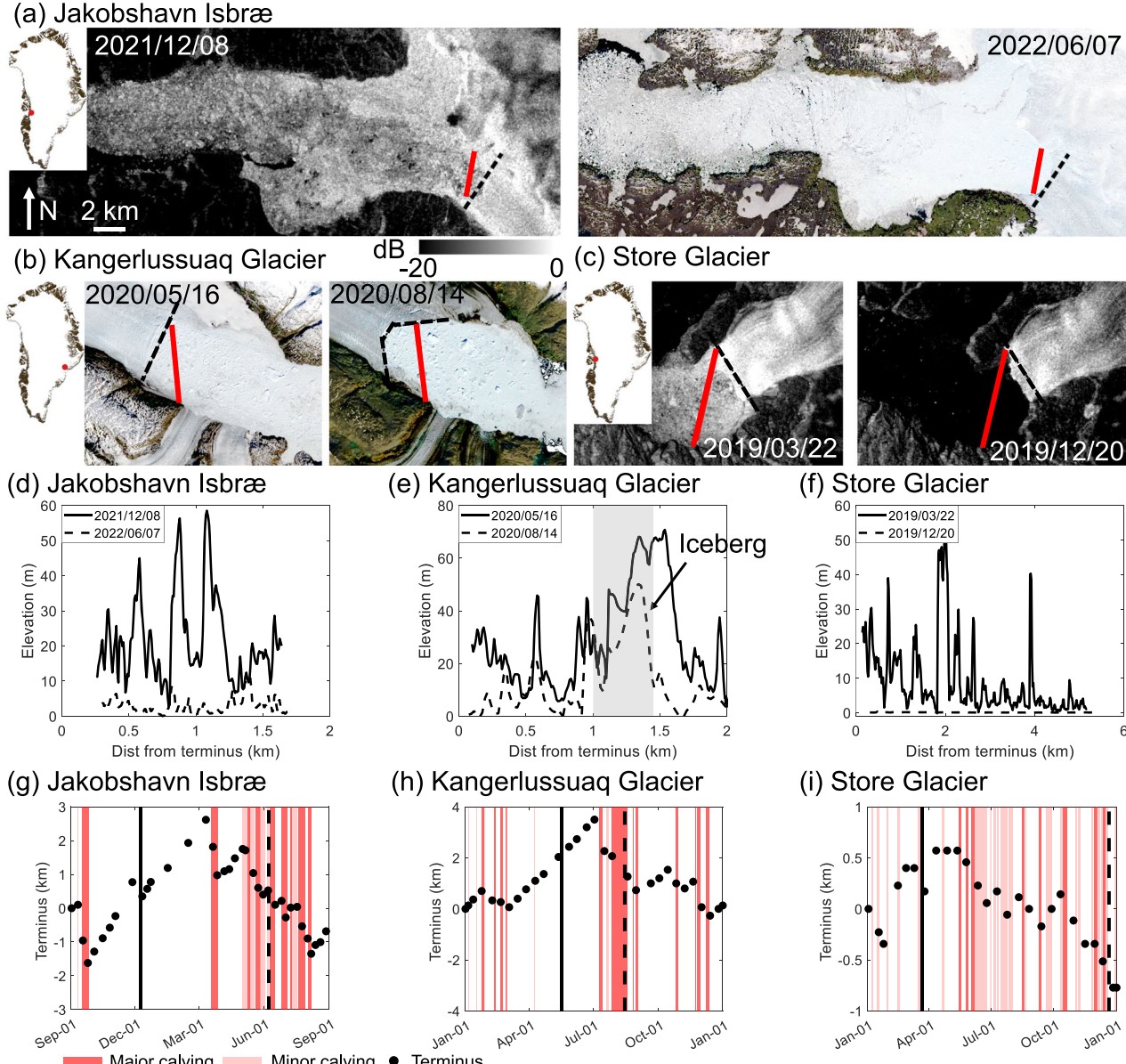

**Fig. 2 | Seasonal changes of mélange thickness are correlated with calving dynamics and terminus position. a** Sentinel-1 HV and Landsat 8 images for Jakobshavn Isbræ on 8 Dec 2021, 7 Jun 2022, respectively. **b** Landsat 8 images for Kangerlussuaq Glacier on 19 May 2020, 14 Aug 2020, respectively. **c** Sentinel-1 HV images for Store Glacier on 22 Mar 2019, 19 Dec 2019, respectively. In (**a**–**c**), the black dashed line indicates the terminus position and the red line across the fjord indicates the ICESat-2 track along which surface elevation data is acquired. The date on the image shows the acquisition date for ICESat-2 data, which is around the same date as the presented satellite image. Upper left inset shows location of the glacier terminus in Greenland. Images are in polar stereographic projection (EPSG: 3413). **d**–**f** Surface elevation profiles extracted along ICESat-2 tracks in (**a**–**c**), after accounting for the local difference between the ellipsoid and the geoid. The horizontal axis shows the distance from the terminus (black dashed line in (**a**–**c**)). Solid and dashed lines represent the surface elevation data acquired from different dates. Note that for Kangerlussuaq on 14 Aug 2020 (dashed line in (**e**)), there is an increase in surface elevation at distance 1–1.5 km from the terminus due to the presence of a large iceberg. **g**–**i** Black dots show the terminus position relative to 1 Sep 2021, 1 Jan 2020, and 1 Jan 2019 for Jakobshavn Isbræ, Kangerlussuaq Glacier, and Store Glacier, respectively. Here, the positive sign indicates terminus advance. Calving events are inferred from satellite images. The vertical solid and dashed lines mark the dates for ICESat-2 data presented in (**d**–**f**).

of variables that are required for describing the continuum models. First, we define the strain rate tensor as $\dot{\epsilon}_{ij} = \frac{1}{2}\left(\frac{\partial u_i}{\partial x_j} + \frac{\partial u_j}{\partial x_i}\right)$, where $u_i$ is the velocity component and $x_i$ is the spatial coordinate. $u$, $v$, $w$ denote the velocity in the $x$, $y$, $z$ components, respectively. The Cauchy stress tensor $\sigma_{ij} = \sigma_{ji}$ partitions into deviatoric stress $\sigma'_{ij}$ and the hydrostatic pressure $p$ via $\sigma_{ij} = -p\delta_{ij} + \sigma'_{ij}$, where $p = -\frac{1}{3}\sigma_{kk}$ and $\delta_{ij}$ is the Kronecker delta. Here, compressive stresses have negative values. The trace of the deviatoric stress tensor is equal to zero, that is, $\sigma'_{xx} + \sigma'_{yy} + \sigma'_{zz} = 0$.

We make the following assumptions: (i) the fjord width is a constant; (ii) the mélange is in a three-dimensional state; (iii) the mélange packing density, thickness, viscosity, and strain rates are uniform across the width of the fjord and across the depth of the mélange, but vary with the distance from terminus; (iv) a viscous constitutive relationship between the mélange deviatoric stress and the strain rate, that is, $\sigma'_{ij} = 2\eta\dot{\epsilon}_{ij}$, where $\eta$ is the effective mélange viscosity. As the trace of the deviatoric stress tensor is equal to zero, the mélange flow is incompressible, that is, $\dot{\epsilon}_{xx} + \dot{\epsilon}_{yy} + \dot{\epsilon}_{zz} = 0$; (v) variations of horizontal velocities across the depth of the mélange are negligible, that is,

**Fig. 3 | Schematic of the glacier-ocean-mélange system.** We use a Cartesian coordinate system, with $x$ starting from the terminus and extending along the fjord, $y$ running across the fjord, and $z$ in the vertical direction, where $z = 0$ is at sea level. $z_b$ and $z_s$ denote the bottom and surface of the mélange, respectively, while $\phi_0$ and $H_0$ represent the mélange packing density and thickness at the terminus. $\rho_i$ and $\rho_w$

are the densities of ice and water, respectively, and $g$ is the acceleration due to gravity. $\phi(x)$ and $H(x)$ indicate the packing density and thickness of the ice mélange varying along the fjord direction. Because the mélange is a porous medium, the skeleton vertical stress, $\sigma_{zz}$, vanishes at its bottom free surface[45-47].

$\frac{\partial w}{\partial x} \sim \frac{\partial w}{\partial y} \ll \frac{\partial u}{\partial z} \sim \frac{\partial v}{\partial z} \cong 0$, and therefore $\sigma'_{xz} = \sigma'_{yz} = 0$; and (vi) the bottom of the mélange is fully permeable and leaves the skeleton stress-free, that is, $\sigma_{zz}|_{z_b} = 0$. Such assumption align with the fact that the effective stress always vanishes at the free surface of the solid skeleton in a porous medium[45-47].

Under steady flow conditions, the vertical force balance for ice mélange states that:

$$\frac{\partial \sigma_{zz}}{\partial z} = \rho_i \phi(x) g', \tag{1}$$

where $\rho_i$ is the density of ice, $\phi(x)$ is the packing density of ice mélange that varies along the fjord direction, and $g'$ is the effective acceleration due to gravity. For ice above the waterline, $g' = g$. For ice below the waterline, $g' = (1 - \frac{\rho_w}{\rho_i})g$. Since the vertical stress in ice mélange equals zero at its top and bottom surface, we arrive at a final expression for vertical stress $\sigma_{zz}$ (Fig. 3):

$$\sigma_{zz}(x, z) = \begin{cases} \rho_i \phi(x) g \left( z - \left(1 - \frac{\rho_i}{\rho_w}\right) H(x) \right), \text{where } 0 < z < (1 - \frac{\rho_i}{\rho_w}) H(x), \\ (\rho_i - \rho_w) \phi(x) g \left( z + \frac{\rho_i}{\rho_w} H(x) \right), \text{where } -\frac{\rho_i}{\rho_w} H(x) < z < 0. \end{cases} \tag{2}$$

where $\rho_w$ is the density of sea water, and $H(x)$ is the mélange thickness that varies along the fjord direction. The equation states that the vertical stress for mélange linearly decreases from zero at the top to $-\rho_i \phi(x) g(1 - \frac{\rho_i}{\rho_w}) H(x)$ at sea level, and then linearly increases to zero at the bottom (Fig. 3). With some algebraic steps we derive the mélange buttressing force ($F$) per unit width ($W$) on the terminus as follows (see SI for derivation details):

$$\frac{F}{W} = \left( \int_{z_b}^{z_s} -\sigma_{xx}(x, z) dz \right) \bigg|_{x=0} = \frac{1}{2} \rho_i \left(1 - \frac{\rho_i}{\rho_w}\right) g \phi_0 H_0^2 - 4 H_0 \left( \eta \frac{\partial u}{\partial x} \right) \bigg|_{x=0} - 2 H_0 \left( \eta \frac{\partial v}{\partial y} \right) \bigg|_{x=0}. \tag{3}$$

where $z_b$, $z_s$ are at the bottom and surface of the mélange and $\phi_0$, $H_0$ are the mélange packing density and thickness at the terminus, respectively.

When the mélange packing density approaches $\phi_0 = 1$, Eq. (3) converges to the expression of ice shelf buttressing[48]. It is well known that an unconfined ice shelf (i.e., ice tongue) provides zero buttressing as the glaciostatic pressure balances out the extensional stress[49]. The horizontal momentum balance equation (Eq. 8 in SI) shows that without lateral confinements from fjord walls and assuming a uniform mélange velocity field, ice mélange cannot thicken near the terminus, and thus also provides negligible buttressing force. In reality, fjords always provide lateral confinements on the mélange. Equation (3) states that the mélange buttressing force has two components: (i) the

glaciostatic pressure induced by mélange thickness ($\propto H_0^2$), and (ii) horizontal deviatoric stresses induced by velocity gradients ($\propto \frac{\partial u}{\partial x}, \frac{\partial v}{\partial y}$). Previous studies have shown that winter velocity fields of the mélange are generally steady and highly uniform in space[28,29], whereas summer velocity fields of the mélange tend to be much more variable and can be uniform, compressional, or extensional[29]. For dense mélange confined within a straight fjord, the velocity gradient along the fjord is much larger than that across the fjord, that is, $\frac{\partial u}{\partial x} \gg \frac{\partial v}{\partial y}$. To characterize the relative magnitude of the horizontal deviatoric stress to the glaciostatic pressure, we substitute representative values for parameters in Eq. (3) and obtain:

$$\frac{|4 H_0 (\eta \frac{\partial u}{\partial x})|_{x=0}|}{\frac{1}{2} \rho_i \left(1 - \frac{\rho_i}{\rho_w}\right) g \phi_0 H_0^2} \in \left[ 3.90 \times 10^{-14}, 1.17 \times 10^{-11} \right] \times \eta, \tag{4}$$

where the range of mélange thickness, $H_0$, is derived from DEM observations. We take $H_0$ to vary from 35 m, which is the minimum size of icebergs detected within the mélange[50,51], to 240 m, which is the thickest mélange observed across 32 Greenland termini in 2013–2022 (See Table 2 in SI). We adopt $\frac{\partial u}{\partial x} \in [\frac{2 \, \text{m/day}}{15 \, \text{km}}, \frac{25 \, \text{m/day}}{10 \, \text{km}}]$[29], $\phi_0 \in [0.64, 1]$[52,53], $\rho_i \in [870 \, \text{kg/m}^3, 920 \, \text{kg/m}^3]$[23], and $\rho_w \in [1020 \, \text{kg/m}^3, 1029 \, \text{kg/m}^3]$[54]. As the mélange acts as a weak granular ice shelf[28], its effective viscosity should be much smaller than the glacier ice viscosity, $\eta \ll \eta_i = 10^{12} - 10^{15}$ Pa·s[55,56]. For mélange with a high viscosity ($\eta > 10^{11}$ Pa·s), we need to consider deviatoric stress effects as has been done in[57]. The mélange in the following discrete element model has an estimated viscosity of $2 \times 10^{10}$ Pa·s (see Section 1 in SI for details). Therefore, for mélange with a low viscosity, glaciostatic pressure dominates and the mélange buttressing force can be approximated as:

$$\frac{F}{W} = \frac{1}{2} \rho_i \left(1 - \frac{\rho_i}{\rho_w}\right) g \phi_0 H_0^2. \tag{5}$$

The "granular ice shelves" depth-averaged horizontal momentum equations for mélange (see SI for derivation and validation against discrete element simulations) resemble that of ice shelves:

$$\frac{\partial}{\partial x}(2 H(x, y) \bar{\sigma}'_{xx}) + \frac{\partial}{\partial x}(H(x, y) \bar{\sigma}'_{yy}) + \frac{\partial}{\partial y}(H(x, y) \bar{\sigma}'_{xy}) = \rho_i g \left(1 - \frac{\rho_i}{\rho_w}\right) H(x, y) \frac{\partial (\phi(x, y) H(x, y))}{\partial x},$$

$$\frac{\partial}{\partial x}(H(x, y) \bar{\sigma}'_{xy}) + \frac{\partial}{\partial y}(H(x, y) \bar{\sigma}'_{xx}) + \frac{\partial}{\partial y}(2 H(x, y) \bar{\sigma}'_{yy}) = \rho_i g \left(1 - \frac{\rho_i}{\rho_w}\right) H(x, y) \frac{\partial (\phi(x, y) H(x, y))}{\partial y}, \tag{6}$$

where the depth-averaged stress $\bar{\sigma}_{ij} = \frac{1}{H} \int_{z_b}^{z_s} \sigma_{ij} dz$. When the mélange packing density approaches $\phi = 1$, Eq. (6) converges to the shallow shelf approximation (SSA)[58]. The mélange momentum balance along the fjord direction reveals three competing forces: compressional/extensional flow from velocity gradients within the mélange (negligible if mélange viscosity is smaller than $10^{11}$ Pa·s), glaciostatic stress from

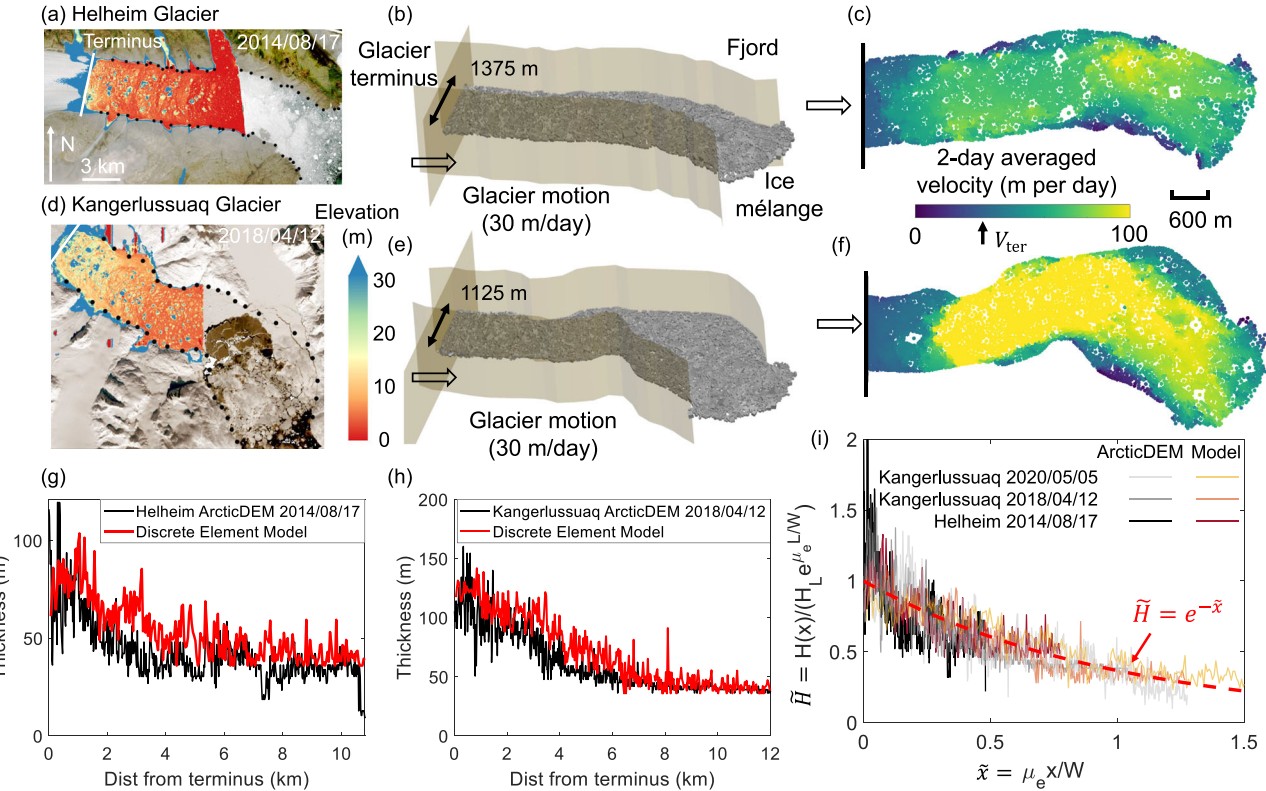

**Fig. 4 | The three-dimensional discrete element model for mélange composed of cubic icebergs with a power-law size distribution and confined within real fjord geometries. a–c** For Helheim glacier, **d–f** for Kangerlussuaq glacier. The real fjord geometry is scaled down by four times for the discrete element model. **a** The mélange elevation above mean sea level from ArcticDEM strip, overlain on satellite images acquired around the same date. The ArcticDEM acquisition date is shown in the top right corner. White line across the fjord indicates glacier front location. The images are in polar stereographic projection (EPSG: 3413). Black dots along fjords are adopted in the model to construct boundary walls that resemble fjord geometries. **b**, **c** are side and top view for iceberg positions and velocities after 16 days

into simulations with steady terminus advance and no calving. The glacier terminus moves at a constant velocity, $V_{ter} = 30$ m/day. We calculate the 2-day averaged velocity of each iceberg element by dividing the iceberg's displacement between 14 and 16 days of terminus motion by the time interval (2 days), which is indicated by filled color in (**c**). **d–f** follow captions of (**a–c**). **g**, **h** The comparison of the mélange thickness profile between ArcticDEM (black lines) and discrete element model (red lines) for Helheim and Kangerlussuaq glacier, respectively. **i** The mélange thickness profile from three ArcticDEM strips and corresponding discrete element models collapse onto the exponential analytical solution (Eq. (7), red dashed line). See supplementary videos for the full temporal evolution of the mélange behaviors.

mélange thickness, and shear stresses on fjords. Therefore, the full thickness profile of the mélange depends on fjord/mélange friction/cohesion properties, velocity gradients and viscosity of the mélange, and the mélange width/length. Using the horizontal momentum balance equations (Eq. (6)) and non-dimensionalization (see SI for details), we further derive the mélange thickness profile, $H(x)$:

$$\tilde{H}(\tilde{x}) = e^{-\tilde{x}}, \tilde{x} \in \left[0, \frac{\mu_e L}{W}\right], \tag{7}$$

where $\mu_e$ is the effective coefficient of friction between the mélange and the fjord wall, which depends on the material friction coefficient and the geometry of the fjord walls, i.e., wall roughness[28,59,60]. Here, we define the mélange length, $L$, as the distance beyond which the mélange thickness decays to a threshold value, $H_L$, where mélange becomes a monolayer of icebergs with thickness dominated by particle size distribution instead of stress balances that give rise to Eq. (7). We define the dimensionless distance, $\tilde{x} = \frac{\mu_e x}{W}$, and the dimensionless thickness, $\tilde{H} = \frac{H(x)}{H_L e^{\frac{\mu_e L}{W}}}$. To quantify the mélange buttressing force per unit width, previous two-dimensional model assuming mélange of uniform thickness required assumptions on fjord/mélange friction properties and the mélange width/length[28]; in our three-dimensional model the mélange thickness and packing density at the terminus are the only parameters needed. As the length and

thickness are coupled by stress balances within the granular material, the mélange thickness build-up at the terminus already encodes the aforementioned material and geometric properties. For instance, thicker mélange can be built up at the terminus with longer fjords, larger fjord friction, or increased mélange rigidity in winter.

In summary, momentum balance equations reveal that the mélange buttressing force per unit width is solely controlled by the packing density $\phi_0$ and mélange thickness $H_0$ at the glacier terminus, and is proportional to the square of mélange thickness (Eq. (5)). Additionally, mélange thickness exponentially decays with its distance from terminus (Eq. (7)).

## A three-dimensional discrete element model of ice mélange

To validate continuum predictions on the mélange buttressing force and thickness profile (Eqs. (5), (7)), we develop a three-dimensional discrete element model on the mélange with a steadily advancing terminus. We adopt fjord geometries from Helheim and Kangerlussuaq glaciers (black dots in Fig. 4a, d), which are scaled down by four times for the discrete element models to save computational costs (Fig. 4b, e). Note that such downscaling does not affect the shape of the dimensionless mélange thickness profile, which will be compared with Eq. (7). Icebergs are modeled as cubic particles with sizes varying from 36 m to 200 m and have a power-law size distribution with an exponent of $-4$[50,51,61]. In the models, we limit the maximum size of icebergs

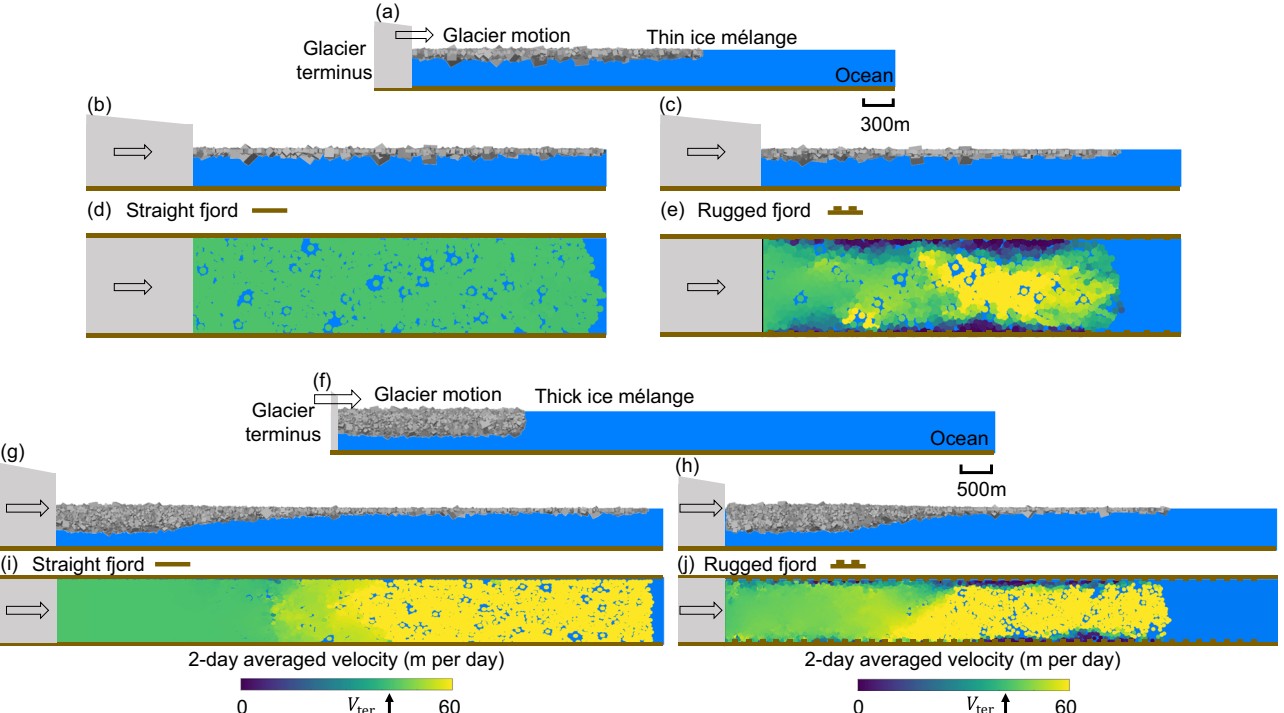

**Fig. 5 | The three-dimensional discrete element model for mélange composed of cubic icebergs with power-law size distribution and confined within simplified fjord geometries. a–e** For a thin mélange, **f–j** for a thick mélange. **a** A side view of the initial condition for the simulation with mélange width $W = 1$ km, length $L = 3$ km, and thickness $H_{ini} = 60$ m. The glacier terminus is shown as a gray block on the left. The ocean floor and fjord walls are plotted in brown. The glacier terminus starts to move at a constant velocity, $V_{ter} = 43.2$ m/day. **b–e** are snapshots for iceberg positions and velocities after 16 days into simulations with steady terminus advance and no calving. **b**, **d** are the side and top view for a straight fjord wall configuration; **c**, **e** are the side and top view for a rugged fjord wall configuration. The two-day averaged velocity of each iceberg element is indicated by filled color in (**d**) and (**e**). **f** A side view of the initial condition for the simulation with $W = 1$ km, $L = 3$ km, and $H_{ini} = 378$ m. **g–j** follow captions of (**b–e**). See supplementary videos for the full temporal evolution of the mélange behaviors.

to be around one-fifth of the fjord width, as commonly observed in both fjords. We initialize the ice mélange thickness with a profile that linearly decays within 1.2 km from the terminus and with the right end open to the ocean. We push the left end of the mélange with an advancing terminus at 30 m per day[62–64] and record the temporal evolution of the buttressing force exerted on the terminus. We present modeling results after 16 days of terminus motion, when the mélange motion has approximately reached a steady state. In a series of simulations, we vary the initial mélange thickness to determine its influence on the steady-state buttressing force. To validate the modeled mélange thickness with observations, we identify ArcticDEM strips at 2-m resolution that cover the mélange regions in front of Helheim and Kangerlussuaq glacier termini (see section "Calving dynamics associated with m´elange buttressing force seasonality across Greenland glacier termini in 2013–2022" and "Terrestrial laser scanner data and uncertainty assessment" for details). The steady-state mélange thickness at the terminus varies from 40 to 135 m at Helheim fjord, and 60–240 m at Kangerlussuaq fjord, that are consistent with ArcticDEM observations (Table 2 in SI). We calculate the two-day averaged velocity of each iceberg element by dividing the iceberg's displacement between 14 and 16 days of terminus motion by the time interval (2 days). The mélange near the terminus moves at the terminus velocity with shear bands developed at fjord walls (Fig. 4c, f). The mélange near the open end becomes loosely packed and more fluidic (See supplementary video for the full temporal evolution of the mélange behaviors.) The modeled velocity field showcases both uniform (Fig. 4c) and extensional (Fig. 4f) flow regimes that are consistent with remote observations[29]. See Fig. 6 in SI for the modeled velocity fields, including instantaneous velocity, velocity averaged over 1 h, 1 day, and 2 days, and the satellite-derived velocity fields of the mélange[29].

The modeled mélange thickness profiles generally align with observations from ArcticDEMs on 17 Aug 2014 at Helheim (Fig. 4a, g) and on 12 Apr 2018 at Kangerlussuaq (Fig. 4d, h). In addition, we conducted another simulation with more icebergs confined in Kangerlussuaq fjord, to produce a thicker steady state mélange thickness profile as observed on 5 May 2020. We set the thickness threshold ($H_L$) as $36 \pm 10$ m for mélange to be considered as three-dimensional, and retrieve the corresponding mélange length ($L$) that varies from 7 to 12 km. The non-dimensional mélange thickness profile from three ArcticDEM strips and the discrete element models collapse onto the exponential analytical solution in Eq. (7), with $\mu_e$ fitted to be 0.3–0.4 for Helheim fjord and 0.4–0.8 for Kangerlussuaq fjord (Fig. 4i, see Table 1 in SI for details). The Kangerlussuaq fjord has a larger $\mu_e$ due to its more rugged fjord geometries.

To further explore the effect of fjord frictional properties on the mélange buttressing force, we adopt two simplified channel configurations that conceptualize complex Greenland fjord geometries. (1) The straight channel configuration (Fig. 5b, d, g, i) has a constant-width fjord. (2) The rugged channel configuration (Fig. 5c, e, h, j) has uniformly-spaced bulges on both sides[28], which can be more clearly seen from a perspective view of the simulation (Fig. 7 in SI). We present modeling results for a thin and thick layer of ice mélange in Fig. 5, initialized with a uniform initial thickness, $H_{ini}$, set to 60 m and 378 m, respectively. The upper limit of the initial thickness refers to the unusually thick mélange wedge observed over 1-month period in early summer 2016 at Jakobshavn Isbræ, which has a thickness of 400 m over one-fifth of the fjord width[23]. A summary of the modeling parameters is given in Table 1. At steady state, the thin layer of mélange expands into a two-dimensional monolayer (Fig. 5b, c). The mélange thickness at a specific position reflects the height of an individual iceberg, which varies in space. The

**Table 1 | Modeling parameters for the three-dimensional discrete element model with simplified fjord geometries**

| Symbol | Value | Unit | Variable |
|---|---|---|---|
| $L$ | 3 | km | Initial length of the ice mélange |
| $H_{ini}$ | [30, 380] with a mean step size 15 | m | Initial thickness of the ice mélange |
| $N_p$ | [1634, 15,264] with a mean step size 1238 | | Total number of icebergs in a simulation |
| $W$ | 1 | km | Fjord width |
| $V_{ter}$ | 43.2 | m/day | Terminus velocity |
| $C_w$ | 0.5 | | Dimensionless drag coefficient for icebergs in seawater |
| $E$ | 2.6 | MPa | Iceberg elastic modulus |
| $a_{min}$ | 17.7 | m | Minimum side length of a cubic iceberg |
| $a_{max}$ | 141.4 | m | Maximum side length of a cubic iceberg |
| $d_r$ | 150 | m | Spacing between bulges on the rugged wall |
| $a_r$ | 60 | m | Side length of bulges on the rugged wall |
| $h_r$ | 20 | m | Thickness of bulges on the rugged wall |
| $\Delta t$ | 0.1 | s | Modeling time step |
| $\delta t_{buoy}$ | 5 | s | Time step to update the buoyant force for icebergs |
| $\mu$ | 0.3 | | Kinetic friction coefficient between the particle and the wall |
| $\mu_p$ | 1.0 | | Kinetic friction coefficient between particles |
| $\beta$ | 0.7 | | Iceberg critical damping ratio |
| $v$ | 0.3 | | Iceberg Poisson's Ratio |
| $\rho_i$ | 910 | kg/m³ | Iceberg density |
| $\rho_w$ | 1028 | kg/m³ | Seawater density |

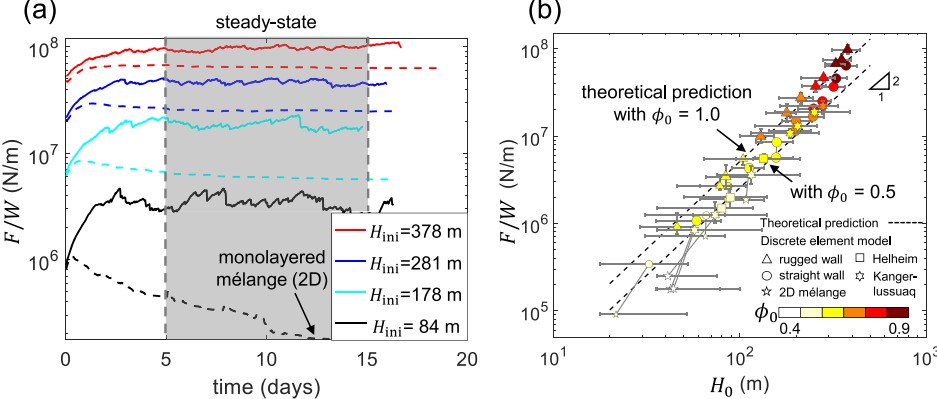

**Fig. 6 | Comparison between discrete element model and continuum predictions of the mélange buttressing force. a** The temporal evolution of the width-averaged mélange buttressing force, $F/W$, during the terminus motion for straight (dashed lines) and rugged (solid lines) fjord walls. The red, blue, cyan, and black colors correspond to mélange with initial thicknesses, $H_{ini}$ = 378 m, 281 m, 178 m, 84 m, respectively. Simulations reach the steady state after 5 days, except for the thinnest mélange ($H_{ini}$ = 84 m). **b** Steady state buttressing force, $F/W$, as a function of steady-state mélange thickness at the terminus, $H_0$. Different markers represent simulations with different fjord geometries. Square markers represent Helheim fjord, hexagram markers represent Kangerlussuaq fjord, circular markers represent straight fjords, and triangular markers represent rugged fjords. The smaller markers indicate simulations with smaller icebergs (half of the original size). $F/W$ is obtained by averaging the total buttressing force on the terminus over the terminus width during simulation time 5–15 days. The marker shows the averaged steady-state value of $F/W$, with a vertical error bar showing its fluctuation. $H_0$ is obtained by averaging the mélange thickness within 200 m of the terminus and over the terminus width. The marker shows the averaged steady-state value of $H_0$, with a horizontal error bar showing its variation over the terminus width brought by the iceberg size polydispersity. For simulations where the mélange collapse into monolayers at the end, we plot both the peak and minimal $F/W$ values and connect them by gray lines. The minimal $F/W$ values for monolayered, two-dimensional (2D) mélange are shown by pentagram markers. All markers are colored by the mélange packing density at the terminus at steady state, ranging from 0.4 to 0.9. The dashed lines represent Eq. (5) with the mélange packing density at the terminus, $\phi_0$ = 0.5 and 1.0, respectively.

thick layer of mélange collapses into a three-dimensional granular heap with a thickness gradient (Fig. 5g, h). The mélange thickness decreases with the distance from terminus, and becomes a two-dimensional monolayer at the open end in the ocean. In the straight channel configuration, the mélange behaves like plug flow with a uniform velocity profile within the fjord (Fig. 5d, i). In the rugged channel configuration, the mélange exhibits shear bands near fjord boundaries with uniform flow near the terminus (Fig. 5e, j), which has also been reported in previous studies[28,29]. The mélange switches between the jammed and

unjammed state, as evidenced by noticeable fluctuations in the velocity and the buttressing force (See Supplementary videos for the full temporal evolution of the mélange behaviors).

**Mélange buttressing force increases with mélange thickness at the glacier terminus**

We present the temporal evolution of the buttressing force for mélange with different initial thicknesses in straight and rugged channel configurations (Fig. 6a). For the same initial mélange thickness, the

buttressing force is always larger in rugged channels (solid lines) than that in straight channels (dashed lines). The bulges in rugged channels increase the shear resistance from fjord walls, which results in larger buttressing forces exerted on the advancing terminus. This is also evidenced by the difference in the mélange length at steady state (Fig. 5). The mélange has a smaller length when confined within rugged channels compared with straight channels. By conservation of mass, the mélange has to be either thicker or more densely-packed (or both) within rugged channels, which leads to a larger buttressing force as predicted in Eq. (5). The thickness and buttressing force of most mélange reach steady-state values after 5 days of simulation. Therefore, we take the time window 5–15 days to calculate their averaged steady-state values.

To validate the continuum theory (Eq. (5)), we plot the steady state buttressing force and mélange thickness at the terminus for all simulations in Fig. 6b. We calculate the averaged steady state buttressing force over the fjord width ($F/W$) with force fluctuations indicated as vertical error bars. We calculate the averaged mélange thickness within 200 m of the terminus ($H_0$), with thickness variations indicated as horizontal error bars. We also compute the packing density of the mélange within 200 m of the terminus ($\phi_0$) and color data markers by the magnitude of $\phi_0$. For simulations that start with thin mélange and collapse into monolayers at the end, we plot both the minimum and maximum $F/W$ values and connect them by gray lines. We compare the buttressing force predicted by the continuum equation (5) with simulations. The modeled buttressing force slightly deviates from the continuum prediction due to extra buttressing force induced by compressional flow that exists in simulations but has been neglected in Eq. (5). However, the overall good match between modeling results and the continuum prediction shows that Eq. (5) is robust and the glaciostatic pressure outweighs deviatoric stresses. A simple scaling analysis between glaciostatic pressure and fjord friction further shows that the mélange viscosity is around $2 \times 10^{10}$ Pa·s (Section 1 in SI), which validates the assumption ($\eta < 10^{11}$ Pa·s) underlying Eq. (5). For the six cases where the mélange collapses into thin monolayers at the end of the simulation (denoted as pentagram markers), the final buttressing forces can be predicted well by the previously developed theory for mélange of a uniform thickness[28] with the yield stress parameter, $\sigma_0$, fitted to be 0.12–0.16 kPa. The modeled buttressing forces in these cases are smaller than in the three-dimensional continuum (Eq. (5); black lines in Fig. 6), because the mélange only has a monolayer and violates the assumption of three-dimensional mélange with a constant packing density throughout its depth. Our modeling results confirm that, for both realistic fjord geometries (Helheim or Kangerlussuaq) and simplified geometries (straight or rugged), the thickness of the mélange at the terminus indicates its buttressing force. As the fjord friction increases, fjords are able to pile up thicker and denser mélange at the glacier terminus. The robustness of Eq. (5) with different fjord properties is the key to interpreting field observations across Greenland glacier termini.

## Calving dynamics associated with mélange buttressing force seasonality across 32 Greenland glacier termini in 2013–2022

Our models reveal that the mélange buttressing force can be predicted solely from remote sensing observations of its thickness at glacier terminus (Eq. 5). However, further investigation is needed to address the question of how does the spatio-temporal variations in mélange buttressing force correlate with calving dynamics in Greenland. Recent studies covering the period from 2015 to 2021 found that among 219 marine-terminating glaciers in Greenland, nearly 80% of them showed significant seasonal variations in terminus position, which retreat in summer and advance in winter[65]. We hypothesize that the seasonal terminus-position variability could be induced by a mélange buttressing force seasonality. To test this hypothesis, we collect available

ArcticDEM strips at Jakobshavn Isbræ in the past decade, and compare DEM acquisition dates to a time series of the terminus position (Fig. 7a). Among the eight DEM strips, five of them (dashed lines) are acquired in late spring-to-summer (May–Aug) when the terminus retreats, and three of them (solid lines) are acquired in fall-to-early spring (Sep–Mar) when the terminus advances. From the corresponding mélange elevation profiles constructed the same way as in Fig. 1 (solid blue lines in Fig. 7f–i), we first confirm that they are not contaminated by large icebergs whose elevation values are above 30 m (Fig. 7b–e). The elevation profile successfully reflects the overall thickness variations within the mélange that piled up from small icebergs. We observe that the freeboard height of the mélange at the terminus ranges from 2.8 to 3.9 m in summer and 19.2 to 26.8 m in fall-to-spring.

We then extend our study to 32 glacier termini, most of which (ID 1–25) are picked from previous studies with strong terminus-position seasonality[36,37], and the rest (ID 26–32) have annual ice discharge larger than 5 Gt/yr[40]. The locations of the termini are marked on a Greenland velocity map in Fig. 8a. We identify 341 ArcticDEM strips at 2-m resolution that cover the mélange regions for the 32 studied termini. For each DEM strip, we investigate terminus position variations[38] during a 2-month time window centering on the DEM acquisition date. If the terminus keeps advancing (or retreating) within the time window, then the DEM potentially represents mélange with a strong (or weak) buttressing force. If the terminus alternates between advancing and retreating within the time window, we discard the corresponding DEM strip because the relationship between mélange and calving dynamics is ambiguous in this case. After filtering all DEM strips through this criterion, we identify 60 DEM strips during terminus advances, and 50 DEM strips during terminus retreats, from February to November 2013–2022. Figure 8 in SI presents an example of the DEM filtering procedure at Helheim Glacier, resulting in two TLS-derived DEMs during terminus advances (Fig. 1c, e), and two ArcticDEM strips during terminus retreats. For every ArcticDEM strip, the mélange thickness is defined as the maximum value obtained from the representative mélange thickness profile within a 200-meter range from the terminus. Table 2 in SI summarizes the observed minimum (or maximum) mélange thickness when terminus retreats (or advances) as $H_0^{\min}$ (or $H_0^{\max}$), with the corresponding DEM acquisition month shown in the bracket.

We also present all observed mélange freeboard heights at the terminus ($Z_0$) in Fig. 8b. A complete catalog of terminus position variations, DEM acquisition dates and mélange freeboard heights for 32 studied termini is summarized in Figs. 17–48 in SI. Assuming the mélange to be densely-packed with $\phi_0 = 0.9^{+0.1}_{-0.26}$, ice density in the plausible range of $910^{+10}_{-40}$ kg/m$^3$ and water density of $1028^{+1}_{-8}$ kg/m$^3$, we arrive at the mélange buttressing force per unit width ($F/W$) through Eqn. (5). For the studied glacier termini, the observed mélange thicknesses when terminus advances (82% in late fall-to-spring, November to May) range from $49^{+19}_{-19}$ m to $240^{+52}_{-69}$ m, with buttressing forces ranging from $1.1^{+0.9}_{-0.7} \times 10^6$ N/m to $2.7^{+11}_{-1.4} \times 10^7$ N/m. Previous force balance analysis of a calving iceberg revealed that for a terminus at floatation, the mélange buttressing force of order $-1.0 \times 10^7$ N/m is sufficient to inhibit calving by preventing iceberg rotation[24]. Finite element models suggested that the mélange buttressing force of this magnitude can also inhibit calving by suppressing fracture propagation[4,25,26,33,34]. Most of our inferred buttressing forces during terminus advance are consistent with the proposed threshold. The observed mélange thicknesses when terminus retreats (78% in summer-to-fall, June–October) range from $1^{+11}_{-1}$ m to $87^{+26}_{-29}$ m, with inferred buttressing forces ranging from $0.1^{+6.2}_{-0.1} \times 10^4$ N/m to $3.5^{+2.1}_{-2.1} \times 10^6$ N/m. Therefore in summer and fall, mélange is generally too thin to inhibit calving. Here we report the buttressing force in N/m, which can be compared to other studies including field observations[57], simulations[27,28], and analytical analysis[24].

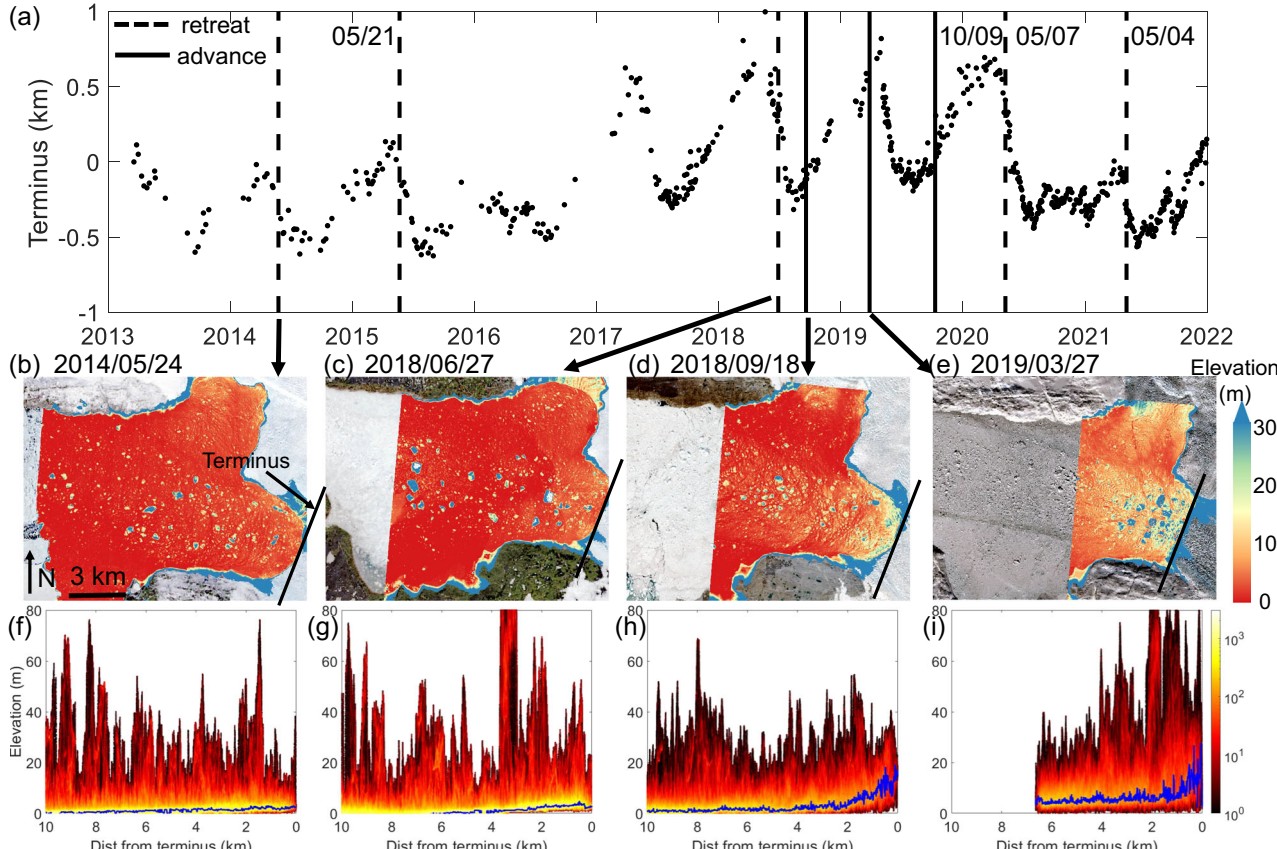

**Fig. 7 | Seasonal variations in mélange thickness coincide with calving dynamics at Jakobshavn Isbræ. a** Terminus position in 2013–2022[38] where positive sign indicates the direction of advance. Eight vertical black lines mark the acquisition dates of available ArcticDEMs, four of which are prensented in (**b**–**e**), corresponding to 24 May 2014, 27 Jun 2018, 18 Sep 2018, and 27 Mar 2019, respectively. Solid black lines mark the dates with terminus advances, and dashed black lines mark the dates with terminus retreats. **b**–**e** The mélange elevation above mean sea level from ArcticDEM strips, overlain on satellite images acquired around the same date. Black line across the fjord indicates the glacier front location. The images are in polar stereographic projection (EPSG: 3413). **f**–**i** Surface elevation profiles for the mélange displayed as density plots (8,649,023–18,183,005 data points in total) constructed the same way as in Fig. 1. For any specific distance from terminus, we find the elevation value that has the maximum number of data points. Solid blue lines connect these elevation values along the distance from terminus as the representative mélange elevation profiles. We observe thick mélange in fall-to-spring when terminus advances, and thin mélange in summer when terminus retreats.

## Discussion

Our discrete element model of ice mélange is composed of realistic cubic icebergs instead of disk-shaped grains, and showcases an exponential decay of the thickness profile in Helheim and Kangerlussuaq fjords, that is consistent with ArcticDEM observations and analytical predictions. Apart from the exponential decay, we also observe other shapes of the mélange thickness profile, such as a plateau near the terminus, a steep drop or a bulge at few kilometers away from the terminus, etc (See SI for ArcticDEM strips at 32 glacier termini). These shapes could be attributed to calving-induced jamming wave propagation, ice-ocean interactions, iceberg size distribution, heterogeneous friction, or cohesion within the mélange or at fjord walls, all of which cannot be captured by the simplified rheology and friction law underneath Eq. (7). Coupled with computational fluid mechanics, our discrete element model can be used to further explore how the mélange thickness at the terminus evolves with ice-ocean interactions that influence calving dynamics, including ocean tides[14], ocean warming[20,21,39], and subglacial plumes[13,66].

Previous research suggests that the presence of ice mélange can reduce iceberg calving by providing "backstress" to the terminus[17–19,22–29]. Our comparisons of time-varying mélange thickness and calving dynamics at Helheim Glacier (Fig. 1b) support the view that the buttressing force increases with the mélange thickness and thus inhibits calving. Our TLS scans indicate a correlation between mélange thickness and calving dynamics, but we cannot determine causality before examining other mechanisms driving calving dynamics. Though we have observed mélange thinning prior to calving (Fig. 1h, i), the mélange thickness and the terminus may react simultaneously but independently to other oceanic and atmospheric forcing. Scanning through 108 ArcticDEM strips, we discover calving dynamics associated with mélange thickness seasonality across 32 Greenland glacier termini in 2013–2022. When termini advance in late fall-to-spring (Nov-May), the average value of all observed mélange thicknesses is $119^{+31}_{-37}$ m, with a corresponding buttressing force $6.5^{+3.4}_{-3.7} \times 10^6$ N/m. When termini retreat in summer-to-fall (Jun-Oct), the average thickness is $34^{+17}_{-15}$ m, with a corresponding buttressing force of $5.2^{+5.9}_{-3.8} \times 10^5$ N/m.

While we have observed strong evidence of correlations between mélange thickness and terminus seasonality, understanding their causality requires considerations of other environmental forcings. Previous research shows that seasonal terminus positions for some central west Greenland glaciers with small-magnitude calving events correlate stronger with glacial runoff than mélange presence or ocean thermal forcing[67]. On the other hand, researchers have observed slowdown and thickening of Jakobshavn since 2016 and attribute it to concurrent cooling of ocean waters[68]. Analytical and numerical models imply that submarine melting can amplify calving by melt-undercutting[7,69]. We note that if submarine melting causes the observed summer thinning of mélange, mélange's buttressing

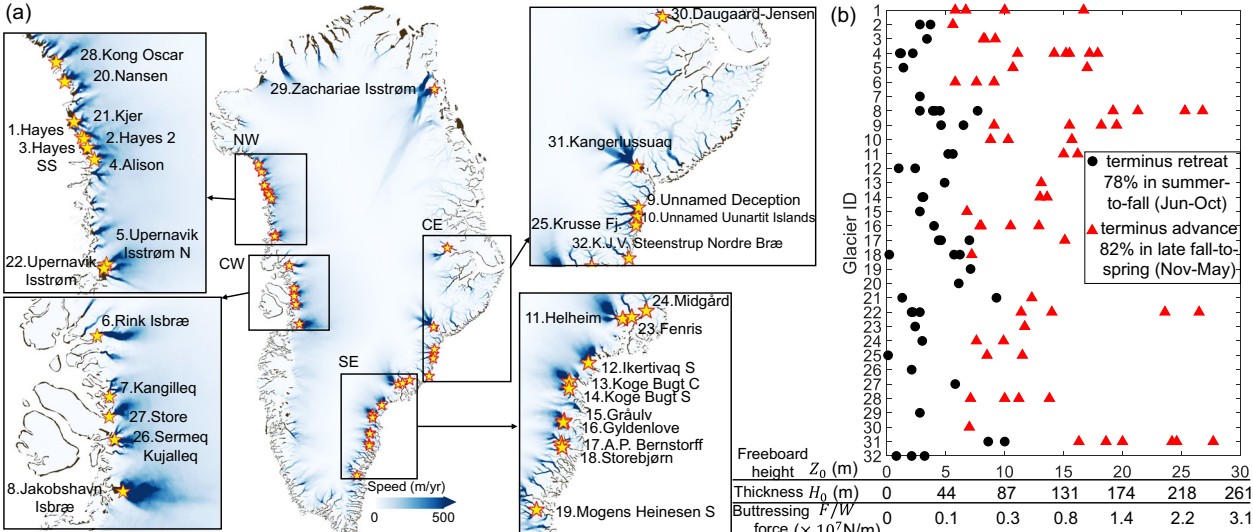

**Fig. 8 | Seasonal changes of mélange thickness and buttressing forces across 32 Greenland glacier termini in 2013–2022. a** Locations of the 32 studied glacier termini are shown as pendular markers on the background Greenland ice velocity map[85–87]. We also present zoomed in views of the studied glacier termini in northwest (NW), central west (CW), southeast (SE), and central east (CE) regions of Greenland. **b** From 2013 to 2022, the observed mélange freeboard heights at the terminus ($Z_0$) from all available ArcticDEM (108 in total). Red triangular markers correspond to DEM acquired when terminus advances (82% in late fall-to-spring, November to May), and black circular markers correspond to DEM acquired when terminus retreats (78% in summer-to-fall, June to October). The horizontal axis contains three variables: the mélange freeboard height directly retrieved from DEM ($Z_0$), the inferred mélange thickness from hydrostatic equilibrium ($H_0$), and the inferred mélange buttressing force from Eq. (5) ($F/W$). Data used to calculate the buttressing forces and their uncertainties are listed in Table 2 in SI.

strength can be strongly tied to submarine melting. The impact of submarine melting on mélange strength can be significant due to the strong dependence of buttressing on mélange thickness inferred in our study.

We note that the hypothesis of summer-runoff induced calving, on its own, can not explain the observations at Koge Bugt C Glacier (ID = 13 in Fig. 8) where the terminus advanced from July to September in 2015 and retreated or remained a plateau during the same period in 2014, 2016, 2017, and 2020[38]. We attribute the summertime terminus advance in 2015 to mélange buttressing from the presence of unusually thick mélange observed on 2 Jul with a thickness of $114^{+30}_{-36}$ m (see Fig. 9 in SI for details), the same as what happened at Jakobshavn Isbræ in Jun 2016[23]. If calving dynamics are controlled by mélange buttressing, then our analysis infers that the minimum buttressing force required to inhibit calving varies across termini from $1.1^{+0.9}_{-0.7} \times 10^6$ (Hayes Glacier 2) to $9.3^{+4.6}_{-5.2} \times 10^6$ N/m (Kangerlussuaq Glacier). Such variations in the buttressing threshold could be attributed to spatial variations in ice velocities, terminus geometry, bed topography, basal friction, oceanic and atmospheric forcings, etc. Our analysis offers a framework to mechanistically study the effects of mélange buttressing and other ice-ocean interactions on calving.

In summary, our continuum and discrete element models offer a way to estimate the mélange buttressing force with a single measurement: freeboard height (or thickness) of the mélange at the terminus. Our data analysis shows that mélange thickness seasonality strongly correlates with calving dynamics across Greenland. As termini keep retreating inland, the emergence of longer fjords could retain more icebergs and potentially enhance mélange thickness (especially in winter), which could slow down the process of overall termini retreat, as has been observed at Steenstrup[39]. Given that mélange thickness dictates its buttressing force, the impacts of submarine melting and subglacial discharge on calving will be amplified by melting and thinning the mélange. On the other hand, cooler ocean and air temperatures in winter enhance mélange rigidity[21], making it easier to pile up thick mélange at the terminus to provide buttressing. How warmer oceans and atmospheric influence the mélange strength is the subject of future work. Lastly, our models provide a simple way to incorporate

mélange effects into large-scale numerical ice sheets models. Knowing the mélange thickness at the terminus, the mélange buttressing force can be calculated by Eq. (5) and imposed as the boundary condition for ice sheet models. Our result indicates that climate change, manifested in lengthening summer seasons, can weaken the mélange buttressing effect, accelerating terminus recession and ice mass loss at tidewater glaciers in Greenland.

## Methods

### Terrestrial laser scanner data and uncertainty assessment

The TLS is located at a latitude of 66.32963°N and a longitude of −38.1739° W, with the location provided in Fig. 1a and is -1–5.8 km from Helheim Glacier's calving face. TLS point clouds are ellipsoidal heights. After removing the geoid height at Helheim Glacier[42], the mélange surface elevations are heights above mean sea level with tidal variations[23]. We then remove the tidal trend of mélange elevations with a tidal model[43]. ATLAS-generated point clouds were gridded at 100 m × 100 m resolution to ensure sufficient point densities per grid cell using the Point Cloud Data Abstraction Library[70] for DEM creation. The resulting DEMs contain a minimum, maximum, and average band where each point which falls into a 100 m × $\sqrt{2}$ radius contributes to a grid cell. Generally, five main sources of uncertainty exist when using TLS data. These sources being registration, atmospheric conditions, scanning geometry, instrument and hardware limitations, rasterization, and surface reflectance properties[71]. The TLS was tied to the global spatial reference frame using an array of 5 cm cylindrical reflective targets. The centroids of these cylinders were surveyed using GNSS receivers, and the reflective exterior surfaces of the cylinders were scanned with the LiDAR scanner. The position and orientation of the scanner were then adjusted to bring the LiDAR-derived cylinder centroids and the GNSS-derived cylinder positions into optimal alignment using a least squares method. This error for the 2019 ATLAS south scanning epoch was 2 cm. Following the uncertainty methods from ref. 71, the total uncertainty per grid cell is defined by $\sigma_{total} = \frac{\sigma_{point}}{\sqrt{\rho_{point}}} + \sigma_{reg}$. From our uncertainty analysis, our REIGL vz6000's uncertainty is dominated by the registration error if the

point density is high. $\sigma_{point}$ depends on instrument hardware error $\sigma_{instrument}$, beam geometry $\sigma_{geo}$, and atmospheric conditions $\sigma_{atm}$. For a REIGL vz-6000 instrument, errors can range from 0.09–0.64[71], depending on the range. Uncertainty varies with range mainly for two reasons: 1) instrument uncertainty is defined by $R \cdot \sin(0.008^\circ)$ where $R$ is the distance the laser pulse traveled to the target surface, and 2) scanning geometry also depends on scanning distance as the laser footprint expands over distance. More about general terrestrial laser scanner uncertainty can be read in ref. 71.

Following the methods outlined in ref. 71, our vertical uncertainty ranges from ~2 to 4.5 cm depending on the distance and point density. Using the upper range of point densities in a 100 m × 100 m grid cell in the melange region consisting of 10,000 points, the uncertainty is closer to 2 cm while the lower end of point density (≈2000 points) will have an uncertainty closer to 4.5 cm. We also vary the $\sigma_{geo}$ uncertainty from 5 to 50 cm, which the upper limit is double the[71] values in our assessment.

To construct the representative mélange elevation profile, we first connect the median or mean elevation values along the distance from the terminus, the shape of which turns out to be sensitive to large icebergs with elevation values larger than 30 m (Fig. 2 in SI). Incorporating large icebergs into the representative mélange elevation profile has several drawbacks. Firstly, instead of representing the elevation piled up from small icebergs and sea ice, it tends to reflect the size distribution of large icebergs. Secondly, the existence and location of large icebergs are sporadic and hard to characterize by a continuum model. Lastly, comparing mélange elevation profiles across different Greenland fjords becomes challenging, as the size and shape distribution of large icebergs are sensitive to calving styles[72] and terminus geometries. To reflect the mélange elevation that is piled up from small icebergs and sea ice, we connect elevation values that have maximum numbers of data points along the distance from terminus as the representative mélange elevation profiles (solid blue lines in Fig. 1c–f in the main text and Fig. 2b, e in SI).

## ICESat-2 data and uncertainty assessment

We use the ATL06 data set from ICESat-2 that provides geolocated, mean land-ice surface heights above the WGS 84 ellipsoid that are averaged along 40 m segments of ground track and spaced 20 m apart[41]. The temporal resolution is 91 days from 14 October 2018 to present. We compute the mélange surface elevation after accounting for the local difference between the ellipsoid and the EIGEN-6C4 geoid[42], and then remove the tidal signal from the surface elevation[43]. We identify three ICESat-2 tracks for Jakobshavn Isbræ, Kangerlussuaq Glacier, and Store Glacier, respectively, and use data from strong beams to compose the mélange elevation profile (Fig. 2). The averaged standard error in the reported elevation data ranges from 0.02 to 0.52 m due to sampling error and first-photon bias correction from the land ice algorithm, as described in the algorithm theoretical basis document (ATBD) for land ice along-track height product (ATL06)[73]. To account for the uncertainty from geoid and tidal corrections, we identify ICESat-2 tracks passing through the ocean area near the three glaciers presented in Fig. 2 and compute the difference between ATL06 measurements over the ocean and the geoid[42] with tidal corrections[43]. We found that the mean error is 0.37 m at Jakobshavn Isbræ, 0.50 m at Kangerlussuaq Glacier, and 0.60 m at Store Glacier (Fig. 10 in SI). In addition, we compare the mélange surface elevation acquired on 10 Sep 2019 from TLS and ICESat-2 at Helheim Glacier (Fig. 11 in SI). The TLS data was acquired at 13:01:56 UTC, and the ICESat-2 data was acquired after 90 min, at 14:32:53 UTC. Overall, the mélange surface elevation profiles acquired from TLS and ICESat-2 show good agreement with a mean absolute difference of 0.94 m at the points of overlap. The mismatch at a distance around 2.5 km might come from mélange motion within the 90 min time window. In

addition, the TLS scans are oblique to the surface and not straight down, so TLS and ICESat-2 do not measure the exact same surface. All these sources of uncertainty are an order of magnitude less than the observed temporal changes in mélange freeboard height presented in Fig. 2, suggesting that our interpretation of seasonal changes in mélange thickness from ICESat-2 is robust.

## ArcticDEM data and uncertainty assessment

Since the DEM strips can have vertical offsets of up to 4 m[74], we co-registered them using two methods to reduce vertical uncertainty. For the 63 DEM strips covering both mélange and the ocean, we adopted the sea level calculation method in[50] by plotting DEM elevation values above the WGS84 ellipsoid in a histogram with 0.25 m bin widths. The peak elevation value (i.e., the most common elevation in the DEM) was supposed to be sea level at the time when the DEM was acquired[50]. The DEM strip was registered to the detected sea level by subtracting the peak elevation value from elevation values relative to the WGS84 ellipsoid. The accuracy of the elevation values above sea level over non-mélange areas varied from 0.13 m to 0.37 m for a DEM strip acquired on 11 Jun 2014 at Helheim Glacier (See Fig. 12 in SI for details). For the 45 DEM strips covering mélange only, we registered each DEM strip with the mosaic DEM[75], which has been registered to ICESat-2. For each glacier terminus, we selected line segments on neighboring rock (Figs. 17–48 in SI) and calculated averaged elevation offsets between individual DEM strips and the mosaic DEM along these line segments. After applying the elevation offset and subtracting the geoid from the ellipsoid with the tidal correction[43], we plotted DEM elevation values above sea level in a histogram with 0.25 m bin widths, making sure its peak was larger than zero at the time when the DEM was acquired[50]. In summary, the elevation offsets applied to the 108 DEM strips were $0.38 \pm 2.23$ m. With this protocol, the elevation accuracy of the DEM strip segment improved from 4 m[74] to 1.06 m[75,76]. In addition, we compare the mélange surface elevation acquired on 10 May 2020 from TLS and ArcticDEM at Helheim Glacier (Fig. 13 in SI). The TLS data was acquired at 13:01:57 UTC, and the ArcticDEM data were acquired after 40 min, at 13:42:45 UTC. Overall, the mélange surface elevation profiles acquired from TLS and ArcticDEM show good agreement with a mean absolute difference of 0.66 m at the points of overlap.

For each glacier terminus, we digitized terminus positions using ArcticDEMs on the dates when the data was acquired. For mélange of length 15 km and width 4 km, there were ~15,000,000 data points available. For each data point in a DEM strip, we calculated its distance from terminus and the surface elevation value after applying the elevation offset. After picking specific values for the number of horizontal and vertical bins, we displayed all data points in a density map where surface elevation was plotted as a function of distance from terminus (Fig. 14 in SI). The accuracy from varying the number of bins of density maps ranged from 0.11 to 0.27 m (Fig. 15 in SI). For any specific distance from terminus, we find the elevation value that had the maximum number of data points (Fig. 14 in SI). We then connected these values along the distance from terminus as the representative mélange elevation profiles (solid blue lines in Fig. 7), $Z(x)$. We calculated the maximum mélange elevation within 200 m from the terminus as $Z_0$. The value $Z_0$ was further divided by $1 - \rho_i/\rho_w$ to obtain the mélange thickness, $H_0$, which was used for calculating the buttressing force, $F/W$, based on Eq. (5). In Table 2 in SI, we report the thickness uncertainty arising from ArcticDEM ($\pm 1.06$ m), ice ($910^{+10}_{-40}$ kg/m³) and water ($1028^{+1}_{-8}$ kg/m³) densities. The uncertainties in ice and water densities, mélange packing density ($\phi_0 = 0.9^{+0.1}_{-0.26}$), and the mélange thickness fed into Eq. (5) to obtain the uncertainty in the buttressing force, $F/W$.

## The three-dimensional discrete element model for ice mélange

We develop a three-dimensional discrete element model for ice mélange with commercial software, PFC3D®[77]. We use the same Cartesian coordinate system as in section "A three-dimensional continuum

model of ice m´elange", with $x$ starting from the terminus and in the direction along the fjord, $y$ in the direction across the fjord, and $z$ in the vertical direction with $z = 0$ at sea level. Iceberg interactions are simulated using a classical Hertzian model for elastic contact between disks with a Coulomb friction law and viscoelastic damping to maintain stability[28]. The Poisson ratio of the particle is set as 0.3[78–80]. The Young's modulus of the particle is set as 2.6 MPa, which gives rise to 1–6% particle overlap with the commonly-observed mélange thickness. We confirm that such overlap does not affect macroscopic behaviors by running a simulation with a Young's modulus of 100 MPa, a mélange thickness of 260 m confined in straight fjords, resulting in a buttressing force difference of less than 7%, that is smaller than steady state force fluctuations. We use a smaller Young's modulus to save computational time, as has been done in sea ice models[80]. The iceberg damping ratio for the viscous dashpot is set as 0.7, taking an uncertainty range of ±0.2 gives rise to a buttressing force difference of less than 8%. The kinetic friction coefficient between the particle and the wall, $\mu = 0.3$, is adopted from[28]. The kinetic friction coefficient between particles is set to $\mu_p = 1.0$. The particles also experience small viscous drag force that is proportional to the iceberg velocity to represent hydrodynamic drag from seawater. The drag coefficient is set to $C_w = 0.5$. Varying the value of $C_w$ from 0.01, 0.1, 0.5, 1.0, to 3.0 gives rise to a buttressing force difference of less than 9% that is smaller than steady state force fluctuations. To impose buoyant force on an individual iceberg, we need to identify its relative position to the sea water level, which is prescribed at $z = 0$. As it is computationally expensive to compute the indentation of a cubic particle into a plane, we instead use a surrogate sphere that has the same center positions and volume of the cubic particle for buoyancy calculations. Assuming the side length of the cubic particle is $a$, then the surrogate sphere has the radius, $r = (\frac{3a^3}{4\pi})^{1/3}$. We calculate the immersed volume of the surrogate sphere in the seawater and obtain the corresponding buoyant force on an individual cubic iceberg. As positions of icebergs are evolving during simulations, we update their buoyant forces on a regular basis, $\delta t_{buoy} = 5$ s. The mechanical timestep is chosen to be the same as in previous two-dimensional discrete element model[28] to maintain mechanical stability, $\Delta t = 0.1$ s.

We use cubic grains which can achieve a higher packing density, thus buttressing forces, than disk-shaped grains. We adopt the iceberg area distribution observed in the mélange of Jakobshavn Isbræ and Helheim Glacier, which is approximated as a power-law distribution with an exponent of −2.0[50,51]. The resulting iceberg size distribution for cubic grains is a power-law distribution with an exponent of −4.0. Taking the simulation with the Helheim fjord for instance (Fig. 4b), the side lengths of cubic icebergs are 36 m, 50 m, 75 m, 100 m, 200 m, and the corresponding numbers of particles are 3120, 838, 166, 52, 3, respectively. To model mélange with different steady-state thicknesses, the total number of particles varies from 2918 to 12,522 for simulations with Helheim and Kangerlussuaq fjords. In simplified fjord geometries, the side lengths of cubic icebergs are 35.4 m, 50 m, 70.7 m, 100 m, 141.4 m, and the corresponding numbers of particles for the thick mélange are 8190, 2045, 510, 125, 30, respectively (Fig. 5f). We vary the total number of particles from 1634 to 15,264 to change the mélange thickness. To confirm that the modeling results are invariant to the particle size, we conduct six more simulations with smaller icebergs, whose sizes are half of original sizes and range from 17.7 m to 70.7 m (small markers in Fig. 6b).

To construct the initial mélange state, we divide the total number of particles into three equal batches. In each batch, iceberg sizes are randomly drawn from the distribution described above. We put a right boundary wall at distance $L$ from the terminus on the left to prescribe the initial length of the mélange. The mélange is confined in $y$ direction by two side walls representing fjords at a distance $W$. To explore the influence of fjord friction properties on mélange behaviors, we have Helheim, Kangerlussuaq, and simplified (straight and rugged) channel

configurations. All configurations have the same kinetic friction coefficient, $\mu$, and rugged channels have cuboid bulges of dimension $a_r \times a_r \times h_r$ that are uniformly spaced at $d_r$ in x and z directions. We deposit icebergs in each batch from the same height and then they settle under gravity and buoyancy. Following pouring, the entire array of cubic particles is permitted to settle until static equilibrium is achieved, as shown in Fig. 5a, f. We then delete the right boundary wall so that the mélange has an open end in the ocean. We move the terminus on the left at a constant velocity, $V_{ter} = 30$ m/day for real fjord geometries[62–64], and 43.2 m/day for simplified fjord geometries[27,28]. To confirm that the averaged steady-state buttressing force is invariant to the terminus velocity, we conducted simulations with mélange thickness 280 m, terminus velocity at 21.6 m/day, 43.2 m/day, and 86.4 m/day, for both straight fjords and rugged fjords configurations (Fig. 16 in SI). The results show that the averaged buttressing force is mostly invariant to the terminus velocity in both fjord configurations. Taking the force fluctuations into account, the maximum buttressing force difference among the chosen velocities is 4% and 8% for straight and rugged fjords, respectively. In rugged fjords, faster terminus motion leads to larger force fluctuations due to larger velocity gradient during stick-slip/jam-unjam cycles. We adopt the terminus velocity of 43.2 m/day to simulate mélange in simplified fjord geometries for the sake of computational efficiency.

## Estimating modeling mélange thickness and packing density at glacier termini and uncertainty assessment

As icebergs have a power-law size distribution, the thickness of mélange is a spatial variable in horizontal directions (x and y). We compute the mélange thickness at each particle position within a sampling cylinder of radius 80 m and capped by icebergs at the top and the bottom of the mélange. We then take an average of thickness values for icebergs within 200 m (i.e., 1/5 of the fjord width) from the terminus and display it as a marker in Fig. 6b, with the horizontal error bar denoting the minimum and maximum thickness values. Therefore, the reported uncertainty of the mélange thickness comes from the iceberg size polydispersity. In comparison, the mélange thickness uncertainty from doubling the sampling cylinder radius is below 15 m, which is smaller than the size of icebergs, and therefore is neglected here.

To compute the packing density of the mélange, we focus on its dependency along the fjord direction and set an interval size ($dx$) of 67 m. We compute the averaged mélange thickness at each interval with the aforementioned method and obtain $H(x)$. At each interval, we divide the total volume of icebergs by the total volume of the mélange ($H(x) \times W \times dx$) and obtain the packing density, $\phi(x)$. We then take an average of the first three intervals to output the packing density at the terminus, $\phi_0$. The uncertainty in $\phi_0$ by doubling the interval size is below 0.05 and therefore we only report the first decimal place for $\phi_0$ in Fig. 6b.

## Data availability

All data sets produced as part of this paper are available through Zenodo at https://zenodo.org/records/13382185[81]. These data include the following: a supplementary word file that lists the ArcticDEM tiles, ICESat-2 laser altimetry tracks, and the Landsat, Sentinel-1 and Sentinel-2 scene numbers used in this study, a supplementary excel file that lists elevation offsets applied on the 108 ArcticDEM strips with corresponding ArcticDEM strip acquisition dates and Mosaic DEM tiles for coregistration, ICESat-2 data cropped by the mélange region of interest for the three glaciers presented in Fig. 2, 32 zipped folders that contain ArcticDEM data cropped by the mélange region of interest for the presented 32 Greenland glacier termini in Fig. 8, and source codes to plot mélange surface elevation (or thickness) profile as a function of distance from terminus from ICESat-2 data and ArcticDEM strips. The map of Greenland in Figs. 1, 2, and 8 was created by overlying an ice mask from the QGreenland collection[82,83] onto the coastline of Greenland[84]. Landsat images were downloaded through the Amazon

Web Services (https://registry.opendata.aws/usgs-landsat). Copernicus Sentinel-1 data acquired in 2019 and 2021, and Copernicus Sentinel-2 data acquired in 2018 and 2019 were provided by European Space Agency and downloaded through the Amazon Web Services (https://registry.opendata.aws/sentinel-1, https://registry.opendata.aws/sentinel-2). TLS data for Helheim Glacier are available upon reasonable request. ICESat-2 laser altimetry tracks are available through the OpenAltimetry portal at https://openaltimetry.org/data/icesat2/ with download services provided by the National Snow and Ice Data Center[41]. ArcticDEM digital elevation models[74,75] are available from the University of Minnesota Polar Geospatial Center (PGC): https://www.pgc.umn.edu/data/arcticdem/. Ice surface velocity and BedMachine Greenland are freely available at the National Snow and Ice Data Center (NSIDC) at https://nsidc.org/data/nsidc-0725/versions/5[85–87] and https://nsidc.org/data/idbmg4/versions/5[88,89], respectively. The time series of Greenland terminus positions is available from ref. 38 at https://zenodo.org/records/10095674.

## Code availability
All source data and MATLAB codes to reproduce figures in the main text are available through Zenodo at https://zenodo.org/records/13382185[81]. The codes used for the three-dimensional discrete element model are available from the corresponding author upon reasonable request. PFC3D®[77] is a software from Itasca Consulting Group, Inc. through a commercial license.

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

## Acknowledgements
C.-Y.L. and Y.M. acknowledge funding from NSF's Office of Polar Programs through OPP-2235051. R.T.C. acknowledges support from the NASA Early Career Investigators Program under award 80NSSC24K1037. L.A.S. and M.G.S. acknowledge funding by the Heising Simons Foundation (HSF) through grant #2017-316. ATLAS instrumentation and processing pipeline was conducted in collaboration with colleagues at the Cold Regions Research and Experimental Laboratory, in particular David Finnegan, Adam LeWinter, and Howard Butler. J.C.B. and K.N. would like to acknowledge support from US National Science Foundation grant 2025795. Geospatial support for this work was provided by the Polar Geospatial Center under NSF-OPP awards 1043681, 1559691, and 2129685. DEMs were provided by the Polar Geospatial Center under NSF-OPP awards 1043681, 1559691, 1542736, 1810976, and 2129685.

## Author contributions
Y.M. led the project and the preparation of the manuscript. Y.M. and C.-Y.L. conceived the study. R.C. provided guidance on data processing from remote observations. M.G.S. and L.A.S. supplied and interpreted TLS data. C.-Y.L., J.B., and K.N. helped with the model development. Y.M., C.-Y.L., R.C., M.G.S., L.A.S., J.B., and K.N. contributed to the scientific interpretation of the results, and the writing of the manuscript.

## Competing interests
The authors declare no competing interests.
