## [Transparent Peer Review file · Nature Communications]

Seasonal Changes of Mélange Thickness Coincide With Greenland Calving Dynamics

Corresponding Author: Dr Yue Meng

Version 0:

Reviewer comments:

Reviewer #1

(Remarks to the Author)

Please see my comments in the attached PDF.

(Remarks on code availability)

I was able to view the code and metadata in the code ocean portal, but could not run the code without creating an account. After creating an account, I received "access denied" notices. While I was unable to directly run the code, the matlab scripts themselves appeared well commented and described, with previous run outputs and figures matching the anticipated format.

Reviewer #2

(Remarks to the Author)

I should start by saying that I mostly enjoyed reading this manuscript. Individual sections were generally well-written, both the observations and modeling approaches were interesting, and the conclusions are potentially quite important to the glaciological community. Having said that, I have several major concerns (see General comments below). The main one is that the manuscript is set-up as observationally-driven modeling study but does not actually use the observations to inform the modeling. I was expecting that the modeling would be carefully constrained with validated ice mélange thicknesses, observed iceberg distributions, and channel configurations that actually resembled real fjords. However, the manuscript never makes it clear where many of the forcing parameters come from and the observations are mostly used in a haphazard manner. Overall, I think this manuscript needs substantial reworking. I have provided some feedback for doing so in my General comments below.

General comments

(1) There is a general mismatch between observations and modeling. After several sections describing how mélange thickness was derived from field measurements and satellite data, the continuum model is just simply forced with mélange thickness of either 75 m or 200 m. There is no mention of why these values were chosen or if they were informed by observations. The discrete element model is then forced with different initial thicknesses of 60 m and 378 m. It is unclear why different values were chosen or whether these values represent Helheim Glacier or one of the many other glaciers mentioned in the manuscript (e.g. Jakobshavn Isbræ, Store, or Kangerlussuaq). After the all the hard work to observe ice mélange thickness in a remote fjord in East Greenland, surely the authors could at least use that data to justify the ice mélange thickness in the models.

(2) The manuscript lacks cohesiveness. This is related to the first point but more about the structure/readability rather than the research. There are five different approaches described in this study. The two modeling approaches (continuum and discrete element) and three observational approaches (TLS, ICESat-2, and ArcticDEM). Given the letter style of this manuscript, each approach has to be described in the Results section. This means that the narrative is interrupted in several places, making it difficult to follow the major arguments. Furthermore, because there are so many approaches, there are many places where there are not enough details. There are several key terms like terminus retreat, terminus quiescence, and major and minor calving events that are not adequately defined and would benefit from more complete descriptions.

(3) The observed mélange thicknesses are lacking validation. The TLS freeboards have stated accuracy of 0.10 m based on a “registration scan” which is never described. The only citation in this section is from a laser scanning study by a completely different group of researchers at Hintereisferner (Austria). The uncertainty from ICESat-2 freeboards is stated as “0.02 m~0.52 m”. It looks like this comes from the ATL06 ATBD but it is not clear. More effort went into correcting and validating ArcticDEM strips. But surely the mélange thicknesses from these different approaches overlap in space/time so it was unclear why they were not compared. Disappointingly, the mélange thicknesses were not provided in the Data Availability statement.

Specific comments

L28-29: Does this sentence refer to projections of Greenland’s contribution to sea level? Or both the Earth’s ice sheets? Given the previous sentences are about Greenland, it would make more sense to constrain this statement to Greenland as well.

L34: “calving retreats” is a strange term. Do you mean “calving front retreat”?

L36-38: It would be useful to clarify whether this is a finding from observations or modeling here. Or both? Secondly, has mélange rigidity actually been observed directly? If not, this would be a good place to clarify what previous observations have observed.

L39: Might use a different word to “buttressing” if you are going to define it in the next sentence.

L52-53: For readability, it would be useful to clarify what this statement implies.

L70-71: Please provide more details about this approach since it looks like the findings from the modeling will hinge upon accurate computation of mélange thickness.

L72-77: This seems unnecessarily complicated. Why not mean or median elevation value for each specific distance?

L78: Why are icebergs removed? Surely the discrete element model described later in the manuscript requires an iceberg size distribution?

L85-88: Can you be more specific about which data was used to quantify calving? It seems from the Xie et al. (2019) reference that this was done using the TLS scans. Regardless, the approach to classify calving events as “major” vs. “minor” seems a little subjective and challenging to replicate. It would be useful to include a figure showing what these two calving events look like in the TLS scans.

L87: How was terminus retreat defined and how was it quantified?

L89: How is this “terminus quiescence” defined? No major or minor calving events? Or just no major calving events.

L90-96: These are all very interesting findings but they need to be developed in more detail to be fully understood. See general comment 2.

L96-102: This kind of interpretation/reflection would probably be a better fit for the discussion section.

L103: Not sure what to suggest here but the abrupt transition from field measurements to satellite remote sensing observations interrupts the flow of the manuscript due to the mismatch in scale. This generally makes it challenging to follow the narrative throughout the manuscript.

L114-115: Which ICESat-2 product? How were freeboards computed? Was it the difference from a sea level product? Or track-to-track differences? Even though some of these details are provided later in the Methods section, more details are needed here to fully interpret the findings.

L116: Which years?

L119-120: Are these averages or ranges?

L127: Is this a new conclusion? If not, please include citations to studies who have already identified this behavior.

Fig. 2: It would be more appropriate to convert these y-axis values in panel D to ice mélange thickness, given that it is the important variable for the modeling/

L130: “quantifying the buttressing”

L175: Are the “winter velocity fields” of the mélange? If so, please clarify.

L182: Are these ice mélange thickness values from the TLS, ICESat-2 or from the references cited? It would be useful to

include a few sentences here showing how the hard-won observational data has been incorporated into the continuum model. Otherwise, the manuscript feels like several different ideas with very little that links them together.

Eq 5: Some sort of summarizing statement after this equation would be useful to remind the reader that the buttressing effect is proportional to the square of mélange thickness. It was mentioned in the introduction but would be good to say it again to tie this section back to the application.

L197-198: It is unclear where the range in initial ice thicknesses come from. Are they from the TLS scans at Helheim or the ICESat-2 data from one of the other big glaciers? 378 m seems a little thick given that the authors are assuming that the density of the ice mélange is uniform with both depth and width. If the observations are to be believed, then a much lower value should be chosen that represents the at least the width-averaged thickness of the ice mélange (e.g. Fig. 1c-f). This statement epitomizes the major problems with this manuscript in that some (good) individual ideas have just been copied and pasted into a single manuscript without much attempt to link them together.

L199-200: Any justification for this value?

L204: Can you show an image of what this looks like? There do not appear to be any uniformly-spaced bulges in Figure 4.

L209-212: Another mismatch between observations and modeling. Surely if the thick mélange is bunching up near the terminus, there would compressive stresses which would appear in the strain rate fields documented by Amundson and Burton (2018)? But their observations demonstrate uniform flow during winter.

L250: It would be useful to report the buttressing force for the thick mélange simulations in these units as well so that the findings can be compared to other studies.

L335: Melting?

L338-345: This is an interesting piece of evidence that would appear to support your argument about mélange control on terminus advance/retreat. However, none of this evidence is presented in the results and there is no accompanying figure to support these surprising findings. I am finding it a little hard to believe that there is 100 m thick mélange in July and August in Greenland. Can you show a figure that demonstrates better demonstrates this finding for these six glaciers? You could also go into more detail. For example, do these glaciers always have mélange at this time of year? If not then you could look at interannual advance/retreat patterns to provide more evidence to support your hypothesis.

Section 3.1: Given that there are no references, it appears that this is the first manuscript that has used data from the TLS instrument. The brevity of this section is therefore a little concerning. It's probably easiest for me to communicate this feedback as questions: where was the TLS located? How far was it from the calving front (nearest/furthest)? Was a DEM of the mélange derived every day during the study period? It looks like it from Fig. 1b but it's difficult to tell. How many points were contained in each 100 x 100 m grid cell? What is the reasoning behind the vertical accuracy choice of 0.10 m? Is the accuracy of the freeboard or ice thickness? How do you know the accuracy varies with distance from the scanner? Is it because there are fewer points to average?

L382: It's a bit misleading to use the term "spatial resolution" for ICESat-2 data. "Segment length" would be a more accurate term.

L383-384: This seems a little fraught given the resolution of the geoid and uncertainty in the tidal corrections. A more convincing way of reporting the uncertainty would be compute the difference between ATL06 over the ocean and the geoid with tidal corrections.

L399-400: Was a minimum advance/retreat used when making this classification? In other words, is a 1 m advance classified the same as a 1,000 m advance?

L404-405: The effort that went into correcting the ArcticDEM strips is appreciated but the actual uncertainty of this approach is a combination of ArcticDEM uncertainty + geoid/tidal correction uncertainty. Currently, the authors assume there is no uncertainty in the geoid/tidal correction which seems unwise. Again it would be useful to compute this difference between the two datasets over a non-mélange area to provide a more robust estimate of the accuracy of this method.

L414: It would make more sense if this paragraph went before the computation of ice thickness.

Section 4: The data availability statement does not comply (at least by my interpretation) with Nature's policy since I am not able to interpret and verify the research in the manuscript. It is not sufficient to provide links to the raw data files since they have been substantially modified. At a minimum, please provide access to the corrected ArcticDEM strips, ICESat-2 ice mélange freeboards/ice thicknesses.

(Remarks on code availability)

Reviewer #3

(Remarks to the Author)

Summary:

This manuscript described the observed mélange thickness changes and glacier calving dynamics based on field and remote sensing data, and developed a three-dimensional discrete element model to simulate the mélange motion. The authors concluded that mélange buttressing force (per unit lateral-width) acted on the glacier is solely controlled by the packing density and thickness of the mélange immediately at the glacier front. The model is carefully designed, and most of the assumptions seem to be reasonable. I see merit in publication of this work. I have some comments I hope the authors could answer/consider, mostly minor, please see details below.

Major comments:

1. While the data suggest a strong correlation between thick mélange and terminus advance, I did not see direct evidence that thick mélange leads to the terminus advance. Alternatively, could the glacier terminus advance be simply caused by a re-configuration process after major calving events? For instance, calving causes terminus retreat, and the new glacier front likely has fewer crevasses and high degree of cohesion. Consequently, it will take a longer period to detach ice blocks from the glacier. Figure 1 appears to support this explanation: the mélange was pretty thick throughout the TLS observations, however, calving events still occurred frequently; and most of the time the terminus advance rate was not affected by the decrease of mélange thickness.

2. I was impressed by Figure 7. While I have no doubt about the data analysis, I am curious about how good can selected ArcticDEM tiles represent the whole scenario. For instance, in Figure 7 I did not find black markers representing Helheim glacier retreat when the mélange freeboard was greater than 10 m. However, Figure 1 clearly illustrates some calving events (should correspond to glacier retreat) when the mélange freeboard was greater than 10 m. Did I miss anything?

3. About the structure of this manuscript: the modeling work is not directly related to the observations. The observations seem to support the hypothesis that the seasonal variabilities in mélange thickness and calving dynamics are correlated, whereas the modeling work does not seem to aim at testing the hypothesis. Perhaps there are some links between the observations and the model. Some rephrasing may be needed to make the two components more tightly related in a paper.

4. For the model, it is reasonable to make many simplifications. I was wondering if the authors were able to test the implications of some of the major simplifications. For example, most fjords do not have regular geometry; remote sensing and field observations indicate that it is not uncommon for the mélange to have highly localized strain rates across the width of the fjord.

5. For the buttressing force, what was estimated in this work was the force per unit lateral-width. It would be good to note that when it was used, or clarify it when first referred.

Other Comments:

Ln 29: Does the variation of 400 mm equivalent sea level rise correspond to iceberg calving in GrIS alone, or in Antarctic & GrIS?

Ln 43-45: Did the authors imply that a complete loss of mélange buttressing would lead to more frequent or greater volume of calving in both summer and winter? What's the difference between winter and summer terminus changes in such a scenario?

Ln 60-62: I thought mélange width is also a needed parameter if "per-unit width" is not specified. And how about the packing ratio?

Ln 71: I don't think the geoid undulation is associated with tides. They are two independent variables. Please clarify (here and in the Methods).

Ln 98: replace "may be reacting" with "react".

Ln 100: delete "completely"?

Ln 113: perhaps only selected tracks were used? ICESat-2 has more tracks passing over glacier termini.

Ln 126-127: thick mélange may influence calving, however, the data described by the authors could not exclude other possible factors. For instance, could temperature, sea ice, or other factors play some roles?

Fig 1: for the elevation profile, instead of using the value that has the maximum number of data points, it is perhaps more robust to use the median value, particularly for high precision data.

Ln 146: delete "the" after "fjord and"

Ln 324-325: are these averaged thicknesses calculated immediately at the glacier front? Or calculated within a certain distance to the glacier front? Better to clarify that.

Ln 349: what is the difference between bed topography and bathymetry?

Ln 354: delete "field"

Ln 382: add "along-track" to the front of "is 20 m".

(Remarks on code availability)

Version 1:

Reviewer comments:

Reviewer #1

(Remarks to the Author)

The manuscript by Meng. et al has undergone substantial revisions and is much improved in its current form. I commend the authors for their tedious and thorough response to the original submission. My primary remaining comment relates to the framing of melange "thinning" coinciding with retreat events, particularly in section 1.1 of the results. While a reduction in the elevations within 1 km of the terminus technically does occur, I think the language stating "melange thinning" implies a rapid melting of the material in place, or a thermodynamic forcing, rather than (what I take to be the case) the calving event disrupting the proglacial melange and flushing the previously adjacent melange further down fjord, and partially replaced with the newly calved material. By the study's defining parameters for melange thickness calculations (at Helheim, for example, the melange within 1 km of the terminus) would sample different areas of melange before and after retreat even if the melange remained completely static, as a retreated terminus would allow new melanged areas of the fjord to meet the "within 1 km" criteria. This is further supported by the statement in the revision that melange thinning was also "accompanied by large increases in fraction of melange area composed of large icebergs." Particularly the January 2020 event, melange is reported to have thinning by > 5m in three days during winter, which would not follow expected seasonal thinning trends. In section 1.1, it would be more helpful to provide a figure for the melange thickness preceding the calving events, as it does appear to show a gradual decrease in elevation following the second major calving event of fall 2019 (Figure 1b) up through the January 2020 calving event.

I also would suggest prefacing the conclusion of section 1.2 (that ICESat-2 data provides strong evidence of correlations between melange thickness and terminus seasonality" with an acknowledgement of how highly variable melange thickness distributions can be across both time and space, as evidenced by earlier results from section 1.1, and as evident in ArcticDEM freeboard heights. The ICESat-2 data provide a snapshot into a profile that may look quite different within days or across a different profile line in the fjord. That is not to say the authors do not provide useful analysis, as it is yet another helpful metric that adds to the study as a whole, but only that generalized conclusions should also include an acknowledgement of spatiotemporal uncertainty.

(Remarks on code availability)

Reviewer #2

(Remarks to the Author)

I thank the authors for largely addressing all my general and specific comments. I think the manuscript is much improved. However, there is a lot of new text and some of it is not as clear as it could be. I've provided some specific comments below. Note that my line numbers refer to the tracked changes version

L49-50: This is a pretty vague statement. What do you mean by "...significant effect on the seasonal range in the terminus position"? More range or less range? Or shifting the range inland? Or both?

L113-114: I understand that the center of the white line is placed on the furthest downstream point of the calving front but how do you determine where the edges of the line intersect the fjord?

L123: Consider clarifying that no "major or minor" calving occurred.

L144: Clarify that this is "satellite" remote sensing

L163: What about "surface elevations represent heights above mean sea level...?"

L163: It's unclear from this statement whether these surface elevations include or exclude tidal variations.

L301-302: The problem for a new reader is that ArcticDEM observations (L301, L314, and L320) have not been introduced yet. Surely L399-431 has to go before you compare the model with the mélange thickness observations from ArcticDEM? I think it be worth reordering some of the text to address this.

L409-410: How do you define “warm” and “cold” months? Here and elsewhere (e.g. L493 and 495)

L459: I thought these were “disk-shaped” in L55 and L713? Or is there a difference between these statements?

L485-487: Can you explain why a glacier that has just calved have fewer crevasses and take longer to calve again?

L487-489: I think you should elaborate on this point a little more. Although I still believe that there is some threshold behavior where mélange rigidity gradually reduces to a point where large calving events can occur. I do still find it plausible that large calving events are responsible for dramatic changes in mélange thickness. How else can you explain the 4-5 m of mélange elevation reduction in 3-5 days that you observe with the TLS?

(Remarks on code availability)

Reviewer #3

(Remarks to the Author)

The authors have adequately addressed almost all my concerns. My only comment for their responses is above the above waterline mélange height calculations described in the 2nd paragraph of subsection 1.2: after removing geoid heights, the mélange surface elevations are [approximately] the heights above mean sea level with tidal variations. This is because the mean sea level can deviate from the geoid by a few tens of centimeters (location-dependent). The relatively small difference might not affect the analysis. However, geodesists may prefer to see an accurate description of the relationship.

Here is another question I have for the authors: is there evidence that the ice blocks in the mélange pile up (e.g., Figure 3)? I understand that many previous studies made similar assumptions. I am just curious to know if there were observations supporting that. To my knowledge, at least for some fjords in Greenland, the pre-existing icebergs do not seem to change their shapes and orientations significantly during some calving events (yes, the new icebergs will likely change the appearance of the fjord, but the pre-existing icebergs can have their above-water geometry largely unchanged). If ice blocks do not pile up, the effective accelerations for ice above and below the waterline could be the same in equation 1.

Thank you!

(Remarks on code availability)

I could only see a piece of code reading a DEM and a function to extract elevation profiles. I am sure the authors will make the essential code available upon publication of the paper.

Version 2:

Reviewer comments:

Reviewer #1

(Remarks to the Author)

The authors have thoroughly addressed my earlier comments - this is an excellent manuscript and I have no further comments or suggestions.

(Remarks on code availability)

Matlab code supports reproducible runs and mirrors conclusions in paper, and includes information (docker file) describing environment for module runs, etc.

Reviewer #2

(Remarks to the Author)

I thank the authors for their response which has fully addressed my remaining comments.

(Remarks on code availability)

Reviewer #3

(Remarks to the Author)

Thank you for your thorough response to my comments. I believe the manuscript merits publication. I have no more questions and I look forward to reading the published version.

(Remarks on code availability)

REPLY TO REFEREE COMMENTS - REFEREE 1.

We thank the referee for a constructive review of our paper. The referee states that “The primary contribution of this work is in the presentation of the author’s new three-dimensional discrete element model of mélange... Most importantly, the authors derive an Equation showing how the mélange buttressing force can be reasonably calculated from observations of near-terminus mélange thickness, which is an important step in making mélange/terminus interactions more readily representable in ice sheet and glacier flow models.” He/she states that “This manuscript presents new and important research that will be valuable to both the observational and modeling glaciological communities, and should, in principle, be suitable for publication in Nature Communications after some suggested revisions below.” He/she makes several constructive suggestions to strengthen and improve the manuscript. We have taken the referee’s comments very seriously, and we have revised the manuscript to address the referee’s comments and suggestions.

In the following, we detail the amendments made to the manuscript (highlighted in red color) in response to the referee’s comments (included in italics).

Comment (1). Remarks on code availability: I was able to view the code and metadata in the code ocean portal, but could not run the code without creating an account. After creating an account, I received "access denied" notices. While I was unable to directly run the code, the matlab scripts themselves appeared well commented and described, with previous run outputs and figures matching the anticipated format.

Response. We have uploaded the data and codes onto Zenodo at <https://zenodo.org/records/13382185> [1]. We have revised the data and code availability sections as follows:

4 Data availability All data sets produced as part of this paper are available through Zenodo at <https://zenodo.org/records/13382185> [1]. These data include the following: a supplementary word file that lists the ArcticDEM tiles, ICESat-2 laser altimetry tracks, and the Landsat, Sentinel-1 and Sentinel-2 scene numbers used in this study, a supplementary excel file that lists elevation offsets applied on the 108 ArcticDEM strips with corresponding ArcticDEM strip acquisition dates and Mosaic DEM tiles for coregistration, ICESat-2 data cropped by the mélange region of interest for the three glaciers presented in Fig. 2, 32 zipped folders that contain ArcticDEM data cropped by the mélange region of interest for the presented 32 Greenland glacier termini in Fig. 8, and source codes to plot mélange surface elevation (or thickness) profile as a function of distance from terminus from ICESat-2 data and ArcticDEM strips. Landsat images were downloaded through the Amazon Web Services (<https://registry.opendata.aws/usgs-landsat>). Copernicus Sentinel-1 data acquired in 2019 and 2021, and Copernicus Sentinel-2 data acquired in

2018 and 2019 were provided by European Space Agency and downloaded through the Amazon Web Services (<https://registry.opendata.aws/sentinel-1>, <https://registry.opendata.aws/sentinel-2>). TLS data for Helheim Glacier are available upon reasonable request. ICESat-2 laser altimetry tracks are available through the OpenAltimetry portal at <https://openaltimetry.org/data/icesat2/> with download services provided by the National Snow and Ice Data Center [2]. ArcticDEM digital elevation models [3, 4] are available from the University of Minnesota Polar Geospatial Center (PGC): <https://www.pgc.umn.edu/data/arcticdem/>. Ice surface velocity and BedMachine Greenland are freely available at the National Snow and Ice Data Center (NSIDC) at <https://nsidc.org/data/nsidc-0725/versions/5> [5] and <https://nsidc.org/data/idbm4/versions/5> [6], respectively. The time series of Greenland terminus positions is available from [7] at <https://zenodo.org/records/10095674>.

5 Code availability All source data and MATLAB codes to reproduce figures in the main text are available through Zenodo at <https://zenodo.org/records/13382185> [1]. The codes used for the three-dimensional discrete element model are available from the corresponding author upon reasonable request. PFC3D[®] [8] is a software from Itasca Consulting Group, Inc. through a commercial license.

Comment (2). Main Comments

Manuscript organization and inclusion or terrestrial laser scanner + ICESat-2 observations. *I found that transition from the abstract/introduction to the first section of the results (on the point cloud observations of Helheim mélange thickness) to be rather abrupt and without sufficient context. For example, I found this section (and the following section where ICESat-2 and satellite imagery are used to study seasonal changes at three tidewater glaciers) both very similar in the goal of the analyses but disjointed from the remainder of the manuscript. I found it challenging to tie in the results from these sections with the later derivation of the relationship between buttressing force and mélange thickness and application of the 3D mélange model. These sections also raised the following questions:*

1.) *Can the authors provide more context for the use of 30m freeboard height exclusion threshold? Was this threshold empirically derived? I am also curious if the authors noticed any temporal bias in how heavily this threshold excluded glaciers (for example, were more thick icebergs present and subsequently excluded in spring versus summer).*

2.) *What was the motivation for calculating mean thickness within 1 km in this section, but later in the manuscript, modeled outputs are averaged within 200 m of the terminus?*

3.) *Line 92, on four dates of pronounced thinning – over what time period is a decrease in elevation calculated? The lower bound of the elevation is given, but not the initial elevation prior to thinning (other than a 15 m average).*

I would suggest adding more organizational context to the introduction to prime readers for the relevance of the initial observational analyses. My interpretation is that they are “proof of concept” or testing available observational datasets to show whether, initially, there is sufficient evidence of seasonal variability in mélange thickness to motivate the

development of the three-dimensional model. Perhaps results from these early studies help inform the model with a realistic range of near-termini mélange thicknesses? It would also be helpful to include a brief description of why the three glaciers were selected for the ICESat-2 work, even if that is simply something similar to there being “large discharging glaciers with identified seasonally varying mélange.”

Response. We answer the specific questions in detail as follows:

1) The 30 m threshold is empirically derived from TLS and ArcticDEM data. We have added a figure in SI (Fig. 2 in SI and Fig. 1 in this document) to show that large icebergs within mélange typically have freeboard heights above 30 m. We have also analyzed the temporal evolution of the existence of large icebergs within the mélange, which strongly correlates with calving dynamics (Fig. 5 in SI and Fig. 4 in this document).

2) The motivation was to calculate mélange thickness within a distance of 1/5 fjord width from the terminus. Helheim glacier terminus has a width around 5 km, and therefore we calculate the mélange thickness within 1 km from the terminus. In the discrete element models, the fjord width is 1 km (Table 1 in the main text), and therefore we calculate the mélange thickness within 200 m from the terminus.

3) We identified four dates where noticeable mélange thinning occurred, which were 3 Sep 2019, 3 Jan 2020 (Fig. 1(d)), 31 May 2020 (Fig. 1(f)), and 14 July 2020. Around these four dates, the mélange elevation at the terminus decreased as follows: by 1.6 m within two days to 10.8 ± 0.05 m, by 5.3 m within three days to 9.4 ± 0.05 m, by 4 m within five days to 10 ± 0.05 m, and by 2.3 m within seven days to 11.6 ± 0.05 m. We include TLS scans during these four major calving events in the supplementary material (Fig. 4 in SI, Fig. 3 in this document).

We have reorganized the manuscript and revised the TLS and ICESat-2 sections as follows:

Motivated by recent observations of weekly to seasonal variations in mélange thickness, we hypothesize that mélange thickness dictates its buttressing force per unit width. To test this hypothesis, we use terrestrial laser scanning and ICESat-2 data to demonstrate that mélange thickness is well-correlated with terminus position and calving dynamics at 4 major Greenland outlet glaciers. Motivated by these observations, we develop a continuum mélange model, validated with a three-dimensional discrete element model, which confirms that mélange thickness and packing density are the only observations required to estimate mélange buttressing force per unit width. We then apply this continuum model to estimate mélange buttressing forces around Greenland from ArcticDEM observations, and show that the inferred buttressing force is highly consistent with observed patterns of terminus advance and retreat.

1 Results

1.1 Mélange thickness associated with calving dynamics at Helheim Glacier in 2019-2020. To investigate the correlation between the mélange thickness and calving dynamics, we derive time series of mélange elevation at the terminus, calving events, and terminus position inferred from TLS data and

satellite images (Sentinel-1, Sentinel-2 and Landsat 8) from 1 Sep 2019 to 1 Sep 2020 at Helheim Glacier (Fig. 1(b)). Throughout 2019, a REIGL VZ-6000 terrestrial laser scanner (TLS), located at the triangle marked in Figure 1(a), scanned the terminus and ice mélange of Helheim Glacier. Fig. 1(a) shows the TLS-measured surface elevation field for ice mélange on 30 Nov 2019 (see Methods for the TLS processing steps). Mélange surface elevation was derived from TLS every 24 hours in the winter months, and every 6 hours in non-winter months. To display the spatial profile of the mélange elevation, we calculate distances from terminus for all data points in the ice mélange and plot them as density maps in Fig. 1(c)-(f). For any specific distance from the terminus, there is a spread of mélange elevation that exhibits a long-tail distribution due to the presence of large icebergs. To reflect the mélange elevation that is piled up from small icebergs and sea ice, we connect elevation values that have maximum numbers of data points along the distance from terminus as the representative mélange elevation profiles (solid blue lines in Fig. 1(c)-(f), see Methods and Fig. 2 in SI for a sensitivity study of this metric). In doing so, we prevent large icebergs from disrupting the mélange elevation profiles without cutting off the data by an arbitrary elevation threshold. The resulting representative mélange elevation profiles are always below 30 m, indicating that it is a reasonable elevation threshold to exclude large icebergs from consideration. To estimate mélange elevation near the glacier terminus (Z_0), we take an average of all data points below the 30 m threshold within 1 km (i.e., 1/5 of the fjord width) of the terminus. We infer thickness of the mélange based on TLS-derived surface elevations and assuming hydrostatic equilibrium.

We digitize the terminus position on each TLS scan with a straight line (the white line in Fig. 1(a)). The terminus advance/retreat refers to the changes in the terminus position, and is quantified by distances between these straight lines. This can indicate whether the glaciers are advancing (growing) or retreating (shrinking). With reference to previous classification of calving events [9], here, we define “major calving” events as those with block size $>0.25 \text{ km}^2$ (observed from TLS scans), and causing significant mélange motion and an overall terminus retreat (observed from satellite images); “minor calving” events are those in which visible blocks calved, but the mélange or terminus position remained largely unchanged. We observe two episodes of calving cessation, from 8 Oct 2019 - 31 Dec 2019 and 1 Mar 2020 - 20 May 2020, when no calving occurred and the terminus advanced steadily for 2 km and 1.7 km, respectively. We found that mélange elevation at the terminus averaged 15 m during these periods. We include TLS scans during these two episodes of calving cessation in the supplementary material (Fig. 3 in SI). We identified four dates where noticeable mélange thinning occurred, which were 3 Sep 2019, 3 Jan 2020 (Fig. 1(d)), 31 May 2020 (Fig. 1(f)), and 14 July 2020. Around these four dates, the mélange elevation at the terminus decreased as follows: by 1.6 m within two days to $10.8 \pm 0.05 \text{ m}$, by 5.3 m within three days to $9.4 \pm 0.05 \text{ m}$,

by 4 m within five days to 10 ± 0.05 m, and by 2.3 m within seven days to 11.6 ± 0.05 m. Major calving happened around these dates, with corresponding retreats at the terminus of 0.5 km, 1.2 km, 1.3 km, and 1.0 km, respectively, accompanied by large increases in fraction of mélange area composed of large icebergs (see SI Fig. 4 & 5 for associated TLS scans and temporal evolution of iceberg fraction).

Apart from the weekly terminus variability of Helheim Glacier presented in this section, remote sensing observations on many Greenland glacier termini have shown significant terminus-position seasonality, with advance from winter to spring and retreat from summer to fall through enhanced calving [10, 11, 7]. Previous studies have attributed seasonal calving dynamics to buttressing from ice mélange [12, 13, 14], which motivates us to further explore seasonal changes of mélange thickness and calving dynamics on other Greenland glaciers.

1.2 Seasonal changes of mélange thickness and calving dynamics. To investigate whether there are correlations between ice mélange thickness and calving dynamics on other glaciers, we use ICESat-2 observations of mélange surface elevation. While this dataset does not provide the temporal resolution to study individual calving events, we can leverage the observed seasonality in terminus advance and retreat at many Greenland glaciers to assess whether mélange thickness is correlated with periods of quiescence versus vigorous calving.

There are very few ICESat-2 tracks passing through the fronts of termini in different seasons, because positions of termini vary seasonally but ICESat-2 tracks are generally fixed in space. We identify ICESat-2 tracks passing over glacier termini in different seasons for Jakobshavn Isbræ (Fig. 2(a)), Kangerlussuaq Glacier (Fig. 2(b)), and Store Glacier (Fig. 2(c)), which are large discharging glaciers contributing to Greenland's mass losses [15]. We use the ATL06 data set from ICESat-2 that provides geolocated, land-ice surface heights above the WGS 84 ellipsoid [2]. After removing the geoid heights at glacier termini [16], the mélange surface elevations are heights above mean sea level with tidal variations [9]. We then remove the tidal trend of mélange elevations with the Greenland 1 kilometer tide model [17]. Surface elevation data is acquired along the ICESat-2 track and displayed as a function of the distance from terminus. We found that mélange was continuously present at Jakobshavn Isbræ from 2021 to 2022 and at Kangerlussuaq Glacier in 2020, while it was seasonally present at Store Glacier in 2019. Where mélange persisted, we calculated seasonally distinct freeboard heights from winter to early spring (solid black lines) and in summer (dashed black lines) (Fig. 2(d)(e)). Near the termini, mélange for the two glaciers both exhibit different ranges of freeboard heights during the two seasons: 20 ~ 35 m in winter or spring, and below 5 m in summer. The seasonal changes in mélange thickness at the terminus may explain the observed calving dynamics and terminus motion: zero or minor calving with

an advancing terminus from winter to spring, and vigorous calving with a retreating terminus from summer to fall (Fig. 2(g)(h)). At Store Glacier, the mélange was present from 1 Jan to 14 June, after which calving resumed and the terminus kept retreating (Fig. 2(i)). The mélange elevation profile on 22 March 2019 exhibits a thickness gradient with a freeboard height of around 30 m near the terminus (Fig. 2(f)). **In summary, ICESat-2 data provides strong evidence of correlations between mélange thickness and terminus seasonality.**

FIGURE 1. Helheim Glacier and ice mélange. (a) TLS-measured elevation map after accounting for local differences between the ellipsoid and geoid with tidal corrections [17], overlain on a Sentinel-1 HV image (both acquired on 30 Nov 2019). The white line across the fjord indicates the glacier front location. The white triangle indicates the TLS location. The upper left inset shows the location of Helheim Glacier in Greenland. The image is in polar stereographic projection (EPSG: 3413). (b) Surface elevation profile for the mélange displayed as a density plot; the colour bar denotes the number of data points that have the same elevation and distance from terminus values. For any specific distance from terminus, we find the elevation value that has the maximum number of data points. The solid blue line connects these elevation values along the distance from terminus as the representative mélange elevation profile. We also calculate the median and mean elevation values for each specific distance from terminus, and connect them by solid black and cyan lines, respectively. (c) The number of data points against the surface elevation value at distance from terminus of 0.5 km (black line), 1 km (blue line), and 2 km (red line). Due to the long-tail distribution of the mélange elevation, the median or mean elevation value approach in (b) tends to incorporate large icebergs into the elevation profile, failing to represent the elevation piled up from small icebergs and sea ice. (d) ArcticDEM measured elevation map after accounting for local differences between the ellipsoid and geoid with tidal corrections [17], overlain on a Landsat 8 image (acquired on 30 Mar 2014). The image is in polar stereographic projection (EPSG: 3413). (e) and (f) follow captions of (b) and (c).

FIGURE 2. The TLS-measured mélange surface elevation map at Helheim Glacier during two episodes of calving cessation, from (a, b) 8 Oct 2019 – 31 Dec 2019, and (c, d) 1 Mar 2020 – 20 May 2020. White lines indicate positions of the terminus. Dates on the images show the acquisition dates for TLS data. TLS scans are overlain on Sentinel-1 and 2 images acquired around the same dates. Images are in polar stereographic projection (EPSG: 3413).

FIGURE 3. The TLS-measured mélange surface elevation map at Helheim Glacier during four major calving events, from (a)-(c) 3 Sep 2019 – 4 Sep 2019, (d)-(f) 31 Dec 2019 – 3 Jan 2020, (g)-(i) 27 May 2020 – 31 May 2020, and (j)-(l) 14 Jul 2020 – 18 Jul 2020. Dashed white lines indicate positions of the terminus before calving. Solid white lines indicate positions of the terminus after calving. Dates on the images show the acquisition dates for TLS data and times are in UTC. TLS scans are overlain on Sentinel-1 and 2 images acquired around the same dates. Images are in polar stereographic projection (EPSG: 3413).

FIGURE 4. Helheim Glacier and ice mélange. (a) Terminus position relative to 1 Sep 2019, where the positive sign indicates terminus advance. Blue dots denote the averaged mélange elevation within 1 km of the terminus, Z_0 . Calving events are inferred from TLS and satellite images. Due to limited temporal sampling of the data, we are not able to determine the exact time of each calving event. Instead, we mark the time period during which a calving event occurs by a red-shade rectangle. Four vertical black lines mark the dates for the TLS-measured elevation data presented in Fig. 1(c)-(f) in the main text, which corresponds to 30 Nov 2019, 3 Jan 2020, 30 Mar 2020, and 31 May 2020, respectively. Solid black lines mark the dates with terminus advances, and dashed black lines mark the dates with terminus retreats. (b) follows legends in (a), except blue dots here denote the ratio between the TLS-measured area of mélange with a freeboard height above 30 m ($A_{z>30 m}$) and the total area of mélange (A_{total}) within 1 km of the terminus.

To better link observations to models, we make major revisions on both continuum and discrete element models. Note that for the continuum model for predicting buttressing stresses, we are already using the mélange thickness at the glacial termini, H_0 , from the satellite-derived observations (ArcticDEM) from 32 glaciers. We take H_0 to vary from 35 m, which is the minimum size of icebergs detected within the mélange [18, 19], to 240 m, which is the thickest mélange observed across 32 Greenland termini (See Table 2 in SI). We have revised the continuum model in the main text as follows to make this more clear (lines 208–220):

To characterize the relative magnitude of the horizontal deviatoric stress to the glaciostatic pressure, we substitute representative values for parameters in Eqn. 3 and obtain:

$$\frac{|4H_0(\eta\frac{\partial u}{\partial x})|_{x=0}}{\frac{1}{2}\rho_i(1-\frac{\rho_i}{\rho_w})g\phi_0H_0^2} \in [3.90 \times 10^{-14}, 1.17 \times 10^{-11}] \times \eta, \quad (1)$$

where the range of mélange thickness, H_0 , is derived from DEM observations. We take H_0 to vary from 35 m, which is the minimum size of icebergs detected within the mélange [18, 19], to 240 m, which is the thickest mélange observed across 32 Greenland termini in 2013–2022 (See Table 2 in SI). We adopt $\frac{\partial u}{\partial x} \in [\frac{2 \text{ m/day}}{15 \text{ km}}, \frac{25 \text{ m/day}}{10 \text{ km}}]$ [20], $\phi_0 \in [0.64, 1]$ [21, 22], $\rho_i \in [870 \text{ kg/m}^3, 920 \text{ kg/m}^3]$ [9], and $\rho_w \in [1020 \text{ kg/m}^3, 1029 \text{ kg/m}^3]$ [23]. As the mélange acts as a weak granular ice shelf [24], its effective viscosity should be much smaller than the glacier ice viscosity, $\eta \ll \eta_i = 10^{12} - 10^{15} \text{ Pa}\cdot\text{s}$ [25, 26]. For mélange with a high viscosity ($\eta > 10^{11} \text{ Pa}\cdot\text{s}$), we need to consider deviatoric stress effects as has been done in [27]. The mélange in the following discrete element model has an estimated viscosity of $2 \times 10^{10} \text{ Pa}\cdot\text{s}$ (see Section 2 in SI for details). Therefore, for mélange with a low viscosity, glaciostatic pressure dominates and the mélange buttressing force can be approximated as:

$$\frac{F}{W} = \frac{1}{2}\rho_i(1-\frac{\rho_i}{\rho_w})g\phi_0H_0^2. \quad (2)$$

To better match our discrete element models with observations, we have conducted ten more simulations with Helheim and Kangerlussuaq fjord geometries, observed iceberg distributions, and mélange thickness from ArcticDEMs. And we have justified the mélange thickness profile from discrete element models with both ArcticDEM observations and the continuum theory.

We first added a derivation of the continuum theory for mélange thickness profile in SI as follows (lines 97–119):

Finally, we derive the expression for the mélange thickness profile, $H(x)$. Eqn. 14 can be reorganized as follows:

$$\frac{\partial}{\partial y}(H(x)\bar{\sigma}'_{xy}) = -\frac{\partial}{\partial x}(H(x)\bar{\sigma}_{zz}) - \frac{\partial}{\partial x}(2H(x)\bar{\sigma}'_{xx}) - \frac{\partial}{\partial x}(H(x)\bar{\sigma}'_{yy}) \quad (3)$$

Because we assume mélange thickness and stresses do not vary with y , we can integrate Eqn. (3) over the y direction as:

$$(H(x)\bar{\sigma}'_{xy})|_{y=W} - (H(x)\bar{\sigma}'_{xy})|_{y=0} = W\left(-\frac{\partial}{\partial x}(H(x)\bar{\sigma}_{zz}) - \frac{\partial}{\partial x}(2H(x)\bar{\sigma}'_{xx}) - \frac{\partial}{\partial x}(H(x)\bar{\sigma}'_{yy})\right) \quad (4)$$

We use Coulomb friction law to calculate the shear stress at the fjord walls as follows:

$$\begin{aligned} \bar{\sigma}'_{xy}|_{y=W} &= \mu_e \bar{\sigma}_{yy}, \\ \bar{\sigma}'_{xy}|_{y=0} &= -\mu_e \bar{\sigma}_{yy}, \end{aligned} \quad (5)$$

where μ_e is the effective coefficient of friction between the mélange and the fjord wall, which depends on the material friction coefficient and the geometry of the fjord walls, i.e., wall roughness [24, 28, 29]. As $\bar{\sigma}_{yy} = \bar{\sigma}_{zz} + \bar{\sigma}'_{xx} + 2\bar{\sigma}'_{yy}$, Eqn. (4) becomes

$$\frac{2H(x)\mu_e}{W}(\bar{\sigma}_{zz} + \bar{\sigma}'_{xx} + 2\bar{\sigma}'_{yy}) = -\frac{\partial}{\partial x}(H(x)\bar{\sigma}_{zz}) - \frac{\partial}{\partial x}(2H(x)\bar{\sigma}'_{xx}) - \frac{\partial}{\partial x}(H(x)\bar{\sigma}'_{yy}) \quad (6)$$

Following the scaling analysis in the main text (Eqn. 4 in section 1.3), we can reasonably assume that $\bar{\sigma}'_{xx}, \bar{\sigma}'_{yy} \ll \bar{\sigma}_{zz}$, where the depth-averaged vertical stress is $\bar{\sigma}_{zz} = \frac{1}{2}\rho_i g \phi(x)(1 - \frac{\rho_i}{\rho_w})H(x)$. We further assume that the mélange packing density remains a constant along fjords. Therefore, Eqn. (6) becomes

$$\frac{\partial H}{\partial x} + \frac{\mu_e}{W}H(x) = 0 \quad (7)$$

which gives

$$H(x) = C e^{-\frac{\mu_e}{W}x} \quad (8)$$

where C is a constant that needs to be constrained by a boundary condition of the thickness profile. The mélange thickness exponentially decays with the distance from terminus. However, Eqn. 14 only holds when mélange can be considered as a three-dimensional material. When the mélange thickness decays to a monolayer of icebergs, its thickness is dictated by the iceberg size distribution, instead of stress balances that give rise to Eqn. (8). We identify the mélange length, L , where the mélange thickness decays to a threshold value, H_L , below which the mélange is considered to be a two-dimensional material. Using the boundary condition, $H(x = L) = H_L$, we arrive at the final expression for the mélange thickness profile:

$$H(x) = H_L e^{\frac{\mu_e L}{W}(1 - \frac{x}{L})}, x \in [0, L] \quad (9)$$

By defining the dimensionless distance, $\tilde{x} = \frac{\mu_e x}{W}$, and the dimensionless thickness, $\tilde{H} = \frac{H(x)}{H_L e^{\frac{\mu_e L}{W}}}$, we arrive at the dimensionless form of the mélange thickness profile:

$$\tilde{H}(\tilde{x}) = e^{-\tilde{x}}, \tilde{x} \in [0, \frac{\mu_e L}{W}]. \quad (10)$$

We then summarized the continuum theory for mélange thickness profile in the main text as follows (lines 221–249):

The “granular ice shelves” depth-averaged horizontal momentum equations for mélange (see SI for derivation and validation against discrete element simulations) resemble that of ice shelves:

$$\begin{aligned} \frac{\partial}{\partial x}(2H(x, y)\bar{\sigma}'_{xx}) + \frac{\partial}{\partial x}(H(x, y)\bar{\sigma}'_{yy}) + \frac{\partial}{\partial y}(H(x, y)\bar{\sigma}'_{xy}) &= \rho_i g(1 - \frac{\rho_i}{\rho_w})H(x, y) \frac{\partial(\phi(x, y)H(x, y))}{\partial x}, \\ \frac{\partial}{\partial x}(H(x, y)\bar{\sigma}'_{xy}) + \frac{\partial}{\partial y}(H(x, y)\bar{\sigma}'_{xx}) + \frac{\partial}{\partial y}(2H(x, y)\bar{\sigma}'_{yy}) &= \rho_i g(1 - \frac{\rho_i}{\rho_w})H(x, y) \frac{\partial(\phi(x, y)H(x, y))}{\partial y}, \end{aligned} \quad (11)$$

where the depth-averaged stress $\bar{\sigma}_{ij} = \frac{1}{H} \int_{z_b}^{z_s} \sigma_{ij} dz$. When the mélange packing density approaches $\phi = 1$, Eqn. (11) converges to the shallow shelf approximation (SSA) [30]. The mélange momentum balance along the fjord direction reveals three competing forces: compressional/extensional flow from velocity gradients within the mélange (negligible if mélange viscosity is smaller than 10^{11} Pa·s), glaciostatic stress from mélange thickness, and shear stresses on fjords. Therefore, the full thickness profile of the mélange depends on fjord/mélange friction/cohesion properties, velocity gradients and viscosity of the mélange, and the mélange width/length. Using the horizontal momentum balance equations (Eqn. 11) and non-dimensionalization (see SI for details), we further derive the mélange thickness profile, $H(x)$:

$$\tilde{H}(\tilde{x}) = e^{-\tilde{x}}, \tilde{x} \in [0, \frac{\mu_e L}{W}], \quad (12)$$

where μ_e is the effective coefficient of friction between the mélange and the fjord wall, which depends on the material friction coefficient and the geometry of the fjord walls, i.e., wall roughness [24, 28, 29]. Here, we define the mélange length, L , as the distance beyond which the mélange thickness decays to a threshold value, H_L , where mélange becomes a monolayer of icebergs with thickness dominated by particle size distribution instead of stress balances that give rise to Eqn. (12). We define the dimensionless distance, $\tilde{x} = \frac{\mu_e x}{W}$, and the dimensionless thickness, $\tilde{H} = \frac{H(x)}{H_L e^{\frac{\mu_e L}{W}}}$. To quantify the mélange buttressing force per unit width, previous two-dimensional model assuming mélange of uniform thickness required assumptions on fjord/mélange friction properties and the mélange width/length [24]; in our three-dimensional model the mélange thickness and packing density at the terminus are the only parameters needed. As the length and thickness are coupled by stress balances within the granular material, the mélange thickness build-up at the terminus already encodes the aforementioned material and geometric properties. For instance, thicker mélange can be built up at the terminus with longer fjords, larger fjord friction, or increased mélange rigidity in winter.

In summary, momentum balance equations reveal that the mélange buttressing force per unit width is solely controlled by the packing density ϕ_0 and mélange thickness H_0 at the glacier terminus, and is proportional to the square of mélange thickness (Eqn. (2)). Additionally, mélange thickness exponentially decays with its distance from terminus (Eqn. (12)).

For the new simulations with Helheim and Kangerlussuaq fjord geometries, we have made a new figure to justify the mélange thickness profile from discrete element models with both ArcticDEM observations and the continuum theory (Fig. 5 in this document and Fig. 4 in the main text). We have also added a supplementary video for modeling mélange in Helheim and Kangerlussuaq fjord geometries. To reflect these changes, we have modified the main text as follows (lines 250–289):

1.4 A three-dimensional discrete element model of ice mélange.

To validate continuum predictions on the mélange buttressing force and thickness profile (Eqn. (2), (12)), we develop a three-dimensional discrete element model on the mélange with a steadily advancing terminus. We adopt fjord geometries from Helheim and Kangerlussuaq glaciers (black dots in Fig. 5(a)(d)), which are scaled down by four times for the discrete element models to save computational costs (Fig. 5(b)(e)). Note that such downscaling does not affect the shape of the dimensionless mélange thickness profile, which will be compared with Eqn. (12). Icebergs are modelled as cubic particles with sizes varying from 36 m to 200 m and have a power-law size distribution with an exponent of -4 [19, 18, 31]. In the models, we limit the maximum size of icebergs to be around one fifth of the fjord width, as commonly observed in both fjords. We initialize the ice mélange thickness with a profile that linearly decays within 1.2 km from the terminus and with the right end open to the ocean. We push the left end of the mélange with an advancing terminus at 30 meters per day [32, 33, 34] and record the temporal evolution of the buttressing force exerted on the terminus. We present modeling results after 16 days of terminus motion, when the mélange motion has approximately reached a steady state. In a series of simulations, we vary the initial mélange thickness to determine its influence on the steady state buttressing force. The steady state mélange thickness at the terminus varies from 40~135 m at Helheim fjord, and 60~240 m at Kangerlussuaq fjord, that are consistent with ArcticDEM observations (Table 2 in SI). We calculate the two-day averaged velocity of each iceberg element by dividing the iceberg’s displacement between 14 and 16 days of terminus motion by the time interval (two days). The mélange near the terminus moves at the terminus velocity with shear bands developed at fjord walls (Fig. 5(c)(f)). The mélange near the open end becomes loosely-packed and more fluidic (See supplementary video for the full temporal evolution of the mélange behaviors.) The modeled velocity field showcases both uniform (Fig. 5(c)) and extensional (Fig. 5(f)) flow regimes that are consistent with

remote observations [20]. See Fig. 6 in SI for the modeled velocity fields, including instantaneous velocity, velocity averaged over one hour, one day, and two days, and the satellite-derived velocity fields of the mélange [20].

The modeled mélange thickness profiles generally align with observations from ArcticDEMs on 17 Aug 2014 at Helheim (Fig. 5(a)(g)) and on 12 Apr 2018 at Kangerlussuaq (Fig. 5(d)(h)). In addition, we conduct another simulation with more icebergs confined in Kangerlussuaq fjord, to produce a thicker steady state mélange thickness profile as observed on 5 May 2020. We set the thickness threshold (H_L) as 36 ± 10 m for mélange to be considered as three-dimensional, and retrieve the corresponding mélange length (L) that varies from 7 to 12 km. The non-dimensional mélange thickness profile from three ArcticDEM strips and the discrete element models collapse onto the exponential analytical solution in Eqn. (12), with μ_e fitted to be $0.3\sim 0.4$ for Helheim fjord and $0.4\sim 0.8$ for Kangerlussuaq fjord (Fig. 5(i), see table 1 in SI for details). The Kangerlussuaq fjord has a larger μ_e due to its more rugged fjord geometries.

To further explore the effect of fjord frictional properties on the mélange buttressing force, we adopt two simplified channel configurations that conceptualize complex Greenland fjord geometries. ...

FIGURE 5. The three-dimensional discrete element model for mélange composed of cubic icebergs with a power-law size distribution and confined within real fjord geometries. (a)-(c) For Helheim glacier, (d)-(f) for Kangerlussuaq glacier. The real fjord geometry is scaled down by four times for the discrete element model. (a) The mélange elevation above mean sea level from ArcticDEM strip, overlain on satellite images acquired around the same date. The ArcticDEM acquisition date is shown at the top right corner. White line across the fjord indicates glacier front location. The images are in polar stereographic projection (EPSG: 3413). Black dots along fjords are adopted in the model to construct boundary walls that resemble fjord geometries. (b), (c) are side and top view for iceberg positions and velocities after 16 days into simulations with steady terminus advance and no calving. The glacier terminus moves at a constant velocity, $V_{\text{ter}} = 30$ m/day. We calculate the two-day averaged velocity of each iceberg element by dividing the iceberg's displacement between 14 and 16 days of terminus motion by the time interval (two days), which is indicated by filled colour in (c). (d)-(f) follow captions of (a)-(c). (g), (h) The comparison of the mélange thickness profile between ArcticDEM (black lines) and discrete element model (red lines) for Helheim and Kangerlussuaq glacier, respectively. (i) The mélange thickness profile from three ArcticDEM strips and corresponding discrete element models collapse onto the exponential analytical solution (Eqn. (12), red dashed line). See supplementary videos for the full temporal evolution of the mélange behaviors.

We have added a table for the fitted fjord effective coefficient of friction (μ_e) in SI as follows:

TABLE 1. The fitted fjord effective coefficient of friction (μ_e) from mélange thickness profile in Fig. 4 in the main text. The uncertainties come from the mélange thickness threshold, $H_L = 36 \pm 10$ m.

Glacier name	ArcticDEM date	μ_e (ArcticDEM)	μ_e (Discrete element model)
Kangerlussuaq	2020/05/05	$0.64^{+0.09}_{-0.06}$	$0.77^{+0.11}_{-0.06}$
Kangerlussuaq	2018/04/12	$0.39^{+0.05}_{-0.04}$	$0.48^{+0.07}_{-0.04}$
Helheim	2014/08/17	$0.25^{+0.03}_{-0.03}$	$0.39^{+0.05}_{-0.05}$

We have modified Fig. 6(b) in the main text (Fig. 6(b) in this document) to incorporate new simulations with Helheim and Kangerlussuaq fjords. We found that with real fjord geometries, iceberg size distribution, and mélange thickness profile, the buttressing forces from discrete element models still show good agreement with the continuum predictions.

FIGURE 6. Comparison between discrete element model and continuum predictions of the mélange buttressing force. (a) The temporal evolution of F/W during the terminus motion for straight (dashed lines) and rugged (solid lines) fjord walls. The red, blue, cyan and black colours correspond to mélange with initial thicknesses, $H_{ini} = 378$ m, 281 m, 178 m, 84 m, respectively. Simulations reach the steady state after 5 days, except for the thinnest mélange ($H_{ini} = 84$ m). (b) Steady state buttressing force, F/W , as a function of steady-state mélange thickness at the terminus, H_0 . Different markers represent simulations with different fjord geometries. Square markers represent Helheim fjord, hexagram markers represent Kangerlussuaq fjord, circular markers represent straight fjords, and triangular markers represent rugged fjords. The smaller markers indicate simulations with smaller icebergs (half of the original size). F/W is obtained by averaging the total buttressing force on the terminus over the terminus width during simulation time 5 ~ 15 days. The marker shows the averaged steady-state value of F/W , with a vertical error bar showing its fluctuation. H_0 is obtained by averaging the mélange thickness within 200 m of the terminus and over the terminus width. The marker shows the averaged steady-state value of H_0 , with a horizontal error bar showing its variation over the terminus width brought by the iceberg size polydispersity. For simulations where the mélange collapse into monolayers at the end, we plot both the peak and minimal F/W values and connect them by gray lines. The minimal F/W values for monolayered, two-dimensional mélange are shown by pentagram markers. All markers are coloured by the mélange packing density at the terminus at steady state, ranging from 0.4 to 0.9. The dashed lines represent Eq. (2) with the mélange packing density at the terminus, $\phi_0 = 0.5$ and 1.0, respectively.

To highlight limitations of current mélange thickness profile analysis, we have added following discussions in the main text as follows (lines 405–418):

2 Discussion Our discrete element model of ice mélange is the first to be composed of realistic cubic icebergs instead of spheres, and showcases an exponential decay of the thickness profile in Helheim and Kangerlussuaq fjords, that is consistent with ArcticDEM observations and analytical predictions. Apart from the exponential decay, we also observe other shapes of the mélange thickness profile, such as a plateau near the terminus, a steep drop or a bulge at few kilometers away from the terminus, etc (See SI for ArcticDEM strips at 32 glacier termini). These shapes could be attributed to calving-induced jamming wave propagation, ice-ocean interactions, iceberg size distribution, heterogeneous friction or cohesion within the mélange or at fjord walls, all of which cannot be captured by the simplified rheology and friction law underneath Eqn. (12). Coupled with computational fluid mechanics, our discrete element model can be used to further explore how the mélange thickness at the terminus evolves with ice-ocean interactions that influence calving dynamics, including ocean tides [35], ocean warming [14, 36, 37], and subglacial plumes [38, 39].

Lastly, we have added details of the new simulations into the Methods section as follows (lines 618–655):

3.4 The three-dimensional discrete element model for ice mélange. ...

We use cubic grains which can achieve a higher packing density, thus butressing forces, than disk-shaped grains. We adopt the iceberg *area* distribution observed in the mélange of Jakobshavn Isbræ and Helheim Glacier, which is approximated as a power-law distribution with an exponent of -2.0 [19, 18]. The resulting iceberg size distribution for cubic grains is a power-law distribution with an exponent of -4.0. Taking the simulation with the Helheim fjord for instance (Fig. 5(b)), the side lengths of cubic icebergs are 36 m, 50 m, 75 m, 100 m, 200 m, and the corresponding numbers of particles are 3120, 838, 166, 52, 3, respectively. To model mélange with different steady state thicknesses, the total number of particles varies from 2918 to 12522 for simulations with Helheim and Kangerlussuaq fjords. In simplified fjord geometries, the side lengths of cubic icebergs are 35.4 m, 50 m, 70.7 m, 100 m, 141.4 m, and the corresponding numbers of particles for the thick mélange are 8190, 2045, 510, 125, 30, respectively (Fig. 5(f)). We vary the total number of particles from 1634 to 15264 to change the mélange thickness. To confirm that the modeling results are invariant to the particle size, we conduct six more simulations with smaller icebergs, whose sizes are half of original sizes and range from 17.7 m to 70.7 m (small markers in Fig. 6(b)).

To construct the initial mélange state, we divide the total number of particles into three equal batches. In each batch, iceberg sizes are randomly drawn from the distribution described above. We put a right boundary wall at distance L from the terminus on the left to prescribe the initial length of the mélange.

The mélange is confined in y direction by two side walls representing fjords at a distance W . To explore influence of fjord friction properties on mélange behaviors, we have Helheim, Kangerlussuaq, and simplified (straight and rugged) channel configurations. All configurations have the same kinetic friction coefficient, μ , and rugged channels have cuboid bulges of dimension $a_r \times a_r \times h_r$ that are uniformly spaced at d_r in x and z directions. We deposit icebergs in each batch from the same height and then they settle under gravity and buoyancy. Following pouring, the entire array of cubic particles is permitted to settle until static equilibrium is achieved, as shown in Fig. 5(a)(f). We then delete the right boundary wall so that the mélange has an open end in the ocean. We move the terminus on the left at a constant velocity, $V_{\text{ter}} = 30$ m/day for real fjord geometries [32, 33, 34], and 43.2 m/day for simplified fjord geometries [40, 24]. To confirm that the averaged steady-state buttressing force is invariant to the terminus velocity, we conducted simulations with mélange thickness 280 m, terminus velocity at 21.6 m/day, 43.2 m/day, and 86.4 m/day, for both straight fjords and rugged fjords configurations (Fig. 16 in SI). The results show that the averaged buttressing force is mostly invariant to the terminus velocity in both fjord configurations. Taking the force fluctuations into account, the maximum buttressing force difference among the chosen velocities is 4% and 8% for straight and rugged fjords, respectively. In rugged fjords, faster terminus motion leads to larger force fluctuations due to larger velocity gradient during stick-slip/jam-unjam cycles. We adopt the terminus velocity of 43.2 m/day to simulate mélange in simplified fjord geometries for the sake of computational efficiency.

Comment (3). Section 1.2 I found it challenging to understand from the given text how comparisons at Jakobshavn and Kangerlussuaq varied from Store. All three seem to show gradients in mélange freeboard heights that coincide to expected terminus activity, yet the text partitions them in a way that implies either the method or interpretation of the results vary between the two groups. I would suggest additional text early in the paragraph similar to “We acquire ICESat-2 elevation data near the terminus for these three glaciers, where mélange was continually present over our study period at X and X, and seasonally present at X. Where mélange persisted, we calculated seasonally distinct freeboard heights...etc”

On line 126, the results are described to support a hypothesis of thick mélange enabling winter terminus advance. Consider changing the language in the first paragraph of this section to better articulate this as a study hypothesis, which as written states a more general objective of identifying if there are ‘correlations between mélange thickness and calving dynamics.’

Response. We have revised Section 1.2 as follows:

1.2 Seasonal changes of mélange thickness and calving dynamics. To investigate whether there are correlations between ice mélange thickness and

calving dynamics on other glaciers, we use ICESat-2 observations of mélange surface elevation. While this dataset does not provide the temporal resolution to study individual calving events, we can leverage the observed seasonality in terminus advance and retreat at many Greenland glaciers to assess whether mélange thickness is correlated with periods of quiescence versus vigorous calving.

There are very few ICESat-2 tracks passing through the fronts of termini in different seasons, because positions of termini vary seasonally but ICESat-2 tracks are generally fixed in space. We identify ICESat-2 tracks passing over glacier termini in different seasons for Jakobshavn Isbræ (Fig. 2(a)), Kangerlussuaq Glacier (Fig. 2(b)), and Store Glacier (Fig. 2(c)), which are large discharging glaciers contributing to Greenland's mass losses [15]. We use the ATL06 data set from ICESat-2 that provides geolocated, land-ice surface heights above the WGS 84 ellipsoid [2]. After removing the geoid heights at glacier termini [16], the mélange surface elevations are heights above mean sea level with tidal variations [9]. We then remove the tidal trend of mélange elevations with the Greenland 1 kilometer tide model [17]. Surface elevation data is acquired along the ICESat-2 track and displayed as a function of the distance from terminus. We found that mélange was continuously present at Jakobshavn Isbræ from 2021 to 2022 and at Kangerlussuaq Glacier in 2020, while it was seasonally present at Store Glacier in 2019. Where mélange persisted, we calculated seasonally distinct freeboard heights from winter to early spring (solid black lines) and in summer (dashed black lines) (Fig. 2(d)(e)). Near the termini, mélange for the two glaciers both exhibit different ranges of freeboard heights during the two seasons: 20 ~ 35 m in winter or spring, and below 5 m in summer. The seasonal changes in mélange thickness at the terminus may explain the observed calving dynamics and terminus motion: zero or minor calving with an advancing terminus from winter to spring, and vigorous calving with a retreating terminus from summer to fall (Fig. 2(g)(h)). At Store Glacier, the mélange was present from 1 Jan to 14 June, after which calving resumed and the terminus kept retreating (Fig. 2(i)). The mélange elevation profile on 22 March 2019 exhibits a thickness gradient with a freeboard height of around 30 m near the terminus (Fig. 2(f)). In summary, ICESat-2 data provides strong evidence of correlations between mélange thickness and terminus seasonality.

Comment (4). *Section 1.6* Line 291 states that during winter advance, the buttressing force varies from 1.7 to 2.7×10^7 N/m (which exceeds the threshold suggested to impede calving at a floating terminus). However, later in the discussion, the average wintertime advance force is given of 6.5×10^6 N/m. What is the reason for this difference?

Response. Line 291 states that “For the studied glacier termini, the observed mélange thicknesses when terminus advances (85% in winter) range from 60_{-23}^{+21} m to 240_{-69}^{+52} m, with buttressing forces ranging from $1.7_{-1.1}^{+1.3} \times 10^6$ N/m to $2.7_{-1.4}^{+1.1} \times 10^7$ N/m.” The lower bound

is $1.7_{-1.1}^{+1.3} \times 10^6$ N/m instead of $1.7_{-1.1}^{+1.3} \times 10^7$ N/m. This can be clearly seen from Fig. 8 in the main text (Fig. 7 in this document). The red triangle markers correspond to buttressing forces ranging from $1.1_{-0.7}^{+0.9} \times 10^6$ N/m (ID 2 Hayes Glacier 2) to $2.7_{-1.4}^{+1.1} \times 10^7$ N/m (ID 31 Kangerlussuaq Glacier). In fact, we have made a mistake in the lower bound, which should be $1.1_{-0.7}^{+0.9} \times 10^6$ N/m instead of $1.7_{-1.1}^{+1.3} \times 10^6$ N/m, and we have revised the main text as follows (lines 392–404):

For the studied glacier termini, the observed mélange thicknesses when terminus advances (82% in late fall-to-spring, November to May) range from 49_{-19}^{+19} m to 240_{-69}^{+52} m, with buttressing forces ranging from $1.1_{-0.7}^{+0.9} \times 10^6$ N/m to $2.7_{-1.4}^{+1.1} \times 10^7$ N/m. Previous force balance analysis of a calving iceberg revealed that for a terminus at floatation, the mélange buttressing force of order $\sim 1.0 \times 10^7$ N/m is sufficient to inhibit calving by preventing iceberg rotation [41]. Finite element models suggested that the mélange buttressing force of this magnitude can also inhibit calving by suppressing fracture propagation [42, 43, 44, 45, 46]. Most of our inferred buttressing forces during terminus advance are consistent with the proposed threshold. The observed mélange thicknesses when terminus retreats (78% in summer-to-fall, June to October) range from 1_{-1}^{+11} m to 87_{-29}^{+26} m, with inferred buttressing forces ranging from $0.1_{-0.1}^{+6.2} \times 10^4$ N/m to $3.5_{-2.1}^{+2.1} \times 10^6$ N/m. Therefore in summer and fall, mélange is generally too thin to inhibit calving. Here we report the buttressing force in N/m, which can be compare to other studies including field observations [27], simulations [40, 24], and analytical analysis [41].

Therefore, the averaged value of the buttressing force observed during terminus advance (red triangles in Fig. 7(b)) is $6.5_{-3.7}^{+3.4} \times 10^6$ N/m (lines 432-435):

When termini advance in winter, the average value of all observed mélange thicknesses is 119_{-37}^{+31} m, with a corresponding buttressing force $6.5_{-3.7}^{+3.4} \times 10^6$ N/m. When termini retreat in summer, the average thickness is 34_{-15}^{+17} m, with a corresponding buttressing force of $5.2_{-3.8}^{+5.9} \times 10^5$ N/m.

We have explained the deviation of observed buttressing force during terminus advance from the theoretical threshold $\sim 1 \times 10^7$ N/m suggested to impede calving at a floating terminus as follow (lines 453–459):

If calving dynamics are controlled by mélange buttressing, then our analysis infers that the minimum buttressing force required to inhibit calving varies across termini from $1.1_{-0.7}^{+0.9} \times 10^6$ (Hayes Glacier 2) to $9.3_{-5.2}^{+4.6} \times 10^6$ N/m (Kangerlussuaq Glacier). Such variations in the buttressing threshold could be attributed to spatial variations in ice velocities, terminus geometry, bed topography, basal friction, oceanic and atmospheric forcings, etc. Our analysis offers a new framework to mechanistically study the effects of mélange buttressing and other ice-ocean interactions on calving.

FIGURE 7. Seasonal changes of mélange thickness and buttressing forces across 32 Greenland glacier termini in 2013-2022. (a) Locations of the 32 studied glacier termini are shown as pendular markers on the background Greenland ice velocity map belonging to 1 Dec 2020 - 30 Nov 2021. We also present zoomed in views of the studied glacier termini in northwest (NW), central west (CW), southeast (SE), and central east (CE) regions of Greenland. (b) From 2013 to 2022, the observed mélange freeboard heights at the terminus (Z_0) from all available ArcticDEM (108 in total). Red triangular markers correspond to DEM acquired when terminus advances (82% in late fall-to-spring, November to May), and black circular markers correspond to DEM acquired when terminus retreats (78% in summer-to-fall, June to October). The horizontal axis contains three variables: the mélange freeboard height directly retrieved from DEM (Z_0), the inferred mélange thickness from hydrostatic equilibrium (H_0), and the inferred mélange buttressing force from Eqn. (2) (F/W). Data used to calculate the buttressing forces and their uncertainties are listed in Table 2 in SI.

We have also added sentences in the discussion section to emphasize that the observed thick mélange correlates with seasonal terminus advances but might not be the cause of it, as follows (lines 419–446):

Previous research suggests that the presence of ice mélange can reduce iceberg calving by providing “backstress” to the terminus [9, 47, 40, 48, 41, 12, 24, 20, 43, 42, 49]. Our comparisons of time-varying mélange thickness and calving dynamics at Helheim Glacier (Fig. 1(b)) support the view that the buttressing force increases with the mélange thickness and thus inhibits calving. Our TLS scans indicate a correlation between mélange thickness and calving dynamics, but we cannot determine causality before examining other mechanisms driving calving dynamics. For instance, the terminus advances right after major

calving events can be explained by the possibility that the new glacier front has fewer crevasses and will take a longer period to calve again. Alternatively, the mélange thickness and the terminus react simultaneously but independently to other oceanic and atmospheric forcing. To establish a causal relationship between thin mélange and calving events, we would need in-situ observations with high temporal resolution in minutes to capture the sequence of a calving event and a mélange thinning event [35, 9]. Scanning through 108 ArcticDEM strips, we discover calving dynamics associated with mélange thickness seasonality across 32 Greenland glacier termini in 2013-2022. When termini advance in cold months, the average value of all observed mélange thicknesses is 119_{-37}^{+31} m, with a corresponding buttressing force $6.5_{-3.7}^{+3.4} \times 10^6$ N/m. When termini retreat in warm months, the average thickness is 34_{-15}^{+17} m, with a corresponding buttressing force of $5.2_{-3.8}^{+5.9} \times 10^5$ N/m.

While we have observed strong evidence of correlations between mélange thickness and terminus seasonality, understanding their causality requires considerations of other environmental forcings. Previous research shows that seasonal terminus positions for some central west Greenland glaciers with small-magnitude calving events correlate stronger with glacial runoff than mélange presence or ocean thermal forcing [50]. On the other hand, researchers observe slowdown and thickening of Jakobshavn since 2016 and attribute it to concurrent cooling of ocean waters [51]. Analytical and numerical models imply that submarine melting can amplify calving by melt-undercutting [52, 53]. We note that if submarine melting causes the observed summer thinning of mélange, mélange's buttressing strength can be strongly tied to submarine melting. The impact of submarine melting on mélange strength can be significant due to the strong dependence of buttressing on mélange thickness inferred in our study.

Comment (5). On filtering ArcticDEM strip data The methods section of the manuscript describes the process of excluding strips when the terminus behavior is variable (both advance and retreat) within a 2-month window surrounding the strip acquisition date. This filtering step reduces the strip number from > 300 to 108. I think this is relevant information to describe earlier in the paper, in section 1.6. I understand why the authors used this approach given the study's main questions, but the fact that the majority of scenes featured episodically changing terminus behavior is important for evaluating when and under what conditions mélange thickness-to-buttressing force calculations are most applicable to understanding terminus change.

Response. We thank the reviewer for the constructive advice. We have added Figure 8 in SI (Fig. 8 in this document), which presents an example of the DEM filtering procedure at Helheim Glacier, resulting in two TLS-derived DEMs during terminus advances, and two ArcticDEM strips during terminus retreats. We have also moved the DEM filtering procedure from the Methods section to the Results section as follows:

1.6 Calving dynamics associated with mélange buttressing force seasonality across 32 Greenland glacier termini in 2013-2022. ...

We then extend our study to 32 glacier termini, most of which (ID 1~25) are picked from previous studies with strong terminus-position seasonality [10, 11], and the rest (ID 26~32) have annual ice discharge larger than 5 Gt/yr [15]. The locations of the termini are marked on a Greenland velocity map in Fig. 7(a). We identify 341 ArcticDEM strips at 2-meter resolution that cover the mélange regions for the 32 studied termini. For each DEM strip, we investigate terminus position variations [7] during a two-month time window centering on the DEM acquisition date. If the terminus keeps advancing (or retreating) within the time window, then the DEM potentially represents mélange with a strong (or weak) buttressing force. If the terminus alternates between advancing and retreating within the time window, we discard the corresponding DEM strip because the relationship between mélange and calving dynamics is ambiguous in this case. After filtering all DEM strips through this criterion, we identify 60 DEM strips during terminus advances, and 50 DEM strips during terminus retreats, from February to November in 2013–2022. Figure 8 in SI presents an example of the DEM filtering procedure at Helheim Glacier, resulting in two TLS-derived DEMs during terminus advances (Fig. 1(c)(e)), and two ArcticDEM strips during terminus retreats. For every ArcticDEM strip, the mélange thickness is defined as the maximum value obtained from the representative mélange thickness profile within a 200-meter range from the terminus. Table. 2 in SI summarizes the observed minimum (or maximum) mélange thickness when terminus retreats (or advances) as H_0^{\min} (or H_0^{\max}), with the corresponding DEM acquisition month shown in the bracket.

FIGURE 8. A time series of terminus position [7] and observed mélange freeboard heights (Z_0) at the terminus of Helheim Glacier. Black dots denote the terminus position where the positive sign indicates terminus advance. Blue dots and squares denote the mélange elevation observed at the terminus, Z_0 , from TLS and ArcticDEM strips, respectively. The solid red (or dashed black) line marks the data acquisition date when the terminus consistently advances (or retreats) within the two-month time window centering on that date, indicating that the DEM potentially represents mélange with a strong (or weak) buttressing force. We exclude DEMs when the terminus behavior alternates between advancing and retreating within the two-month time window, because the relationship between mélange and calving dynamics is ambiguous in this case. We present only the filtered DEM data (i.e., four data points at Helheim Glacier) in Figure 8 in the main text.

Comment (6). Minor comments/requests for clarification

-Heading of section 1.5, consider editing to “at the glacier terminus”.

Response. We have changed the heading as follows:

1.5 Mélange buttressing force increases with mélange thickness at the glacier terminus.

Comment (7). -It may be helpful to specify the months corresponding to the seasonal descriptions (spring, winter, summer...). These may be the obvious 3-month definitions, but there are places in the text (for example, that 85% of advance occurred in winter) where it was unclear whether winter indicated DJF or more general “cold” vs “warm” season.

Response. We have revised the figure caption of Figure 7 in the main text (Fig. 9 in this document), and the legend and caption of Figure 8 in the main text (Fig. 7 in this document). We have also revised the main text as follows:

1.6 Calving dynamics associated with mélange thickness seasonality across 32 Greenland glacier termini in 2013-2022. Our models reveal that the mélange buttressing force can be predicted solely from remote sensing observations of its thickness at glacier terminus (Eqn. (2)). However, further investigation is needed to address the question of how does the spatio-temporal variations in mélange thickness correlate with calving dynamics in Greenland. Recent studies covering the period from 2015 to 2021 found that among 219

marine-terminating glaciers in Greenland, nearly 80% of them showed significant seasonal variations in terminus position, which retreat in summer and advance in winter [54]. We hypothesize that the seasonal terminus-position variability could be induced by a mélange thickness seasonality. To test this hypothesis, we collect available ArcticDEM strips at Jakobshavn Isbræ in the past decade, and compare DEM acquisition dates to a time series of the terminus position (Fig. 9(a)). Among the eight DEM strips, five of them (dashed lines) are acquired in **late spring-to-summer (May-Aug)** when the terminus retreats, and three of them (solid lines) are acquired in **fall-to-early spring (Sep-Mar)** when the terminus advances. From the corresponding mélange elevation profiles constructed the same way as in Fig. 1 (solid blue lines in Fig. 9(f)-(i)), we first confirm that they are not contaminated by large icebergs whose elevation values are above 30 m (Fig. 9(b)-(e)). The elevation profile successfully reflects the overall thickness variations within the mélange that piled up from small icebergs. We observe that the freeboard height of the mélange at the terminus ranges from 2.8 ~ 3.9 m in **warm months** and 19.2 ~ 26.8 m in **cold months**.

FIGURE 9. Seasonal variations in mélange thickness coincide with calving dynamics at Jakobshavn Isbræ. (a) Terminus position in 2013–2022 [7] where positive sign indicates the direction of advance. Eight vertical black lines mark the acquisition dates of available ArcticDEMs, four of which are presented in (b)–(e), corresponding to 24 May 2014, 27 Jun 2018, 18 Sep 2018, and 27 Mar 2019, respectively. Solid black lines mark the dates with terminus advances, and dashed black lines mark the dates with terminus retreats. (b)–(e) The mélange elevation above mean sea level from ArcticDEM strips, overlain on satellite images acquired around the same date. Black line across the fjord indicates glacier front location. The images are in polar stereographic projection (EPSG: 3413). (f)–(i) Surface elevation profiles for the mélange displayed as density plots (8,649,023 ~ 18,183,005 data points in total) constructed the same way as in Fig. 1. **For any specific distance from terminus, we find the elevation value that has the maximum number of data points. Solid blue lines connect these elevation values along the distance from terminus as the representative mélange elevation profiles. We observe thick mélange in cold months when terminus advances, and thin mélange in warm months when terminus retreats.**

We then extend our study to 32 glacier termini, most of which (ID 1~25) are picked from previous studies with strong terminus-position seasonality [10, 11], and the rest (ID 26~32) have annual ice discharge larger than 5 Gt/yr [15]. The locations of the termini are marked on a Greenland velocity map in Fig. 7(a). We identify 341 ArcticDEM strips at 2-meter resolution that cover the mélange regions for the 32 studied termini. For each DEM strip, we investigate terminus position variations [7] during a two-month time window centering on the DEM acquisition date. If the terminus keeps advancing (or retreating) within the time window, then the DEM potentially represents mélange with a strong (or weak) buttressing force. If the terminus alternates between advancing and retreating within the time window, we discard the corresponding DEM strip because the relationship between mélange and calving dynamics is ambiguous in this case. After filtering all DEM strips through this criterion, we identify 60 DEM strips during terminus advances, and 50 DEM strips during terminus retreats, from February to November in 2013–2022. Figure 7 in SI presents an example of the DEM filtering procedure at Helheim Glacier, resulting in two TLS-derived DEMs during terminus advances (Fig. 1(c)(e)), and two ArcticDEM strips during terminus retreats. Table. 2 in SI summarizes the observed minimum (or maximum) mélange thickness when terminus retreats (or advances) as H_0^{\min} (or H_0^{\max}), with the corresponding DEM acquisition month shown in the bracket. We also present all observed mélange freeboard heights at the terminus (Z_0) in Fig. 7(b). A complete catalog of terminus position variations, DEM acquisition dates and mélange freeboard heights for 32 studied termini is summarized in Fig. 16~47 in SI. Assuming the mélange to be densely-packed with $\phi_0 = 0.9_{-0.26}^{+0.1}$, ice density in the plausible range of 910_{-40}^{+10} kg/m³ and water density of 1028_{-8}^{+1} kg/m³, we arrive at the mélange buttressing force per unit width (F/W) through Eqn. (2). For the studied glacier termini, the observed mélange thicknesses when terminus advances (**82% in late fall-to-spring, November to May**) range from 49_{-19}^{+19} m to 240_{-69}^{+52} m, with buttressing forces ranging from $1.1_{-0.7}^{+0.9} \times 10^6$ N/m to $2.7_{-1.4}^{+1.1} \times 10^7$ N/m. Previous force balance analysis of a calving iceberg revealed that for a terminus at floatation, the mélange buttressing force of order $\sim 1.0 \times 10^7$ N/m is sufficient to inhibit calving by preventing iceberg rotation [41]. Finite element models suggested that the mélange buttressing force of this magnitude can also inhibit calving by suppressing fracture propagation [42, 43, 44, 45, 46]. Most of our inferred buttressing forces during terminus advance are consistent with the proposed threshold. The observed mélange thicknesses when terminus retreats (**78% in summer-to-fall, June to October**) range from 1_{-1}^{+11} m to 87_{-29}^{+26} m, with inferred buttressing forces ranging from $0.1_{-0.1}^{+6.2} \times 10^4$ N/m to $3.5_{-2.1}^{+2.1} \times 10^6$ N/m. Therefore **in summer and fall**, mélange is generally too thin to inhibit calving.

We have also revised sentences in the discussion section as follows (lines 432–435):

When termini advance **in cold months**, the average value of all observed mélange thicknesses is 119_{-37}^{+31} m, with a corresponding buttressing force $6.5_{-3.7}^{+3.4} \times 10^6$ N/m. When termini retreat **in warm months**, the average thickness is 34_{-15}^{+17} m, with a corresponding buttressing force of $5.2_{-3.8}^{+5.9} \times 10^5$ N/m.

Comment (8). *Line 339 through 342: consider adding glacier ID's in parentheses in the discussion text for easy comparison to figure 7.*

Response. We have made major revisions on the referred paragraph based on another reviewer's comment as follows:

“L338-345: This is an interesting piece of evidence that would appear to support your argument about mélange control on terminus advance/retreat. However, none of this evidence is presented in the results and there is no accompanying figure to support these surprising findings. I am finding it a little hard to believe that there is 100 m thick mélange in July and August in Greenland. Can you show a figure that demonstrates better demonstrates this finding for these six glaciers? You could also go into more detail. For example, do these glaciers always have mélange at this time of year? If not then you could look at interannual advance/retreat patterns to provide more evidence to support your hypothesis.”

Response. The reviewer suggests that to better support our argument that thick mélange in summertime leads to terminus advance at the six glaciers (Hayes SS, Alison, Unnamed Deception, Unnamed Uunartit Islands, Koge Bugt C, and Kong Oscar), we should examine interannual advance/retreat patterns. Specifically, we should check whether unusually thick mélange is observed in June/July only in the years when terminus advances in summer, and thin mélange is observed in June/July in other years when terminus retreats in summer. After checking the available ArcticDEM data and terminus interannual advance/retreat patterns, we found abundant evidence at Koge Bugt C glacier (ID=13 in Fig. 8 in the main text) to support our argument. We found that the terminus advanced from July to October in 2015 and retreated during the same period in 2014, 2016 and 2017 [7]. From ArcticDEM strips acquired in June/July in 2014–2017, we only observe thick mélange in 2015, and thin mélange in other years. We attribute the summertime terminus advance in 2015 to mélange buttressing from the presence of unusually thick mélange observed on 2 Jul with a thickness of 114_{-36}^{+30} m (see Fig. 9 in SI and Fig. 10 in this document), the same as what happened at Jakobshavn Isbræ in Jun 2016 [9]. We have modified the main text as follows (lines 447–459):

We note that the hypothesis of summer-runoff induced calving, on its own, can not explain the observations at **Koge Bugt C Glacier (ID=13 in Fig. 7) where the terminus advanced from July to September in 2015 and retreated or remained a plateau during the same period in 2014, 2016, 2017 and 2020 [7]. We attribute the summertime terminus advance in 2015 to mélange buttressing from the presence of unusually thick mélange observed on 2 Jul with a thickness of 114_{-36}^{+30} m (see Fig. 9 in SI for details), the same as what happened at Jakobshavn Isbræ in Jun 2016 [9].** If calving dynamics are controlled by mélange

buttressing, then our analysis infers that the minimum buttressing force required to inhibit calving varies across termini from $1.1_{-0.7}^{+0.9} \times 10^6$ (Hayes Glacier 2) to $9.3_{-5.2}^{+4.6} \times 10^6$ N/m (Kangerlussuaq Glacier). Such variations in the buttressing threshold could be attributed to spatial variations in ice velocities, terminus geometry, bed topography, basal friction, oceanic and atmospheric forcings, etc. Our analysis offers a new framework to mechanistically study the effects of mélange buttressing and other ice-ocean interactions on calving.

We also examined ArcticDEM data and terminus interannual advance/retreat patterns for the other five glaciers. We found that the termini advance from May to July in multiple years and there are very few ArcticDEM strips acquired in June/July, which prevents us from revealing any correlation between mélange thickness and terminus dynamics in summer (Fig. 11–15 in this document). Therefore, we have excluded them from the discussion section.

FIGURE 10. (a)-(e) The mélange elevation map above mean sea level at Koge Bugt C Glacier from ArcticDEM observations, acquired in June and July from 2014–2020, overlain on the satellite image acquired around the same date. The image is in polar stereographic projection (EPSG: 3413). The date of ArcticDEM acquisition is shown on the upper left corner. (f)-(j) The surface elevation profile for the mélange presented in (a)–(e) displayed as a density plot. The solid blue line is the representative mélange elevation profile as a function of distance from terminus. (k) A time series of terminus position [7], ArcticDEM acquisition dates and observed mélange freeboard heights (Z_0) at the terminus of Koge Bugt C Glacier. The solid red line marks the ArcticDEM acquisition date when an unusually thick mélange is observed at the terminus in July with the terminus advancing from May to October in 2015. The solid black lines mark ArcticDEM acquisition dates around the same time (late June to July) in different years (2014, 2016, 2017, 2020) when thin mélange is observed at the terminus with the terminus retreating or remaining a plateau from July to September (the shaded gray regions).

FIGURE 11. (a),(c) The mélangé elevation map above mean sea level at Hayes Glacier SS from ArcticDEM observations, acquired on 28 May 2016 and 5 Jun 2018, overlain on satellite images acquired around the same dates. Images are in polar stereographic projection (EPSG: 3413). The date of ArcticDEM acquisition is shown on the upper left corner. (b),(d) The surface elevation profile for the mélangé presented in (a),(c) displayed as a density plot. The solid blue line is the representative mélangé elevation profile as a function of distance from terminus. (e) A time series of terminus position [7], ArcticDEM acquisition dates and observed mélangé freeboard heights (Z_0) at the terminus of Hayes Glacier SS.

FIGURE 12. (a),(c) The mélange elevation map above mean sea level at Alison Glacier from ArcticDEM observations, acquired on 18 Jun 2015 and 16 Jun 2017, overlain on satellite images acquired around the same dates. Images are in polar stereographic projection (EPSG: 3413). The date of ArcticDEM acquisition is shown on the upper left corner. (b),(d) The surface elevation profile for the mélange presented in (a),(c) displayed as a density plot. The solid blue line is the representative mélange elevation profile as a function of distance from terminus. (e) A time series of terminus position [7], ArcticDEM acquisition dates and observed mélange freeboard heights (Z_0) at the terminus of Alison Glacier.

FIGURE 13. (a) The mélange elevation map above mean sea level at Unnamed Deception Glacier from ArcticDEM observations, acquired on 21 Jun 2016, overlain on a satellite image acquired around the same date. The image is in polar stereographic projection (EPSG: 3413). The date of ArcticDEM acquisition is shown on the upper left corner. (b) The surface elevation profile for the mélange presented in (a) displayed as a density plot. The solid blue line is the representative mélange elevation profile as a function of distance from terminus. (c) A time series of terminus position [7], ArcticDEM acquisition date and observed mélange freeboard height (Z_0) at the terminus of Unnamed Deception Glacier.

FIGURE 14. (a) The mélange elevation map above mean sea level at Unnamed Uunartit Islands from ArcticDEM observations, acquired on 7 Aug 2018, overlain on a satellite image acquired around the same date. The image is in polar stereographic projection (EPSG: 3413). The date of ArcticDEM acquisition is shown on the upper left corner. (b) The surface elevation profile for the mélange presented in (a) displayed as a density plot. The solid blue line is the representative mélange elevation profile as a function of distance from terminus. (c) A time series of terminus position [7], ArcticDEM acquisition date and observed mélange freeboard height (Z_0) at the terminus of Unnamed Uunartit Islands.

FIGURE 15. (a)–(e) The mélangé elevation map above mean sea level at Kong Oscar Glacier from ArcticDEM observations, acquired in July from 2014–2020, overlain on the satellite image acquired around the same date. The image is in polar stereographic projection (EPSG: 3413). The date of ArcticDEM acquisition is shown on the upper left corner. (f)–(j) The surface elevation profile for the mélangé presented in (a)–(e) displayed as a density plot. The solid blue line is the representative mélangé elevation profile as a function of distance from terminus. (k) A time series of terminus position [7], ArcticDEM acquisition dates and observed mélangé freeboard heights (Z_0) at the terminus of Kong Oscar Glacier.

Comment (9). Line 267 correct “We hypothesis” to “we hypothesize”

Response. We have corrected the word as follows:

We **hypothesize** that the seasonal terminus-position variability could be induced by a mélangé thickness seasonality.

Comment (10). Figure 6: Consider again including a note that the blue line follows the highest density elevation in the figure caption.

Response. We have revised the figure caption as suggested by the reviewer. See Figure 9 in this document.

In summary, we thank the referee for making these constructive suggestions. We hope that our responses have resolved the ambiguities he/she has pointed out. We believe these additions have improved our manuscript significantly.

REFERENCES

- [1] Yue Meng, Ching-Yao Lai, Riley Culberg, Michael G. Shahin, Leigh A. Stearns, Justin C. Burton, and Kavinda Nissanka. Supporting data – seasonal changes of mélange thickness coincide with greenland calving dynamics, 2024.
- [2] B. Smith, S. Adusumilli, B. M. Csathó, D. Felikson, H. A. Fricker, A. S. Gardner, N. Holschuh, J. Lee, J. Nilsson, F. Paolo, M. R. Siegfried, T. Sutterley, and the ICESat-2 Science Team. Atlas/icesat-2 l3a land ice height, version 6, 2023.
- [3] Claire Porter, Ian Howat, Myoung-Jon Noh, Erik Husby, Samuel Khuvis, Evan Danish, Karen Tomko, Judith Gardiner, Adelaide Negrete, Bidhyananda Yadav, James Klassen, Cole Kelleher, Michael Cloutier, Jesse Bakker, Jeremy Enos, Galen Arnold, Greg Bauer, and Paul Morin. ArcticDEM - Strips, Version 4.1, 2022.
- [4] Claire Porter, Ian Howat, Myoung-Jon Noh, Erik Husby, Samuel Khuvis, Evan Danish, Karen Tomko, Judith Gardiner, Adelaide Negrete, Bidhyananda Yadav, James Klassen, Cole Kelleher, Michael Cloutier, Jesse Bakker, Jeremy Enos, Galen Arnold, Greg Bauer, and Paul Morin. ArcticDEM - Mosaics, Version 4.1, 2023.
- [5] I. Joughin. Measures greenland annual ice sheet velocity mosaics from sar and landsat, version 5, 2023.
- [6] M. et al. Morlighem. Icebridge bedmachine greenland, version 5, 2022.
- [7] Enze Zhang, Ginny Catania, and Daniel T Trugman. Autoterm: an automated pipeline for glacier terminus extraction using machine learning and a “big data” repository of greenland glacier termini. *The Cryosphere*, 17(8):3485–3503, 2023.
- [8] Itasca Consulting Group, Inc. *PFC — Particle Flow Code, Ver. 7.0*. Minneapolis: Itasca, 2021.
- [9] Surui Xie, Timothy H Dixon, David M Holland, Denis Voytenko, and Irena Vaňková. Rapid iceberg calving following removal of tightly packed pro-glacial mélange. *Nature communications*, 10(1):3250, 2019.
- [10] Twila Moon, Ian Joughin, Ben Smith, Michiel R Van Den Broeke, Willem Jan Van De Berg, Brice Noël, and Mika Usher. Distinct patterns of seasonal greenland glacier velocity. *Geophysical research letters*, 41(20):7209–7216, 2014.
- [11] Saurabh Vijay, Shfaqat Abbas Khan, Anders Kusk, Anne M Solgaard, Twila Moon, and Anders Anker Bjørk. Resolving seasonal ice velocity of 45 greenlandic glaciers with very high temporal details. *Geophysical Research Letters*, 46(3):1485–1495, 2019.
- [12] Ryan Cassotto, Mark Fahnestock, Jason M Amundson, Martin Truffer, and Ian Joughin. Seasonal and interannual variations in ice melange and its impact on terminus stability, jakobshavn isbræ, greenland. *Journal of Glaciology*, 61(225):76–88, 2015.
- [13] Adrien Wehrlé, Martin P Lüthi, and Andreas Vieli. The control of short-term ice mélange weakening episodes on calving activity at major greenland outlet glaciers. *The Cryosphere*, 17(1):309–326, 2023.
- [14] Thomas R Chudley, Ian M Howat, Michalea D King, and Adelaide Negrete. Atlantic water intrusion triggers rapid retreat and regime change at previously stable greenland glacier. *Nature Communications*, 14(1):2151, 2023.

- [15] Jérémie Mouginot, Eric Rignot, Anders A Bjørk, Michiel Van den Broeke, Romain Millan, Mathieu Morlighem, Brice Noël, Bernd Scheuchl, and Michael Wood. Forty-six years of greenland ice sheet mass balance from 1972 to 2018. *Proceedings of the national academy of sciences*, 116(19):9239–9244, 2019.
- [16] Förste Ch, Sean L Bruinsma, Oleg Abrikosov, Jean-Michel Lemoine, T Schaller, HJ Gtze, J Ebbing, JC Marty, F Flechtner, G Balmino, et al. Eigen-6c4 the latest combined global gravity field model including goce data up to degree and order 2190 of gfz potsdam and grgs toulouse. *GFZ Data Services*, 10(10.5880), 2014.
- [17] Susan L Howard and Laurie Padman. Gr1kmtm: Greenland 1 kilometer tide model, 2021.
- [18] Connor J Shiggins, James M Lea, and Stephen Brough. Automated arcticdem ice-berg detection tool: insights into area and volume distributions, and their potential application to satellite imagery and modelling of glacier–iceberg–ocean systems. *The Cryosphere*, 17(1):15–32, 2023.
- [19] Ellyn M Enderlin, Gordon S Hamilton, Fiammetta Straneo, and David A Sutherland. Iceberg meltwater fluxes dominate the freshwater budget in greenland’s ice-berg-congested glacial fjords. *Geophysical Research Letters*, 43(21):11–287, 2016.
- [20] Jason M Amundson and JC Burton. Quasi-static granular flow of ice mélange. *Journal of Geophysical Research: Earth Surface*, 123(9):2243–2257, 2018.
- [21] Lufeng Liu, Zhuoran Li, Yang Jiao, and Shuixiang Li. Maximally dense random packings of cubes and cuboids via a novel inverse packing method. *Soft matter*, 13(4):748–757, 2017.
- [22] ShuiXiang Li, Jian Zhao, Peng Lu, and Yu Xie. Maximum packing densities of basic 3d objects. *Chinese Science Bulletin*, 55(2):114–119, 2010.
- [23] Kishor G Nayar, Mostafa H Sharqawy, Leonardo D Banchik, et al. Thermophysical properties of seawater: A review and new correlations that include pressure dependence. *Desalination*, 390:1–24, 2016.
- [24] Justin C Burton, Jason M Amundson, Ryan Cassotto, Chin-Chang Kuo, and Michael Dennin. Quantifying flow and stress in ice mélange, the world’s largest granular material. *Proceedings of the National Academy of Sciences*, 115(20):5105–5110, 2018.
- [25] A. C. Fowler. Glaciers and ice sheets. In Jesús Idefonso Díaz, editor, *The Mathematics of Models for Climatology and Environment*, pages 301–336, Berlin, Heidelberg, 1997. Springer Berlin Heidelberg.
- [26] Jeremy N Bassis and Samuel B Kachuck. Beyond the stokes approximation: shallow visco-elastic ice-sheet models. *Journal of Glaciology*, pages 1–12, 2023.
- [27] Jae Hun Kim, Eric Rignot, David Holland, and Denise Holland. Seawater intrusion at the grounding line of jakobshavn isbræ, greenland, from terrestrial radar interferometry. *Geophysical Research Letters*, 51(6):e2023GL106181, 2024.
- [28] Adeline Favier de Coulomb, Mehdi Bouzid, Philippe Claudin, Eric Clément, and Bruno Andreotti. Rheology of granular flows across the transition from soft to rigid particles. *Physical Review Fluids*, 2(10):102301, 2017.
- [29] M Yasinul Karim and Eric I Corwin. Eliminating friction with friction: 2d janssen effect in a friction-driven system. *Physical Review Letters*, 112(18):188001, 2014.

- [30] Douglas R MacAyeal. Large-scale ice flow over a viscous basal sediment: Theory and application to ice stream b, antarctica. *Journal of Geophysical Research: Solid Earth*, 94(B4):4071–4087, 1989.
- [31] Daniel J Sulak, David A Sutherland, Ellyn M Enderlin, Leigh A Stearns, and Gordon S Hamilton. Iceberg properties and distributions in three greenlandic fjords using satellite imagery. *Annals of Glaciology*, 58(74):92–106, 2017.
- [32] Laura M Kehrl, Ian Joughin, David E Shean, Dana Floricioiu, and Lukas Krieger. Seasonal and interannual variabilities in terminus position, glacier velocity, and surface elevation at helheim and kangerlussuaq glaciers from 2008 to 2016. *Journal of Geophysical Research: Earth Surface*, 122(9):1635–1652, 2017.
- [33] Gong Cheng, Mathieu Morlighem, Jérémie Mouginot, and Daniel Cheng. Helheim glacier’s terminus position controls its seasonal and inter-annual ice flow variability. *Geophysical Research Letters*, 49(5):e2021GL097085, 2022.
- [34] Lizz Ultee, Denis Felikson, Brent Minchew, Leigh A Stearns, and Bryan Riel. Helheim glacier ice velocity variability responds to runoff and terminus position change at different timescales. *Nature Communications*, 13(1):6022, 2022.
- [35] Ryan K Cassotto, Justin C Burton, Jason M Amundson, Mark A Fahnestock, and Martin Truffer. Granular decoherence precedes ice mélange failure and glacier calving at jakobshavn isbræ. *Nature Geoscience*, 14(6):417–422, 2021.
- [36] Suzanne L Bevan, Adrian J Luckman, Douglas I Benn, Tom Cowton, and Joe Todd. Impact of warming shelf waters on ice mélange and terminus retreat at a large se greenland glacier. *The Cryosphere*, 13(9):2303–2315, 2019.
- [37] Ian Joughin, David E Shean, Benjamin E Smith, and Dana Floricioiu. A decade of variability on jakobshavn isbræ: ocean temperatures pace speed through influence on mélange rigidity. *The Cryosphere*, 14(1):211–227, 2020.
- [38] Sierra M Melton, Richard B Alley, Sridhar Anandakrishnan, Byron R Parizek, Michael G Shahin, Leigh A Stearns, Adam L LeWinter, and David C Finnegan. Meltwater drainage and iceberg calving observed in high-spatiotemporal resolution at helheim glacier, greenland. *Journal of Glaciology*, 68(270):812–828, 2022.
- [39] Jason M Amundson, Christian Kienholz, Alexander O Hager, Rebecca H Jackson, Roman J Motyka, Jonathan D Nash, and David A Sutherland. Formation, flow and break-up of ephemeral ice mélange at leconte glacier and bay, alaska. *Journal of Glaciology*, 66(258):577–590, 2020.
- [40] Alexander A Robel. Thinning sea ice weakens buttressing force of iceberg mélange and promotes calving. *Nature Communications*, 8(1):14596, 2017.
- [41] Jason M Amundson, Mark Fahnestock, Martin Truffer, Jed Brown, Martin P Lüthi, and Roman J Motyka. Ice mélange dynamics and implications for terminus stability, jakobshavn isbræ, greenland. *Journal of Geophysical Research: Earth Surface*, 115(F1), 2010.
- [42] Joe Todd and Poul Christoffersen. Are seasonal calving dynamics forced by buttressing from ice mélange or undercutting by melting? outcomes from full-stokes simulations of store glacier, west greenland. *The Cryosphere*, 8(6):2353–2365, 2014.

- [43] J Krug, G Durand, O Gagliardini, and J Weiss. Modelling the impact of submarine frontal melting and ice mélange on glacier dynamics. *The Cryosphere*, 9(3):989–1003, 2015.
- [44] Joe Todd, Poul Christoffersen, Thomas Zwinger, Peter Råback, Nolwenn Chauché, Doug Benn, Adrian Luckman, Johnny Ryan, Nick Toberg, Donald Slater, et al. A full-stokes 3-d calving model applied to a large greenlandic glacier. *Journal of Geophysical Research: Earth Surface*, 123(3):410–432, 2018.
- [45] Joe Todd, Poul Christoffersen, Thomas Zwinger, Peter Råback, and Douglas I Benn. Sensitivity of a calving glacier to ice–ocean interactions under climate change: new insights from a 3-d full-stokes model. *The Cryosphere*, 13(6):1681–1694, 2019.
- [46] Jamie Barnett, Felicity A Holmes, and Nina Kirchner. Modelled dynamic retreat of kangerlussuaq glacier, east greenland, strongly influenced by the consecutive absence of an ice mélange in kangerlussuaq fjord. *Journal of Glaciology*, 69(275):433–444, 2023.
- [47] Twila Moon, Ian Joughin, and Ben Smith. Seasonal to multiyear variability of glacier surface velocity, terminus position, and sea ice/ice mélange in northwest greenland. *Journal of Geophysical Research: Earth Surface*, 120(5):818–833, 2015.
- [48] Steve Foga, Leigh A Stearns, and CJ Van der Veen. Application of satellite remote sensing techniques to quantify terminus and ice mélange behavior at helheim glacier, east greenland. *Marine Technology Society Journal*, 48(5):81–91, 2014.
- [49] Jacob I Walter, Jason E Box, Slawek Tulaczyk, Emily E Brodsky, Ian M Howat, Yushin Ahn, and Abel Brown. Oceanic mechanical forcing of a marine-terminating greenland glacier. *Annals of Glaciology*, 53(60):181–192, 2012.
- [50] MJ Fried, GA Catania, LA Stearns, DA Sutherland, TC Bartholomaus, E Shroyer, and J Nash. Reconciling drivers of seasonal terminus advance and retreat at 13 central west greenland tidewater glaciers. *Journal of Geophysical Research: Earth Surface*, 123(7):1590–1607, 2018.
- [51] Ala Khazendar, Ian G Fenty, Dustin Carroll, Alex Gardner, Craig M Lee, Ichiro Fukumori, Ou Wang, Hong Zhang, Hélène Seroussi, Delwyn Moller, et al. Interruption of two decades of jakobshavn isbrae acceleration and thinning as regional ocean cools. *Nature Geoscience*, 12(4):277–283, 2019.
- [52] Douglas I Benn, JAN Åström, Thomas Zwinger, JOE Todd, Faezeh M Nick, Susan Cook, Nicholas RJ Hulton, and Adrian Luckman. Melt-under-cutting and buoyancy-driven calving from tidewater glaciers: new insights from discrete element and continuum model simulations. *Journal of Glaciology*, 63(240):691–702, 2017.
- [53] DA Slater, DI Benn, TR Cowton, JN Bassis, and JA Todd. Calving multiplier effect controlled by melt undercut geometry. *Journal of Geophysical Research: Earth Surface*, 126(7):e2021JF006191, 2021.
- [54] Taryn E Black and Ian Joughin. Weekly to monthly terminus variability of greenland’s marine-terminating outlet glaciers. *The Cryosphere*, 17(1):1–13, 2023.

REPLY TO REFEREE COMMENTS - REFEREE 2.

We thank the referee for a constructive review of our paper. The referee states that “Individual sections were generally well-written, both the observations and modeling approaches were interesting, and the conclusions are potentially quite important to the glaciological community.” He/she states that “The main concern is that the manuscript is set-up as observationally-driven modeling study but does not actually use the observations to inform the modeling. I was expecting that the modeling would be carefully constrained with validated ice mélange thicknesses, observed iceberg distributions, and channel configurations that actually resembled real fjords. However, the manuscript never makes it clear where many of the forcing parameters come from and the observations are mostly used in a haphazard manner.” He/she makes several constructive suggestions to strengthen and improve the manuscript. We have taken the referee’s comments very seriously, and we have revised the manuscript to address the referee’s comments and suggestions.

In the following, we detail the amendments made to the manuscript (highlighted in red color) in response to the referee’s comments (included in italics).

Comment (1). General comments

There is a general mismatch between observations and modeling. After several sections describing how mélange thickness was derived from field measurements and satellite data, the continuum model is just simply forced with mélange thickness of either 75 m or 200 m. There is no mention of why these values were chosen or if they were informed by observations. The discrete element model is then forced with different initial thicknesses of 60 m and 378 m. It is unclear why different values were chosen or whether these values represent Helheim Glacier or one of the many other glaciers mentioned in the manuscript (e.g. Jakobshavn Isbræ, Store, or Kangerlussuaq). After the all the hard work to observe ice mélange thickness in a remote fjord in East Greenland, surely the authors could at least use that data to justify the ice mélange thickness in the models.

Response. Note that for the continuum model for predicting buttressing stresses, we are already using the mélange thickness at the glacial termini, H_0 , from the satellite-derived observations (ArcticDEM) from 32 glaciers. We take H_0 to vary from 35 m, which is the minimum size of icebergs detected within the mélange [1, 2], to 240 m, which is the thickest mélange observed across 32 Greenland termini (See Table 2 in SI). We have revised the continuum model in the main text as follows to make this more clear (lines 208–220):

To characterize the relative magnitude of the horizontal deviatoric stress to the glaciostatic pressure, we substitute representative values for parameters in Eqn. 3 and obtain:

$$\frac{|4H_0(\eta\frac{\partial u}{\partial x})|_{x=0}}{\frac{1}{2}\rho_i(1 - \frac{\rho_i}{\rho_w})g\phi_0H_0^2} \in [3.90 \times 10^{-14}, 1.17 \times 10^{-11}] \times \eta, \quad (1)$$

where the range of mélange thickness, H_0 , is derived from DEM observations. We take H_0 to vary from 35 m, which is the minimum size of icebergs detected within the mélange [1, 2], to 240 m, which is the thickest mélange observed across 32 Greenland termini in 2013–2022 (See Table 2 in SI). We adopt $\frac{\partial u}{\partial x} \in [\frac{2 \text{ m/day}}{15 \text{ km}}, \frac{25 \text{ m/day}}{10 \text{ km}}]$ [3], $\phi_0 \in [0.64, 1]$ [4, 5], $\rho_i \in [870 \text{ kg/m}^3, 920 \text{ kg/m}^3]$ [6], and $\rho_w \in [1020 \text{ kg/m}^3, 1029 \text{ kg/m}^3]$ [7]. As the mélange acts as a weak granular ice shelf [8], its effective viscosity should be much smaller than the glacier ice viscosity, $\eta \ll \eta_i = 10^{12} - 10^{15} \text{ Pa}\cdot\text{s}$ [9, 10]. For mélange with a high viscosity ($\eta > 10^{11} \text{ Pa}\cdot\text{s}$), we need to consider deviatoric stress effects as has been done in [11]. The mélange in the following discrete element model has an estimated viscosity of $2 \times 10^{10} \text{ Pa}\cdot\text{s}$ (see Section 2 in SI for details). Therefore, for mélange with a low viscosity, glaciostatic pressure dominates and the mélange buttressing force can be approximated as:

$$\frac{F}{W} = \frac{1}{2} \rho_i \left(1 - \frac{\rho_i}{\rho_w}\right) g \phi_0 H_0^2. \quad (2)$$

To better match our discrete element models with observations, we have conducted ten more simulations with Helheim and Kangerlussuaq fjord geometries, observed iceberg distributions, and mélange thickness from ArcticDEMs. And we have justified the mélange thickness profile from discrete element models with both ArcticDEM observations and the continuum theory.

We first added a derivation of the continuum theory for mélange thickness profile in SI as follows (lines 97–119):

Finally, we derive the expression for the mélange thickness profile, $H(x)$. Eqn. 14 can be reorganized as follows:

$$\frac{\partial}{\partial y} (H(x) \bar{\sigma}'_{xy}) = -\frac{\partial}{\partial x} (H(x) \bar{\sigma}_{zz}) - \frac{\partial}{\partial x} (2H(x) \bar{\sigma}'_{xx}) - \frac{\partial}{\partial x} (H(x) \bar{\sigma}'_{yy}) \quad (3)$$

Because we assume mélange thickness and stresses do not vary with y , we can integrate Eqn. (3) over the y direction as:

$$(H(x) \bar{\sigma}'_{xy})|_{y=W} - (H(x) \bar{\sigma}'_{xy})|_{y=0} = W \left(-\frac{\partial}{\partial x} (H(x) \bar{\sigma}_{zz}) - \frac{\partial}{\partial x} (2H(x) \bar{\sigma}'_{xx}) - \frac{\partial}{\partial x} (H(x) \bar{\sigma}'_{yy}) \right) \quad (4)$$

We use Coulomb friction law to calculate the shear stress at the fjord walls as follows:

$$\begin{aligned} \bar{\sigma}'_{xy}|_{y=W} &= \mu_e \bar{\sigma}_{yy}, \\ \bar{\sigma}'_{xy}|_{y=0} &= -\mu_e \bar{\sigma}_{yy}, \end{aligned} \quad (5)$$

where μ_e is the effective coefficient of friction between the mélange and the fjord wall, which depends on the material friction coefficient and the geometry of the fjord walls, i.e., wall roughness [8, 12, 13]. As $\bar{\sigma}_{yy} = \bar{\sigma}_{zz} + \bar{\sigma}'_{xx} + 2\bar{\sigma}'_{yy}$,

Eqn. (4) becomes

$$\frac{2H(x)\mu_e}{W}(\bar{\sigma}_{zz} + \bar{\sigma}'_{xx} + 2\bar{\sigma}'_{yy}) = -\frac{\partial}{\partial x}(H(x)\bar{\sigma}_{zz}) - \frac{\partial}{\partial x}(2H(x)\bar{\sigma}'_{xx}) - \frac{\partial}{\partial x}(H(x)\bar{\sigma}'_{yy}) \quad (6)$$

Following the scaling analysis in the main text (Eqn. 4 in section 1.3), we can reasonably assume that $\bar{\sigma}'_{xx}, \bar{\sigma}'_{yy} \ll \bar{\sigma}_{zz}$, where the depth-averaged vertical stress is $\bar{\sigma}_{zz} = \frac{1}{2}\rho_i g \phi(x)(1 - \frac{\rho_i}{\rho_w})H(x)$. We further assume that the mélange packing density remains a constant along fjords. Therefore, Eqn. (6) becomes

$$\frac{\partial H}{\partial x} + \frac{\mu_e}{W}H(x) = 0 \quad (7)$$

which gives

$$H(x) = Ce^{-\frac{\mu_e}{W}x} \quad (8)$$

where C is a constant that needs to be constrained by a boundary condition of the thickness profile. The mélange thickness exponentially decays with the distance from terminus. However, Eqn. 14 only holds when mélange can be considered as a three-dimensional material. When the mélange thickness decays to a monolayer of icebergs, its thickness is dictated by the iceberg size distribution, instead of stress balances that give rise to Eqn. (8). We identify the mélange length, L , where the mélange thickness decays to a threshold value, H_L , below which the mélange is considered to be a two-dimensional material. Using the boundary condition, $H(x = L) = H_L$, we arrive at the final expression for the mélange thickness profile:

$$H(x) = H_L e^{\frac{\mu_e L}{W}(1 - \frac{x}{L})}, x \in [0, L] \quad (9)$$

By defining the dimensionless distance, $\tilde{x} = \frac{\mu_e x}{W}$, and the dimensionless thickness, $\tilde{H} = \frac{H(x)}{H_L e^{\frac{\mu_e L}{W}}}$, we arrive at the dimensionless form of the mélange thickness profile:

$$\tilde{H}(\tilde{x}) = e^{-\tilde{x}}, \tilde{x} \in [0, \frac{\mu_e L}{W}]. \quad (10)$$

We then summarized the continuum theory for mélange thickness profile in the main text as follows (lines 221–249):

The “granular ice shelves” depth-averaged horizontal momentum equations for mélange (see SI for derivation and validation against discrete element simulations) resemble that of ice shelves:

$$\begin{aligned} \frac{\partial}{\partial x}(2H(x, y)\bar{\sigma}'_{xx}) + \frac{\partial}{\partial x}(H(x, y)\bar{\sigma}'_{yy}) + \frac{\partial}{\partial y}(H(x, y)\bar{\sigma}'_{xy}) &= \rho_i g(1 - \frac{\rho_i}{\rho_w})H(x, y) \frac{\partial(\phi(x, y)H(x, y))}{\partial x}, \\ \frac{\partial}{\partial x}(H(x, y)\bar{\sigma}'_{xy}) + \frac{\partial}{\partial y}(H(x, y)\bar{\sigma}'_{xx}) + \frac{\partial}{\partial y}(2H(x, y)\bar{\sigma}'_{yy}) &= \rho_i g(1 - \frac{\rho_i}{\rho_w})H(x, y) \frac{\partial(\phi(x, y)H(x, y))}{\partial y}, \end{aligned} \quad (11)$$

where the depth-averaged stress $\bar{\sigma}_{ij} = \frac{1}{H} \int_{z_b}^{z_s} \sigma_{ij} dz$. When the mélange packing density approaches $\phi = 1$, Eqn. (11) converges to the shallow shelf approximation (SSA) [14]. The mélange momentum balance along the fjord direction reveals three competing forces: compressional/extensional flow from velocity gradients within the mélange (negligible if mélange viscosity is smaller than 10^{11} Pa-s), glaciostatic stress from mélange thickness, and shear stresses on fjords. Therefore, the full thickness profile of the mélange depends on fjord/mélange friction/cohesion properties, velocity gradients and viscosity of the mélange, and the mélange width/length. Using the horizontal momentum balance equations (Eqn. 11) and non-dimensionalization (see SI for details), we further derive the mélange thickness profile, $H(x)$:

$$\tilde{H}(\tilde{x}) = e^{-\tilde{x}}, \tilde{x} \in [0, \frac{\mu_e L}{W}], \quad (12)$$

where μ_e is the effective coefficient of friction between the mélange and the fjord wall, which depends on the material friction coefficient and the geometry of the fjord walls, i.e., wall roughness [8, 12, 13]. Here, we define the mélange length, L , as the distance beyond which the mélange thickness decays to a threshold value, H_L , where mélange becomes a monolayer of icebergs with thickness dominated by particle size distribution instead of stress balances that give rise to Eqn. (12). We define the dimensionless distance, $\tilde{x} = \frac{\mu_e x}{W}$, and the dimensionless thickness, $\tilde{H} = \frac{H(x)}{H_L e^{\frac{\mu_e L}{W}}}$. To quantify the mélange buttressing force per unit width, previous two-dimensional model assuming mélange of uniform thickness required assumptions on fjord/mélange friction properties and the mélange width/length [8]; in our three-dimensional model the mélange thickness and packing density at the terminus are the only parameters needed. As the length and thickness are coupled by stress balances within the granular material, the mélange thickness build-up at the terminus already encodes the aforementioned material and geometric properties. For instance, thicker mélange can be built up at the terminus with longer fjords, larger fjord friction, or increased mélange rigidity in winter.

In summary, momentum balance equations reveal that the mélange buttressing force per unit width is solely controlled by the packing density ϕ_0 and mélange thickness H_0 at the glacier terminus, and is proportional to the square of mélange thickness (Eqn. (2)). Additionally, mélange thickness exponentially decays with its distance from terminus (Eqn. (12)).

For the new simulations with Helheim and Kangerlussuaq fjord geometries, we have made a new figure to justify the mélange thickness profile from discrete element models with both ArcticDEM observations and the continuum theory (Fig. 1 in this document and Fig. 4 in the main text). We have also added a supplementary video for modeling mélange in Helheim and Kangerlussuaq fjord geometries. To reflect these changes, we have modified the main text as follows (lines 250–289):

1.4 A three-dimensional discrete element model of ice mélange.

To validate continuum predictions on the mélange buttressing force and thickness profile (Eqn. (2), (12)), we develop a three-dimensional discrete element model on the mélange with a steadily advancing terminus. We adopt fjord geometries from Helheim and Kangerlussuaq glaciers (black dots in Fig. 1(a)(d)), which are scaled down by four times for the discrete element models to save computational costs (Fig. 1(b)(e)). Note that such downscaling does not affect the shape of the dimensionless mélange thickness profile, which will be compared with Eqn. (12). Icebergs are modelled as cubic particles with sizes varying from 36 m to 200 m and have a power-law size distribution with an exponent of -4 [2, 1, 15]. In the models, we limit the maximum size of icebergs to be around one fifth of the fjord width, as commonly observed in both fjords. We initialize the ice mélange thickness with a profile that linearly decays within 1.2 km from the terminus and with the right end open to the ocean. We push the left end of the mélange with an advancing terminus at 30 meters per day [16, 17, 18] and record the temporal evolution of the buttressing force exerted on the terminus. We present modeling results after 16 days of terminus motion, when the mélange motion has approximately reached a steady state. In a series of simulations, we vary the initial mélange thickness to determine its influence on the steady state buttressing force. The steady state mélange thickness at the terminus varies from 40~135 m at Helheim fjord, and 60~240 m at Kangerlussuaq fjord, that are consistent with ArcticDEM observations (Table 2 in SI). We calculate the two-day averaged velocity of each iceberg element by dividing the iceberg's displacement between 14 and 16 days of terminus motion by the time interval (two days). The mélange near the terminus moves at the terminus velocity with shear bands developed at fjord walls (Fig. 1(c)(f)). The mélange near the open end becomes loosely-packed and more fluidic (See supplementary video for the full temporal evolution of the mélange behaviors.) The modeled velocity field showcases both uniform (Fig. 1(c)) and extensional (Fig. 1(f)) flow regimes that are consistent with remote observations [3]. See Fig. 6 in SI for the modeled velocity fields, including instantaneous velocity, velocity averaged over one hour, one day, and two days, and the satellite-derived velocity fields of the mélange [3].

The modeled mélange thickness profiles generally align with observations from ArcticDEMs on 17 Aug 2014 at Helheim (Fig. 1(a)(g)) and on 12 Apr 2018 at Kangerlussuaq (Fig. 1(d)(h)). In addition, we conduct another simulation with more icebergs confined in Kangerlussuaq fjord, to produce a thicker steady state mélange thickness profile as observed on 5 May 2020. We set the thickness threshold (H_L) as 36 ± 10 m for mélange to be considered as three-dimensional, and retrieve the corresponding mélange length (L) that varies from 7 to 12 km. The non-dimensional mélange thickness

profile from three ArcticDEM strips and the discrete element models collapse onto the exponential analytical solution in Eqn. (12), with μ_e fitted to be 0.3~0.4 for Helheim fjord and 0.4~0.8 for Kangerlussuaq fjord (Fig. 1(i), see table 1 in SI for details). The Kangerlussuaq fjord has a larger μ_e due to its more rugged fjord geometries.

To further explore the effect of fjord frictional properties on the mélange buttressing force, we adopt two simplified channel configurations that conceptualize complex Greenland fjord geometries. ...

FIGURE 1. The three-dimensional discrete element model for mélange composed of cubic icebergs with a power-law size distribution and confined within real fjord geometries. (a)-(c) For Helheim glacier, (d)-(f) for Kangerlussuaq glacier. The real fjord geometry is scaled down by four times for the discrete element model. (a) The mélange elevation above mean sea level from ArcticDEM strip, overlain on satellite images acquired around the same date. The ArcticDEM acquisition date is shown at the top right corner. White line across the fjord indicates glacier front location. The images are in polar stereographic projection (EPSG: 3413). Black dots along fjords are adopted in the model to construct boundary walls that resemble fjord geometries. (b), (c) are side and top view for iceberg positions and velocities after 16 days into simulations with steady terminus advance and no calving. The glacier terminus moves at a constant velocity, $V_{\text{ter}} = 30$ m/day. We calculate the two-day averaged velocity of each iceberg element by dividing the iceberg's displacement between 14 and 16 days of terminus motion by the time interval (two days), which is indicated by filled colour in (c). (d)-(f) follow captions of (a)-(c). (g), (h) The comparison of the mélange thickness profile between ArcticDEM (black lines) and discrete element model (red lines) for Helheim and Kangerlussuaq glacier, respectively. (i) The mélange thickness profile from three ArcticDEM strips and corresponding discrete element models collapse onto the exponential analytical solution (Eqn. (12), red dashed line). See supplementary videos for the full temporal evolution of the mélange behaviors.

We have added a table for the fitted fjord effective coefficient of friction (μ_e) in SI as follows:

TABLE 1. The fitted fjord effective coefficient of friction (μ_e) from mélange thickness profile in Fig. 4 in the main text. The uncertainties come from the mélange thickness threshold, $H_L = 36 \pm 10$ m.

Glacier name	ArcticDEM date	μ_e (ArcticDEM)	μ_e (Discrete element model)
Kangerlussuaq	2020/05/05	$0.64^{+0.09}_{-0.06}$	$0.77^{+0.11}_{-0.06}$
Kangerlussuaq	2018/04/12	$0.39^{+0.05}_{-0.04}$	$0.48^{+0.07}_{-0.04}$
Helheim	2014/08/17	$0.25^{+0.03}_{-0.03}$	$0.39^{+0.05}_{-0.05}$

We have modified Fig. 6(b) in the main text (Fig. 2(b) in this document) to incorporate new simulations with Helheim and Kangerlussuaq fjords. We found that with real fjord geometries, iceberg size distribution, and mélange thickness profile, the buttressing forces from discrete element models still show good agreement with the continuum predictions.

FIGURE 2. Comparison between discrete element model and continuum predictions of the mélange buttressing force. (a) The temporal evolution of F/W during the terminus motion for straight (dashed lines) and rugged (solid lines) fjord walls. The red, blue, cyan and black colours correspond to mélange with initial thicknesses, $H_{ini} = 378$ m, 281 m, 178 m, 84 m, respectively. Simulations reach the steady state after 5 days, except for the thinnest mélange ($H_{ini} = 84$ m). (b) Steady state buttressing force, F/W , as a function of steady-state mélange thickness at the terminus, H_0 . Different markers represent simulations with different fjord geometries. Square markers represent Helheim fjord, hexagram markers represent Kangerlussuaq fjord, circular markers represent straight fjords, and triangular markers represent rugged fjords. The smaller markers indicate simulations with smaller icebergs (half of the original size). F/W is obtained by averaging the total buttressing force on the terminus over the terminus width during simulation time 5 ~ 15 days. The marker shows the averaged steady-state value of F/W , with a vertical error bar showing its fluctuation. H_0 is obtained by averaging the mélange thickness within 200 m of the terminus and over the terminus width. The marker shows the averaged steady-state value of H_0 , with a horizontal error bar showing its variation over the terminus width brought by the iceberg size polydispersity. For simulations where the mélange collapse into monolayers at the end, we plot both the peak and minimal F/W values and connect them by gray lines. The minimal F/W values for monolayered, two-dimensional mélange are shown by pentagram markers. All markers are coloured by the mélange packing density at the terminus at steady state, ranging from 0.4 to 0.9. The dashed lines represent Eq. (2) with the mélange packing density at the terminus, $\phi_0 = 0.5$ and 1.0, respectively.

To highlight limitations of current mélange thickness profile analysis, we have added following discussions in the main text as follows (lines 405–418):

2 Discussion Our discrete element model of ice mélange is the first to be composed of realistic cubic icebergs instead of spheres, and showcases an exponential decay of the thickness profile in Helheim and Kangerlussuaq fjords, that is consistent with ArcticDEM observations and analytical predictions. Apart from the exponential decay, we also observe other shapes of the mélange thickness profile, such as a plateau near the terminus, a steep drop or a bulge at few kilometers away from the terminus, etc (See SI for ArcticDEM strips at 32 glacier termini). These shapes could be attributed to calving-induced jamming wave propagation, ice-ocean interactions, iceberg size distribution, heterogeneous friction or cohesion within the mélange or at fjord walls, all of which cannot be captured by the simplified rheology and friction law underneath Eqn. (12). Coupled with computational fluid mechanics, our discrete element model can be used to further explore how the mélange thickness at the terminus evolves with ice-ocean interactions that influence calving dynamics, including ocean tides [19], ocean warming [20, 21, 22], and subglacial plumes [23, 24].

Lastly, we have added details of the new simulations into the Methods section as follows (lines 618–655):

3.4 The three-dimensional discrete element model for ice mélange. ...

We use cubic grains which can achieve a higher packing density, thus butressing forces, than disk-shaped grains. We adopt the iceberg area distribution observed in the mélange of Jakobshavn Isbræ and Helheim Glacier, which is approximated as a power-law distribution with an exponent of -2.0 [2, 1]. The resulting iceberg size distribution for cubic grains is a power-law distribution with an exponent of -4.0. Taking the simulation with the Helheim fjord for instance (Fig. 1(b)), the side lengths of cubic icebergs are 36 m, 50 m, 75 m, 100 m, 200 m, and the corresponding numbers of particles are 3120, 838, 166, 52, 3, respectively. To model mélange with different steady state thicknesses, the total number of particles varies from 2918 to 12522 for simulations with Helheim and Kangerlussuaq fjords. In simplified fjord geometries, the side lengths of cubic icebergs are 35.4 m, 50 m, 70.7 m, 100 m, 141.4 m, and the corresponding numbers of particles for the thick mélange are 8190, 2045, 510, 125, 30, respectively (Fig. 10(f)). We vary the total number of particles from 1634 to 15264 to change the mélange thickness. To confirm that the modeling results are invariant to the particle size, we conduct six more simulations with smaller icebergs, whose sizes are half of original sizes and range from 17.7 m to 70.7 m (small markers in Fig. 2(b)).

To construct the initial mélange state, we divide the total number of particles into three equal batches. In each batch, iceberg sizes are randomly drawn from the distribution described above. We put a right boundary wall at distance L from the terminus on the left to prescribe the initial length of the mélange.

The mélange is confined in y direction by two side walls representing fjords at a distance W . To explore influence of fjord friction properties on mélange behaviors, we have Helheim, Kangerlussuaq, and simplified (straight and rugged) channel configurations. All configurations have the same kinetic friction coefficient, μ , and rugged channels have cuboid bulges of dimension $a_r \times a_r \times h_r$ that are uniformly spaced at d_r in x and z directions. We deposit icebergs in each batch from the same height and then they settle under gravity and buoyancy. Following pouring, the entire array of cubic particles is permitted to settle until static equilibrium is achieved, as shown in Fig. 10(a)(f). We then delete the right boundary wall so that the mélange has an open end in the ocean. We move the terminus on the left at a constant velocity, $V_{\text{ter}} = 30$ m/day for real fjord geometries [16, 17, 18], and 43.2 m/day for simplified fjord geometries [25, 8]. To confirm that the averaged steady-state buttressing force is invariant to the terminus velocity, we conducted simulations with mélange thickness 280 m, terminus velocity at 21.6 m/day, 43.2 m/day, and 86.4 m/day, for both straight fjords and rugged fjords configurations (Fig. 16 in SI). The results show that the averaged buttressing force is mostly invariant to the terminus velocity in both fjord configurations. Taking the force fluctuations into account, the maximum buttressing force difference among the chosen velocities is 4% and 8% for straight and rugged fjords, respectively. In rugged fjords, faster terminus motion leads to larger force fluctuations due to larger velocity gradient during stick-slip/jam-unjam cycles. We adopt the terminus velocity of 43.2 m/day to simulate mélange in simplified fjord geometries for the sake of computational efficiency.

Comment (2). The manuscript lacks cohesiveness. This is related to the first point but more about the structure/readability rather than the research. There are five different approaches described in this study. The two modeling approaches (continuum and discrete element) and three observational approaches (TLS, ICESat-2, and ArcticDEM). Given the letter style of this manuscript, each approach has to be described in the Results section. This means that the narrative is interrupted in several places, making it difficult to follow the major arguments. Furthermore, because there are so many approaches, there are many places where there are not enough details. There are several key terms like terminus retreat, terminus quiescence, and major and minor calving events that are not adequately defined and would benefit from more complete descriptions.

Response. We thank the referee for pointing out the lack of cohesiveness between different subsections in the Results section. We have added several transition/conclusion sentences between subsections, and we have clarified key terms (terminus retreat, major and minor calving events) in section 1.1 when they first appear. To avoid ambiguities, we replace “terminus quiescence” with “calving cessation” in section 1.1. We have added sentences before the Results section and modified the Results section as follows:

Motivated by recent observations of weekly to seasonal variations in mélange thickness, we hypothesize that mélange thickness dictates its buttressing force per unit width. To test this hypothesis, we use terrestrial laser scanning and ICESat-2 data to demonstrate that mélange thickness is well-correlated with terminus position and calving dynamics at 4 major Greenland outlet glaciers. Motivated by these observations, we develop a continuum mélange model, validated with a three-dimensional discrete element model, which confirms that mélange thickness and packing density are the only observations required to estimate mélange buttressing force per unit width. We then apply this continuum model to estimate mélange buttressing forces around Greenland from ArcticDEM observations, and show that the inferred buttressing force is highly consistent with observed patterns of terminus advance and retreat.

1 Results

1.1 Mélange thickness associated with calving dynamics at Helheim Glacier in 2019-2020. To investigate the correlation between the mélange thickness and calving dynamics, we derive time series of mélange elevation at the terminus, calving events, and terminus position inferred from TLS data and satellite images (Sentinel-1, Sentinel-2 and Landsat 8) from 1 Sep 2019 to 1 Sep 2020 at Helheim Glacier (Fig. 1(b)). Throughout 2019, a REIGL VZ-6000 terrestrial laser scanner (TLS), located at the triangle marked in Figure 1(a), scanned the terminus and ice mélange of Helheim Glacier. Fig. 1(a) shows the TLS-measured surface elevation field for ice mélange on 30 Nov 2019 (see Methods for the TLS processing steps). Mélange surface elevation was derived from TLS every 24 hours in the winter months, and every 6 hours in non-winter months. To display the spatial profile of the mélange elevation, we calculate distances from terminus for all data points in the ice mélange and plot them as density maps in Fig. 1(c)-(f). For any specific distance from the terminus, there is a spread of mélange elevation that exhibits a long-tail distribution due to the presence of large icebergs. To reflect the mélange elevation that is piled up from small icebergs and sea ice, we connect elevation values that have maximum numbers of data points along the distance from terminus as the representative mélange elevation profiles (solid blue lines in Fig. 1(c)-(f), see Methods and Fig. 2 in SI for a sensitivity study of this metric). In doing so, we prevent large icebergs from disrupting the mélange elevation profiles without cutting off the data by an arbitrary elevation threshold. The resulting representative mélange elevation profiles are always below 30 m, indicating that it is a reasonable elevation threshold to exclude large icebergs from consideration. To estimate mélange elevation near the glacier terminus (Z_0), we take an average of all data points below the 30 m threshold within 1 km (i.e., 1/5 of the fjord width) of the terminus. We infer thickness of the mélange based on TLS-derived surface elevations and assuming hydrostatic equilibrium.

We digitize the terminus position on each TLS scan with a straight line (the white line in Fig. 1(a)). The terminus advance/retreat refers to the changes

in the terminus position, and is quantified by distances between these straight lines. This can indicate whether the glaciers are advancing (growing) or retreating (shrinking). With reference to previous classification of calving events [6], here, we define “major calving” events as those with block size >0.25 km² (observed from TLS scans), and causing significant mélange motion and an overall terminus retreat (observed from satellite images); “minor calving” events are those in which visible blocks calved, but the mélange or terminus position remained largely unchanged. We observe two episodes of calving cessation, ...

Apart from the weekly terminus variability of Helheim Glacier presented in this section, remote sensing observations on many Greenland glacier termini have shown significant terminus-position seasonality, with advance from winter to spring and retreat from summer to fall through enhanced calving [26, 27, 28]. Previous studies have attributed seasonal calving dynamics to buttressing from ice mélange [29, 30, 20], which motivates us to further explore seasonal changes of mélange thickness and calving dynamics on other Greenland glaciers.

1.2 Seasonal changes of mélange thickness and calving dynamics. To investigate whether there are correlations between ice mélange thickness and calving dynamics on other glaciers, we use ICESat-2 observations of mélange surface elevation. While this dataset does not provide the temporal resolution to study individual calving events, we can leverage the observed seasonality in terminus advance and retreat at many Greenland glaciers to assess whether mélange thickness is correlated with periods of quiescence versus vigorous calving.

... In summary, ICESat-2 data provides strong evidence of correlations between mélange thickness and terminus seasonality.

1.3 A three-dimensional continuum model of ice mélange. Remote sensing observations reveal a strong correlation between mélange thickness and calving dynamics. As a result, quantifying the buttressing force of mélange in terms of its thickness is the first step to better representing ice-ocean interactions and developing process-based calving models. ...

In summary, momentum balance equations reveal that the mélange buttressing force per unit width is solely controlled by the packing density ϕ_0 and mélange thickness H_0 at the glacier terminus, and is proportional to the square of mélange thickness (Eqn. (2)). Additionally, mélange thickness exponentially decays with its distance from terminus (Eqn. (12)).

1.4 A three-dimensional discrete element model of ice mélange. To validate continuum predictions on the mélange buttressing force and thickness profile (Eqn. (2), (12)), we develop a three-dimensional discrete element model on the mélange with a steadily advancing terminus. We adopt fjord geometries from Helheim and Kangerlussuaq glaciers (black dots in Fig. 1(a)(d)), which are scaled down by four times for the discrete element models to save computational costs (Fig. 1(b)(e)). ...

1.5 Mélange buttressing force increases with mélange thickness at the glacier terminus. ... The robustness of Eqn. (2) with different fjord properties is the key to interpreting field observations across Greenland glacier termini.

1.6 Calving dynamics associated with mélange buttressing force seasonality across 32 Greenland glacier termini in 2013-2022. Our models reveal that the mélange buttressing force can be predicted solely from remote sensing observations of its thickness at glacier terminus (Eqn. (2)). However, further investigation is needed to address the question of how does the spatio-temporal variations in mélange buttressing force correlate with calving dynamics in Greenland...

Comment (3). The observed mélange thicknesses are lacking validation. The TLS freeboards have stated accuracy of 0.10 m based on a “registration scan” which is never described. The only citation in this section is from a laser scanning study by a completely different group of researchers at Hintereisferner (Austria). The uncertainty from ICESat-2 freeboards is stated as “0.02 m~0.52 m”. It looks like this comes from the ATL06 ATBD but it is not clear. More effort went into correcting and validating ArcticDEM strips. But surely the mélange thicknesses from these different approaches overlap in space/time so it was unclear why they were not compared. Disappointingly, the mélange thicknesses were not provided in the Data Availability statement.

Response. We thank the reviewer for the constructive comment, which inspires us to validate the mélange thicknesses from TLS, ICESat-2, and ArcticDEM. We have provided the uncertainty assessment of the TLS data, with an updated accuracy of 0.045 m. We have added following paragraphs to the Methods section 3.1 as follows:

3 Methods

3.1 Terrestrial laser scanner data and uncertainty assessment. The TLS is located at a latitude of 66.32963° N and a longitude of -38.1739° W, with the location provided in Figure 1(a) and is approximately 1 km - 5.8 km from Helheim Glacier’s calving face. TLS point clouds are ellipsoidal heights. After removing the geoid height at Helheim Glacier [31], the mélange surface elevations are heights above mean sea level with tidal variations [6]. We then remove the tidal trend of mélange elevations with a tidal model [32]. ATLAS generated point clouds were gridded at 100 m × 100 m resolution to insure sufficient point densities per grid cell using the Point Cloud Data Abstraction Library (PDAL) [33] for DEM creation. The resulting DEMs contain a minimum, maximum, and average band where each point which falls into a 100 m × $\sqrt{2}$ radius contributes to a grid cell. Generally, five main sources of uncertainty exist when using terrestrial laser scanner (TLS) data. These sources being registration, atmospheric conditions, scanning geometry, instrument and hardware limitations, rasterization, and surface reflectance properties [34]. The TLS was tied to the global spatial reference frame using an array of 5 cm cylindrical reflective targets. The centroids of these cylinders were surveyed

using GNSS receivers, and the reflective exterior surfaces of the cylinders were scanned with the LiDAR scanner. The position and orientation of the scanner was then adjusted to bring the LiDAR-derived cylinder centroids and the GNSS-derived cylinder positions into optimal alignment using a least squares method. This error for the 2019 ATLAS south scanning epoch was 2 cm. Following the uncertainty methods from [34], the total uncertainty per grid cell is defined by $\sigma_{total} = \frac{\sigma_{point}}{\sqrt{\rho_{point}}} + \sigma_{reg}$. From our uncertainty analysis, our REIGL vz6000's uncertainty is dominated by the registration error if the point density is high. σ_{point} depends on instrument hardware error $\sigma_{instrument}$, beam geometry σ_{geo} , and atmospheric conditions σ_{atm} . For a REIGL vz-6000 instrument, errors can range from 0.09 - 0.64 [34], depending on the range. Uncertainty varies with range mainly for two reasons: 1) instrument uncertainty is defined by $R \cdot \sin(0.008^\circ)$ where R is the distance the laser pulse traveled to the target surface, and 2) scanning geometry also depends on scanning distance as the laser footprint expands over distance. More about general terrestrial laser scanner uncertainty can be read in [34].

Following the methods outlined in [34], our vertical uncertainty ranges from approximately 2 cm - 4.5 cm depending on the distance and point density. Using the upper range of point densities in a 100 m x 100 m grid cell in the mélange region consisting of 10,000 points, the uncertainty is closer to 2 cm while the lower end of point density ($\approx 2,000$ points) will have an uncertainty closer to 4.5 cm. We also vary the σ_{geo} uncertainty from 5 - 50 cm, which the upper limit is double of the [34] values in our assessment.

To construct the representative mélange elevation profile, we first connect the median or mean elevation values along the distance from the terminus, the shape of which turns out to be sensitive to large icebergs with elevation values larger than 30 m (Fig. 2 in SI). Incorporating large icebergs into the representative mélange elevation profile has several drawbacks. Firstly, instead of representing the elevation piled up from small icebergs and sea ice, it tends to reflect the size distribution of large icebergs. Secondly, the existence and location of large icebergs are sporadic and hard to characterize by a continuum model. Lastly, comparing mélange elevation profiles across different Greenland fjords becomes challenging, as the size and shape distribution of large icebergs are sensitive to calving styles [35] and terminus geometries. To reflect the mélange elevation that is piled up from small icebergs and sea ice, we connect elevation values that have maximum numbers of data points along the distance from terminus as the representative mélange elevation profiles (solid blue lines in Fig. 1(c)-(f) in the main text and Fig. 2(b)(e) in SI).

The uncertainty from ICESat-2 freeboards indeed comes from the ATL06 ATBD. We compare the mélange surface elevation acquired on 10 Sep 2019 from TLS and ICESat-2 at Helheim Glacier (Fig. 11 in SI, Fig. 4 in this document). The TLS data was acquired at 13:01:56 UTC, and the ICESat-2 data was acquired after 90 minutes, at 14:32:53 UTC.

Overall, the mélange surface elevation profiles acquired from TLS and ICESat-2 show good agreement, with a mismatch at a distance around 2.5 km that might come from mélange motion within the 90 minutes time window. To account for the uncertainty from geoid [31] and tidal corrections [32], we identify ICESat-2 tracks passing through the ocean area near the three glaciers reported in Fig. 2 in the main text (Jakobshavn Isbræ, Kangerlussuaq Glacier, Store Glacier), and compute the difference between ATL06 measurements over the ocean and the geoid [31] with tidal corrections [32]. We have added figures in the SI (Fig. 10&11 in SI and Fig. 3&4 in this document) and added following sentences in the Methods section:

3.2 ICESat-2 data and uncertainty assessment. We use the ATL06 data set from ICESat-2 that provides geolocated, mean land-ice surface heights above the WGS 84 ellipsoid that are averaged along 40 m segments of ground track and spaced 20 m apart [36]. The temporal resolution is 91 days from 14 October 2018 to present. We compute the mélange surface elevation after accounting for the local difference between the ellipsoid and the EIGEN-6C4 geoid [31], and then remove the tidal signal from the surface elevation [32]. We identify three ICESat-2 tracks for Jakobshavn Isbræ, Kangerlussuaq Glacier and Store Glacier, respectively, and use data from strong beams to compose the mélange elevation profile (Fig. 2). The averaged standard error in the reported elevation data ranges from 0.02 m~0.52 m due to sampling error and first-photon bias correction from the land ice algorithm, as described in the algorithm theoretical basis document (ATBD) for land ice along-track height product (ATL06) [37]. To account for the uncertainty from geoid and tidal corrections, we identify ICESat-2 tracks passing through the ocean area near the three glaciers presented in Fig. 2 and compute the difference between ATL06 measurements over the ocean and the geoid [31] with tidal corrections [32]. We found that the mean error is 0.37 m at Jakobshavn Isbræ, 0.50 m at Kangerlussuaq Glacier, and 0.60 m at Store Glacier (Fig. 10 in SI). In addition, we compare the mélange surface elevation acquired on 10 Sep 2019 from TLS and ICESat-2 at Helheim Glacier (Fig. 11 in SI). The TLS data was acquired at 13:01:56 UTC, and the ICESat-2 data was acquired after 90 minutes, at 14:32:53 UTC. Overall, the mélange surface elevation profiles acquired from TLS and ICESat-2 show good agreement with a mean absolute difference of 0.94 m at the points of overlap. The mismatch at a distance around 2.5 km might come from mélange motion within the 90 minutes time window. In addition, the TLS scans are oblique to the surface and not straight down, so TLS and ICESat-2 are not measuring the exact same surface. All these sources of uncertainty are an order of magnitude less than the observed temporal changes in mélange freeboard height presented in Fig. 2, suggesting that our interpretation of seasonal changes in mélange thickness from ICESat-2 are robust.

FIGURE 3. The mélange elevation data uncertainty by computing the difference between ATL06 measurements over the ocean and the geoid [31] with tidal corrections [32], at Jakobshavn Isbræ (a)(b), Kangerlussuaq Glacier (c)(d), and Store Glacier (e)(f). (a) The Sentinel-2 image for Jakobshavn Isbræ on 17 Aug 2021. In (a)(c)(e), the white lines indicate the ICESat-2 track along which ATL06 surface elevation data is acquired. The date and time on the image shows the acquisition time for ICESat-2 data, which is around the same date of the presented satellite image. Images are in polar stereographic projection (EPSG: 3413). The ICESat-2 track and beam IDs are presented in white texts. (c) The Sentinel-2 image for Kangerlussuaq Glacier on 19 Jul 2020. (e) The Sentinel-1 HV image for Store Glacier on 19 Dec 2019. (b)(d)(f) The difference between ATL06 measurements over the ocean and the geoid [31] with tidal corrections [32] as a function of the distance along the ICESat-2 track in (a)(c)(e).

FIGURE 4. Comparison of mélange surface elevation from TLS and ICESat-2 at Helheim Glacier. (a) TLS-measured elevation map after accounting for local differences between the ellipsoid and geoid, overlain on a Sentinel-2 image (both acquired on 10 Sep 2019). The black line across the fjord indicates the ICESat-2 track along which surface elevation data is acquired. Images are in polar stereographic projection (EPSG: 3413). (b) Surface elevation profiles extracted along the black line in (a). Black cross markers indicate data acquired from TLS on 10 Sep 2019, 13:01:56 UTC, and red dot markers indicate data acquired from ICESat-2 on 10 Sep 2019, 14:32:53 UTC. The horizontal axis shows the distance from point A in (a).

We have also compared the mélange surface elevation acquired on 10 May 2020 from TLS and ArcticDEM at Helheim Glacier (Fig. 13 in SI, Fig. 6 in this document). The TLS data was acquired at 13:01:57 UTC, and the ArcticDEM data was acquired after 40 minutes, at 13:42:45 UTC. Overall, the mélange surface elevation profiles acquired from TLS and ArcticDEM show good agreement. For the uncertainties in geoid and tidal corrections, we have already followed procedures in [1] to incorporate sea level registration for ArcticDEM strips, as pointed out in the submitted main text “After applying the elevation offset and subtracting the geoid from the ellipsoid with the tidal correction [32], we plotted DEM elevation values above mean sea level in a histogram with 0.25 m bin widths, making sure its peak (i.e. the most common elevation above mean sea level in the DEM) was larger or equal to sea level at the time when the DEM was acquired [1].” We have added a figure in SI to illustrate this procedure (Fig. 12 in SI and Fig. 5 in this document), and have modified the Methods section for clarification as follows:

3.3 ArcticDEM data and uncertainty assessment. Since the DEM strips can have vertical offsets of up to 4 m [38], we co-registered them using two methods to reduce vertical uncertainty. For the 63 DEM strips covering both mélange and the ocean, we adopted the sea level calculation method in [1] by plotting DEM elevation values above the WGS84 ellipsoid in a histogram with 0.25 m bin widths. The peak elevation value (i.e. the most common elevation in the DEM) was supposed to be sea level at the time when the DEM was acquired [1]. The DEM strip was registered to the detected sea level

by subtracting the peak elevation value from elevation values relative to the WGS84 ellipsoid. The accuracy of the elevation values above sea level over non-mélange areas varied from 0.13 m to 0.37 m for a DEM strip acquired on 11 Jun 2014 at Helheim Glacier (See Fig. 12 in SI for details). For the 45 DEM strips covering mélange only, we registered each DEM strip with the mosaic DEM [39], which has been registered to ICESat-2. For each glacier terminus, we selected line segments on neighboring rock (Fig. 17 – 48 in SI) and calculated averaged elevation offsets between individual DEM strips and the mosaic DEM along these line segments. After applying the elevation offset and subtracting the geoid from the ellipsoid with the tidal correction [32], we plotted DEM elevation values above sea level in a histogram with 0.25 m bin widths, making sure its peak was larger than zero at the time when the DEM was acquired [1]. In summary, the elevation offsets applied to the 108 DEM strips were 0.38 ± 2.23 m. With this protocol, the elevation accuracy of the DEM strip segment improved from 4 m [38] to 1.06 m [39, 40]. In addition, we compare the mélange surface elevation acquired on 10 May 2020 from TLS and ArcticDEM at Helheim Glacier (Fig. 13 in SI). The TLS data was acquired at 13:01:57 UTC, and the ArcticDEM data was acquired after 40 minutes, at 13:42:45 UTC. Overall, the mélange surface elevation profiles acquired from TLS and ArcticDEM show good agreement with a mean absolute difference of 0.66 m at the points of overlap.

For each glacier terminus, we digitized terminus positions using ArcticDEMs on the dates when the data was acquired. For mélange of length 15 km and width 4 km, there were approximately 15,000,000 data points available. For each data point in a DEM strip, we calculated its distance from terminus and the surface elevation value **after applying the elevation offset**. After picking specific values for the number of horizontal and vertical bins, we displayed all data points in a density map where surface elevation was plotted as a function of distance from terminus (Fig. 14 in SI). The accuracy from varying the number of bins of density maps ranged from 0.11~0.27 m (Fig. 15 in SI). For any specific distance from terminus, we find the elevation value that had the maximum number of data points (Fig. 14 in SI). We then connected these values along the distance from terminus as the representative mélange elevation profiles (solid blue lines in Fig. 7), $Z(x)$. We calculated the maximum mélange elevation within 200 m from the terminus as Z_0 . The value Z_0 was further divided by $1 - \rho_i/\rho_w$ to obtain the mélange thickness, H_0 , which was used for calculating the buttressing force, F/W , based on Eqn. (2). In Table. 2 in SI, we report the thickness uncertainty arising from ArcticDEM (± 1.06 m), ice (910_{-40}^{+10} kg/m³) and water (1028_{-8}^{+1} kg/m³) densities. The uncertainties in ice and water densities, mélange packing density ($\phi_0 = 0.9_{-0.26}^{+0.1}$), and the mélange thickness fed into Eqn. (2) to obtain the uncertainty in the buttressing force, F/W .

FIGURE 5. The calculation of sea level for an ArcticDEM strip that covers mélangé and ocean areas. (a) Example of a histogram of elevation pixel count in an ArcticDEM image at Helheim Glacier acquired on 11 Jun 2014. The elevation with the highest pixel count is automatically selected as the sea level for that scene. In this example sea level would be 51.5 m relative to the WGS84 ellipsoid. (b) The Landsat 8 image for Helheim Glacier on 11 Jun 2014 in polar stereographic projection (EPSG: 3413). (c) The mélangé elevation above sea level from the ArcticDEM strip acquired on 11 Jun 2014 overlain on the satellite image in (b). The ArcticDEM strip is registered to the detected sea level in (a) by subtracting 51.5 m from the elevation values relative to the WGS84 ellipsoid. (d) The surface elevation profile for the mélangé displayed as a density plot. The solid blue line is the representative mélangé elevation profile. (e) A zoom-in view of the representative mélangé elevation profile with the distance from terminus varying from 10 km to 14 km. The region is mostly ocean area (i.e. 0 m above sea level) with the detected elevation above sea level varying from -0.13 m to 0.37 m.

FIGURE 6. Comparison of mélange surface elevation from TLS and ArcticDEM at Helheim Glacier. (a), (b) are TLS, ArcticDEM measured elevation map after accounting for local differences between the ellipsoid and geoid, overlain on a Sentinel-2 image (acquired on 10 May 2020). Images are in polar stereographic projection (EPSG: 3413). (c) Surface elevation profiles extracted along the black line in (a)-(b). Black cross markers indicate data acquired from TLS on 10 May 2020, 13:01:57 UTC, and red dot markers indicate data acquired from ArcticDEM on 10 May 2020, 13:42:45 UTC. The horizontal axis shows the distance from point A in (a). (d), (e) are TLS and ArcticDEM measured surface elevation profiles for the mélange displayed as density plots; the color bar denotes the number of data points that have the same elevation and distance from terminus values. For any specific distance from terminus, we find the elevation value that has the maximum number of data points. Solid blue lines connect these elevation values along the distance from terminus as the representative mélange elevation profiles.

Lastly, we have added data and pipeline to plot the mélange thickness profile in the data and code availability sections as follows:

4 Data availability All data sets produced as part of this paper are available through Zenodo at <https://zenodo.org/records/13382185> [41]. These data include the following: a supplementary word file that lists the ArcticDEM tiles, ICESat-2 laser altimetry tracks, and the Landsat, Sentinel-1 and

Sentinel-2 scene numbers used in this study, a supplementary excel file that lists elevation offsets applied on the 108 ArcticDEM strips with corresponding ArcticDEM strip acquisition dates and Mosaic DEM tiles for coregistration, ICESat-2 data cropped by the mélange region of interest for the three glaciers presented in Fig. 2, 32 zipped folders that contain ArcticDEM data cropped by the mélange region of interest for the presented 32 Greenland glacier termini in Fig. 8, and source codes to plot mélange surface elevation (or thickness) profile as a function of distance from terminus from ICESat-2 data and ArcticDEM strips. Landsat images were downloaded through the Amazon Web Services (<https://registry.opendata.aws/usgs-landsat>). Copernicus Sentinel-1 data acquired in 2019 and 2021, and Copernicus Sentinel-2 data acquired in 2018 and 2019 were provided by European Space Agency and downloaded through the Amazon Web Services (<https://registry.opendata.aws/sentinel-1>, <https://registry.opendata.aws/sentinel-2>). TLS data for Helheim Glacier are available upon reasonable request. ICESat-2 laser altimetry tracks are available through the OpenAltimetry portal at <https://openaltimetry.org/data/icesat2/> with download services provided by the National Snow and Ice Data Center [36]. ArcticDEM digital elevation models [38, 39] are available from the University of Minnesota Polar Geospatial Center (PGC): <https://www.pgc.umn.edu/data/arcticdem/>. Ice surface velocity and BedMachine Greenland are freely available at the National Snow and Ice Data Center (NSIDC) at <https://nsidc.org/data/nsidc-0725/versions/5> [42] and <https://nsidc.org/data/idbmg4/versions/5> [43], respectively. The time series of Greenland terminus positions is available from [28] at <https://zenodo.org/records/10095674>.

5 Code availability All source data and MATLAB codes to reproduce figures in the main text are available through Zenodo at <https://zenodo.org/records/13382185> [41]. The codes used for the three-dimensional discrete element model are available from the corresponding author upon reasonable request. PFC3D[®] [44] is a software from Itasca Consulting Group, Inc. through a commercial license.

Comment (4). Specific comments

L28-29: Does this sentence refer to projections of Greenland’s contribution to sea level? Or both the Earth’s ice sheets? Given the previous sentences are about Greenland, it would make more sense to constrain this statement to Greenland as well.

Response. We have modified the aforementioned sentence for clarification as follows (lines 26–30):

Under high carbon emission scenario, the GrIS is projected to contribute about 79–167 mm of sea-level rise by 2100, 30% to 60% of which comes from iceberg calving at marine-terminating glaciers [45, 46]. Projections of sea level rise by 2100 can vary by 374 mm depending on the rate of iceberg calving at Greenland ice sheet margins [47].

Comment (5). L34: “calving retreats” is a strange term. Do you mean “calving front retreat”?

Response. Yes. We have modified the aforementioned sentence as follows (lines 34–35):

Recent large calving **front** retreats at some Greenland outlet glaciers have been correlated with rapid breakup of mélange,...

Comment (6). L36-38: It would be useful to clarify whether this is a finding from observations or modeling here. Or both? Secondly, has mélange rigidity actually been observed directly? If not, this would be a good place to clarify what previous observations have observed.

Response. It is a finding from observations only. Rigid, immobile mélange has been observed from either persistently low surface temperatures [29] or a coherent and uniform velocity field at the terminus measured by feature tracking [48, 49, 21, 22]. We have modified the sentences as follows (lines 36–41):

Remote observations reveal that seasonal advance and retreat of glacier termini coincides with the formation and disappearance of mélange [50], and variations in the mélange rigidity induced by sea ice that grows in winter and decays in summer [48, 29, 49, 21, 22]. Rigid, immobile mélange has been observed from either persistently low surface temperatures [29] or a coherent and uniform velocity field at the terminus measured by feature tracking [48, 49, 21, 22].

Comment (7). L39: Might use a different word to “buttressing” if you are going to define it in the next sentence.

Response. We have modified the sentence as follows:

These observations suggest that the presence of rigid mélange can mitigate iceberg calving by **providing a back force at the glacier terminus** [51, 6, 52, 53, 48, 49, 54, 29, 25, 8, 3]. The force exerted by the mélange to support the glacier terminus is called the mélange buttressing force [8].

Comment (8). L52-53: For readability, it would be useful to clarify what this statement implies.

Response. We have modified the sentence as follows (lines 55–57):

In early summer 2016 for Jakobshavn Isbræ, an unusually thick mélange wedge at the glacier front coincided with a one-month glacier quiescence period, **implying that thick mélange can inhibit calving** [6].

Comment (9). L70-71: Please provide more details about this approach since it looks like the findings from the modeling will hinge upon accurate computation of mélange thickness.

Response. See response to Comment 3 for details.

Comment (10). L72-77: This seems unnecessarily complicated. Why not mean or median elevation value for each specific distance?

Response. Due to the long-tail distribution of the number of data points against the mélange elevation at any specific distance from terminus, the proposed median or mean elevation value approach tends to incorporate large icebergs into the elevation profile, failing to represent the elevation piled up from small icebergs and sea ice. We have added a figure to compare the elevation profile from connecting the mean and median elevation values, and the elevation values with maximum numbers of data points, from both TLS and ArcticDEM data at Helheim Glacier (Fig. 2 in SI and Fig. 7 in this document). We have also added the following sentences in the main text to justify our method of constructing the mélange elevation profile:

1.1 Mélange thickness associated with calving dynamics at Helheim Glacier in 2019-2020. ... To display the spatial profile of the mélange elevation, we calculate distances from terminus for all data points in the ice mélange and plot them as density maps in Fig. 1(c)-(f). For any specific distance from the terminus, there is a spread of mélange elevation that exhibits a long-tail distribution due to the presence of large icebergs. To reflect the mélange elevation that is piled up from small icebergs and sea ice, we connect elevation values that have maximum numbers of data points along the distance from terminus as the representative mélange elevation profiles (solid blue lines in Fig. 1(c)-(f), see Methods and Fig. 2 in SI for a sensitivity study of this metric). In doing so, we prevent large icebergs from disrupting the mélange elevation profiles without cutting off the data by an arbitrary elevation threshold. The resulting representative mélange elevation profiles are always below 30 m, indicating that it is a reasonable elevation threshold to exclude large icebergs from consideration. To estimate mélange elevation near the glacier terminus (Z_0), we take an average of all data points below the 30 m threshold within 1 km (i.e., 1/5 of the fjord width) of the terminus. We infer thickness of the mélange based on TLS-derived surface elevations and assuming hydrostatic equilibrium.

We have added a paragraph in the Methods section for justification:

3.1 Terrestrial laser scanner data and uncertainty assessment. ... To construct the representative mélange elevation profile, we first connect the median or mean elevation values along the distance from the terminus, the shape of which turns out to be sensitive to large icebergs with elevation values larger than 30 m (Fig. 2 in SI). Incorporating large icebergs into the representative mélange elevation profile has several drawbacks. Firstly, instead of representing the elevation piled up from small icebergs and sea ice, it tends to reflect the size distribution of large icebergs. Secondly, the existence and location of large icebergs are sporadic and hard to characterize by a continuum model. Lastly, comparing mélange elevation profiles across different Greenland fjords becomes challenging, as the size and shape distribution of large icebergs are sensitive to calving styles [35] and terminus geometries. To reflect the mélange elevation that is piled up from small icebergs and sea ice, we connect elevation values

that have maximum numbers of data points along the distance from terminus as the representative mélange elevation profiles (solid blue lines in Fig. 1(c)-(f) in the main text and Fig. 2(b)(e) in SI).

FIGURE 7. Helheim Glacier and ice mélange. (a) TLS-measured elevation map after accounting for local differences between the ellipsoid and geoid, overlain on a Sentinel-1 HV image (both acquired on 30 Nov 2019). The white line across the fjord indicates the glacier front location. The white triangle indicates the TLS location. The upper left inset shows the location of Helheim Glacier in Greenland. The image is in polar stereographic projection (EPSG: 3413). (b) Surface elevation profile for the mélange displayed as a density plot; the colour bar denotes the number of data points that have the same elevation and distance from terminus values. For any specific distance from terminus, we find the elevation value that has the maximum number of data points. The solid blue line connects these elevation values along the distance from terminus as the representative mélange elevation profile. We also calculate the median and mean elevation values for each specific distance from terminus, and connect them by solid black and cyan lines, respectively. (c) The number of data points against the surface elevation value at distance from terminus of 0.5 km (black line), 1 km (blue line), and 2 km (red line). Due to the long-tail distribution of the mélange elevation, the median or mean elevation value approach in (b) tends to incorporate large icebergs into the elevation profile, failing to represent the elevation piled up from small icebergs and sea ice. (d) Arctic-DEM measured elevation map after accounting for local differences between the ellipsoid and geoid, overlain on a Landsat 8 image (acquired on 30 Mar 2014). The image is in polar stereographic projection (EPSG: 3413). (e) and (f) follow captions of (b) and (c).

Comment (11). *L78: Why are icebergs removed? Surely the discrete element model described later in the manuscript requires an iceberg size distribution?*

Response. See the response to comment 10. The mélange surface elevation map confirms that by excluding icebergs with freeboard heights above 30 m (blue regions in Fig. 7(a)(d) in this document), the elevation profile better reflects piling up of small icebergs and sea ice, instead of the size of individual large icebergs. In our discrete element model, we use the iceberg size distribution observed from satellite images, with an upper bound of 200 m (lines 618–629):

We use cubic grains which can achieve a higher packing density, thus buttressing forces, than disk-shaped grains. We adopt the iceberg **area** distribution observed in the mélange of Jakobshavn Isbræ and Helheim Glacier, which is approximated as a power-law distribution with an exponent of -2.0 [2, 1]. **The resulting iceberg size distribution for cubic grains is a power-law distribution with an exponent of -4.0. Taking the simulation with the Helheim fjord for instance (Fig. 1(b)), the side lengths of cubic icebergs are 36 m, 50 m, 75 m, 100 m, 200 m, and the corresponding numbers of particles are 3120, 838, 166, 52, 3, respectively.** To model mélange with different steady state thicknesses, the total number of particles varies from 2918 to 12522 for simulations with Helheim and Kangerlussuaq fjords. In simplified fjord geometries, the side lengths of cubic icebergs are 35.4 m, 50 m, 70.7 m, 100 m, 141.4 m, and the corresponding numbers of particles for the thick mélange are 8190, 2045, 510, 125, 30, respectively (Fig. 10(f)). We vary the total number of particles from 1634 to 15264 to change the mélange thickness.

Comment (12). *L85-88: Can you be more specific about which data was used to quantify calving? It seems from the Xie et al. (2019) reference that this was done using the TLS scans. Regardless, the approach to classify calving events as “major” vs. “minor” seems a little subjective and challenging to replicate. It would be useful to include a figure showing what these two calving events look like in the TLS scans.*

Response. With reference to previous classification of calving events [6], here, we define major calving events as those with block size $>0.25 \text{ km}^2$ (observed from TLS scans), and causing significant mélange motion and an overall terminus retreat (observed from satellite images); minor calving events are those in which visible blocks calved, but the mélange or terminus position remained largely unchanged. Though TLS has a higher temporal resolution than satellite images, satellite images (Sentinel-1, Sentinel-2 and Landsat 8) are still useful to check the overall mélange motion that cannot be fully captured by TLS. Xie et al (2019) also inferred calving events from TRI and satellite images (Fig. 4 in their paper [6]). We have added two figures of TLS scans during two episodes of calving cessation (Fig. 3 in SI and Fig. 8 in this document) and four major calving events (Fig. 4 in SI and Fig. 9 in this document). We have modified Section 1.1 in the main text as follows:

1.1 Mélange thickness associated with calving dynamics at Helheim Glacier in 2019-2020.... **We digitize the terminus position on each TLS scan**

with a straight line (the white line in Fig. 1(a)). The terminus advance/retreat refers to the changes in the terminus position, and is quantified by distances between these straight lines. This can indicate whether the glaciers are advancing (growing) or retreating (shrinking). With reference to previous classification of calving events [6], here, we define “major calving” events as those with block size $>0.25 \text{ km}^2$ (observed from TLS scans), and causing significant mélange motion and an overall terminus retreat (observed from satellite images); “minor calving” events are those in which visible blocks calved, but the mélange or terminus position remained largely unchanged. We observe two episodes of calving cessation, from 8 Oct 2019 - 31 Dec 2019 and 1 Mar 2020 - 20 May 2020, when no calving occurred and the terminus advanced steadily for 2 km and 1.7 km, respectively. We found that mélange elevation at the terminus averaged 15 m during these periods. We include TLS scans during these two episodes of calving cessation in the supplementary material (Fig. 3 in SI). We identified four dates where noticeable mélange thinning occurred, which were 3 Sep 2019, 3 Jan 2020 (Fig. 1(d)), 31 May 2020 (Fig. 1(f)), and 14 July 2020. Around these four dates, the mélange elevation at the terminus decreased as follows: by 1.6 m within two days to $10.8 \pm 0.05 \text{ m}$, by 5.3 m within three days to $9.4 \pm 0.05 \text{ m}$, by 4 m within five days to $10 \pm 0.05 \text{ m}$, and by 2.3 m within seven days to $11.6 \pm 0.05 \text{ m}$. Major calving happened around these dates, with corresponding retreats at the terminus of 0.5 km, 1.2 km, 1.3 km, and 1.0 km, respectively, accompanied by large increases in fraction of mélange area composed of large icebergs (see SI Fig. 4 & 5 for associated TLS scans and temporal evolution of iceberg fraction).

FIGURE 8. The TLS-measured mélange surface elevation map at Helheim Glacier during two episodes of calving cessation, from (a, b) 8 Oct 2019 – 31 Dec 2019, and (c, d) 1 Mar 2020 – 20 May 2020. White lines indicate positions of the terminus. Dates on the images show the acquisition dates for TLS data. TLS scans are overlain on Sentinel-1 and 2 images acquired around the same dates. Images are in polar stereographic projection (EPSG: 3413).

FIGURE 9. The TLS-measured mélange surface elevation map at Helheim Glacier during four major calving events, from (a)-(c) 3 Sep 2019 – 4 Sep 2019, (d)-(f) 31 Dec 2019 – 3 Jan 2020, (g)-(i) 27 May 2020 – 31 May 2020, and (j)-(l) 14 Jul 2020 – 18 Jul 2020. Dashed white lines indicate positions of the terminus before calving. Solid white lines indicate positions of the terminus after calving. Dates on the images show the acquisition dates for TLS data and times are in UTC. TLS scans are overlain on Sentinel-1 and 2 images acquired around the same dates. Images are in polar stereographic projection (EPSG: 3413).

Comment (13). *L87: How was terminus retreat defined and how was it quantified?*

Response. We digitize the terminus position on each TLS scan with a straight line (the white line in Fig. 1(a) in the main text). The terminus advance/retreat refers to the changes in the terminus position, and is quantified by distances between these straight lines. This can indicate whether the glaciers are advancing (growing) or retreating (shrinking). We have modified Section 1.1 in the main text (see response to comment 12).

Comment (14). *L89: How is this “terminus quiescence” defined? No major or minor calving events? Or just no major calving events.*

Response. To avoid ambiguities, we replace “terminus quiescence” with “calving cessation”, which refers to no major or minor calving events. We have modified Section 1.1 in the main text (see response to comment 12).

Comment (15). *L90-96: These are all very interesting findings but they need to be developed in more detail to be fully understood. See general comment 2.*

Response. We have included TLS scans during these four major calving events in the supplementary material (Fig. 4 in SI and Fig. 9 in this document). We have modified Section 1.1 in the main text (see response to comment 12).

Comment (16). *L96-102: This kind of interpretation/reflection would probably be a better fit for the discussion section.*

Response. We have moved the L96–102 interpretation/reflection to the discussion section as follows (lines 419–430):

2 Discussion ... Previous research suggests that the presence of ice mélange can reduce iceberg calving by providing “backstress” to the terminus [6, 49, 25, 48, 52, 29, 8, 3, 54, 53, 51]. Our comparisons of time-varying mélange thickness and calving dynamics at Helheim Glacier (Fig. 1(b)) support the view that the buttressing force increases with the mélange thickness and thus inhibits calving. Our TLS scans indicate a correlation between mélange thickness and calving dynamics, but we cannot determine causality before examining other mechanisms driving calving dynamics. For instance, the terminus advances right after major calving events can be explained by the possibility that the new glacier front has fewer crevasses and will take a longer period to calve again. Alternatively, the mélange thickness and the terminus react simultaneously but independently to other oceanic and atmospheric forcing. To establish a causal relationship between thin mélange and calving events, we would need in-situ observations with high temporal resolution in minutes to capture the sequence of a calving event and a mélange thinning event [19, 6]. ...

Comment (17). *L103: Not sure what to suggest here but the abrupt transition from field measurements to satellite remote sensing observations interrupts the flow of the manuscript due to the mismatch in scale. This generally makes it challenging to follow the narrative throughout the manuscript.*

Response. We have added a transition paragraph in Section 1.1 as follows:

1.1 Mélange thickness associated with calving dynamics at Helheim Glacier in 2019-2020. ... Apart from the weekly terminus variability of Helheim Glacier presented in this section, remote sensing observations on many Greenland glacier termini have shown significant terminus-position seasonality, with advance in winter and retreat in spring to summer through enhanced calving [26, 27, 28]. Previous studies have attributed seasonal calving dynamics to buttressing from ice mélange [29, 30, 20], which motivates us to further explore seasonal changes of mélange thickness and calving dynamics on other Greenland glaciers.

1.2 Seasonal changes of mélange thickness and calving dynamics. To investigate whether there are correlations between ice mélange thickness and

calving dynamics on other glaciers, we use ICESat-2 observations of mélange surface elevation. ...

Comment (18). L114-115: *Which ICESat-2 product? How were freeboards computed? Was it the difference from a sea level product? Or track-to-track differences? Even though some of these details are provided later in the Methods section, more details are needed here to fully interpret the findings.*

Response. We thank the reviewer for pointing out the ambiguities. We have revised Section 1.2 in the main text as follows (lines 134–140):

We identify ICESat-2 tracks passing over glacier termini in different seasons for Jakobshavn Isbræ (Fig. 2(a)), Kangerlussuaq Glacier (Fig. 2(b)), and Store Glacier (Fig. 2(c)). We use the ATL06 data set from ICESat-2 that provides geolocated, land-ice surface heights above the WGS 84 ellipsoid [55]. After removing the geoid heights at glacier termini [31], the mélange surface elevations are heights above mean sea level with tidal variations [6]. We then remove the tidal trend of mélange elevations with a tidal model [32]. Surface elevation data is acquired along the ICESat-2 track and displayed as a function of the distance from terminus.

Comment (19). L116: *Which years?*

Response. We have added the years as follows (lines 142–145):

We found that mélange was continuously present at Jakobshavn Isbræ from 2021 to 2022 and at Kangerlussuaq Glacier in 2020, while it was seasonally present at Store Glacier in 2019. Where mélange persisted, we calculated seasonally distinct freeboard heights from winter to early spring (solid black lines) and in summer (dashed black lines) (Fig. 2(d)(e)).

Comment (20). L119-120: *Are these averages or ranges?*

Response. They refer to ranges, and we have modified the main text as follows (lines 145–147):

Near the termini, mélange for the two glaciers both exhibit different ranges of freeboard heights during the two seasons: 20 ~ 35 m in winter or spring, and below 5 m in summer.

Comment (21). L127: *Is this a new conclusion? If not, please include citations to studies who have already identified this behavior.*

Response. L127: “In summary, the available data supports our hypothesis that thick mélange in winter inhibits calving and leads to the seasonal terminus advance” is a new conclusion. Previous research has identified seasonal calving dynamics and attributed them to the presence of rigid mélange in winter observed from either persistently low surface temperatures [29] or a coherent and uniform velocity field at the terminus measured by feature tracking [48, 49, 21, 22]. Our study provides new insights into why rigid mélange in winter provides a larger buttressing force. This is because the mélange is thicker in

winter, as revealed by our systematic analysis of ICESat-2 and ArcticDEM data. Previous research on seasonal calving dynamics has been included in the introduction section as follows (lines 36–41):

Remote observations reveal that seasonal advance and retreat of glacier termini coincides with the formation and disappearance of mélange [50], and variations in the mélange rigidity induced by sea ice that grows in winter and decays in summer [48, 29, 49, 21, 22]. Rigid, immobile mélange has been observed from either persistently low surface temperatures [29] or a coherent and uniform velocity field at the terminus measured by feature tracking [48, 49, 21, 22]. These observations suggest that the presence of rigid mélange can mitigate iceberg calving by providing a back force at the glacier terminus [51, 6, 52, 53, 48, 49, 54, 29, 25, 8, 3]. The force exerted by the mélange to support the glacier terminus is called the mélange buttressing force [8]. Prescribing a periodic change in the magnitude of the mélange buttressing force in ice-sheet models successfully reproduces observed seasonal calving dynamics [56, 57, 58, 53, 54, 59, 60, 46].

We have also modified the summary sentence in Section 1.2 as follows:

In summary, ICESat-2 data provides strong evidence of correlations between mélange thickness and terminus seasonality.

Comment (22). Fig. 2: It would be more appropriate to convert these y-axis values in panel D to ice mélange thickness, given that it is the important variable for the modeling.

Response. We thank the reviewer for the constructive advice. Observations from TLS at Helheim Glacier in 2019–2020 (Section 1.1), and ICESat-2 at Jakobshavn Isbræ in 2021–2022, Kangerlussuaq Glacier in 2020, and Store Glacier in 2019 (Section 1.2), reveal that mélange thickness associates with monthly and seasonal calving dynamics, which motivates us to develop three-dimensional models for ice mélange (Sections 1.3 to 1.5). With the developed continuum model for the mélange buttressing force, we further investigate calving dynamics associated with mélange thickness seasonality across 32 Greenland glacier termini in 2013–2022 from ArcticDEM observations (Section 1.6). For generality, we decided to use observations from ArcticDEM strips in Section 1.6 (Table 2 in SI or Fig. 8 in the main text) to inform the range of mélange thickness in continuum and discrete element models, which have better spatial and temporal coverage than TLS or ICESat-2 data. Due to the structure of the result section (Section 1.1 to 1.6), we provide mélange freeboard heights directly retrieved from TLS, ICESat-2, and ArcticDEM data, together with inferred mélange thickness (assuming hydrostatic equilibrium) and the buttressing force from ArcticDEM data only.

Comment (23). L130: “quantifying the buttressing”

Response. We have revised the sentence as follows:

As a result, quantifying the buttressing force of *mélange* in terms of its thickness is the first step to better representing ice-ocean interactions and developing process-based calving models.

Comment (24). L175: Are the “winter velocity fields” of the *mélange*? If so, please clarify.

Response. Yes, we have modified the sentence as follows (lines 203–206):

Previous studies have shown that winter velocity fields of the *mélange* are generally steady and highly uniform in space [3, 8], whereas summer velocity fields of the *mélange* tend to be much more variable and can be uniform, compressional, or extensional [3].

Comment (25). L182: Are these ice *mélange* thickness values from the TLS, ICESat-2 or from the references cited? It would be useful to include a few sentences here showing how the hard-won observational data has been incorporated into the continuum model. Otherwise, the manuscript feels like several different ideas with very little that links them together.

Response. For generality, we decided to use observations from ArcticDEM strips in Section 1.6 (Table 2 in SI or Fig. 8 in the main text) to inform the range of *mélange* thickness in continuum and discrete element models, which have better spatial and temporal coverage than TLS or ICESat-2 data. For the coherence of the result section, see response to comment 22. We have modified sentences in the continuum model section as follows:

To characterize the relative magnitude of the horizontal deviatoric stress to the glaciostatic pressure, we substitute representative values for parameters in Eqn. 3 and obtain:

$$\frac{|4H_0(\eta\frac{\partial u}{\partial x})|_{x=0}}{\frac{1}{2}\rho_i(1 - \frac{\rho_i}{\rho_w})g\phi_0H_0^2} \in [3.90 \times 10^{-14}, 1.17 \times 10^{-11}] \times \eta, \quad (13)$$

where the range of *mélange* thickness, H_0 , is derived from DEM observations. We take H_0 to vary from 35 m, which is the minimum size of icebergs detected within the *mélange* [1, 2], to 240 m, which is the thickest *mélange* observed across 32 Greenland termini in 2013–2022 (See Table 2 in SI).

Comment (26). Eq 5: Some sort of summarizing statement after this equation would be useful to remind the reader that the buttressing effect is proportional to the square of *mélange* thickness. It was mentioned in the introduction but would be good to say it again to tie this section back to the application.

Response. We have added a paragraph at the end of the continuum model section as follows (lines 246–249):

In summary, momentum balance equations reveal that the *mélange* buttressing force per unit width is solely controlled by the packing density ϕ_0 and *mélange* thickness H_0 at the glacier terminus, and is proportional to the square of

mélange thickness (Eqn. (2)). Additionally, mélange thickness exponentially decays with its distance from terminus (Eqn. (12)).

Comment (27). L197-198: It is unclear where the range in initial ice thicknesses come from. Are they from the TLS scans at Helheim or the ICESat-2 data from one of the other big glaciers? 378 m seems a little thick given that the authors are assuming that the density of the ice mélange is uniform with both depth and width. If the observations are to be believed, then a much lower value should be chosen that represents the at least the width-averaged thickness of the ice mélange (e.g. Fig. 1c-f). This statement epitomizes the major problems with this manuscript in that some (good) individual ideas have just been copied and pasted into a single manuscript without much attempt to link them together.

Response. See response to Comment (1) on the modified discrete element models with Helheim and Kangerlussuaq fjord geometries (Section 1.4). In particular, we have modified the mélange initial thickness in the model as follows:

1.4 A three-dimensional discrete element model of ice mélange. ... We adopt fjord geometries from Helheim and Kangerlussuaq glaciers (black dots in Fig. 1(a)(d))... In a series of simulations, we vary the initial mélange thickness to determine its influence on the steady state buttressing force. The steady state mélange thickness at the terminus varies from 40~135 m at Helheim fjord, and 60~240 m at Kangerlussuaq fjord, that are consistent with ArcticDEM observations (Table 2 in SI).

For previous simulations with simplified channel configurations (straight and rugged), we have provided justification for the imposed thick mélange (378 m) (lines 287–296):

To further explore the effect of fjord frictional properties on the mélange buttressing force, we adopt two simplified channel configurations that conceptualize complex Greenland fjord geometries. (1) The straight channel configuration (Fig. 10(b)(d)(g)(i)) has a constant-width fjord. (2) The rugged channel configuration (Fig. 10(c)(e)(h)(j)) has uniformly-spaced bulges on both sides [8], which can be more clearly seen from a perspective view of the simulation (Fig. 7 in SI). We present modeling results for a thin and thick layer of ice mélange in Fig. 10, initialized with a uniform initial thickness, H_{ini} , set to 60 m and 378 m, respectively. **The upper limit of the initial thickness refers to the unusually thick mélange wedge observed over a one-month period in early summer 2016 at Jakobshavn Isbræ, which has a thickness of 400 m over one-fifth of the fjord width [6].**

Comment (28). L199-200: Any justification for this value?

Response. In the new discrete element simulations with Helheim and Kangerlussuaq fjords, we adopted a terminus velocity of 30 m/day from remote observations [16, 17, 18]. We revised the main text as follows:

We push the left end of the mélange with an advancing terminus at 30 meters per day [16, 17, 18] and record the temporal evolution of the buttressing force exerted on the terminus.

For previous simulations in simplified fjord geometries (straight and rugged), we used a terminus velocity of 43.2 m/day and provided justification in the Method section (Section 3.4) as follows:

We move the terminus on the left at a constant velocity, $V_{\text{ter}} = 30$ m/day for real fjord geometries [16, 17, 18], and 43.2 m/day for simplified fjord geometries [25, 8]. To confirm that the averaged steady-state buttressing force is invariant to the terminus velocity, we conducted simulations with mélange thickness 280 m, terminus velocity at 21.6 m/day, 43.2 m/day, and 86.4 m/day, for both straight fjords and rugged fjords configurations (Fig. 14 in SI). The results show that the averaged buttressing force is mostly invariant to the terminus velocity in both fjord configurations. Taking the force fluctuations into account, the maximum buttressing force difference among the chosen velocities is 4% and 8% for straight and rugged fjords, respectively. In rugged fjords, faster terminus motion leads to larger force fluctuations due to larger velocity gradient during stick-slip/jam-unjam cycles. We adopt the terminus velocity of 43.2 m/day to simulate mélange in simplified fjord geometries for the sake of computational efficiency.

Comment (29). L204: Can you show an image of what this looks like? There do not appear to be any uniformly-spaced bulges in Figure 4.

Response. Figure 4 already has bulges on the fjord walls. As bulges (thickness of 20 m and length of 60 m) are much smaller than the mélange length (around 6 km) and Figure 4 is drawn to scale, we have to zoom in to see the bulges in Figure 4 (e)(j). We have added schematics near the texts “straight fjord” and “rugged fjord” to highlight the bulges (Fig. 10 in this document). To better present the bulges, we have also included a perspective view of the simulation in rugged fjords in the supplemental material (Fig. 7 in SI and Fig. 11 in this document). We have revised the main text as follows:

The rugged channel configuration (Fig. 10(c)(e)(h)(j)) has uniformly-spaced bulges on both sides [8], which can be more clearly seen from a perspective view of the simulation (Fig. 7 in SI).

FIGURE 10. The three-dimensional discrete element model for mélange composed of cubic icebergs with a power-law size distribution and confined within simplified fjord geometries. (a)-(e) For a thin mélange, (f)-(j) for a thick mélange. (a) A side view of the initial condition for the simulation with $W = 1$ km, $L = 3$ km, and $H_{\text{ini}} = 60$ m. The glacier terminus is shown as a grey block on the left. The ocean floor and fjord walls are plotted in brown. The glacier terminus starts to move at a constant velocity, $V_{\text{ter}} = 43.2$ m/day. (b)-(e) are snapshots for iceberg positions and velocities after 16 days into simulations with steady terminus advance and no calving. (b), (d) are the side and top view for a straight fjord wall configuration; (c), (e) are the side and top view for a rugged fjord wall configuration. **The two-day averaged velocity** of each iceberg element is indicated by filled colour in (d) and (e). (f) A side view of the initial condition for the simulation with $W = 1$ km, $L = 3$ km, and $H_{\text{ini}} = 378$ m. (g)-(j) follow captions of (b)-(e). See supplementary videos for the full temporal evolution of the mélange behaviors.

FIGURE 11. A perspective view of the three-dimensional discrete element model for the thick *mélange* confined within the rugged fjord wall configuration (the simulation shown in Fig. 5 (h)(j) in the main text).

Comment (30). L209-212: *Another mismatch between observations and modeling. Surely if the thick *mélange* is bunching up near the terminus, there would compressive stresses which would appear in the strain rate fields documented by Amundson and Burton (2018)? But their observations demonstrate uniform flow during winter.*

Response. The reviewer points out the modeled *mélange* velocity field (compressive flow in rugged channels) is inconsistent with the remote observations (uniform flow in winter) [3], which motivates us to check our calculation of the *mélange* velocity. Previously, we presented the iceberg velocity as the instantaneous velocity directly output from the discrete element model, which is the particle’s velocity at each mechanical timestep (0.1 s). However, the observed *mélange* velocity field is retrieved by applying particle image velocimetry (PIV) to satellite imagery, with the time interval between image pairs ranging from 2 to 30 days [3]. Therefore, we analyze modeled velocity fields, including instantaneous velocity, velocity averaged over one hour, one day, and two days, and compare results with the satellite-derived velocity fields of the *mélange* [3] (Fig. 6 in SI and Fig. 12 in this document). The comparison shows that the two-day averaged velocity field, which is calculated by dividing the iceberg’s displacement over two days of terminus motion by the time interval, showcases both uniform (Fig. 12(e)) and extensional (Fig. 12(k)) flow regimes that are consistent with remote observations [3]. We have presented the two-day

averaged velocity for the simulations with Helheim and Kangerlussuaq fjords (Fig. 4 in the main text and Fig. 1 in this document). We have modified the main text as follows (lines 268–276):

We calculate the two-day averaged velocity of each iceberg element by dividing the iceberg's displacement between 14 and 16 days of terminus motion by the time interval (two days). The mélange near the terminus moves at the terminus velocity with shear bands developed at fjord walls (Fig. 4(c)(f)). The mélange near the open end becomes loosely-packed and more fluidic (See supplementary video for the full temporal evolution of the mélange behaviors.) The modeled velocity field showcases both uniform (Fig. 4(c)) and extensional (Fig. 4(f)) flow regimes that are consistent with remote observations [3]. See Fig. 6 in SI for the modeled velocity fields, including instantaneous velocity, velocity averaged over one hour, one day, and two days, and the satellite-derived velocity fields of the mélange [3].

FIGURE 12. The three-dimensional discrete element model for mélange composed of cubic icebergs with a power-law size distribution and confined within real fjord geometries. (a)-(e) For Helheim glacier, (g)-(k) for Kangerlussuaq glacier. The real fjord geometry is scaled down by four times for the discrete element model. (a) The side view for iceberg positions after 16 days into simulations with steady terminus advance and no calving. The glacier terminus moves at a constant velocity, $V_{\text{ter}} = 30$ m/day. (b) The top view of the instantaneous velocity of iceberg element indicated by filled colour. We also calculate the time averaged velocity of each iceberg element by dividing the iceberg's displacement over one hour, one day, and two days of terminus motion by the corresponding time intervals. We indicate the 1-hour averaged (c), 1-day averaged (d), and 2-day averaged (e) velocity field by filled colour. (g)-(k) follow captions of (a)-(e). See supplementary videos for the full temporal evolution of the mélange behaviors. Remote observations of uniform (f) and extensional flow (l) regimes of ice mélange adapted with permission from [3]. Velocities were calculated over 3–12 February 2014 at Helheim Glacier (f), and 7–9 August 2014 at Kangerlussuaq Glacier (l). Gray regions indicate where image cross-correlation results were discarded due to low signal-to-noise ratios.

We have also replaced instantaneous velocity fields for straight and rugged fjords with two-day averaged velocity fields (Fig. 5 in the main text and Fig. 10 in this document), and modified the main text as follows (lines 301–305):

In the straight channel configuration, the mélange behaves like plug flow with a uniform velocity profile within the fjord (Fig. 10(d)(i)). In the rugged channel configuration, the mélange exhibits shear bands near fjord boundaries **with uniform flow near the terminus** (Fig. 10(e)(j)), which has also been reported in previous studies [8, 3].

Comment (31). L250: It would be useful to report the buttressing force for the thick mélange simulations in these units as well so that the findings can be compared to other studies.

Response. The reviewer suggests us to convert the reported mélange buttressing force over the fjord width (N/m) into the mélange backstress (Pa). Previous research found that the mélange applies a supporting pressure equivalent to a backstress of 30–60 kPa acting on the entire face of the Store Glacier terminus [51]. To calculate the mélange buttressing force over the fjord width and convert it to the mélange backstress, we need to know both the mélange thickness and the terminus thickness [53], the latter of which is absent in our simulations. In fact, it is more convenient to compare to other studies with the current unit (N/m), including field observations [11], simulations [25, 8], and analytical analysis [52].

Comment (32). L335: Melting?

Response. Yes, we have modified the main text as follows (lines 443–446):

We note that if submarine melting causes the observed summer thinning of mélange, mélange's buttressing strength can be strongly tied to submarine **melting**. The impact of submarine **melting** on mélange strength can be significant due to the strong dependence of buttressing on mélange thickness inferred in our study.

Comment (33). L338–345: This is an interesting piece of evidence that would appear to support your argument about mélange control on terminus advance/retreat. However, none of this evidence is presented in the results and there is no accompanying figure to support these surprising findings. I am finding it a little hard to believe that there is 100 m thick mélange in July and August in Greenland. Can you show a figure that demonstrates better demonstrates this finding for these six glaciers? You could also go into more detail. For example, do these glaciers always have mélange at this time of year? If not then you could look at interannual advance/retreat patterns to provide more evidence to support your hypothesis.

Response. The reviewer suggests that to better support our argument that thick mélange in summertime leads to terminus advance at the six glaciers (Hayes SS, Alison, Unnamed Deception, Unnamed Unartit Islands, Koge Bugt C, and Kong Oscar), we should examine interannual advance/retreat patterns. Specifically, we should check

whether unusually thick mélange is observed in June/July only in the years when terminus advances in summer, and thin mélange is observed in June/July in other years when terminus retreats in summer. After checking the available ArcticDEM data and terminus interannual advance/retreat patterns, we found abundant evidence at Koge Bugt C glacier to support our argument. We found that the terminus advanced from July to September in 2015 and retreated or remained a plateau during the same period in 2014, 2016, 2017 and 2020 [28]. From ArcticDEM strips acquired in June/July in 2014–2020, we only observe thick mélange in 2015, and thin mélange in other years. We attribute the summertime terminus advance in 2015 to mélange buttressing from the presence of unusually thick mélange observed on 2 Jul with a thickness of 114_{-36}^{+30} m (see Fig. 9 in SI and Fig. 13 in this document), the same as what happened at Jakobshavn Isbræ in Jun 2016 [6]. We have modified the main text as follows (lines 447–453):

We note that the hypothesis of summer-runoff induced calving, on its own, can not explain the observations at Koge Bugt C Glacier (ID=13 in Fig. 8) where the terminus advanced from July to September in 2015 and retreated or remained a plateau during the same period in 2014, 2016, 2017 and 2020 [28]. We attribute the summertime terminus advance in 2015 to mélange buttressing from the presence of unusually thick mélange observed on 2 Jul with a thickness of 114_{-36}^{+30} m (see Fig. 9 in SI for details), the same as what happened at Jakobshavn Isbræ in Jun 2016 [6].

We also examined ArcticDEM data and terminus interannual advance/retreat patterns for the other five glaciers. We found that the termini advance from May to July in multiple years and there are very few ArcticDEM strips acquired in June/July, which prevents us from revealing any correlation between mélange thickness and terminus dynamics in summer (Fig. 14–18 in this document).

FIGURE 13. (a)-(e) The mélangé elevation map above mean sea level at Koge Bugt C Glacier from ArcticDEM observations, acquired in June and July from 2014–2020, overlain on the satellite image acquired around the same date. The image is in polar stereographic projection (EPSG: 3413). The date of ArcticDEM acquisition is shown on the upper left corner. (f)-(j) The surface elevation profile for the mélangé presented in (a)–(e) displayed as a density plot. The solid blue line is the representative mélangé elevation profile as a function of distance from terminus. (k) A time series of terminus position [28], ArcticDEM acquisition dates and observed mélangé freeboard heights (Z_0) at the terminus of Koge Bugt C Glacier. The solid red line marks the ArcticDEM acquisition date when an unusually thick mélangé is observed at the terminus in July with the terminus advancing from May to October in 2015. The solid black lines mark ArcticDEM acquisition dates around the same time (late June to July) in different years (2014, 2016, 2017, 2020) when thin mélangé is observed at the terminus with the terminus retreating or remaining a plateau from July to September (the shaded gray regions).

FIGURE 14. (a),(c) The mélangé elevation map above mean sea level at Hayes Glacier SS from ArcticDEM observations, acquired on 28 May 2016 and 5 Jun 2018, overlain on satellite images acquired around the same dates. Images are in polar stereographic projection (EPSG: 3413). The date of ArcticDEM acquisition is shown on the upper left corner. (b),(d) The surface elevation profile for the mélangé presented in (a),(c) displayed as a density plot. The solid blue line is the representative mélangé elevation profile as a function of distance from terminus. (e) A time series of terminus position [28], ArcticDEM acquisition dates and observed mélangé freeboard heights (Z_0) at the terminus of Hayes Glacier SS.

FIGURE 15. (a),(c) The mélangé elevation map above mean sea level at Alison Glacier from ArcticDEM observations, acquired on 18 Jun 2015 and 16 Jun 2017, overlain on satellite images acquired around the same dates. Images are in polar stereographic projection (EPSG: 3413). The date of ArcticDEM acquisition is shown on the upper left corner. (b),(d) The surface elevation profile for the mélangé presented in (a),(c) displayed as a density plot. The solid blue line is the representative mélangé elevation profile as a function of distance from terminus. (e) A time series of terminus position [28], ArcticDEM acquisition dates and observed mélangé freeboard heights (Z_0) at the terminus of Alison Glacier.

FIGURE 16. (a) The *mélange* elevation map above mean sea level at Unnamed Deception Glacier from ArcticDEM observations, acquired on 21 Jun 2016, overlain on a satellite image acquired around the same date. The image is in polar stereographic projection (EPSG: 3413). The date of ArcticDEM acquisition is shown on the upper left corner. (b) The surface elevation profile for the *mélange* presented in (a) displayed as a density plot. The solid blue line is the representative *mélange* elevation profile as a function of distance from terminus. (c) A time series of terminus position [28], ArcticDEM acquisition date and observed *mélange* freeboard height (Z_0) at the terminus of Unnamed Deception Glacier.

FIGURE 17. (a) The *mélange* elevation map above mean sea level at Unnamed Uunartit Islands from ArcticDEM observations, acquired on 7 Aug 2018, overlain on a satellite image acquired around the same date. The image is in polar stereographic projection (EPSG: 3413). The date of ArcticDEM acquisition is shown on the upper left corner. (b) The surface elevation profile for the *mélange* presented in (a) displayed as a density plot. The solid blue line is the representative *mélange* elevation profile as a function of distance from terminus. (c) A time series of terminus position [28], ArcticDEM acquisition date and observed *mélange* freeboard height (Z_0) at the terminus of Unnamed Uunartit Islands.

FIGURE 18. (a)–(e) The mélange elevation map above mean sea level at Kong Oscar Glacier from ArcticDEM observations, acquired in July from 2014–2020, overlain on the satellite image acquired around the same date. The image is in polar stereographic projection (EPSG: 3413). The date of ArcticDEM acquisition is shown on the upper left corner. (f)–(j) The surface elevation profile for the mélange presented in (a)–(e) displayed as a density plot. The solid blue line is the representative mélange elevation profile as a function of distance from terminus. (k) A time series of terminus position [28], ArcticDEM acquisition dates and observed mélange freeboard heights (Z_0) at the terminus of Kong Oscar Glacier.

Comment (34). *Section 3.1: Given that there are no references, it appears that this is the first manuscript that has used data from the TLS instrument. The brevity of this section is therefore a little concerning. It’s probably easiest for me to communicate this feedback as questions: where was the TLS located? How far was it from the calving front (nearest/furthest)? Was a DEM of the mélange derived every day during the study period? It looks like it from Fig. 1b but it’s difficult to tell. How many points were contained in each 100 x 100 m grid cell? What is the reasoning behind the vertical accuracy choice of 0.10 m? Is the accuracy of the freeboard or ice thickness? How do you know the accuracy varies with distance from the scanner? Is it because there are fewer points to average?*

Response. The TLS is located at a latitude of 66.32963° N and a longitude of -38.1739° W, with the location now provided in Figure 1a (Fig. 19 in this document) and is approximately 1 km - 5.8 km from Helheim Glacier’s calving face and is 471 meters above sea level

sampled from ArcticDEM. DEMS in the melange were derived every 6 hours from 2019-09-01 to 2019-10-13 and from 2020-05-01 to 2020-09-01. Daily DEMs were derived from 2019-10-14 to 2020-04-30. This record contains some data gaps due to power constraints and atmospheric conditions. Points per 100 x 100 m grid cell in the melange can range from ~ 2000 to ≥ 10000 points. The vertical accuracy is for the freeboard height. More of a detailed error analysis is provided in the response to comment 3. The revised Results section on TLS data (Section 1.1) is provided in the response to comment 2.

FIGURE 19. Helheim Glacier and ice mélange. (a) TLS-measured elevation map after accounting for local differences between the ellipsoid and geoid, overlain on a Sentinel-1 HV image (both acquired on 30 Nov 2019). The white line across the fjord indicates the glacier front location. **The white triangle indicates the TLS location.** The upper left inset shows the location of Helheim Glacier in Greenland. The image is in polar stereographic projection (EPSG: 3413). (b) Terminus position relative to 1 Sep 2019, where the positive sign indicates terminus advance. Blue dots denote the averaged mélange elevation within 1 km of the terminus, Z_0 . Calving events are inferred from TLS and satellite images. Due to limited temporal sampling of the data, we are not able to determine the exact time of each calving event. Instead, we mark the time period during which a calving event occurs by a red-shade rectangle. Four vertical black lines mark the dates for the TLS-measured elevation data presented in (c)-(f), which corresponds to 30 Nov 2019, 3 Jan 2020, 30 Mar 2020, and 31 May 2020, respectively. Solid black lines mark the dates with terminus advances, and dashed black lines mark the dates with terminus retreats. (c)-(f) Surface elevation profiles for the mélange displayed as density plots (1510~1859 data points in total); the colour bar denotes the number of data points that have the same elevation and distance from terminus values. For any specific distance from terminus, we find the elevation value that has the maximum number of data points. Solid blue lines connect these elevation values along the distance from terminus as the representative mélange elevation profiles. Mélange thinning on 3 Jan (d) and 31 May (f) coincided with more calving activities and terminus retreats as shown in (b).

Comment (35). L382: It's a bit misleading to use the term "spatial resolution" for ICESat-2 data. "Segment length" would be a more accurate term.

Response. Yes, we have modified the main text as follows:

3.2 ICESat-2 data and uncertainty assessment. We use the ATL06 data set from ICESat-2 that provides geolocated, mean land-ice surface heights above the WGS 84 ellipsoid that are averaged along 40 m segments of ground track and spaced 20 m apart [55].

Comment (36). L383-384: This seems a little fraught given the resolution of the geoid and uncertainty in the tidal corrections. A more convincing way of reporting the uncertainty would be compute the difference between ATL06 over the ocean and the geoid with tidal corrections.

Response. See response to Comment 3.

Comment (37). L399-400: Was a minimum advance/retreat used when making this classification? In other words, is a 1 m advance classified the same as a 1,000 m advance?

Response. We did not use any minimum advance/retreat distance to make the classification, because the terminus velocity varies across Greenland and it is hard to have a universal advance/retreat distance threshold for the studied 32 Greenland termini (see Fig. 17–48 in SI). Instead, we define "ArcticDEM acquired during terminus advance/retreat" as "when the terminus kept advancing/retreating within a two-month time window centering on the DEM acquisition date". If terminus alternated between advancing and retreating within the time window, we discarded the corresponding DEM strip because the relationship between mélange and calving dynamics was ambiguous in this case. The classification has been presented in the main text as follows (lines 368–386):

We then extend our study to 32 glacier termini, most of which (ID 1~25) are picked from previous studies with strong terminus-position seasonality [26, 27], and the rest (ID 26~32) have annual ice discharge larger than 5 Gt/yr [61]. The locations of the termini are marked on a Greenland velocity map in Fig. 8(a). We identify 341 ArcticDEM strips at 2-meter resolution that cover the mélange regions for the 32 studied termini. For each DEM strip, we investigate terminus position variations [28] during a two-month time window centering on the DEM acquisition date. If the terminus keeps advancing (or retreating) within the time window, then the DEM potentially represents mélange with a strong (or weak) buttressing force. If the terminus alternates between advancing and retreating within the time window, we discard the corresponding DEM strip because the relationship between mélange and calving dynamics is ambiguous in this case. After filtering all DEM strips through this criterion, we identify 60 DEM strips during terminus advances, and 50 DEM strips during terminus retreats, from February to November in 2013–2022. Figure 8 in SI presents an example of the DEM filtering procedure at Helheim Glacier, resulting in two TLS-derived DEMs during terminus advances (Fig. 1(c)(e)), and two ArcticDEM strips during terminus retreats. For every ArcticDEM strip, the mélange thickness is defined as

the maximum value obtained from the representative mélange thickness profile within a 200-meter range from the terminus. Table. 2 in SI summarizes the observed minimum (or maximum) mélange thickness when terminus retreats (or advances) as H_0^{\min} (or H_0^{\max}), with the corresponding DEM acquisition month shown in the bracket.

Comment (38). L404-405: *The effort that went into correcting the ArcticDEM strips is appreciated but the actual uncertainty of this approach is a combination of ArcticDEM uncertainty + geoid/tidal correction uncertainty. Currently, the authors assume there is no uncertainty in the geoid/tidal correction which seems unwise. Again it would be useful to compute this difference between the two datasets over a non-mélange area to provide a more robust estimate of the accuracy of this method.*

Response. See response to Comment 3.

Comment (39). L414: *It would make more sense if this paragraph went before the computation of ice thickness.*

Response. We have reorganized the Method section for ArcticDEM as suggested by the reviewer. See response to Comment 3.

Comment (40). Section 4: *The data availability statement does not comply (at least by my interpretation) with Nature’s policy since I am not able to interpret and verify the research in the manuscript. It is not sufficient to provide links to the raw data files since they have been substantially modified. At a minimum, please provide access to the corrected ArcticDEM strips, ICESat-2 ice mélange freeboards/ice thicknesses.*

Response. We have included the elevation offsets on the ICESat-2 and ArcticDEM data, together with pipelines to plot figures in the main text at <https://zenodo.org/records/13382185> [41]. See response to Comment 3 for the revised data and code availability sections.

In summary, we thank the referee for making these constructive suggestions. We hope that our responses have resolved the ambiguities he/she has pointed out. We believe these additions have improved our manuscript significantly.

REFERENCES

- [1] Connor J Shiggins, James M Lea, and Stephen Brough. Automated arcticdem ice-berg detection tool: insights into area and volume distributions, and their potential application to satellite imagery and modelling of glacier–iceberg–ocean systems. *The Cryosphere*, 17(1):15–32, 2023.
- [2] Ellyn M Enderlin, Gordon S Hamilton, Fiammetta Straneo, and David A Sutherland. Iceberg meltwater fluxes dominate the freshwater budget in greenland’s ice-berg-congested glacial fjords. *Geophysical Research Letters*, 43(21):11–287, 2016.
- [3] Jason M Amundson and JC Burton. Quasi-static granular flow of ice mélange. *Journal of Geophysical Research: Earth Surface*, 123(9):2243–2257, 2018.

- [4] Lufeng Liu, Zhuoran Li, Yang Jiao, and Shuixiang Li. Maximally dense random packings of cubes and cuboids via a novel inverse packing method. *Soft matter*, 13(4):748–757, 2017.
- [5] ShuiXiang Li, Jian Zhao, Peng Lu, and Yu Xie. Maximum packing densities of basic 3d objects. *Chinese Science Bulletin*, 55(2):114–119, 2010.
- [6] Surui Xie, Timothy H Dixon, David M Holland, Denis Voytenko, and Irena Vaňková. Rapid iceberg calving following removal of tightly packed pro-glacial mélange. *Nature communications*, 10(1):3250, 2019.
- [7] Kishor G Nayar, Mostafa H Sharqawy, Leonardo D Banchik, et al. Thermophysical properties of seawater: A review and new correlations that include pressure dependence. *Desalination*, 390:1–24, 2016.
- [8] Justin C Burton, Jason M Amundson, Ryan Cassotto, Chin-Chang Kuo, and Michael Dennin. Quantifying flow and stress in ice mélange, the world’s largest granular material. *Proceedings of the National Academy of Sciences*, 115(20):5105–5110, 2018.
- [9] A. C. Fowler. Glaciers and ice sheets. In Jesús Ildefonso Díaz, editor, *The Mathematics of Models for Climatology and Environment*, pages 301–336, Berlin, Heidelberg, 1997. Springer Berlin Heidelberg.
- [10] Jeremy N Bassis and Samuel B Kachuck. Beyond the stokes approximation: shallow visco-elastic ice-sheet models. *Journal of Glaciology*, pages 1–12, 2023.
- [11] Jae Hun Kim, Eric Rignot, David Holland, and Denise Holland. Seawater intrusion at the grounding line of jakobshavn isbræ, greenland, from terrestrial radar interferometry. *Geophysical Research Letters*, 51(6):e2023GL106181, 2024.
- [12] Adeline Favier de Coulomb, Mehdi Bouzid, Philippe Claudin, Eric Clément, and Bruno Andreotti. Rheology of granular flows across the transition from soft to rigid particles. *Physical Review Fluids*, 2(10):102301, 2017.
- [13] M Yasinul Karim and Eric I Corwin. Eliminating friction with friction: 2d janssen effect in a friction-driven system. *Physical Review Letters*, 112(18):188001, 2014.
- [14] Douglas R MacAyeal. Large-scale ice flow over a viscous basal sediment: Theory and application to ice stream b, antarctica. *Journal of Geophysical Research: Solid Earth*, 94(B4):4071–4087, 1989.
- [15] Daniel J Sulak, David A Sutherland, Ellyn M Enderlin, Leigh A Stearns, and Gordon S Hamilton. Iceberg properties and distributions in three greenlandic fjords using satellite imagery. *Annals of Glaciology*, 58(74):92–106, 2017.
- [16] Laura M Kehrl, Ian Joughin, David E Shean, Dana Floricioiu, and Lukas Krieger. Seasonal and interannual variabilities in terminus position, glacier velocity, and surface elevation at helheim and kangerlussuaq glaciers from 2008 to 2016. *Journal of Geophysical Research: Earth Surface*, 122(9):1635–1652, 2017.
- [17] Gong Cheng, Mathieu Morlighem, Jérémie Mouginot, and Daniel Cheng. Helheim glacier’s terminus position controls its seasonal and inter-annual ice flow variability. *Geophysical Research Letters*, 49(5):e2021GL097085, 2022.
- [18] Lizz Ultee, Denis Felikson, Brent Minchew, Leigh A Stearns, and Bryan Riel. Helheim glacier ice velocity variability responds to runoff and terminus position change at different timescales. *Nature Communications*, 13(1):6022, 2022.

- [19] Ryan K Cassotto, Justin C Burton, Jason M Amundson, Mark A Fahnestock, and Martin Truffer. Granular decoherence precedes ice mélange failure and glacier calving at jakobshavn isbræ. *Nature Geoscience*, 14(6):417–422, 2021.
- [20] Thomas R Chudley, Ian M Howat, Michalea D King, and Adelaide Negrete. Atlantic water intrusion triggers rapid retreat and regime change at previously stable greenland glacier. *Nature Communications*, 14(1):2151, 2023.
- [21] Suzanne L Bevan, Adrian J Luckman, Douglas I Benn, Tom Cowton, and Joe Todd. Impact of warming shelf waters on ice mélange and terminus retreat at a large se greenland glacier. *The Cryosphere*, 13(9):2303–2315, 2019.
- [22] Ian Joughin, David E Shean, Benjamin E Smith, and Dana Floricioiu. A decade of variability on jakobshavn isbræ: ocean temperatures pace speed through influence on mélange rigidity. *The Cryosphere*, 14(1):211–227, 2020.
- [23] Sierra M Melton, Richard B Alley, Sridhar Anandakrishnan, Byron R Parizek, Michael G Shahin, Leigh A Stearns, Adam L LeWinter, and David C Finnegan. Meltwater drainage and iceberg calving observed in high-spatiotemporal resolution at helheim glacier, greenland. *Journal of Glaciology*, 68(270):812–828, 2022.
- [24] Jason M Amundson, Christian Kienholz, Alexander O Hager, Rebecca H Jackson, Roman J Motyka, Jonathan D Nash, and David A Sutherland. Formation, flow and break-up of ephemeral ice mélange at leconte glacier and bay, alaska. *Journal of Glaciology*, 66(258):577–590, 2020.
- [25] Alexander A Robel. Thinning sea ice weakens buttressing force of iceberg mélange and promotes calving. *Nature Communications*, 8(1):14596, 2017.
- [26] Twila Moon, Ian Joughin, Ben Smith, Michiel R Van Den Broeke, Willem Jan Van De Berg, Brice Noël, and Mika Usher. Distinct patterns of seasonal greenland glacier velocity. *Geophysical research letters*, 41(20):7209–7216, 2014.
- [27] Saurabh Vijay, Shfaqat Abbas Khan, Anders Kusk, Anne M Solgaard, Twila Moon, and Anders Anker Bjørk. Resolving seasonal ice velocity of 45 greenlandic glaciers with very high temporal details. *Geophysical Research Letters*, 46(3):1485–1495, 2019.
- [28] Enze Zhang, Ginny Catania, and Daniel T Trugman. Autoterm: an automated pipeline for glacier terminus extraction using machine learning and a “big data” repository of greenland glacier termini. *The Cryosphere*, 17(8):3485–3503, 2023.
- [29] Ryan Cassotto, Mark Fahnestock, Jason M Amundson, Martin Truffer, and Ian Joughin. Seasonal and interannual variations in ice melange and its impact on terminus stability, jakobshavn isbræ, greenland. *Journal of Glaciology*, 61(225):76–88, 2015.
- [30] Adrien Wehrlé, Martin P Lüthi, and Andreas Vieli. The control of short-term ice mélange weakening episodes on calving activity at major greenland outlet glaciers. *The Cryosphere*, 17(1):309–326, 2023.
- [31] Förste Ch, Sean L Bruinsma, Oleg Abrikosov, Jean-Michel Lemoine, T Schaller, HJ Gtze, J Ebbing, JC Marty, F Flechtner, G Balmimo, et al. Eigen-6c4 the latest combined global gravity field model including goce data up to degree and order 2190 of gfz potsdam and grgs toulouse. *GFZ Data Services*, 10(10.5880), 2014.

- [32] Susan L Howard and Laurie Padman. Gr1kmtm: Greenland 1 kilometer tide model, 2021.
- [33] Howard Butler, Bradley Chambers, Preston Hartzell, and Craig Glennie. PDAL: An open source library for the processing and analysis of point clouds. *Computers and Geosciences*, 148(December 2020):104680, 2021.
- [34] Annelies Voordendag, Brigitta Goger, Christoph Klug, Rainer Prinz, Martin Rutzinger, Tobias Sauter, and Georg Kaser. Uncertainty assessment of a permanent long-range terrestrial laser scanning system for the quantification of snow dynamics on Hintereisferner (Austria). *Frontiers in Earth Science*, 11, mar 2023.
- [35] Christopher Miele and Timothy Bartholomaeus. Greenland ice sheet’s distinct calving styles are identified in terminus change timeseries. *Authorea Preprints*, 2024.
- [36] B. Smith, S. Adusumilli, B. M. Csathó, D. Felikson, H. A. Fricker, A. S. Gardner, N. Holschuh, J. Lee, J. Nilsson, F. Paolo, M. R. Siegfried, T. Sutterley, and the ICESat-2 Science Team. Atlas/icesat-2 l3a land ice height, version 6, 2023.
- [37] B Smith, D Hancock, K Harbeck, L Roberts, T Neumann, Kelly Brunt, H Fricker, A Gardner, M Siegfried, S Adusumilli, et al. Algorithm theoretical basis document (ATBD) for land ice along-track height product (ATL06). *Ice, Cloud, and land elevation satellite-2 (ICESat-2) project*, 2019.
- [38] Claire Porter, Ian Howat, Myoung-Jon Noh, Erik Husby, Samuel Khuvis, Evan Danish, Karen Tomko, Judith Gardiner, Adelaide Negrete, Bidhyananda Yadav, James Klassen, Cole Kelleher, Michael Cloutier, Jesse Bakker, Jeremy Enos, Galen Arnold, Greg Bauer, and Paul Morin. ArcticDEM - Strips, Version 4.1, 2022.
- [39] Claire Porter, Ian Howat, Myoung-Jon Noh, Erik Husby, Samuel Khuvis, Evan Danish, Karen Tomko, Judith Gardiner, Adelaide Negrete, Bidhyananda Yadav, James Klassen, Cole Kelleher, Michael Cloutier, Jesse Bakker, Jeremy Enos, Galen Arnold, Greg Bauer, and Paul Morin. ArcticDEM - Mosaics, Version 4.1, 2023.
- [40] I. Howat, A. Negrete, and B. Smith. Measures greenland ice mapping project (grimp) digital elevation model from geoeye and worldview imagery, version 2, 2022.
- [41] Yue Meng, Ching-Yao Lai, Riley Culberg, Michael G. Shahin, Leigh A. Stearns, Justin C. Burton, and Kavinda Nissanka. Supporting data – seasonal changes of mélange thickness coincide with greenland calving dynamics, 2024.
- [42] I. Joughin. Measures greenland annual ice sheet velocity mosaics from sar and landsat, version 5, 2023.
- [43] M. et al. Morlighem. Icebridge bedmachine greenland, version 5, 2022.
- [44] Itasca Consulting Group, Inc. *PFC — Particle Flow Code, Ver. 7.0*. Minneapolis: Itasca, 2021.
- [45] Youngmin Choi, Mathieu Morlighem, Eric Rignot, and Michael Wood. Ice dynamics will remain a primary driver of greenland ice sheet mass loss over the next century. *Communications Earth & Environment*, 2(1):26, 2021.
- [46] Joe Todd, Poul Christoffersen, Thomas Zwinger, Peter Råback, and Douglas I Benn. Sensitivity of a calving glacier to ice–ocean interactions under climate change: new insights from a 3-d full-stokes model. *The Cryosphere*, 13(6):1681–1694, 2019.

- [47] W Tad Pfeffer, Joel T Harper, and Shad O’Neel. Kinematic constraints on glacier contributions to 21st-century sea-level rise. *Science*, 321(5894):1340–1343, 2008.
- [48] Steve Foga, Leigh A Stearns, and CJ Van der Veen. Application of satellite remote sensing techniques to quantify terminus and ice mélange behavior at helheim glacier, east greenland. *Marine Technology Society Journal*, 48(5):81–91, 2014.
- [49] Twila Moon, Ian Joughin, and Ben Smith. Seasonal to multiyear variability of glacier surface velocity, terminus position, and sea ice/ice mélange in northwest greenland. *Journal of Geophysical Research: Earth Surface*, 120(5):818–833, 2015.
- [50] Ian M Howat, Jason E Box, Yushin Ahn, Adam Herrington, and Ellyn M McFADDEN. Seasonal variability in the dynamics of marine-terminating outlet glaciers in greenland. *Journal of Glaciology*, 56(198):601–613, 2010.
- [51] Jacob I Walter, Jason E Box, Slawek Tulaczyk, Emily E Brodsky, Ian M Howat, Yushin Ahn, and Abel Brown. Oceanic mechanical forcing of a marine-terminating greenland glacier. *Annals of Glaciology*, 53(60):181–192, 2012.
- [52] Jason M Amundson, Mark Fahnestock, Martin Truffer, Jed Brown, Martin P Lüthi, and Roman J Motyka. Ice mélange dynamics and implications for terminus stability, jakobshavn isbræ, greenland. *Journal of Geophysical Research: Earth Surface*, 115(F1), 2010.
- [53] Joe Todd and Poul Christoffersen. Are seasonal calving dynamics forced by buttressing from ice mélange or undercutting by melting? outcomes from full-stokes simulations of store glacier, west greenland. *The Cryosphere*, 8(6):2353–2365, 2014.
- [54] J Krug, G Durand, O Gagliardini, and J Weiss. Modelling the impact of submarine frontal melting and ice mélange on glacier dynamics. *The Cryosphere*, 9(3):989–1003, 2015.
- [55] B Smith, S Adusumilli, BM Csathó, D Felikson, HA Fricker, A Gardner, N Holschuh, J Lee, J Nilsson, FS Paolo, et al. the ICESat-2 Science Team: ATLAS/ICESat-2 L3A Land Ice Height, Version 5, NASA National Snow and Ice Data Center Distributed Active Archive Center, Boulder, Colorado, USA [dataset], 2021.
- [56] Faezeh M Nick, Cornelis J Van der Veen, Andreas Vieli, and Douglas I Benn. A physically based calving model applied to marine outlet glaciers and implications for the glacier dynamics. *Journal of Glaciology*, 56(199):781–794, 2010.
- [57] Andreas Vieli and Faezeh M Nick. Understanding and modelling rapid dynamic changes of tidewater outlet glaciers: issues and implications. *Surveys in geophysics*, 32:437–458, 2011.
- [58] Susan Cook, IC Rutt, T Murray, A Luckman, T Zwinger, N Selmes, A Goldsack, and TD James. Modelling environmental influences on calving at helheim glacier in eastern greenland. *The Cryosphere*, 8(3):827–841, 2014.
- [59] Jamie Barnett, Felicity A Holmes, and Nina Kirchner. Modelled dynamic retreat of kangerlussuaq glacier, east greenland, strongly influenced by the consecutive absence of an ice mélange in kangerlussuaq fjord. *Journal of Glaciology*, 69(275):433–444, 2023.
- [60] Joe Todd, Poul Christoffersen, Thomas Zwinger, Peter Råback, Nolwenn Chauché, Doug Benn, Adrian Luckman, Johnny Ryan, Nick Toberg, Donald Slater, et al. A full-stokes 3-d calving model applied to a large greenlandic glacier. *Journal of Geophysical*

- Research: Earth Surface*, 123(3):410–432, 2018.
- [61] Jérémie Mouginot, Eric Rignot, Anders A Bjørk, Michiel Van den Broeke, Romain Millan, Mathieu Morlighem, Brice Noël, Bernd Scheuchl, and Michael Wood. Forty-six years of greenland ice sheet mass balance from 1972 to 2018. *Proceedings of the national academy of sciences*, 116(19):9239–9244, 2019.

REPLY TO REFEREE COMMENTS - REFEREE 3.

We thank the referee for a constructive review of our paper. The referee states that “This manuscript described the observed mélange thickness changes and glacier calving dynamics based on field and remote sensing data, and developed a three-dimensional discrete element model to simulate the mélange motion... The model is carefully designed, and most of the assumptions seem to be reasonable. I see merit in publication of this work.” He/she states that “I have some comments I hope the authors could answer/consider, mostly minor, please see details below.” He/she makes several constructive suggestions to strengthen and improve the manuscript. We have taken the referee’s comments very seriously, and we have revised the manuscript to address the referee’s comments and suggestions.

In the following, we detail the amendments made to the manuscript (highlighted in red color) in response to the referee’s comments (included in italics).

Comment (1). Major comments

While the data suggest a strong correlation between thick mélange and terminus advance, I did not see direct evidence that thick mélange leads to the terminus advance. Alternatively, could the glacier terminus advance be simply caused by a re-configuration process after major calving events? For instance, calving causes terminus retreat, and the new glacier front likely has fewer crevasses and high degree of cohesion. Consequently, it will take a longer period to detach ice blocks from the glacier. Figure 1 appears to support this explanation: the mélange was pretty thick throughout the TLS observations, however, calving events still occurred frequently; and most of the time the terminus advance rate was not affected by the decrease of mélange thickness.

Response. We thank the reviewer for the constructive advice, which motivates us to better present and interpret the TLS data. To emphasize the strong correlation between mélange thickness and calving dynamics (i.e. thin mélange when terminus retreats with vigorous calving, thick mélange when terminus advances without calving), we have added TLS scans during calving cessation (Fig. 3 in SI and Fig. 1 in this document) and major calving (Fig. 4 in SI and Fig. 2 in this document) periods. We have modified the main text as follows:

1.1 Mélange thickness associated with calving dynamics at Helheim Glacier in 2019-2020. ...We observe **two episodes of calving cessation**, from 8 Oct 2019 - 31 Dec 2019 and 1 Mar 2020 - 20 May 2020, when no calving occurred and the terminus advanced steadily **for 2 km and 1.7 km, respectively**. We found that mélange elevation at the terminus averaged 15 m during these periods. **We include TLS scans during these two episodes of calving cessation in the supplementary material (Fig. 3 in SI)**. We identified four dates where noticeable mélange thinning occurred, which were 3 Sep 2019, 3 Jan 2020

(Fig. 1(d)), 31 May 2020 (Fig. 1(f)), and 14 July 2020. Around these four dates, the mélange elevation at the terminus decreased as follows: by 1.6 m within two days to 10.8 ± 0.05 m, by 5.3 m within three days to 9.4 ± 0.05 m, by 4 m within five days to 10 ± 0.05 m, and by 2.3 m within seven days to 11.6 ± 0.05 m. Major calving happened around these dates, with corresponding retreats at the terminus of 0.5 km, 1.2 km, 1.3 km, and 1.0 km, respectively, accompanied by large increases in fraction of mélange area composed of large icebergs (see SI Fig. 4 & 5 for associated TLS scans and temporal evolution of iceberg fraction).

FIGURE 1. The TLS-measured mélange surface elevation map at Helheim Glacier during two episodes of calving cessation, from (a, b) 8 Oct 2019 – 31 Dec 2019, and (c, d) 1 Mar 2020 – 20 May 2020. White lines indicate positions of the terminus. Dates on the images show the acquisition dates for TLS data. TLS scans are overlain on Sentinel-1 and 2 images acquired around the same dates. Images are in polar stereographic projection (EPSG: 3413).

FIGURE 2. The TLS-measured mélange surface elevation map at Helheim Glacier during four major calving events, from (a)-(c) 3 Sep 2019 – 4 Sep 2019, (d)-(f) 31 Dec 2019 – 3 Jan 2020, (g)-(i) 27 May 2020 – 31 May 2020, and (j)-(l) 14 Jul 2020 – 18 Jul 2020. Dashed white lines indicate positions of the terminus before calving. Solid white lines indicate positions of the terminus after calving. Dates on the images show the acquisition dates for TLS data and times are in UTC. TLS scans are overlain on Sentinel-1 and 2 images acquired around the same dates. Images are in polar stereographic projection (EPSG: 3413).

We agree with the reviewer that there is no direct evidence that thick mélange causes the observed terminus advance at Helheim Glacier, as there are other mechanisms driving calving dynamics. As suggested by the reviewer, the terminus advances right after major calving events can be explained by the possibility that the new glacier front has fewer crevasses and will take a longer period to calve again. To emphasize that TLS scans cannot prove thick mélange causes terminus advances, we have added following sentences in the discussion as follows (lines 419–430):

2 Discussion ...Previous research suggests that the presence of ice mélange can reduce iceberg calving by providing “backstress” to the terminus [1, 2, 3, 4, 5, 6, 7, 8, 9, 10, 11]. Our comparisons of time-varying mélange thickness and calving dynamics at Helheim Glacier (Fig. 4(b)) support the view that the buttressing force increases with the mélange thickness and thus inhibits calving. Our TLS scans indicate a correlation between mélange thickness and calving dynamics, but we cannot determine causality before examining other

mechanisms driving calving dynamics. For instance, the terminus advances right after major calving events can be explained by the possibility that the new glacier front has fewer crevasses and will take a longer period to calve again. Alternatively, the mélange thickness and the terminus react simultaneously but independently to other oceanic and atmospheric forcing. To establish a causal relationship between thin mélange and calving events, we would need in-situ observations with high temporal resolution in minutes to capture the sequence of a calving event and a mélange thinning event [12, 1].

Though we have observed seasonal mélange thickness and calving dynamics from ArcticDEMs (Fig. 8 in the main text), the mélange thickness and the terminus may be reacting simultaneously but independently to other oceanic and atmospheric forcing. For instance, the seasonal calving dynamics can be explained by many factors in summer that can potentially weaken the terminus (more crevasses, submarine melting, melt-undercutting, etc), instead of the reduced buttressing from thin mélange observed from ArcticDEMs. Therefore, we focus on finding a case of summertime terminus advance with unusually thick mélange in front of the terminus, as reported in [1]. Specifically, we should check whether unusually thick mélange is observed in June/July only in the years when terminus advances in summer, and thin mélange is observed in June/July in other years when terminus retreats in summer. After checking the available ArcticDEM data and terminus interannual advance/retreat patterns, we found abundant evidence at Koge Bugt C glacier to support our argument. We found that the terminus advanced from July to October in 2015 and retreated during the same period in 2014, 2016 and 2017 [13]. From ArcticDEM strips acquired in June/July in 2014–2017, we only observe thick mélange in 2015, and thin mélange in other years. We attribute the summertime terminus advance in 2015 to mélange buttressing from the presence of unusually thick mélange observed on 2 Jul with a thickness of 114_{-36}^{+30} m (see Fig. 9 in SI and Fig. 3 in this document), the same as what happened at Jakobshavn Isbræ in Jun 2016 [1]. We have added this piece of evidence in the discussion session as follows:

2 Discussion... Scanning through 108 ArcticDEM strips, we discover calving dynamics associated with mélange thickness seasonality across 32 Greenland glacier termini in 2013-2022. When termini advance in cold months, the average value of all observed mélange thicknesses is 119_{-37}^{+31} m, with a corresponding buttressing force $6.5_{-3.7}^{+3.4} \times 10^6$ N/m. When termini retreat in warm months, the average thickness is 34_{-15}^{+17} m, with a corresponding buttressing force of $5.2_{-3.8}^{+5.9} \times 10^5$ N/m.

While we have observed strong evidence of correlations between mélange thickness and terminus seasonality, understanding their causality requires considerations of other environmental forcings. Previous research shows that seasonal terminus positions for some central west Greenland glaciers with small-magnitude calving events correlate stronger with glacial runoff than mélange presence or ocean thermal forcing [14]. On the other hand, researchers observe

slowdown and thickening of Jakobshavn since 2016 and attribute it to concurrent cooling of ocean waters [15]. Analytical and numerical models imply that submarine melting can amplify calving by melt-undercutting [16, 17]. We note that if submarine melting causes the observed summer thinning of mélange, mélange's buttressing strength can be strongly tied to submarine melting. The impact of submarine melting on mélange strength can be significant due to the strong dependence of buttressing on mélange thickness inferred in our study.

We note that the hypothesis of summer-runoff induced calving, on its own, can not explain the observations at Koge Bugt C Glacier (ID=13 in Fig. 8) where the terminus advanced from July to September in 2015 and retreated or remained a plateau during the same period in 2014, 2016, 2017 and 2020 [13]. We attribute the summertime terminus advance in 2015 to mélange buttressing from the presence of unusually thick mélange observed on 2 Jul with a thickness of 114^{+30}_{-36} m (see Fig. 9 in SI for details), the same as what happened at Jakobshavn Isbræ in Jun 2016 [1]. If calving dynamics are controlled by mélange buttressing, then our analysis infers that the minimum buttressing force required to inhibit calving varies across termini from $1.1^{+0.9}_{-0.7} \times 10^6$ (Hayes Glacier 2) to $9.3^{+4.6}_{-5.2} \times 10^6$ N/m (Kangerlussuaq Glacier). Such variations in the buttressing threshold could be attributed to spatial variations in ice velocities, terminus geometry, bed topography, basal friction, oceanic and atmospheric forcings, etc. Our analysis offers a new framework to mechanistically study the effects of mélange buttressing and other ice-ocean interactions on calving.

FIGURE 3. (a)-(e) The mélange elevation map above mean sea level at Koge Bugt C Glacier from ArcticDEM observations, acquired in June and July from 2014–2020, overlain on the satellite image acquired around the same date. The image is in polar stereographic projection (EPSG: 3413). The date of ArcticDEM acquisition is shown on the upper left corner. (f)-(j) The surface elevation profile for the mélange presented in (a)–(e) displayed as a density plot. The solid blue line is the representative mélange elevation profile as a function of distance from terminus. (k) A time series of terminus position [13], ArcticDEM acquisition dates and observed mélange freeboard heights (Z_0) at the terminus of Koge Bugt C Glacier. The solid red line marks the ArcticDEM acquisition date when an unusually thick mélange is observed at the terminus in July with the terminus advancing from May to October in 2015. The solid black lines mark ArcticDEM acquisition dates around the same time (late June to July) in different years (2014, 2016, 2017, 2020) when thin mélange is observed at the terminus with the terminus retreating or remaining a plateau from July to September (the shaded gray regions).

Comment (2). I was impressed by Figure 7. While I have no doubt about the data analysis, I am curious about how good can selected ArcticDEM tiles represent the whole scenario. For instance, in Figure 7 I did not find black markers representing Helheim glacier retreat when the mélange freeboard was greater than 10 m. However, Figure 1 clearly illustrates some calving events (should correspond to glacier retreat) when the mélange freeboard was greater than 10 m. Did I miss anything?

Response. We thank the reviewer for the careful check of Figure 7 in the main text. The methods section (Section 3.3) of the manuscript describes the process of filtering ArcticDEM strip data as follows: “We identify 341 ArcticDEM strips at 2-meter resolution that cover the mélange regions for the 32 studied termini. For each DEM strip, we investigate terminus position variations [13] during a two-month time window centering on the DEM acquisition date. If the terminus keeps advancing (or retreating) within the time window, then the DEM potentially represents mélange with a strong (or weak) buttressing force. If the terminus alternates between advancing and retreating within the time window, we discard the corresponding DEM strip because the relationship between mélange and calving dynamics is ambiguous in this case. After filtering all DEM strips through this criterion, we identify 60 DEM strips during terminus advances, and 48 DEM strips during terminus retreats, from March to October in 2013–2022.” We used the same filtering criterion for the DEM data from TLS at Helheim Glacier. During the period when TLS data was retrieved (1 Sep 2019 – 1 Sep 2020), we observe two episodes of terminus advances that lasted longer than two months (from 8 Oct 2019 - 31 Dec 2019 and 1 Mar 2020 - 20 May 2020, Fig. 4(b) in this document). Calving occurred sporadically and we do not observe episodes of terminus retreats that lasted longer than two months, and therefore there is no TLS data reported during terminus retreats at Helheim Glacier. We have added Figure 8 in SI (Fig. 5 in this document), which presents an example of the DEM filtering procedure at Helheim Glacier, resulting in two TLS-derived DEMs during terminus advances (Fig. 4(c)(e) in this document), and two ArcticDEM strips during terminus retreats. We have also moved the DEM filtering procedure from the Methods section to the Results section as follows:

1.6 Calving dynamics associated with mélange buttressing force seasonality across 32 Greenland glacier termini in 2013-2022. ...

We then extend our study to 32 glacier termini, most of which (ID 1~25) are picked from previous studies with strong terminus-position seasonality [18, 19], and the rest (ID 26~32) have annual ice discharge larger than 5 Gt/yr [20]. The locations of the termini are marked on a Greenland velocity map in Fig. 8(a). We identify 341 ArcticDEM strips at 2-meter resolution that cover the mélange regions for the 32 studied termini. For each DEM strip, we investigate terminus position variations [13] during a two-month time window centering on the DEM acquisition date. If the terminus keeps advancing (or retreating) within the time window, then the DEM potentially represents mélange with a strong (or weak) buttressing force. If the terminus alternates between advancing and retreating within the time window, we discard the corresponding DEM strip because the relationship between mélange and calving dynamics is ambiguous in this case. After filtering all DEM strips through this criterion, we identify 60 DEM strips during terminus advances, and 50 DEM strips during terminus retreats, from February to November in 2013–2022. Figure 8 in SI presents an example of the DEM filtering procedure at Helheim Glacier, resulting in two TLS-derived DEMs

during terminus advances (Fig. 1(c)(e)), and two ArcticDEM strips during terminus retreats. For every ArcticDEM strip, the mélange thickness is defined as the maximum value obtained from the representative mélange thickness profile within a 200-meter range from the terminus. Table. 2 in SI summarizes the observed minimum (or maximum) mélange thickness when terminus retreats (or advances) as H_0^{\min} (or H_0^{\max}), with the corresponding DEM acquisition month shown in the bracket.

FIGURE 4. Helheim Glacier and ice mélange. (a) TLS-measured elevation map after accounting for local differences between the ellipsoid and geoid, overlain on a Sentinel-1 HV image (both acquired on 30 Nov 2019). The white line across the fjord indicates the glacier front location. The white triangle indicates the TLS location. The upper left inset shows the location of Helheim Glacier in Greenland. The image is in polar stereographic projection (EPSG: 3413). (b) Terminus position relative to 1 Sep 2019, where the positive sign indicates terminus advance. Blue dots denote the averaged mélange elevation within 1 km of the terminus, Z_0 . Calving events are inferred from TLS and satellite images. Due to limited temporal sampling of the data, we are not able to determine the exact time of each calving event. Instead, we mark the time period during which a calving event occurs by a red-shade rectangle. Four vertical black lines mark the dates for the TLS-measured elevation data presented in (c)-(f), which corresponds to 30 Nov 2019, 3 Jan 2020, 30 Mar 2020, and 31 May 2020, respectively. Solid black lines mark the dates with terminus advances, and dashed black lines mark the dates with terminus retreats. (c)-(f) Surface elevation profiles for the mélange displayed as density plots (1510~1859 data points in total); the colour bar denotes the number of data points that have the same elevation and distance from terminus values. For any specific distance from terminus, we find the elevation value that has the maximum number of data points. Solid blue lines connect these elevation values along the distance from terminus as the representative mélange elevation profiles. Mélange thinning on 3 Jan (d) and 31 May (f) coincided with more calving activities and terminus retreats as shown in (b).

FIGURE 5. A time series of terminus position [13] and observed mélange freeboard heights (Z_0) at the terminus of Helheim Glacier. Black dots denote the terminus position where the positive sign indicates terminus advance. Blue dots and squares denote the mélange elevation observed at the terminus, Z_0 , from TLS and ArcticDEM strips, respectively. The solid red (or dashed black) line marks the data acquisition date when the terminus consistently advances (or retreats) within the two-month time window centering on that date, indicating that the DEM potentially represents mélange with a strong (or weak) buttressing force. We exclude DEMs when the terminus behavior alternates between advancing and retreating within the two-month time window, because the relationship between mélange and calving dynamics is ambiguous in this case. We present only the filtered DEM data (i.e., four data points at Helheim Glacier) in Figure 8 in the main text.

To further confirm that we can combine the mélange elevation data across different datasets (ArcticDEM and TLS for Helheim glacier), we have compared the mélange surface elevation acquired on 10 May 2020 from TLS and ArcticDEM at Helheim Glacier (Fig. 13 in SI, Fig. 6 in this document). The TLS data was acquired at 13:01:57 UTC, and the ArcticDEM data was acquired after 40 minutes, at 13:42:45 UTC. Overall, the mélange surface elevation profiles acquired from TLS and ArcticDEM show good agreement. We have added the comparison in the Method section as follows:

3.3 ArcticDEM data and uncertainty assessment. ... In addition, we compare the mélange surface elevation acquired on 10 May 2020 from TLS and ArcticDEM at Helheim Glacier (Fig. 13 in SI). The TLS data was acquired at 13:01:57 UTC, and the ArcticDEM data was acquired after 40 minutes, at 13:42:45 UTC. Overall, the mélange surface elevation profiles acquired from TLS and ArcticDEM show good agreement with a mean absolute difference of 0.66 m at the points of overlap.

FIGURE 6. Comparison of mélange surface elevation from TLS and ArcticDEM at Helheim Glacier. (a), (b) are TLS, ArcticDEM measured elevation map after accounting for local differences between the ellipsoid and geoid, overlain on a Sentinel-2 image (acquired on 10 May 2020). Images are in polar stereographic projection (EPSG: 3413). (c) Surface elevation profiles extracted along the black line in (a)-(b). Black cross markers indicate data acquired from TLS on 10 May 2020, 13:01:57 UTC, and red dot markers indicate data acquired from ArcticDEM on 10 May 2020, 13:42:45 UTC. The horizontal axis shows the distance from point A in (a). (d), (e) are TLS and ArcticDEM measured surface elevation profiles for the mélange displayed as density plots; the color bar denotes the number of data points that have the same elevation and distance from terminus values. For any specific distance from terminus, we find the elevation value that has the maximum number of data points. Solid blue lines connect these elevation values along the distance from terminus as the representative mélange elevation profiles.

Comment (3). About the structure of this manuscript: the modeling work is not directly related to the observations. The observations seem to support the hypothesis that the seasonal variabilities in mélange thickness and calving dynamics are correlated, whereas the modeling work does not seem to aim at testing the hypothesis. Perhaps there are some links between the observations and the model. Some rephrasing may be needed to make the two components more tightly related in a paper.

Response. The reviewer is correct that our models do not directly test the hypothesis that seasonal variabilities in mélange thickness and calving dynamics are correlated because we do not model the calving processes. As pointed out in Section 1.3, “Remote sensing observations reveal a strong correlation between mélange thickness and calving dynamics. As a result, quantifying the buttressing force of mélange in terms of its thickness is the first step to better representing ice-ocean interactions and developing process-based calving models”. Note that for the continuum model for predicting buttressing stresses, we are already using the mélange thickness at the glacial termini, H_0 , from the satellite-derived observations (ArcticDEM) from 32 glaciers. We take H_0 to vary from 35 m, which is the minimum size of icebergs detected within the mélange [21, 22], to 240 m, which is the thickest mélange observed across 32 Greenland termini (See Table 2 in SI). We have revised the continuum model in the main text as follows to make this more clear (lines 208–220):

To characterize the relative magnitude of the horizontal deviatoric stress to the glaciostatic pressure, we substitute representative values for parameters in Eqn. 3 and obtain:

$$\frac{|4H_0(\eta\frac{\partial u}{\partial x})|_{x=0}}{\frac{1}{2}\rho_i(1 - \frac{\rho_i}{\rho_w})g\phi_0H_0^2} \in [3.90 \times 10^{-14}, 1.17 \times 10^{-11}] \times \eta, \quad (1)$$

where the range of mélange thickness, H_0 , is derived from DEM observations. We take H_0 to vary from 35 m, which is the minimum size of icebergs detected within the mélange [21, 22], to 240 m, which is the thickest mélange observed across 32 Greenland termini in 2013–2022 (See Table 2 in SI). We adopt $\frac{\partial u}{\partial x} \in [\frac{2 \text{ m/day}}{15 \text{ km}}, \frac{25 \text{ m/day}}{10 \text{ km}}]$ [8], $\phi_0 \in [0.64, 1]$ [23, 24], $\rho_i \in [870 \text{ kg/m}^3, 920 \text{ kg/m}^3]$ [1], and $\rho_w \in [1020 \text{ kg/m}^3, 1029 \text{ kg/m}^3]$ [25]. As the mélange acts as a weak granular ice shelf [7], its effective viscosity should be much smaller than the glacier ice viscosity, $\eta \ll \eta_i = 10^{12} - 10^{15} \text{ Pa}\cdot\text{s}$ [26, 27]. For mélange with a high viscosity ($\eta > 10^{11} \text{ Pa}\cdot\text{s}$), we need to consider deviatoric stress effects as has been done in [28]. The mélange in the following discrete element model has an estimated viscosity of $2 \times 10^{10} \text{ Pa}\cdot\text{s}$ (see Section 2 in SI for details). Therefore, for mélange with a low viscosity, glaciostatic pressure dominates and the mélange buttressing force can be approximated as:

$$\frac{F}{W} = \frac{1}{2}\rho_i(1 - \frac{\rho_i}{\rho_w})g\phi_0H_0^2. \quad (2)$$

To better match our discrete element models with observations, we have conducted ten more simulations with Helheim and Kangerlussuaq fjord geometries, observed iceberg distributions, and mélange thickness from ArcticDEMs. And we have justified the mélange thickness profile from discrete element models with both ArcticDEM observations and the continuum theory.

We first added a derivation of the continuum theory for mélange thickness profile in SI as follows (lines 97–119):

Finally, we derive the expression for the mélange thickness profile, $H(x)$. Eqn. 14 can be reorganized as follows:

$$\frac{\partial}{\partial y}(H(x)\bar{\sigma}'_{xy}) = -\frac{\partial}{\partial x}(H(x)\bar{\sigma}_{zz}) - \frac{\partial}{\partial x}(2H(x)\bar{\sigma}'_{xx}) - \frac{\partial}{\partial x}(H(x)\bar{\sigma}'_{yy}) \quad (3)$$

Because we assume mélange thickness and stresses do not vary with y , we can integrate Eqn. (3) over the y direction as:

$$(H(x)\bar{\sigma}'_{xy})|_{y=W} - (H(x)\bar{\sigma}'_{xy})|_{y=0} = W\left(-\frac{\partial}{\partial x}(H(x)\bar{\sigma}_{zz}) - \frac{\partial}{\partial x}(2H(x)\bar{\sigma}'_{xx}) - \frac{\partial}{\partial x}(H(x)\bar{\sigma}'_{yy})\right) \quad (4)$$

We use Coulomb friction law to calculate the shear stress at the fjord walls as follows:

$$\begin{aligned} \bar{\sigma}'_{xy}|_{y=W} &= \mu_e \bar{\sigma}_{yy}, \\ \bar{\sigma}'_{xy}|_{y=0} &= -\mu_e \bar{\sigma}_{yy}, \end{aligned} \quad (5)$$

where μ_e is the effective coefficient of friction between the mélange and the fjord wall, which depends on the material friction coefficient and the geometry of the fjord walls, i.e., wall roughness [7, 29, 30]. As $\bar{\sigma}_{yy} = \bar{\sigma}_{zz} + \bar{\sigma}'_{xx} + 2\bar{\sigma}'_{yy}$, Eqn. (4) becomes

$$\frac{2H(x)\mu_e}{W}(\bar{\sigma}_{zz} + \bar{\sigma}'_{xx} + 2\bar{\sigma}'_{yy}) = -\frac{\partial}{\partial x}(H(x)\bar{\sigma}_{zz}) - \frac{\partial}{\partial x}(2H(x)\bar{\sigma}'_{xx}) - \frac{\partial}{\partial x}(H(x)\bar{\sigma}'_{yy}) \quad (6)$$

Following the scaling analysis in the main text (Eqn. 4 in section 1.3), we can reasonably assume that $\bar{\sigma}'_{xx}, \bar{\sigma}'_{yy} \ll \bar{\sigma}_{zz}$, where the depth-averaged vertical stress is $\bar{\sigma}_{zz} = \frac{1}{2}\rho_i g \phi(x)(1 - \frac{\rho_i}{\rho_w})H(x)$. We further assume that the mélange packing density remains a constant along fjords. Therefore, Eqn. (6) becomes

$$\frac{\partial H}{\partial x} + \frac{\mu_e}{W}H(x) = 0 \quad (7)$$

which gives

$$H(x) = Ce^{-\frac{\mu_e}{W}x} \quad (8)$$

where C is a constant that needs to be constrained by a boundary condition of the thickness profile. The mélange thickness exponentially decays with the distance from terminus. However, Eqn. 14 only holds when mélange can be considered as a three-dimensional material. When the mélange thickness decays to a monolayer of icebergs, its thickness is dictated by the iceberg size distribution, instead of stress balances that give rise to Eqn. (8). We identify the mélange length, L , where the mélange thickness decays to a threshold value, H_L , below which the mélange is considered to be a two-dimensional material. Using the boundary condition, $H(x = L) = H_L$, we arrive at the final expression for the mélange thickness profile:

$$H(x) = H_L e^{\frac{\mu_e L}{W}(1 - \frac{x}{L})}, x \in [0, L] \quad (9)$$

By defining the dimensionless distance, $\tilde{x} = \frac{\mu_e x}{W}$, and the dimensionless thickness, $\tilde{H} = \frac{H(x)}{H_L e^{\frac{\mu_e L}{W}}}$, we arrive at the dimensionless form of the mélange thickness profile:

$$\tilde{H}(\tilde{x}) = e^{-\tilde{x}}, \tilde{x} \in [0, \frac{\mu_e L}{W}]. \quad (10)$$

We then summarized the continuum theory for mélange thickness profile in the main text as follows (lines 221–249):

The “granular ice shelves” depth-averaged horizontal momentum equations for mélange (see SI for derivation and validation against discrete element simulations) resemble that of ice shelves:

$$\begin{aligned} \frac{\partial}{\partial x}(2H(x, y)\bar{\sigma}'_{xx}) + \frac{\partial}{\partial x}(H(x, y)\bar{\sigma}'_{yy}) + \frac{\partial}{\partial y}(H(x, y)\bar{\sigma}'_{xy}) &= \rho_i g(1 - \frac{\rho_i}{\rho_w})H(x, y) \frac{\partial(\phi(x, y)H(x, y))}{\partial x}, \\ \frac{\partial}{\partial x}(H(x, y)\bar{\sigma}'_{xy}) + \frac{\partial}{\partial y}(H(x, y)\bar{\sigma}'_{xx}) + \frac{\partial}{\partial y}(2H(x, y)\bar{\sigma}'_{yy}) &= \rho_i g(1 - \frac{\rho_i}{\rho_w})H(x, y) \frac{\partial(\phi(x, y)H(x, y))}{\partial y}, \end{aligned} \quad (11)$$

where the depth-averaged stress $\bar{\sigma}_{ij} = \frac{1}{H} \int_{z_b}^{z_s} \sigma_{ij} dz$. When the mélange packing density approaches $\phi = 1$, Eqn. (11) converges to the shallow shelf approximation (SSA) [31]. The mélange momentum balance along the fjord direction reveals three competing forces: compressional/extensional flow from velocity gradients within the mélange (negligible if mélange viscosity is smaller than 10^{11} Pa-s), glaciostatic stress from mélange thickness, and shear stresses on fjords. Therefore, the full thickness profile of the mélange depends on fjord/mélange friction/cohesion properties, velocity gradients and viscosity of the mélange, and the mélange width/length. Using the horizontal momentum balance equations (Eqn. 11) and non-dimensionalization (see SI for details), we further derive the mélange thickness profile, $H(x)$:

$$\tilde{H}(\tilde{x}) = e^{-\tilde{x}}, \tilde{x} \in [0, \frac{\mu_e L}{W}], \quad (12)$$

where μ_e is the effective coefficient of friction between the mélange and the fjord wall, which depends on the material friction coefficient and the geometry of the fjord walls, i.e., wall roughness [7, 29, 30]. Here, we define the mélange length, L , as the distance beyond which the mélange thickness decays to a threshold value, H_L , where mélange becomes a monolayer of icebergs with thickness dominated by particle size distribution instead of stress balances that give rise to Eqn. (12). We define the dimensionless distance, $\tilde{x} = \frac{\mu_e x}{W}$, and the dimensionless thickness, $\tilde{H} = \frac{H(x)}{H_L e^{\frac{\mu_e L}{W}}}$. To quantify the mélange buttressing force per unit width, previous two-dimensional model assuming mélange of uniform thickness required assumptions on fjord/mélange friction properties and the mélange width/length [7]; in our three-dimensional model the mélange

thickness and packing density at the terminus are the only parameters needed. As the length and thickness are coupled by stress balances within the granular material, the mélange thickness build-up at the terminus already encodes the aforementioned material and geometric properties. For instance, thicker mélange can be built up at the terminus with longer fjords, larger fjord friction, or increased mélange rigidity in winter.

In summary, momentum balance equations reveal that the mélange buttressing force per unit width is solely controlled by the packing density ϕ_0 and mélange thickness H_0 at the glacier terminus, and is proportional to the square of mélange thickness (Eqn. (2)). Additionally, mélange thickness exponentially decays with its distance from terminus (Eqn. (12)).

For the new simulations with Helheim and Kangerlussuaq fjord geometries, we have made a new figure to justify the mélange thickness profile from discrete element models with both ArcticDEM observations and the continuum theory (Fig. 7 in this document and Fig. 4 in the main text). We have also added a supplementary video for modeling mélange in Helheim and Kangerlussuaq fjord geometries. To reflect these changes, we have modified the main text as follows (lines 250–289):

1.4 A three-dimensional discrete element model of ice mélange.

To validate continuum predictions on the mélange buttressing force and thickness profile (Eqn. (2), (12)), we develop a three-dimensional discrete element model on the mélange with a steadily advancing terminus. We adopt fjord geometries from Helheim and Kangerlussuaq glaciers (black dots in Fig. 7(a)(d)), which are scaled down by four times for the discrete element models to save computational costs (Fig. 7(b)(e)). Note that such downscaling does not affect the shape of the dimensionless mélange thickness profile, which will be compared with Eqn. (12). Icebergs are modelled as cubic particles with sizes varying from 36 m to 200 m and have a power-law size distribution with an exponent of -4 [22, 21, 32]. In the models, we limit the maximum size of icebergs to be around one fifth of the fjord width, as commonly observed in both fjords. We initialize the ice mélange thickness with a profile that linearly decays within 1.2 km from the terminus and with the right end open to the ocean. We push the left end of the mélange with an advancing terminus at 30 meters per day [33, 34, 35] and record the temporal evolution of the buttressing force exerted on the terminus. We present modeling results after 16 days of terminus motion, when the mélange motion has approximately reached a steady state. In a series of simulations, we vary the initial mélange thickness to determine its influence on the steady state buttressing force. The steady state mélange thickness at the terminus varies from 40~135 m at Helheim fjord, and 60~240 m at Kangerlussuaq fjord, that are consistent with ArcticDEM observations (Table 2 in SI). We calculate the two-day averaged velocity of each iceberg element by dividing the iceberg's displacement between 14 and 16 days of

terminus motion by the time interval (two days). The mélange near the terminus moves at the terminus velocity with shear bands developed at fjord walls (Fig. 7(c)(f)). The mélange near the open end becomes loosely-packed and more fluidic (See supplementary video for the full temporal evolution of the mélange behaviors.) The modeled velocity field showcases both uniform (Fig. 7(c)) and extensional (Fig. 7(f)) flow regimes that are consistent with remote observations [8]. See Fig. 6 in SI for the modeled velocity fields, including instantaneous velocity, velocity averaged over one hour, one day, and two days, and the satellite-derived velocity fields of the mélange [8].

The modeled mélange thickness profiles generally align with observations from ArcticDEMs on 17 Aug 2014 at Helheim (Fig. 7(a)(g)) and on 12 Apr 2018 at Kangerlussuaq (Fig. 7(d)(h)). In addition, we conduct another simulation with more icebergs confined in Kangerlussuaq fjord, to produce a thicker steady state mélange thickness profile as observed on 5 May 2020. We set the thickness threshold (H_L) as 36 ± 10 m for mélange to be considered as three-dimensional, and retrieve the corresponding mélange length (L) that varies from 7 to 12 km. The non-dimensional mélange thickness profile from three ArcticDEM strips and the discrete element models collapse onto the exponential analytical solution in Eqn. (12), with μ_e fitted to be 0.3~0.4 for Helheim fjord and 0.4~0.8 for Kangerlussuaq fjord (Fig. 7(i), see table 1 in SI for details). The Kangerlussuaq fjord has a larger μ_e due to its more rugged fjord geometries.

To further explore the effect of fjord frictional properties on the mélange buttressing force, we adopt two simplified channel configurations that conceptualize complex Greenland fjord geometries. ...

FIGURE 7. The three-dimensional discrete element model for mélange composed of cubic icebergs with a power-law size distribution and confined within real fjord geometries. (a)-(c) For Helheim glacier, (d)-(f) for Kangerlussuaq glacier. The real fjord geometry is scaled down by four times for the discrete element model. (a) The mélange elevation above mean sea level from ArcticDEM strip, overlain on satellite images acquired around the same date. The ArcticDEM acquisition date is shown at the top right corner. White line across the fjord indicates glacier front location. The images are in polar stereographic projection (EPSG: 3413). Black dots along fjords are adopted in the model to construct boundary walls that resemble fjord geometries. (b), (c) are side and top view for iceberg positions and velocities after 16 days into simulations with steady terminus advance and no calving. The glacier terminus moves at a constant velocity, $V_{\text{ter}} = 30$ m/day. We calculate the two-day averaged velocity of each iceberg element by dividing the iceberg's displacement between 14 and 16 days of terminus motion by the time interval (two days), which is indicated by filled colour in (c). (d)-(f) follow captions of (a)-(c). (g), (h) The comparison of the mélange thickness profile between ArcticDEM (black lines) and discrete element model (red lines) for Helheim and Kangerlussuaq glacier, respectively. (i) The mélange thickness profile from three ArcticDEM strips and corresponding discrete element models collapse onto the exponential analytical solution (Eqn. (12), red dashed line). See supplementary videos for the full temporal evolution of the mélange behaviors.

We have added a table for the fitted fjord effective coefficient of friction (μ_e) in SI as follows:

TABLE 1. The fitted fjord effective coefficient of friction (μ_e) from mélange thickness profile in Fig. 4 in the main text. The uncertainties come from the mélange thickness threshold, $H_L = 36 \pm 10$ m.

Glacier name	ArcticDEM date	μ_e (ArcticDEM)	μ_e (Discrete element model)
Kangerlussuaq	2020/05/05	$0.64^{+0.09}_{-0.06}$	$0.77^{+0.11}_{-0.06}$
Kangerlussuaq	2018/04/12	$0.39^{+0.05}_{-0.04}$	$0.48^{+0.07}_{-0.04}$
Helheim	2014/08/17	$0.25^{+0.03}_{-0.03}$	$0.39^{+0.05}_{-0.05}$

We have modified Fig. 6(b) in the main text (Fig. 8(b) in this document) to incorporate new simulations with Helheim and Kangerlussuaq fjords. We found that with real fjord geometries, iceberg size distribution, and mélange thickness profile, the buttressing forces from discrete element models still show good agreement with the continuum predictions.

FIGURE 8. Comparison between discrete element model and continuum predictions of the mélange buttressing force. (a) The temporal evolution of F/W during the terminus motion for straight (dashed lines) and rugged (solid lines) fjord walls. The red, blue, cyan and black colours correspond to mélange with initial thicknesses, $H_{ini} = 378$ m, 281 m, 178 m, 84 m, respectively. Simulations reach the steady state after 5 days, except for the thinnest mélange ($H_{ini} = 84$ m). (b) Steady state buttressing force, F/W , as a function of steady-state mélange thickness at the terminus, H_0 . Different markers represent simulations with different fjord geometries. Square markers represent Helheim fjord, hexagram markers represent Kangerlussuaq fjord, circular markers represent straight fjords, and triangular markers represent rugged fjords. The smaller markers indicate simulations with smaller icebergs (half of the original size). F/W is obtained by averaging the total buttressing force on the terminus over the terminus width during simulation time 5 ~ 15 days. The marker shows the averaged steady-state value of F/W , with a vertical error bar showing its fluctuation. H_0 is obtained by averaging the mélange thickness within 200 m of the terminus and over the terminus width. The marker shows the averaged steady-state value of H_0 , with a horizontal error bar showing its variation over the terminus width brought by the iceberg size polydispersity. For simulations where the mélange collapse into monolayers at the end, we plot both the peak and minimal F/W values and connect them by gray lines. The minimal F/W values for monolayered, two-dimensional mélange are shown by pentagram markers. All markers are coloured by the mélange packing density at the terminus at steady state, ranging from 0.4 to 0.9. The dashed lines represent Eq. (2) with the mélange packing density at the terminus, $\phi_0 = 0.5$ and 1.0, respectively.

To highlight limitations of current mélange thickness profile analysis, we have added following discussions in the main text as follows (lines 405–418):

2 Discussion Our discrete element model of ice mélange is the first to be composed of realistic cubic icebergs instead of spheres, and showcases an exponential decay of the thickness profile in Helheim and Kangerlussuaq fjords, that is consistent with ArcticDEM observations and analytical predictions. Apart from the exponential decay, we also observe other shapes of the mélange thickness profile, such as a plateau near the terminus, a steep drop or a bulge at few kilometers away from the terminus, etc (See SI for ArcticDEM strips at 32 glacier termini). These shapes could be attributed to calving-induced jamming wave propagation, ice-ocean interactions, iceberg size distribution, heterogeneous friction or cohesion within the mélange or at fjord walls, all of which cannot be captured by the simplified rheology and friction law underneath Eqn. (12). Coupled with computational fluid mechanics, our discrete element model can be used to further explore how the mélange thickness at the terminus evolves with ice-ocean interactions that influence calving dynamics, including ocean tides [12], ocean warming [36, 37, 38], and subglacial plumes [39, 40].

Lastly, we have added details of the new simulations into the Methods section as follows (lines 618–655):

3.4 The three-dimensional discrete element model for ice mélange. ...

We use cubic grains which can achieve a higher packing density, thus butressing forces, than disk-shaped grains. We adopt the iceberg *area* distribution observed in the mélange of Jakobshavn Isbræ and Helheim Glacier, which is approximated as a power-law distribution with an exponent of -2.0 [22, 21]. The resulting iceberg size distribution for cubic grains is a power-law distribution with an exponent of -4.0. Taking the simulation with the Helheim fjord for instance (Fig. 7(b)), the side lengths of cubic icebergs are 36 m, 50 m, 75 m, 100 m, 200 m, and the corresponding numbers of particles are 3120, 838, 166, 52, 3, respectively. To model mélange with different steady state thicknesses, the total number of particles varies from 2918 to 12522 for simulations with Helheim and Kangerlussuaq fjords. In simplified fjord geometries, the side lengths of cubic icebergs are 35.4 m, 50 m, 70.7 m, 100 m, 141.4 m, and the corresponding numbers of particles for the thick mélange are 8190, 2045, 510, 125, 30, respectively (Fig. 10(f)). We vary the total number of particles from 1634 to 15264 to change the mélange thickness. To confirm that the modeling results are invariant to the particle size, we conduct six more simulations with smaller icebergs, whose sizes are half of original sizes and range from 17.7 m to 70.7 m (small markers in Fig. 8(b)).

To construct the initial mélange state, we divide the total number of particles into three equal batches. In each batch, iceberg sizes are randomly drawn from the distribution described above. We put a right boundary wall at distance L from the terminus on the left to prescribe the initial length of the mélange.

The mélange is confined in y direction by two side walls representing fjords at a distance W . To explore influence of fjord friction properties on mélange behaviors, we have Helheim, Kangerlussuaq, and simplified (straight and rugged) channel configurations. All configurations have the same kinetic friction coefficient, μ , and rugged channels have cuboid bulges of dimension $a_r \times a_r \times h_r$ that are uniformly spaced at d_r in x and z directions. We deposit icebergs in each batch from the same height and then they settle under gravity and buoyancy. Following pouring, the entire array of cubic particles is permitted to settle until static equilibrium is achieved, as shown in Fig. 10(a)(f). We then delete the right boundary wall so that the mélange has an open end in the ocean. We move the terminus on the left at a constant velocity, $V_{\text{ter}} = 30$ m/day for real fjord geometries [33, 34, 35], and 43.2 m/day for simplified fjord geometries [3, 7]. To confirm that the averaged steady-state buttressing force is invariant to the terminus velocity, we conducted simulations with mélange thickness 280 m, terminus velocity at 21.6 m/day, 43.2 m/day, and 86.4 m/day, for both straight fjords and rugged fjords configurations (Fig. 16 in SI). The results show that the averaged buttressing force is mostly invariant to the terminus velocity in both fjord configurations. Taking the force fluctuations into account, the maximum buttressing force difference among the chosen velocities is 4% and 8% for straight and rugged fjords, respectively. In rugged fjords, faster terminus motion leads to larger force fluctuations due to larger velocity gradient during stick-slip/jam-unjam cycles. We adopt the terminus velocity of 43.2 m/day to simulate mélange in simplified fjord geometries for the sake of computational efficiency.

Comment (4). For the model, it is reasonable to make many simplifications. I was wondering if the authors were able to test the implications of some of the major simplifications. For example, most fjords do not have regular geometry; remote sensing and field observations indicate that it is not uncommon for the mélange to have highly localized strain rates across the width of the fjord.

Response. We have conducted ten more simulations with Helheim and Kangerlussuaq fjord geometries. See response to Comment 3 for details. The reviewer points out the modeled mélange velocity fields (plug flow in straight channels and compressive flow in rugged channels) is inconsistent with the remote observations (highly localized strain rates across the width of the fjord) [8], which motivates us to check our calculation of the mélange velocity. Previously, we presented the iceberg velocity as the instantaneous velocity directly output from the discrete element model, which is the particle's velocity at each mechanical timestep (0.1 s). However, the observed mélange velocity field is retrieved by applying particle image velocimetry (PIV) to satellite imagery, with the time interval between image pairs ranging from 2 to 30 days [8]. Therefore, we analyze modeled velocity fields, including instantaneous velocity, velocity averaged over one hour, one day, and two days, and compare results with the satellite-derived velocity fields of the mélange [8] (Fig. 6 in SI and Fig. 9 in this document). The comparison shows that the two-day averaged velocity field, which is

calculated by dividing the iceberg's displacement over two days of terminus motion by the time interval, showcases both uniform (Fig. 9(e)) and extensional (Fig. 9(k)) flow regimes that are consistent with remote observations [8]. We have presented the two-day averaged velocity for the simulations with Helheim and Kangerlussuaq fjords (Fig. 4 in the main text and Fig. 7 in this document). We have modified the main text as follows (lines 268–276):

1.4 A three-dimensional discrete element model of ice mélange.... We calculate the two-day averaged velocity of each iceberg element by dividing the iceberg's displacement between 14 and 16 days of terminus motion by the time interval (two days). The mélange near the terminus moves at the terminus velocity with shear bands developed at fjord walls (Fig. 7(c)(f)). The mélange near the open end becomes loosely-packed and more fluidic (See supplementary video for the full temporal evolution of the mélange behaviors.) The modeled velocity field showcases both uniform (Fig. 7(c)) and extensional (Fig. 7(f)) flow regimes that are consistent with remote observations [8]. See Fig. 6 in SI for the modeled velocity fields, including instantaneous velocity, velocity averaged over one hour, one day, and two days, and the satellite-derived velocity fields of the mélange [8].

FIGURE 9. The three-dimensional discrete element model for mélange composed of cubic icebergs with a power-law size distribution and confined within real fjord geometries. (a)-(e) For Helheim glacier, (g)-(k) for Kangerlussuaq glacier. The real fjord geometry is scaled down by four times for the discrete element model. (a) The side view for iceberg positions after 16 days into simulations with steady terminus advance and no calving. The glacier terminus moves at a constant velocity, $V_{\text{ter}} = 30$ m/day. (b) The top view of the instantaneous velocity of iceberg element indicated by filled colour. We also calculate the time averaged velocity of each iceberg element by dividing the iceberg’s displacement over one hour, one day, and two days of terminus motion by the corresponding time intervals. We indicate the 1-hour averaged (c), 1-day averaged (d), and 2-day averaged (e) velocity field by filled colour. (g)-(k) follow captions of (a)-(e). See supplementary videos for the full temporal evolution of the mélange behaviors. Remote observations of uniform (f) and extensional flow (l) regimes of ice mélange adapted with permission from [8]. Velocities were calculated over 3–12 February 2014 at Helheim Glacier (f), and 7–9 August 2014 at Kangerlussuaq Glacier (l). Gray regions indicate where image cross-correlation results were discarded due to low signal-to-noise ratios.

We have also replaced instantaneous velocity fields for straight and rugged fjords with two-day averaged velocity fields (Fig. 5 in the main text and Fig. 10 in this document), and modified the main text as follows (lines 301–305):

In the straight channel configuration, the mélange behaves like plug flow with a uniform velocity profile within the fjord (Fig. 10(d)(i)). **In the rugged channel configuration, the mélange exhibits shear bands near fjord boundaries with uniform flow near the terminus** (Fig. 10(e)(j)), which has also been reported in previous studies [7, 8].

FIGURE 10. The three-dimensional discrete element model for mélange composed of cubic icebergs with a power-law size distribution and confined within simplified fjord geometries. (a)-(e) For a thin mélange, (f)-(j) for a thick mélange. (a) A side view of the initial condition for the simulation with $W = 1$ km, $L = 3$ km, and $H_{\text{ini}} = 60$ m. The glacier terminus is shown as a grey block on the left. The ocean floor and fjord walls are plotted in brown. The glacier terminus starts to move at a constant velocity, $V_{\text{ter}} = 43.2$ m/day. (b)-(e) are snapshots for iceberg positions and velocities after 16 days into simulations with steady terminus advance and no calving. (b), (d) are the side and top view for a straight fjord wall configuration; (c), (e) are the side and top view for a rugged fjord wall configuration. **The two-day averaged velocity** of each iceberg element is indicated by filled colour in (d) and (e). (f) A side view of the initial condition for the simulation with $W = 1$ km, $L = 3$ km, and $H_{\text{ini}} = 378$ m. (g)-(j) follow captions of (b)-(e). See supplementary videos for the full temporal evolution of the mélange behaviors.

Comment (5). For the buttressing force, what was estimated in this work was the force per unit lateral-width. It would be good to note that when it was used, or clarify it when first referred.

Response. The reviewer suggests us to highlight that the reported mélange buttressing force is averaged over the fjord width in the unit of N/m, which differs from the mélange backstress in the unit of Pa. Previous research found that the mélange applies a supporting

pressure equivalent to a backstress of 30~60 kPa acting on the entire face of the Store Glacier terminus [11]. To calculate the mélange buttressing force over the fjord width and convert it to the mélange backstress, we need to know both the mélange thickness and the terminus thickness [10], the latter of which is absent in our simulations. In fact, it is more convenient to compare to other studies with the current unit (N/m), including field observations [28], simulations [3, 7], and analytical analysis [5]. We have made clarifications in the main text as follows:

1.3 A three-dimensional continuum model of ice mélange. ... With some algebraic steps we derive the mélange buttressing force (F) per unit width (W) on the terminus as follows (see SI for derivation details):

... In summary, momentum balance equations reveal that the mélange buttressing force per unit width is solely controlled by the packing density ϕ_0 and mélange thickness H_0 at the glacier terminus, and is proportional to the square of mélange thickness (Eqn. (2)). Additionally, mélange thickness exponentially decays with its distance from terminus (Eqn. (12)).

We have also emphasized the unit of the buttressing force after the analysis of ArcticDEM data (lines 403–404):

1.6 Calving dynamics associated with mélange buttressing force seasonality across 32 Greenland glacier termini in 2013-2022. ... Here we report the buttressing force in N/m, which can be compare to other studies including field observations [28], simulations [3, 7], and analytical analysis [5].

Comment (6). Other Comments

Ln 29: Does the variation of 400 mm equivalent sea level rise correspond to iceberg calving in GrIS alone, or in Antarctic & GrIS?

Response. We have modified the sentence as follows (lines 28–30):

Projections of sea level rise by 2100 can vary by 374 mm depending on the rate of iceberg calving at Greenland ice sheet margins [41].

Comment (7). Ln 43-45: Did the authors imply that a complete loss of mélange buttressing would lead to more frequent or greater volume of calving in both summer and winter? What's the difference between winter and summer terminus changes in such a scenario?

Response. The sentence was paraphrased from the cited paper [42]. Figure 4(c) in [42] shows that a complete loss of mélange buttressing results in a plateau in the spring terminus position that would have advanced for 500 m with a mélange thickness of 140 m. We have modified the sentence to avoid ambiguities as follows (lines 46–48):

In a warming climate, a complete loss of mélange buttressing may prevent terminus advances in winter and spring, which has a significant effect on the seasonal range in the terminus position [42].

Comment (8). Ln 60-62: I thought mélange width is also a needed parameter if “per-unit width” is not specified. And how about the packing ratio?

Response. Correct. We have modified the sentence as follows (lines 68–74):

Motivated by these observations, we develop a continuum mélange model, validated with a three-dimensional discrete element model, which confirms that mélange thickness and packing density at the terminus are the only observations required to estimate mélange buttressing force per unit width. We then apply this continuum model to estimate mélange buttressing forces around Greenland from ArcticDEM observations, and show that the inferred buttressing force is highly consistent with observed patterns of terminus advance and retreat.

Comment (9). Ln 71: I don’t think the geoid undulation is associated with tides. They are two independent variables. Please clarify (here and in the Methods).

Response. Yes, that is correct. We have re-written the sentence as follows:

1.1 Mélange thickness associated with calving dynamics at Helheim Glacier in 2019-2020. ...TLS point clouds are ellipsoidal heights. After removing the geoid height at Helheim Glacier [43], the mélange surface elevations are heights above mean sea level with tidal variations [1]. We then remove the tidal trend of mélange elevations with a tidal model [44].

Comment (10). Ln 98: replace “may be reacting” with “react”.

Response. We have moved the sentence to the discussion section and revised it as follows (lines 420–430):

Our comparisons of time-varying mélange thickness and calving dynamics at Helheim Glacier (Fig. 4(b)) support the view that the buttressing force increases with the mélange thickness and thus inhibits calving. Our TLS scans indicate a correlation between mélange thickness and calving dynamics, but we cannot determine causality before examining other mechanisms driving calving dynamics. For instance, the terminus advances right after major calving events can be explained by the possibility that the new glacier front has fewer crevasses and will take a longer period to calve again. Alternatively, the mélange thickness and the terminus react simultaneously but independently to other oceanic and atmospheric forcing. To establish a causal relationship between thin mélange and calving events, we would need in-situ observations with high temporal resolution in minutes to capture the sequence of a calving event and a mélange thinning event [12, 1].

Comment (11). Ln 100: delete “completely”?

Response. We replace “To derive a completely unambiguous explanation” with “To establish a causal relationship between thin mélange and calving events”. See response to Comment 10 for details.

Comment (12). *Ln 113: perhaps only selected tracks were used? ICESat-2 has more tracks passing over glacier termini.*

Response. Yes, there are many ICESat-2 tracks that pass over glacier termini. However, very few of them pass through the front of the same terminus in different seasons, because the terminus position varies seasonally but ICESat-2 tracks are generally fixed in space. We have moved the description from Methods section to the Results section as follows (lines 132–136):

1.2 Seasonal changes of mélange thickness and calving dynamics. ...

There are very few ICESat-2 tracks passing through the fronts of termini in different seasons, because positions of termini vary seasonally but ICESat-2 tracks are generally fixed in space. We identify ICESat-2 tracks passing over glacier termini in different seasons for Jakobshavn Isbræ (Fig. 2(a)), Kangerlussuaq Glacier (Fig. 2(b)), and Store Glacier (Fig. 2(c)), **which are large discharging glaciers contributing to Greenland's mass losses [20].**

Comment (13). *Ln 126-127: thick mélange may influence calving, however, the data described by the authors could not exclude other possible factors. For instance, could temperature, sea ice, or other factors play some roles?*

Response. The reviewer is correct that the data only shows correlation between mélange thickness and calving dynamics, but we cannot conclude causality from it. We have modified the sentence as follows (lines 153–154):

In summary, ICESat-2 data provides strong evidence of correlations between mélange thickness and terminus seasonality.

We have also discussed mechanisms other than mélange buttressing that lead to seasonal calving dynamics in the Discussion section as follows (lines 436–446):

While we have observed strong evidence of correlations between mélange thickness and terminus seasonality, understanding their causality requires considerations of other environmental forcings. Previous research shows that seasonal terminus positions for some central west Greenland glaciers with small-magnitude calving events correlate stronger with glacial runoff than mélange presence or ocean thermal forcing [14]. On the other hand, researchers observe slowdown and thickening of Jakobshavn since 2016 and attribute it to concurrent cooling of ocean waters [15]. Analytical and numerical models imply that submarine melting can amplify calving by melt-undercutting [16, 17]. We note that if submarine melting causes the observed summer thinning of mélange, mélange's buttressing strength can be strongly tied to submarine melting. The impact of submarine melting on mélange strength can be significant due to the strong dependence of buttressing on mélange thickness inferred in our study.

Comment (14). *Fig 1: for the elevation profile, instead of using the value that has the maximum number of data points, it is perhaps more robust to use the median value, particularly for high precision data.*

Response. Due to the long-tail distribution of the number of data points against the mélange elevation at any specific distance from terminus, the proposed median or mean elevation value approach tends to incorporate large icebergs into the elevation profile, failing to represent the elevation piled up from small icebergs and sea ice. We have added a figure to compare the elevation profile from connecting the mean and median elevation values, and the elevation values with maximum numbers of data points, from both TLS and ArcticDEM data at Helheim Glacier (Fig. 2 in SI and Fig. 11 in this document). We have also added the following sentences in the main text to justify our method of constructing the mélange elevation profile:

1.1 Mélange thickness associated with calving dynamics at Helheim Glacier in 2019-2020. ... To display the spatial profile of the mélange elevation, we calculate distances from terminus for all data points in the ice mélange and plot them as density maps in Fig. 1(c)-(f). For any specific distance from the terminus, there is a spread of mélange elevation that exhibits a long-tail distribution due to the presence of large icebergs. To reflect the mélange elevation that is piled up from small icebergs and sea ice, we connect elevation values that have maximum numbers of data points along the distance from terminus as the representative mélange elevation profiles (solid blue lines in Fig. 1(c)-(f), see Methods and Fig. 2 in SI for a sensitivity study of this metric). In doing so, we prevent large icebergs from disrupting the mélange elevation profiles without cutting off the data by an arbitrary elevation threshold. The resulting representative mélange elevation profiles are always below 30 m, indicating that it is a reasonable elevation threshold to exclude large icebergs from consideration. To estimate mélange elevation near the glacier terminus (Z_0), we take an average of all data points below the 30 m threshold within 1 km (i.e., 1/5 of the fjord width) of the terminus. We infer thickness of the mélange based on TLS-derived surface elevations and assuming hydrostatic equilibrium.

We have added a paragraph in the Methods section for justification:

3.1 Terrestrial laser scanner data and uncertainty assessment. ... To construct the representative mélange elevation profile, we first connect the median or mean elevation values along the distance from the terminus, the shape of which turns out to be sensitive to large icebergs with elevation values larger than 30 m (Fig. 2 in SI). Incorporating large icebergs into the representative mélange elevation profile has several drawbacks. Firstly, instead of representing the elevation piled up from small icebergs and sea ice, it tends to reflect the size distribution of large icebergs. Secondly, the existence and location of large icebergs are sporadic and hard to characterize by a continuum model. Lastly, comparing mélange elevation profiles across different Greenland fjords becomes challenging, as the size and shape distribution of large icebergs are sensitive to calving styles [45] and terminus geometries. To reflect the mélange elevation that is piled up from small icebergs and sea ice, we connect elevation values

that have maximum numbers of data points along the distance from terminus as the representative mélange elevation profiles (solid blue lines in Fig. 1(c)-(f) in the main text and Fig. 2(b)(e) in SI).

FIGURE 11. Helheim Glacier and ice mélange. (a) TLS-measured elevation map after accounting for local differences between the ellipsoid and geoid, overlain on a Sentinel-1 HV image (both acquired on 30 Nov 2019). The white line across the fjord indicates the glacier front location. The white triangle indicates the TLS location. The upper left inset shows the location of Helheim Glacier in Greenland. The image is in polar stereographic projection (EPSG: 3413). (b) Surface elevation profile for the mélange displayed as a density plot; the colour bar denotes the number of data points that have the same elevation and distance from terminus values. For any specific distance from terminus, we find the elevation value that has the maximum number of data points. The solid blue line connects these elevation values along the distance from terminus as the representative mélange elevation profile. We also calculate the median and mean elevation values for each specific distance from terminus, and connect them by solid black and cyan lines, respectively. (c) The number of data points against the surface elevation value at distance from terminus of 0.5 km (black line), 1 km (blue line), and 2 km (red line). Due to the long-tail distribution of the mélange elevation, the median or mean elevation value approach in (b) tends to incorporate large icebergs into the elevation profile, failing to represent the elevation piled up from small icebergs and sea ice. (d) Arctic-DEM measured elevation map after accounting for local differences between the ellipsoid and geoid, overlain on a Landsat 8 image (acquired on 30 Mar 2014). The image is in polar stereographic projection (EPSG: 3413). (e) and (f) follow captions of (b) and (c).

Comment (15). Ln 146: delete “the” after “fjord and”

Response. It has been deleted as follows:

We make the following assumptions: ... (iii) the mélange packing density, thickness, viscosity, and strain rates are uniform across the width of the fjord and across the depth of the mélange, but vary with the distance from terminus;

Comment (16). Ln 324-325: are these averaged thicknesses calculated immediately at the glacier front? Or calculated within a certain distance to the glacier front? Better to clarify that.

Response. We have already included the definition of mélange thickness for ArcticDEM strips in the Methods section. For clarification, we have reiterated the definition in the Results section as follows (lines 368–386):

1.6 Calving dynamics associated with mélange buttressing force seasonality across 32 Greenland glacier termini in 2013-2022. We then extend our study to 32 glacier termini, most of which (ID 1~25) are picked from previous studies with strong terminus-position seasonality [18, 19], and the rest (ID 26~32) have annual ice discharge larger than 5 Gt/yr [20]. The locations of the termini are marked on a Greenland velocity map in Fig. 8(a). We identify 341 ArcticDEM strips at 2-meter resolution that cover the mélange regions for the 32 studied termini. For each DEM strip, we investigate terminus position variations [13] during a two-month time window centering on the DEM acquisition date. If the terminus keeps advancing (or retreating) within the time window, then the DEM potentially represents mélange with a strong (or weak) buttressing force. If the terminus alternates between advancing and retreating within the time window, we discard the corresponding DEM strip because the relationship between mélange and calving dynamics is ambiguous in this case. After filtering all DEM strips through this criterion, we identify 60 DEM strips during terminus advances, and 50 DEM strips during terminus retreats, from February to November in 2013–2022. Figure 8 in SI presents an example of the DEM filtering procedure at Helheim Glacier, resulting in two TLS-derived DEMs during terminus advances (Fig. 1(c)(e)), and two ArcticDEM strips during terminus retreats. For every ArcticDEM strip, the mélange thickness is defined as the maximum value obtained from the representative mélange thickness profile within a 200-meter range from the terminus. Table. 2 in SI summarizes the observed minimum (or maximum) mélange thickness when terminus retreats (or advances) as H_0^{\min} (or H_0^{\max}), with the corresponding DEM acquisition month shown in the bracket.

Comment (17). Ln 349: what is the difference between bed topography and bathymetry?

Response. We have deleted bathymetry to avoid repetition.

Comment (18). Ln 354: delete “field”

Response. We have deleted “field” as follows:

Our data analysis show that mélange thickness seasonality strongly correlates with calving dynamics across Greenland.

Comment (19). Ln 382: add “along-track” to the front of “is 20 m”.

Response. We have modified the sentence as follows:

3.2 ICESat-2 data and uncertainty assessment. We use the ATL06 data set from ICESat-2 that provides geolocated, mean land-ice surface heights above the WGS 84 ellipsoid that are averaged along 40 m segments of ground track and spaced 20 m apart [46].

In summary, we thank the referee for making these constructive suggestions. We hope that our responses have resolved the ambiguities he/she has pointed out. We believe these additions have improved our manuscript significantly.

REFERENCES

- [1] Surui Xie, Timothy H Dixon, David M Holland, Denis Voytenko, and Irena Vaňková. Rapid iceberg calving following removal of tightly packed pro-glacial mélange. *Nature communications*, 10(1):3250, 2019.
- [2] Twila Moon, Ian Joughin, and Ben Smith. Seasonal to multiyear variability of glacier surface velocity, terminus position, and sea ice/ice mélange in northwest greenland. *Journal of Geophysical Research: Earth Surface*, 120(5):818–833, 2015.
- [3] Alexander A Robel. Thinning sea ice weakens buttressing force of iceberg mélange and promotes calving. *Nature Communications*, 8(1):14596, 2017.
- [4] Steve Foga, Leigh A Stearns, and CJ Van der Veen. Application of satellite remote sensing techniques to quantify terminus and ice mélange behavior at helheim glacier, east greenland. *Marine Technology Society Journal*, 48(5):81–91, 2014.
- [5] Jason M Amundson, Mark Fahnestock, Martin Truffer, Jed Brown, Martin P Lüthi, and Roman J Motyka. Ice mélange dynamics and implications for terminus stability, jakobshavn isbræ, greenland. *Journal of Geophysical Research: Earth Surface*, 115(F1), 2010.
- [6] Ryan Cassotto, Mark Fahnestock, Jason M Amundson, Martin Truffer, and Ian Joughin. Seasonal and interannual variations in ice melange and its impact on terminus stability, jakobshavn isbræ, greenland. *Journal of Glaciology*, 61(225):76–88, 2015.
- [7] Justin C Burton, Jason M Amundson, Ryan Cassotto, Chin-Chang Kuo, and Michael Dennin. Quantifying flow and stress in ice mélange, the world’s largest granular material. *Proceedings of the National Academy of Sciences*, 115(20):5105–5110, 2018.
- [8] Jason M Amundson and JC Burton. Quasi-static granular flow of ice mélange. *Journal of Geophysical Research: Earth Surface*, 123(9):2243–2257, 2018.

- [9] J Krug, G Durand, O Gagliardini, and J Weiss. Modelling the impact of submarine frontal melting and ice mélange on glacier dynamics. *The Cryosphere*, 9(3):989–1003, 2015.
- [10] Joe Todd and Poul Christoffersen. Are seasonal calving dynamics forced by buttressing from ice mélange or undercutting by melting? outcomes from full-stokes simulations of store glacier, west greenland. *The Cryosphere*, 8(6):2353–2365, 2014.
- [11] Jacob I Walter, Jason E Box, Slawek Tulaczyk, Emily E Brodsky, Ian M Howat, Yushin Ahn, and Abel Brown. Oceanic mechanical forcing of a marine-terminating greenland glacier. *Annals of Glaciology*, 53(60):181–192, 2012.
- [12] Ryan K Cassotto, Justin C Burton, Jason M Amundson, Mark A Fahnestock, and Martin Truffer. Granular decoherence precedes ice mélange failure and glacier calving at jakobshavn isbræ. *Nature Geoscience*, 14(6):417–422, 2021.
- [13] Enze Zhang, Ginny Catania, and Daniel T Trugman. Autoterm: an automated pipeline for glacier terminus extraction using machine learning and a “big data” repository of greenland glacier termini. *The Cryosphere*, 17(8):3485–3503, 2023.
- [14] MJ Fried, GA Catania, LA Stearns, DA Sutherland, TC Bartholomaus, E Shroyer, and J Nash. Reconciling drivers of seasonal terminus advance and retreat at 13 central west greenland tidewater glaciers. *Journal of Geophysical Research: Earth Surface*, 123(7):1590–1607, 2018.
- [15] Ala Khazendar, Ian G Fenty, Dustin Carroll, Alex Gardner, Craig M Lee, Ichiro Fukumori, Ou Wang, Hong Zhang, Hélène Seroussi, Delwyn Moller, et al. Interruption of two decades of jakobshavn isbrae acceleration and thinning as regional ocean cools. *Nature Geoscience*, 12(4):277–283, 2019.
- [16] Douglas I Benn, JAN Åström, Thomas Zwinger, JOE Todd, Faezeh M Nick, Susan Cook, Nicholas RJ Hulton, and Adrian Luckman. Melt-under-cutting and buoyancy-driven calving from tidewater glaciers: new insights from discrete element and continuum model simulations. *Journal of Glaciology*, 63(240):691–702, 2017.
- [17] DA Slater, DI Benn, TR Cowton, JN Bassis, and JA Todd. Calving multiplier effect controlled by melt undercut geometry. *Journal of Geophysical Research: Earth Surface*, 126(7):e2021JF006191, 2021.
- [18] Twila Moon, Ian Joughin, Ben Smith, Michiel R Van Den Broeke, Willem Jan Van De Berg, Brice Noël, and Mika Usher. Distinct patterns of seasonal greenland glacier velocity. *Geophysical research letters*, 41(20):7209–7216, 2014.
- [19] Saurabh Vijay, Shfaqat Abbas Khan, Anders Kusk, Anne M Solgaard, Twila Moon, and Anders Anker Bjørk. Resolving seasonal ice velocity of 45 greenlandic glaciers with very high temporal details. *Geophysical Research Letters*, 46(3):1485–1495, 2019.
- [20] Jérémie Mouginot, Eric Rignot, Anders A Bjørk, Michiel Van den Broeke, Romain Millan, Mathieu Morlighem, Brice Noël, Bernd Scheuchl, and Michael Wood. Forty-six years of greenland ice sheet mass balance from 1972 to 2018. *Proceedings of the national academy of sciences*, 116(19):9239–9244, 2019.
- [21] Connor J Shiggins, James M Lea, and Stephen Brough. Automated arcticdem ice-berg detection tool: insights into area and volume distributions, and their potential application to satellite imagery and modelling of glacier–iceberg–ocean systems. *The*

- Cryosphere*, 17(1):15–32, 2023.
- [22] Ellyn M Enderlin, Gordon S Hamilton, Fiammetta Straneo, and David A Sutherland. Iceberg meltwater fluxes dominate the freshwater budget in greenland’s iceberg-congested glacial fjords. *Geophysical Research Letters*, 43(21):11–287, 2016.
- [23] Lufeng Liu, Zhuoran Li, Yang Jiao, and Shuixiang Li. Maximally dense random packings of cubes and cuboids via a novel inverse packing method. *Soft matter*, 13(4):748–757, 2017.
- [24] ShuiXiang Li, Jian Zhao, Peng Lu, and Yu Xie. Maximum packing densities of basic 3d objects. *Chinese Science Bulletin*, 55(2):114–119, 2010.
- [25] Kishor G Nayar, Mostafa H Sharqawy, Leonardo D Banchik, et al. Thermophysical properties of seawater: A review and new correlations that include pressure dependence. *Desalination*, 390:1–24, 2016.
- [26] A. C. Fowler. Glaciers and ice sheets. In Jesús Idefonso Díaz, editor, *The Mathematics of Models for Climatology and Environment*, pages 301–336, Berlin, Heidelberg, 1997. Springer Berlin Heidelberg.
- [27] Jeremy N Bassis and Samuel B Kachuck. Beyond the stokes approximation: shallow visco-elastic ice-sheet models. *Journal of Glaciology*, pages 1–12, 2023.
- [28] Jae Hun Kim, Eric Rignot, David Holland, and Denise Holland. Seawater intrusion at the grounding line of jakobshavn isbræ, greenland, from terrestrial radar interferometry. *Geophysical Research Letters*, 51(6):e2023GL106181, 2024.
- [29] Adeline Favier de Coulomb, Mehdi Bouzid, Philippe Claudin, Eric Clément, and Bruno Andreotti. Rheology of granular flows across the transition from soft to rigid particles. *Physical Review Fluids*, 2(10):102301, 2017.
- [30] M Yasinul Karim and Eric I Corwin. Eliminating friction with friction: 2d janssen effect in a friction-driven system. *Physical Review Letters*, 112(18):188001, 2014.
- [31] Douglas R MacAyeal. Large-scale ice flow over a viscous basal sediment: Theory and application to ice stream b, antarctica. *Journal of Geophysical Research: Solid Earth*, 94(B4):4071–4087, 1989.
- [32] Daniel J Sulak, David A Sutherland, Ellyn M Enderlin, Leigh A Stearns, and Gordon S Hamilton. Iceberg properties and distributions in three greenlandic fjords using satellite imagery. *Annals of Glaciology*, 58(74):92–106, 2017.
- [33] Laura M Kehrl, Ian Joughin, David E Shean, Dana Floricioiu, and Lukas Krieger. Seasonal and interannual variabilities in terminus position, glacier velocity, and surface elevation at helheim and kangerlussuaq glaciers from 2008 to 2016. *Journal of Geophysical Research: Earth Surface*, 122(9):1635–1652, 2017.
- [34] Gong Cheng, Mathieu Morlighem, Jérémie Mouginot, and Daniel Cheng. Helheim glacier’s terminus position controls its seasonal and inter-annual ice flow variability. *Geophysical Research Letters*, 49(5):e2021GL097085, 2022.
- [35] Lizz Ultee, Denis Felikson, Brent Minchew, Leigh A Stearns, and Bryan Riel. Helheim glacier ice velocity variability responds to runoff and terminus position change at different timescales. *Nature Communications*, 13(1):6022, 2022.
- [36] Thomas R Chudley, Ian M Howat, Michalea D King, and Adelaide Negrete. Atlantic water intrusion triggers rapid retreat and regime change at previously stable greenland

- glacier. *Nature Communications*, 14(1):2151, 2023.
- [37] Suzanne L Bevan, Adrian J Luckman, Douglas I Benn, Tom Cowton, and Joe Todd. Impact of warming shelf waters on ice mélange and terminus retreat at a large se greenland glacier. *The Cryosphere*, 13(9):2303–2315, 2019.
- [38] Ian Joughin, David E Shean, Benjamin E Smith, and Dana Floricioiu. A decade of variability on jakobshavn isbræ: ocean temperatures pace speed through influence on mélange rigidity. *The Cryosphere*, 14(1):211–227, 2020.
- [39] Sierra M Melton, Richard B Alley, Sridhar Anandakrishnan, Byron R Parizek, Michael G Shahin, Leigh A Stearns, Adam L LeWinter, and David C Finnegan. Meltwater drainage and iceberg calving observed in high-spatiotemporal resolution at helheim glacier, greenland. *Journal of Glaciology*, 68(270):812–828, 2022.
- [40] Jason M Amundson, Christian Kienholz, Alexander O Hager, Rebecca H Jackson, Roman J Motyka, Jonathan D Nash, and David A Sutherland. Formation, flow and break-up of ephemeral ice mélange at leconte glacier and bay, alaska. *Journal of Glaciology*, 66(258):577–590, 2020.
- [41] W Tad Pfeffer, Joel T Harper, and Shad O’Neel. Kinematic constraints on glacier contributions to 21st-century sea-level rise. *Science*, 321(5894):1340–1343, 2008.
- [42] Joe Todd, Poul Christoffersen, Thomas Zwinger, Peter Råback, and Douglas I Benn. Sensitivity of a calving glacier to ice–ocean interactions under climate change: new insights from a 3-d full-stokes model. *The Cryosphere*, 13(6):1681–1694, 2019.
- [43] Förste Ch, Sean L Bruinsma, Oleg Abrikosov, Jean-Michel Lemoine, T Schaller, HJ Gtze, J Ebbing, JC Marty, F Flechtner, G Balmino, et al. Eigen-6c4 the latest combined global gravity field model including goce data up to degree and order 2190 of gfz potsdam and grgs toulouse. *GFZ Data Services*, 10(10.5880), 2014.
- [44] Susan L Howard and Laurie Padman. Gr1kmtm: Greenland 1 kilometer tide model, 2021.
- [45] Christopher Miele and Timothy Bartholomäus. Greenland ice sheet’s distinct calving styles are identified in terminus change timeseries. *Authorea Preprints*, 2024.
- [46] B Smith, S Adusumilli, BM Csathó, D Felikson, HA Fricker, A Gardner, N Holschuh, J Lee, J Nilsson, FS Paolo, et al. the ICESat-2 Science Team: ATLAS/ICESat-2 L3A Land Ice Height, Version 5, NASA National Snow and Ice Data Center Distributed Active Archive Center, Boulder, Colorado, USA [dataset], 2021.

REPLY TO REFEREE COMMENTS - REFEREE 1.

We thank the referee for a constructive review of our paper. The referee states that “The manuscript by Meng. et al has undergone substantial revisions and is much improved in its current form. I commend the authors for their tedious and thorough response to the original submission.” He/she makes several constructive suggestions to strengthen and improve the manuscript. We have taken the referee’s comments very seriously, and we have revised the manuscript to address the referee’s comments and suggestions.

In the following, we detail the amendments made to the manuscript (highlighted in red color) in response to the referee’s comments (included in italics). Note that the line numbers refer to the tracked changes version.

Comment (1). My primary remaining comment relates to the framing of melange “thinning” coinciding with retreat events, particularly in section 1.1 of the results. While a reduction in the elevations within 1 km of the terminus technically does occur, I think the language stating “melange thinning” implies a rapid melting of the material in place, or a thermodynamic forcing, rather than (what I take to be the case) the calving event disrupting the proglacial melange and flushing the previously adjacent melange further down fjord, and partially replaced with the newly calved material. By the study’s defining parameters for melange thickness calculations (at Helheim, for example, the melange within 1 km of the terminus) would sample different areas of melange before and after retreat even if the melange remained completely static, as a retreated terminus would allow new melanged areas of the fjord to meet the “within 1 km” criteria. This is further supported by the statement in the revision that melange thinning was also “accompanied by large increases in fraction of melange area composed of large icebergs.” Particularly the January 2020 event, melange is reported to have thinning by ≈ 5 m in three days during winter, which would not follow expected seasonal thinning trends. In section 1.1, it would be more helpful to provide a figure for the melange thickness preceding the calving events, as it does appear to show a gradual decrease in elevation following the second major calving event of fall 2019 (Figure 1b) up through the January 2020 calving event.

Response. The reviewer raised an excellent point that calving can induce significant thinning of mélanges, as has been reported in [1]. We therefore attribute the observed mélanges thinning events to two mechanisms: melting or thermodynamic forcing-induced thinning that occurred prior to calving, and calving-induced divergent motion within the mélanges. We identified two periods during which noticeable mélanges thinning occurred prior to calving, and added TLS scans in Fig. 1 in the main text to reflect this (Fig. 1(g)-(j) in this document). The mélanges elevation at the terminus decreased by 1.6 m to 10.8 ± 0.05 m from 1 Sep to 3 Sep 2019, and by 2.3 m to 11.6 ± 0.05 m from 7 Jul to 14 Jul 2020 (Fig. 1(g)(h)). Major calving occurred on 4 Sep 2019 and 15 Jul 2020 (Fig. 1(i))

with corresponding retreats at the terminus of 0.5 km and 1.0 km, respectively. We have modified the main text as follows (Section 1.1):

1.1 Mélange thickness associated with calving dynamics at Helheim Glacier in 2019-2020. To investigate the correlation between the mélange thickness and calving dynamics, we derive time series of mélange elevation at the terminus, calving events, and terminus position inferred from TLS data and satellite images (Sentinel-1, Sentinel-2 and Landsat 8) from 1 Sep 2019 to 1 Sep 2020 at Helheim Glacier (Fig. 1(b)). Throughout 2019, a REIGL VZ-6000 terrestrial laser scanner (TLS), located at the triangle marked in Figure 1(a), scanned the terminus and ice mélange of Helheim Glacier. Fig. 1(a) shows the TLS-measured surface elevation field for ice mélange on 30 Nov 2019 (see Methods for the TLS processing steps). Mélange surface elevation was derived from TLS every 24 hours in the winter months, and every 6 hours in non-winter months. To display the spatial profile of the mélange elevation, we calculate distances from terminus for all data points in the ice mélange and plot them as density maps in Fig. 1(c)-(f). For any specific distance from the terminus, there is a spread of mélange elevation that exhibits a long-tail distribution due to the presence of large icebergs. To reflect the mélange elevation that is piled up from small icebergs and sea ice, we connect elevation values that have maximum numbers of data points along the distance from terminus as the representative mélange elevation profiles (solid blue lines in Fig. 1(c)-(f), see Methods and Fig. 2 in SI for a sensitivity study of this metric). In doing so, we prevent large icebergs from disrupting the mélange elevation profiles without cutting off the data by an arbitrary elevation threshold. The resulting representative mélange elevation profiles are always below 30 m, indicating that it is a reasonable elevation threshold to exclude large icebergs from consideration. To estimate mélange elevation near the glacier terminus (Z_0), we take an average of all data points below the 30 m threshold within 1 km (i.e., 1/5 of the fjord width) of the terminus. We infer thickness of the mélange based on TLS-derived surface elevations and assuming hydrostatic equilibrium.

We digitize the terminus position on each TLS scan with a straight line (the white line in Fig. 1(a)). The terminus advance/retreat refers to the changes in the terminus position, and is quantified by distances between these straight lines. This can indicate whether the glaciers are advancing (growing) or retreating (shrinking). With reference to previous classification of calving events [1], here, we define “major calving” events as those with block size >0.25 km² (observed from TLS scans), and causing significant mélange motion and an overall terminus retreat (observed from satellite images); “minor calving” events are those in which visible blocks calved, but the mélange or terminus position remained largely unchanged. We observe two episodes of calving cessation, from 8 Oct 2019 - 31 Dec 2019 and 1 Mar 2020 - 20 May 2020, when no calving occurred and the terminus advanced steadily for 2 km and 1.7 km,

respectively. We found that mélange elevation at the terminus averaged 15 m during these periods. We include TLS scans during these two episodes of calving cessation in the supplementary material (Fig. 3 in SI). We identified two periods during which noticeable mélange thinning occurred after calving. The mélange elevation at the terminus decreased by 5.3 m to 9.4 ± 0.05 m from 31 Dec 2019 to 3 Jan 2020 (Fig. 1(d)), and by 4 m to 10 ± 0.05 m from 26 May 2020 to 31 May 2020 (Fig. 1(f)). Major calving occurred on 2 Jan 2020 and 28 May 2020 with corresponding retreats at the terminus of 1.2 km and 1.3 km, respectively, accompanied by large increases in fraction of mélange area composed of large icebergs (see SI Fig. 4 & 5 for associated TLS scans and temporal evolution of iceberg fraction). We attribute the noticeable mélange thinning to calving-induced divergent motion within the mélange that helped to advect ice away [1]. This was followed by one to two months of active calving events from Jan to Mar, and Jun to Jul in 2020 (Fig. 1(b)). In addition, we identified two periods during which noticeable mélange thinning occurred prior to calving. The mélange elevation at the terminus decreased by 1.6 m to 10.8 ± 0.05 m from 1 Sep to 3 Sep 2019, and by 2.3 m to 11.6 ± 0.05 m from 7 Jul to 14 Jul 2020 (Fig. 1(g)(h)). Major calving occurred on 4 Sep 2019 and 15 Jul 2020 (Fig. 1(i)) with corresponding retreats at the terminus of 0.5 km and 1.0 km, respectively.

FIGURE 1. Helheim Glacier and ice mélange. (a) TLS-measured elevation map after accounting for local differences between the ellipsoid and geoid, overlain on a Sentinel-1 HV image (both acquired on 30 Nov 2019). The white line across the fjord indicates the glacier front location. The white triangle indicates the TLS location. The upper left inset shows the location of Helheim Glacier in Greenland. The image is in polar stereographic projection (EPSG: 3413). (b) Terminus position relative to 1 Sep 2019, where the positive sign indicates terminus advance. Blue dots denote the averaged mélange elevation within 1 km of the terminus, Z_0 . Calving events are inferred from TLS and satellite images. Due to limited temporal sampling of the data, we are not able to determine the exact time of each calving event. Instead, we mark the time period during which a calving event occurs by a red-shade rectangle. Four vertical black lines mark the dates for the TLS-measured elevation data presented in (c)-(f), which corresponds to 30 Nov 2019, 3 Jan 2020, 30 Mar 2020, and 31 May 2020, respectively. Solid black lines mark the dates with terminus advances, and dashed black lines mark the dates with terminus retreats. (c)-(f) Surface elevation profiles for the mélange displayed as density plots (1510~1859 data points in total); the colour bar denotes the number of data points that have the same elevation and distance from terminus values. For any specific distance from terminus, we find the elevation value that has the maximum number of data points. Solid blue lines connect these elevation values along the distance from terminus as the representative mélange elevation profiles. **Major calving occurred on 2 Jan 2020 and 28 May 2020 led to noticeable mélange thinning on 3 Jan (d) and 31 May (f), both of which were followed by one to two months of active calving events.** (g)-(j) The TLS-measured mélange surface elevation map from 7 Jul to 18 Jul 2020. Dashed white lines indicate positions of the terminus before calving. Solid white lines indicate positions of the terminus after calving. Dates on the images show the acquisition dates for TLS data and times are in UTC. TLS scans are overlain on Sentinel-2 images acquired around the same dates. Images are in polar stereographic projection (EPSG: 3413). **Mélange thinning was observed from 7 Jul to 14 Jul 2020, followed by a major calving event occurred between 14 Jul and 15 Jul 2020.**

Comment (2). I also would suggest prefacing the conclusion of section 1.2 (that ICESat-2 data provides strong evidence of correlations between mélange thickness and terminus seasonality” with an acknowledgement of how highly variable mélange thickness distributions can be across both time and space, as evidenced by earlier results from section 1.1, and as evident in ArcticDEM freeboard heights. The ICESat-2 data provide a snapshot into a profile that may look quite different within days or across a different profile line in the fjord. That is not to say the authors do not provide useful analysis, as it is yet another helpful metric that adds to the study as a whole, but only that generalized conclusions should also include an acknowledgement of spatiotemporal uncertainty.

Response. We have prefaced the conclusion of section 1.2 as suggested by the reviewer as follows:

1.2 Seasonal changes of mélange thickness and calving dynamics. To investigate whether there are correlations between ice mélange thickness and calving dynamics on other glaciers, we use ICESat-2 observations of mélange surface elevation. While this dataset does not provide the temporal resolution to study individual calving events, we can leverage the observed seasonality in terminus advance and retreat at many Greenland glaciers to assess whether mélange thickness is correlated with periods of quiescence versus vigorous calving.

There are very few ICESat-2 tracks passing through the fronts of termini in different seasons, because positions of termini vary seasonally but ICESat-2 tracks are generally fixed in space. We identify ICESat-2 tracks passing over glacier termini in different seasons for Jakobshavn Isbræ (Fig. 2(a)), Kangerlussuaq Glacier (Fig. 2(b)), and Store Glacier (Fig. 2(c)), which are large discharging glaciers contributing to Greenland’s mass losses [2]. We use the ATL06 data set from ICESat-2 that provides geolocated, land-ice surface heights above the WGS 84 ellipsoid [3]. After removing the geoid heights at glacier termini [4], the mélange surface elevations are heights above mean sea level with tidal variations [1]. We then remove the tidal trend of mélange elevations with the Greenland 1 kilometer tide model [5]. Surface elevation data is acquired along the ICESat-2 track and displayed as a function of the distance from terminus. We found that mélange was continuously present at Jakobshavn Isbræ from 2021 to 2022 and at Kangerlussuaq Glacier in 2020, while it was seasonally present at Store Glacier in 2019. Where mélange persisted, we calculated seasonally distinct freeboard heights from winter to early spring (solid black lines) and in summer (dashed black lines) (Fig. 2(d)(e)). Near the termini, mélange for the two glaciers both exhibit different ranges of freeboard heights during the two seasons: 20 ~ 35 m in winter or spring, and below 5 m in summer. The seasonal changes in mélange thickness at the terminus may explain the observed calving dynamics and terminus motion: zero or minor calving with an advancing terminus from winter to spring, and vigorous calving with a retreating terminus from summer to fall (Fig. 2(g)(h)). At Store Glacier, the mélange was present

from 1 Jan to 14 June, after which calving resumed and the terminus kept retreating (Fig. 2(i)). The mélange elevation profile on 22 March 2019 exhibits a thickness gradient with a freeboard height of around 30 m near the terminus (Fig. 2(f)). ICESat-2 data provide a snapshot of the mélange elevation profile, which may vary significantly within days or across different profile lines in the fjord, as evidenced by results from Section 1.1. Although ICESat-2 data cannot fully capture the spatiotemporal variability in mélange thickness distributions, it still offers valuable insights into the correlations between mélange thickness and terminus seasonality.

In summary, we thank the referee for making these constructive suggestions. We hope that our responses have resolved the ambiguities he/she has pointed out. We believe these additions have improved our manuscript significantly.

REFERENCES

- [1] Surui Xie, Timothy H Dixon, David M Holland, Denis Voytenko, and Irena Vaňková. Rapid iceberg calving following removal of tightly packed pro-glacial mélange. *Nature communications*, 10(1):3250, 2019.
- [2] Jérémie Mouginot, Eric Rignot, Anders A Bjørk, Michiel Van den Broeke, Romain Millan, Mathieu Morlighem, Brice Noël, Bernd Scheuchl, and Michael Wood. Forty-six years of greenland ice sheet mass balance from 1972 to 2018. *Proceedings of the national academy of sciences*, 116(19):9239–9244, 2019.
- [3] B. Smith, S. Adusumilli, B. M. Csathó, D. Felikson, H. A. Fricker, A. S. Gardner, N. Holschuh, J. Lee, J. Nilsson, F. Paolo, M. R. Siegfried, T. Sutterley, and the ICESat-2 Science Team. Atlas/icesat-2 l3a land ice height, version 6, 2023.
- [4] Förste Ch, Sean L Bruinsma, Oleg Abrikosov, Jean-Michel Lemoine, T Schaller, HJ Gtze, J Ebbing, JC Marty, F Flechtner, G Balmino, et al. Eigen-6c4 the latest combined global gravity field model including goce data up to degree and order 2190 of gfg potsdam and grgs toulouse. *GFZ Data Services*, 10(10.5880), 2014.
- [5] Susan L Howard and Laurie Padman. Gr1kmtm: Greenland 1 kilometer tide model, 2021.

REPLY TO REFEREE COMMENTS - REFEREE 2.

We thank the referee for a constructive review of our paper. The referee states that “I thank the authors for largely addressing all my general and specific comments. I think the manuscript is much improved. ” He/she makes several constructive suggestions to strengthen and improve the manuscript. We have taken the referee’s comments very seriously, and we have revised the manuscript to address the referee’s comments and suggestions.

In the following, we detail the amendments made to the manuscript (highlighted in red color) in response to the referee’s comments (included in italics). Note that the line numbers refer to the tracked changes version.

***Comment (1).** L49-50: This is a pretty vague statement. What do you mean by “...significant effect on the seasonal range in the terminus position”? More range or less range? Or shifting the range inland? Or both?*

Response. In a warming climate, a complete loss of mélange buttressing may prevent terminus advances in winter and spring while exacerbating summer retreats, and therefore it is uncertain whether the seasonal range in the terminus position becomes larger or smaller. At Store Glacier, the seasonal range in the terminus position is predicted to decrease from 750 m to 300 m when mélange is entirely absent [1], but we do not want to make a general conclusion for all glacier termini based on this single case. Instead, the seasonal range in the terminus position will certainly be shifted inland. We have clarified the sentence as follows (lines 48–49):

In a warming climate, a complete loss of mélange buttressing may prevent terminus advances in winter and spring **while exacerbating summer retreats, shifting the seasonal range in the terminus position inland [1].**

***Comment (2).** L113-114: I understand that the center of the white line is placed on the furthest downstream point of the calving front but how do you determine where the edges of the line intersect the fjord?*

Response. The white line is placed to intersect and be tangent to the furthest upstream (the most inland) point of the calving front. We have clarified the sentence as follows (lines 101–102):

We digitize the terminus position on each TLS scan with a straight line that **intersects and is tangent to the furthest upstream point of the calving front** (the white line in Fig. 1(a)).

***Comment (3).** L123: Consider clarifying that no “major or minor” calving occurred.*

Response. We have revised the sentence as follows (lines 109–111):

We observe two episodes of calving cessation, from 8 Oct 2019 - 31 Dec 2019 and 1 Mar 2020 - 20 May 2020, when no **major or minor** calving occurred and the terminus advanced steadily for 2 km and 1.7 km, respectively.

Comment (4). *L144: Clarify that this is “satellite” remote sensing*

Response. We have revised the sentence as follows (line 133):

Apart from the weekly terminus variability of Helheim Glacier presented in this section, **satellite** remote sensing observations on many Greenland glacier termini have shown significant terminus-position seasonality, with advance from winter to spring and retreat from summer to fall through enhanced calving [2, 3, 4].

Comment (5). *L163: What about “surface elevations represent heights above mean sea level...”?*

Response. We have revised the sentence as follows (lines 150–154):

After subtracting the geoid heights at the glacier termini [5], the mélange surface elevations represent heights above mean sea level [6]. Next, we correct for the tidal influence on the mélange elevations using the Greenland 1-kilometer tide model [7], ensuring that the adjusted surface elevations represent heights above sea level at the time of data acquisition.

Comment (6). *L163: It’s unclear from this statement whether these surface elevations include or exclude tidal variations.*

Response. See response to comment 5.

Comment (7). *L301-302: The problem for a new reader is that ArcticDEM observations (L301, L314, and L320) have not been introduced yet. Surely L399-431 has to go before you compare the model with the mélange thickness observations from ArcticDEM? I think it be worth reordering some of the text to address this.*

Response. The structure of the results section is organized as follows: Remote observations from TLS (Section 1.1) and ICESat-2 (Section 1.2) at four glacier termini reveal correlations between mélange thickness and calving dynamics, leading us to quantify the mélange buttressing force in terms of thickness using continuum (Section 1.3) and discrete element models (Section 1.4). After showing that mélange thickness controls its buttressing force in our models (Section 1.5), we extend the investigation to 32 Greenland glacier termini by analyzing calving dynamics and mélange buttressing force using terminus positions and ArcticDEM data (Section 1.6). Thus, Lines 399-431 in Section 1.6, which include the analysis of terminus positions and ArcticDEM data at 32 Greenland glacier termini, would be excessive to move to Section 1.4, as that section only covers ArcticDEM data from two glacier termini. We aim to avoid overwhelming Section 1.4 with remote data descriptions in order to 1) prevent redundancy with Section 1.6, and 2) place more emphasis on the discrete element model.

To briefly introduce ArcticDEM data in Section 1.4, we have added the following sentence (lines 284–286):

1.4 A three-dimensional discrete element model of ice mélange. ... In a series of simulations, we vary the initial mélange thickness to determine its influence on the steady state buttressing force. **To validate the modeled mélange thickness with observations, we identify ArcticDEM strips at 2-meter resolution that cover the mélange regions in front of Helheim and Kangerlussuaq glacier termini (see Section 1.6 and 3.3 for details).** The steady state mélange thickness at the terminus varies from 40~135 m at Helheim fjord, and 60~240 m at Kangerlussuaq fjord, that are consistent with ArcticDEM observations (Table 2 in SI). ...

Comment (8). L409-410: How do you define “warm” and “cold” months? Here and elsewhere (e.g. L493 and 495)

Response. We have clarified the sentence as follows (lines 387–388):

We observe that the freeboard height of the mélange at the terminus ranges from 2.8 ~ 3.9 m **in summer** and 19.2 ~ 26.8 m **in fall-to-spring**.

We have clarified the sentence in the discussion section as follows (lines 452–458):

Scanning through 108 ArcticDEM strips, we discover calving dynamics associated with mélange thickness seasonality across 32 Greenland glacier termini in 2013–2022. When termini advance in **late fall-to-spring (Nov-May)**, the average value of all observed mélange thicknesses is 119_{-37}^{+31} m, with a corresponding buttressing force $6.5_{-3.7}^{+3.4} \times 10^6$ N/m. When termini retreat in **summer-to-fall (Jun-Oct)**, the average thickness is 34_{-15}^{+17} m, with a corresponding buttressing force of $5.2_{-3.8}^{+5.9} \times 10^5$ N/m.

Comment (9). L459: I thought these were “disk-shaped” in L55 and L713? Or is there a difference between these statements?

Response. The reviewer is correct that previous simulations are using disk-shaped grains, not spheres. We have revised the sentence as follows (lines 427–429):

Our discrete element model of ice mélange is the first to be composed of realistic cubic icebergs instead of **disk-shaped grains**, and showcases an exponential decay of the thickness profile in Helheim and Kangerlussuaq fjords, that is consistent with ArcticDEM observations and analytical predictions.

Comment (10). L485-487: Can you explain why a glacier that has just calved have fewer crevasses and take longer to calve again?

Response. This sentence is indeed misleading and therefore we decide to delete it. See response to comment 11 for details.

Comment (11). L487-489: I think you should elaborate on this point a little more. Although I still believe that there is some threshold behavior where mélange rigidity gradually reduces to a point where large calving events can occur. I do still find it plausible that large calving events are responsible for dramatic changes in mélange thickness. How else can

you explain the 4-5 m of mélange elevation reduction in 3-5 days that you observe with the TLS?

Response. The reviewer raised an excellent point that calving can induce significant thinning of mélange, as has been reported in [6]. We therefore attribute the observed mélange thinning events to two mechanisms: melting or thermodynamic forcing-induced thinning that occurred prior to calving, and calving-induced divergent motion within the mélange. We identified two periods during which noticeable mélange thinning occurred prior to calving, and added TLS scans in Fig. 1 in the main text to reflect this (Fig. 1(g)-(j) in this document). The mélange elevation at the terminus decreased by 1.6 m to 10.8 ± 0.05 m from 1 Sep to 3 Sep 2019, and by 2.3 m to 11.6 ± 0.05 m from 7 Jul to 14 Jul 2020 (Fig. 1(g)(h)). Major calving occurred on 4 Sep 2019 and 15 Jul 2020 (Fig. 1(i)) with corresponding retreats at the terminus of 0.5 km and 1.0 km, respectively. We have modified the main text as follows (Section 1.1):

1.1 Mélange thickness associated with calving dynamics at Helheim Glacier in 2019-2020....We digitize the terminus position on each TLS scan with a straight line (the white line in Fig. 1(a)). The terminus advance/retreat refers to the changes in the terminus position, and is quantified by distances between these straight lines. This can indicate whether the glaciers are advancing (growing) or retreating (shrinking). With reference to previous classification of calving events [6], here, we define “major calving” events as those with block size $>0.25 \text{ km}^2$ (observed from TLS scans), and causing significant mélange motion and an overall terminus retreat (observed from satellite images); “minor calving” events are those in which visible blocks calved, but the mélange or terminus position remained largely unchanged. We observe two episodes of calving cessation, from 8 Oct 2019 - 31 Dec 2019 and 1 Mar 2020 - 20 May 2020, when no calving occurred and the terminus advanced steadily for 2 km and 1.7 km, respectively. We found that mélange elevation at the terminus averaged 15 m during these periods. We include TLS scans during these two episodes of calving cessation in the supplementary material (Fig. 3 in SI). **We identified two periods during which noticeable mélange thinning occurred after calving.** The mélange elevation at the terminus decreased by 5.3 m to 9.4 ± 0.05 m from 31 Dec 2019 to 3 Jan 2020 (Fig. 1(d)), and by 4 m to 10 ± 0.05 m from 26 May 2020 to 31 May 2020 (Fig. 1(f)). Major calving occurred on 2 Jan 2020 and 28 May 2020 with corresponding retreats at the terminus of 1.2 km and 1.3 km, respectively, accompanied by large increases in fraction of mélange area composed of large icebergs (see SI Fig. 4 & 5 for associated TLS scans and temporal evolution of iceberg fraction). We attribute the noticeable mélange thinning to calving-induced divergent motion within the mélange that helped to advect ice away [6]. This was followed by one to two months of active calving events from Jan to Mar, and Jun to Jul in 2020 (Fig. 1(b)). In addition, we identified two periods during which noticeable mélange thinning occurred prior to calving. The mélange elevation at the terminus decreased by 1.6 m to

10.8±0.05 m from 1 Sep to 3 Sep 2019, and by 2.3 m to 11.6±0.05 m from 7 Jul to 14 Jul 2020 (Fig. 1(g)(h)). Major calving occurred on 4 Sep 2019 and 15 Jul 2020 (Fig. 1(i)) with corresponding retreats at the terminus of 0.5 km and 1.0 km, respectively.

FIGURE 1. Helheim Glacier and ice mélange. (a) TLS-measured elevation map after accounting for local differences between the ellipsoid and geoid, overlain on a Sentinel-1 HV image (both acquired on 30 Nov 2019). The white line across the fjord indicates the glacier front location. The white triangle indicates the TLS location. The upper left inset shows the location of Helheim Glacier in Greenland. The image is in polar stereographic projection (EPSG: 3413). (b) Terminus position relative to 1 Sep 2019, where the positive sign indicates terminus advance. Blue dots denote the averaged mélange elevation within 1 km of the terminus, Z_0 . Calving events are inferred from TLS and satellite images. Due to limited temporal sampling of the data, we are not able to determine the exact time of each calving event. Instead, we mark the time period during which a calving event occurs by a red-shade rectangle. Four vertical black lines mark the dates for the TLS-measured elevation data presented in (c)-(f), which corresponds to 30 Nov 2019, 3 Jan 2020, 30 Mar 2020, and 31 May 2020, respectively. Solid black lines mark the dates with terminus advances, and dashed black lines mark the dates with terminus retreats. (c)-(f) Surface elevation profiles for the mélange displayed as density plots (1510~1859 data points in total); the colour bar denotes the number of data points that have the same elevation and distance from terminus values. For any specific distance from terminus, we find the elevation value that has the maximum number of data points. Solid blue lines connect these elevation values along the distance from terminus as the representative mélange elevation profiles. **Major calving occurred on 2 Jan 2020 and 28 May 2020 led to noticeable mélange thinning on 3 Jan (d) and 31 May (f), both of which were followed by one to two months of active calving events.** (g)-(j) The TLS-measured mélange surface elevation map from 7 Jul to 18 Jul 2020. Dashed white lines indicate positions of the terminus before calving. Solid white lines indicate positions of the terminus after calving. Dates on the images show the acquisition dates for TLS data and times are in UTC. TLS scans are overlain on Sentinel-2 images acquired around the same dates. Images are in polar stereographic projection (EPSG: 3413). **Mélange thinning was observed from 7 Jul to 14 Jul 2020, followed by a major calving event occurred between 14 Jul and 15 Jul 2020.**

We have revised the discussion section as follows (lines 440–458):

Previous research suggests that the presence of ice mélange can reduce iceberg calving by providing “backstress” to the terminus [6, 8, 9, 10, 11, 12, 13, 14, 15, 16, 17]. Our comparisons of time-varying mélange thickness and calving dynamics at Helheim Glacier (Fig. 1(b)) support the view that the buttressing force increases with the mélange thickness and thus inhibits calving. Our TLS scans indicate a correlation between mélange thickness and calving dynamics, but we cannot determine causality before examining other mechanisms driving calving dynamics. **Though we have observed mélange thinning prior to calving (Fig. 1(h)(i)), the mélange thickness and the terminus may react simultaneously but independently to other oceanic and atmospheric forcing.** Scanning through 108 ArcticDEM strips, we discover calving dynamics associated with mélange thickness seasonality across 32 Greenland glacier termini in 2013–2022. When termini advance in late fall-to-spring (Nov–May), the average value of all observed mélange thicknesses is 119_{-37}^{+31} m, with a corresponding buttressing force $6.5_{-3.7}^{+3.4} \times 10^6$ N/m. When termini retreat in summer-to-fall (Jun–Oct), the average thickness is 34_{-15}^{+17} m, with a corresponding buttressing force of $5.2_{-3.8}^{+5.9} \times 10^5$ N/m.

In summary, we thank the referee for making these constructive suggestions. We hope that our responses have resolved the ambiguities he/she has pointed out. We believe these additions have improved our manuscript significantly.

REFERENCES

- [1] Joe Todd, Poul Christoffersen, Thomas Zwinger, Peter Råback, and Douglas I Benn. Sensitivity of a calving glacier to ice–ocean interactions under climate change: new insights from a 3-d full-stokes model. *The Cryosphere*, 13(6):1681–1694, 2019.
- [2] Twila Moon, Ian Joughin, Ben Smith, Michiel R Van Den Broeke, Willem Jan Van De Berg, Brice Noël, and Mika Usher. Distinct patterns of seasonal greenland glacier velocity. *Geophysical research letters*, 41(20):7209–7216, 2014.
- [3] Saurabh Vijay, Shfaqat Abbas Khan, Anders Kusk, Anne M Solgaard, Twila Moon, and Anders Anker Bjørk. Resolving seasonal ice velocity of 45 greenlandic glaciers with very high temporal details. *Geophysical Research Letters*, 46(3):1485–1495, 2019.
- [4] Enze Zhang, Ginny Catania, and Daniel T Trugman. Autoterm: an automated pipeline for glacier terminus extraction using machine learning and a “big data” repository of greenland glacier termini. *The Cryosphere*, 17(8):3485–3503, 2023.
- [5] Förste Ch, Sean L Bruinsma, Oleg Abrikosov, Jean-Michel Lemoine, T Schaller, HJ Gtze, J Ebbing, JC Marty, F Flechtner, G Balmino, et al. Eigen-6c4 the latest combined global gravity field model including goce data up to degree and order 2190 of gfz potsdam and grgs toulouse. *GFZ Data Services*, 10(10.5880), 2014.

- [6] Surui Xie, Timothy H Dixon, David M Holland, Denis Voytenko, and Irena Vaňková. Rapid iceberg calving following removal of tightly packed pro-glacial mélange. *Nature communications*, 10(1):3250, 2019.
- [7] Susan L Howard and Laurie Padman. Gr1kmtm: Greenland 1 kilometer tide model, 2021.
- [8] Twila Moon, Ian Joughin, and Ben Smith. Seasonal to multiyear variability of glacier surface velocity, terminus position, and sea ice/ice mélange in northwest greenland. *Journal of Geophysical Research: Earth Surface*, 120(5):818–833, 2015.
- [9] Alexander A Robel. Thinning sea ice weakens buttressing force of iceberg mélange and promotes calving. *Nature Communications*, 8(1):14596, 2017.
- [10] Steve Foga, Leigh A Stearns, and CJ Van der Veen. Application of satellite remote sensing techniques to quantify terminus and ice mélange behavior at helheim glacier, east greenland. *Marine Technology Society Journal*, 48(5):81–91, 2014.
- [11] Jason M Amundson, Mark Fahnestock, Martin Truffer, Jed Brown, Martin P Lüthi, and Roman J Motyka. Ice mélange dynamics and implications for terminus stability, jakobshavn isbræ, greenland. *Journal of Geophysical Research: Earth Surface*, 115(F1), 2010.
- [12] Ryan Cassotto, Mark Fahnestock, Jason M Amundson, Martin Truffer, and Ian Joughin. Seasonal and interannual variations in ice melange and its impact on terminus stability, jakobshavn isbræ, greenland. *Journal of Glaciology*, 61(225):76–88, 2015.
- [13] Justin C Burton, Jason M Amundson, Ryan Cassotto, Chin-Chang Kuo, and Michael Dennin. Quantifying flow and stress in ice mélange, the world’s largest granular material. *Proceedings of the National Academy of Sciences*, 115(20):5105–5110, 2018.
- [14] Jason M Amundson and JC Burton. Quasi-static granular flow of ice mélange. *Journal of Geophysical Research: Earth Surface*, 123(9):2243–2257, 2018.
- [15] J Krug, G Durand, O Gagliardini, and J Weiss. Modelling the impact of submarine frontal melting and ice mélange on glacier dynamics. *The Cryosphere*, 9(3):989–1003, 2015.
- [16] Joe Todd and Poul Christoffersen. Are seasonal calving dynamics forced by buttressing from ice mélange or undercutting by melting? outcomes from full-stokes simulations of store glacier, west greenland. *The Cryosphere*, 8(6):2353–2365, 2014.
- [17] Jacob I Walter, Jason E Box, Slawek Tulaczyk, Emily E Brodsky, Ian M Howat, Yushin Ahn, and Abel Brown. Oceanic mechanical forcing of a marine-terminating greenland glacier. *Annals of Glaciology*, 53(60):181–192, 2012.

REPLY TO REFEREE COMMENTS - REFEREE 3.

We thank the referee for a constructive review of our paper. The referee states that “The authors have adequately addressed almost all my concerns.” He/she makes several constructive suggestions to strengthen and improve the manuscript. We have taken the referee’s comments very seriously, and we have revised the manuscript to address the referee’s comments and suggestions.

In the following, we detail the amendments made to the manuscript (highlighted in red color) in response to the referee’s comments (included in italics). Note that the line numbers refer to the tracked changes version.

Comment (1). My only comment for their responses is above the above waterline mélange height calculations described in the 2nd paragraph of subsection 1.2: after removing geoid heights, the mélange surface elevations are [approximately] the heights above mean sea level with tidal variations. This is because the mean sea level can deviate from the geoid by a few tens of centimeters (location-dependent). The relatively small difference might not affect the analysis. However, geodesists may prefer to see an accurate description of the relationship.

Response. We have revised the main text as follows (lines 150–154):

After subtracting the geoid heights at the glacier termini [1], the mélange surface elevations are **approximately heights above mean sea level [2]. Next, we correct for the tidal influence on the mélange elevations using the Greenland 1-kilometer tide model [3], ensuring that the adjusted surface elevations represent heights above sea level at the time of data acquisition.**

To account for the uncertainty from geoid [1] and tidal corrections [3], we identify ICESat-2 tracks passing through the ocean area near the three glaciers reported in Fig. 2 in the main text (Jakobshavn Isbræ, Kangerlussuaq Glacier, Store Glacier), and compute the difference between ATL06 measurements over the ocean and the geoid [1] with tidal corrections [3]. We have added a figure in the SI (Fig. 10 in SI and Fig. 1 in this document) and added following sentences in the Methods section:

3.2 ICESat-2 data and uncertainty assessment. We use the ATL06 data set from ICESat-2 that provides geolocated, mean land-ice surface heights above the WGS 84 ellipsoid that are averaged along 40 m segments of ground track and spaced 20 m apart [4]. The temporal resolution is 91 days from 14 October 2018 to present. We compute the mélange surface elevation after accounting for the local difference between the ellipsoid and the EIGEN-6C4 geoid [1], and then remove the tidal signal from the surface elevation [3]. We identify three ICESat-2 tracks for Jakobshavn Isbræ, Kangerlussuaq Glacier and Store Glacier,

respectively, and use data from strong beams to compose the mélange elevation profile (Fig. 2). The averaged standard error in the reported elevation data ranges from 0.02 m~0.52 m due to sampling error and first-photon bias correction from the land ice algorithm, as described in the algorithm theoretical basis document (ATBD) for land ice along-track height product (ATL06) [5]. To account for the uncertainty from geoid and tidal corrections, we identify ICESat-2 tracks passing through the ocean area near the three glaciers presented in Fig. 2 and compute the difference between ATL06 measurements over the ocean and the geoid [1] with tidal corrections [3]. We found that the mean error is 0.37 m at Jakobshavn Isbræ, 0.50 m at Kangerlussuaq Glacier, and 0.60 m at Store Glacier (Fig. 10 in SI)... All these sources of uncertainty are an order of magnitude less than the observed temporal changes in mélange freeboard height presented in Fig. 2, suggesting that our interpretation of seasonal changes in mélange thickness from ICESat-2 are robust.

FIGURE 1. The mélangé elevation data uncertainty by computing the difference between ICESat-2 ATL06 measurements over the ocean and the geoid [1] with tidal corrections [3], at Jakobshavn Isbræ (a)(b), Kangerlussuaq Glacier (c)(d), and Store Glacier (e)(f). (a) The Sentinel-2 image for Jakobshavn Isbræ on 17 Aug 2021. In (a)(c)(e), the white lines indicate ICESat-2 tracks along which ATL06 surface elevation data was acquired. The date and time on the image shows the acquisition time for ICESat-2 data, which is around the same date of the presented satellite image. Images are in polar stereographic projection (EPSG: 3413). The ICESat-2 track and beam IDs are presented in white texts. (c) The Sentinel-2 image for Kangerlussuaq Glacier on 19 Jul 2020. (e) The Sentinel-1 HV image for Store Glacier on 19 Dec 2019. (b)(d)(f) The difference between ATL06 measurements over the ocean and the geoid [1] with tidal corrections [3] as a function of the distance along the ICESat-2 track in (a)(c)(e).

Comment (2). *Here is another question I have for the authors: is there evidence that the ice blocks in the mélange pile up (e.g., Figure 3)? I understand that many previous studies made similar assumptions. I am just curious to know if there were observations supporting that. To my knowledge, at least for some fjords in Greenland, the pre-existing icebergs do not seem to change their shapes and orientations significantly during some calving events (yes, the new icebergs will likely change the appearance of the fjord, but the pre-existing icebergs can have their above-water geometry largely unchanged). If ice blocks do not pile up, the effective accelerations for ice above and below the waterline could be the same in equation 1.*

Response. Observing the cross-sectional structure of mélange remains something of an open challenge. Ice-penetrating radar may be able to provide some insights, but the interpretation is not straightforward. While this far from a publication-ready result, below we show and discuss some ice-penetrating radar measurements of the mélange in front of Jakobshavn Glacier that we believe provides some initial perspective on the question of ice block pile-ups.

The data in Fig. 2 in this document were collected as part of Operation IceBridge on April 19, 2014 (the same date as the ArcticDEM strip shown in Supplementary Fig. 14). The pink dashed lines show the surface as measured by an airborne laser altimeter on the same aircraft as the radar, as well as the bottom of the mélange as inferred from that surface elevation using the hydrostatic equilibrium assumption. The inferred mélange bottom is well-aligned with radar bottom echoes from the largest icebergs, suggesting that these large icebergs are not piled up. However, in between the large icebergs, the radar cannot detect the base of the mélange at all, even though the surface elevation suggests that it should be 20-60 m thick in these regions. Since this radar system can image through a few thousand meters of meteoric ice and can clearly see the bottoms of tabular icebergs a few hundred meters thick, this suggests that there is incredibly high radio wave attenuation within the main body of the mélange. Such attenuation typically results from either entrained liquid water, very warm or salty ice, and/or high scattering losses from a material with heterogeneities on the scale of the radar wavelength (~ 5 m) [6, 7]. For example, radar systems commonly fail to image through accreted marine ice on the bottom of ice shelves where the ice temperature is near the freezing point and there is a high concentration of entrained salts and liquid brine pockets [8]. Therefore, one reasonable explanation for our observations is that the main body of the mélange consists of a porous mass of agglomerated ice chunks that is partially or fully infiltrated with sea water or has sea ice binding the particles together. Thus, these observations may offer some tentative, preliminary support for our model, though it is far beyond the scope of this paper to fully investigate the radar signatures. At a minimum, the radar observations do provide some nuance to our assumption that icebergs will pile up. The background mélange height gradient we observe is not possible without a “piling up” of smaller ice chunks and icebergs. If small icebergs do not pile up, the observed mélange thickness gradient indicates that the size of icebergs within the mélange decays as a function of distance from terminus, which is not true based on satellite images. But larger, more tabular bergs do not seem to pile up and are just packed in by the surrounding mélange. Therefore, in a natural system, the degree of piling

up likely depends on the individual iceberg geometry, e.g. large tabular icebergs may not pile up but may be cemented into the *mélange* by a piled-up mass of small icebergs/bergy-bits.

FIGURE 2. A radar cross-section Jakobshavn Glacier collected on 2014-04-19 (same date as the ArcticDEM strip shown in Supplementary Fig. 14). The pink lines show the surface as measured by laser altimetry from the same aircraft as the radar, as well as the bottom of the *mélange* that we would infer from that surface elevation assuming hydrostatic equilibrium. The inferred *mélange* bottom matches up nicely with radar bottom echoes from the biggest icebergs, suggesting that they are not piled up. But in between the big icebergs, the radar cannot detect the *mélange* bottom, which implies that the *mélange* was a porous mass of agglomerated ice chunks.

Previous observations at Jakobshavn Isbræ show that inferred thickness of the *mélange* (based on TRI-derived surface elevations and assuming hydrostatic equilibrium) near the

glacier terminus can exceed 400 m, and decay to 87 m within 1 km from the terminus [2]. The study also reveals that, calving-like collapse events or divergent motion within the mélange helps to advect ice away, resulting in the divergent thinning of mélange, whose rate is of the same order of magnitude as melt thinning. These observations also support the view that icebergs in the mélange can pile up, resulting in the observed mélange thickness gradient. If icebergs do not pile up, 1) the total thinning rate should all be attributed to melt thinning, which is not true as seen in the 1st supplemental video of [2], 2) the observed mélange thickness gradient indicates that the size of icebergs within the mélange decays as a function of distance from terminus, which is not true based on satellite images in [2] or in our manuscript.

The reviewer is correct that our three-dimensional continuum equations only work if icebergs within the mélange pile up, and we have highlighted this point in the main text as follows (lines 357–363):

For the six cases where the mélange collapses into thin monolayers at the end of the simulation (denoted as pentagram markers), the final buttressing forces can be predicted well by the previously developed theory for mélange of a uniform thickness [9] with the yield stress parameter, σ_0 , fitted to be 0.12 kPa \sim 0.16 kPa. The modeled buttressing forces in these cases are smaller than in the three-dimensional continuum (Eqn. 5; black lines in Figure 6), because the mélange only has a monolayer and violates the assumption of three-dimensional mélange with a constant packing density throughout its depth.

Note that all available remote observations are mélange elevation above sea level, and we retrieve mélange thickness by assuming hydrostatic equilibrium, as in [2, 10]. There is no direct field or remote observations of mélange thickness. It will be exciting future work to obtain Greenland mélange thickness by underwater robots (i.e., Icefin for Antarctica [11]), to validate the hydrostatic assumption of mélange.

***Comment (3).** (Remarks on code availability): I could only see a piece of code reading a DEM and a function to extract elevation profiles. I am sure the authors will make the essential code available upon publication of the paper.*

Response. We have revised the code availability section as follows:

5. Code availability All source data and MATLAB codes to reproduce figures in the main text are available through Zenodo at <https://zenodo.org/records/13382185> [12]. The codes used for the three-dimensional discrete element model are available from the corresponding author upon reasonable request. PFC3D® [13] is a software from Itasca Consulting Group, Inc. through a commercial license.

In summary, we thank the referee for making these constructive suggestions. We hope that our responses have resolved the ambiguities he/she has pointed out. We believe these additions have improved our manuscript significantly.

REFERENCES

- [1] Förste Ch, Sean L Bruinsma, Oleg Abrikosov, Jean-Michel Lemoine, T Schaller, HJ Gtze, J Ebbing, JC Marty, F Flechtner, G Balmino, et al. Eigen-6c4 the latest combined global gravity field model including goce data up to degree and order 2190 of gfz potsdam and grgs toulouse. *GFZ Data Services*, 10(10.5880), 2014.
- [2] Surui Xie, Timothy H Dixon, David M Holland, Denis Voytenko, and Irena Vaňková. Rapid iceberg calving following removal of tightly packed pro-glacial mélange. *Nature communications*, 10(1):3250, 2019.
- [3] Susan L Howard and Laurie Padman. Gr1kmtm: Greenland 1 kilometer tide model, 2021.
- [4] B. Smith, S. Adusumilli, B. M. Csathó, D. Felikson, H. A. Fricker, A. S. Gardner, N. Holschuh, J. Lee, J. Nilsson, F. Paolo, M. R. Siegfried, T. Sutterley, and the ICESat-2 Science Team. Atlas/icesat-2 l3a land ice height, version 6, 2023.
- [5] B Smith, D Hancock, K Harbeck, L Roberts, T Neumann, Kelly Brunt, H Fricker, A Gardner, M Siegfried, S Adusumilli, et al. Algorithm theoretical basis document (ATBD) for land ice along-track height product (ATL06). *Ice, Cloud, and land elevation satellite-2 (ICESat-2) project*, 2019.
- [6] Fawwaz T. Ulaby and David G. Long. *Microwave Radar and Radiometric Remote Sensing*. The University of Michigan Press, Ann Arbor, MI, 2014.
- [7] Kenichi Matsuoka, Joseph A. MacGregor, and Frank Pattyn. Predicting radar attenuation within the Antarctic ice sheet. *Earth and Planetary Science Letters*, 359–360:173–183, December 2012.
- [8] Helen Amanda Fricker, Sergey Popov, I. Allison, and N Young. Distribution of marine ice beneath the Amery Ice Shelf. *Geophysical Research Letters*, 28(11):2241–2244, 2001.
- [9] Justin C Burton, Jason M Amundson, Ryan Cassotto, Chin-Chang Kuo, and Michael Dennin. Quantifying flow and stress in ice mélange, the world’s largest granular material. *Proceedings of the National Academy of Sciences*, 115(20):5105–5110, 2018.
- [10] Jae Hun Kim, Eric Rignot, David Holland, and Denise Holland. Seawater intrusion at the grounding line of jakobshavn isbræ, greenland, from terrestrial radar interferometry. *Geophysical Research Letters*, 51(6):e2023GL106181, 2024.
- [11] Anthony Spears, Michael West, Matthew Meister, Jacob Buffo, Catherine Walker, Thomas Riley Collins, Ayanna Howard, and Britney Schmidt. Under ice in antarctica: The icefin unmanned underwater vehicle development and deployment. *IEEE Robotics & Automation Magazine*, 23(4):30–41, 2016.
- [12] Yue Meng, Ching-Yao Lai, Riley Culberg, Michael G. Shahin, Leigh A. Stearns, Justin C. Burton, and Kavinda Nissanka. Supporting data – seasonal changes of mélange thickness coincide with greenland calving dynamics, 2024.
- [13] Itasca Consulting Group, Inc. *PFC — Particle Flow Code, Ver. 7.0*. Minneapolis: Itasca, 2021.

Review of
Seasonal Changes of Mélange Thickness Coincide with Greenland Calving Dynamics,
by Meng, Yue et al., 2024.

Summary

This manuscript investigates the role of variable proglacial mélange thickness in determining its buttressing force and ability to suppress calving at tidewater glaciers in Greenland. The authors show how mélange thickness varies seasonally using several observational datasets, including terrestrial laser scanning point-cloud data for one site (Helheim Glacier), ICESat-2 derived elevation profiles for three glaciers, and later a larger scale analyses of mélange thicknesses derived from ArcticDEM strips at 32 tidewater glaciers. The primary contribution of this work, however, is in the presentation of the author's new three-dimensional discrete element model of mélange, which incorporates variable mélange thickness with distance from the terminus, and is more sophisticated than previous models which assumed a static mélange thickness field. It is also noteworthy that their model features the inclusion of more realistic cubic shaped icebergs rather than spheres. Most importantly, the authors derive an Equation showing how the mélange buttressing force can be reasonably calculated from observations of near-terminus mélange thickness, which is an important step in making mélange/terminus interactions more readily representable in ice sheet and glacier flow models. The submission also features richly valuable supplementary material, including movies of mélange simulations for different terminus behavior and simplified fjord shapes (one featuring straight, uniform width walls and one with more rugged and variable fjord walls). The paper is well-written and the main science questions adequately motivated by cited prior research showing that mélange is linked to the timing of terminus advance and retreat and paramount to realistically modeling seasonal glacier dynamics. This manuscript presents new and important research that will be valuable to both the observational and modeling glaciological communities, and should, in principle, be suitable for publication in *Nature Communications* after some suggested revisions below. My primary comments address the organization of the overall manuscript to improve readability and comprehensive of the study's methods and science objectives. The main comments are given in more detail below, followed by several minor comments and suggestions.

Main

Manuscript organization and inclusion of terrestrial laser scanner + ICESat-2 observations.

I found that transition from the abstract/introduction to the first section of the results (on the point cloud observations of Helheim mélange thickness) to be rather abrupt and without sufficient context. For example, I found this section (and the following section where ICESat-2 and satellite imagery are used to study seasonal changes at three tidewater glaciers) both very similar in the goal of the analyses but disjointed from the remainder of the manuscript. I found it challenging to tie in the results from these sections with the later derivation of the relationship between buttressing force and mélange thickness and application of the 3D mélange model. These sections also raised the following questions:

- 1.) Can the authors provide more context for the use of 30m freeboard height exclusion threshold? Was this threshold empirically derived? I am also curious if the authors noticed any temporal bias in how heavily this threshold excluded glaciers (for example, were more thick icebergs present and subsequently excluded in spring versus summer).
- 2.) What was the motivation for calculating mean thickness within 1 km in this section, but later in the manuscript, modeled outputs are averaged within 200 m of the terminus?
- 3.) Line 92, on four dates of pronounced thinning – over what time period is a decrease in elevation calculated? The lower bound of the elevation is given, but not the initial elevation prior to thinning (other than a 15 m average).

I would suggest adding more organizational context to the introduction to prime readers for the relevance of the initial observational analyses. My interpretation is that they are “proof of concept” or testing available observational datasets to show whether, initially, there is sufficient evidence of seasonal variability in mélange thickness to motivate the development of the three-dimensional model. Perhaps results from these early studies help inform the model with a realistic range of near-termini mélange thicknesses? It would also be helpful to include a brief description of why the three glaciers were selected for the ICESat-2 work, even if that is simply something similar to there being “large discharging glaciers with identified seasonally varying mélange.”

Section 1.2

I found it challenging to understand from the given text how comparisons at Jakobshavn and Kangerlussuaq varied from Store. All three seem to show gradients in mélange freeboard heights that coincide to expected terminus activity, yet the text partitions them in a way that implies either the method or interpretation of the results vary between the two groups. I would suggest additional text early in the paragraph similar to “*We acquire ICESat-2 elevation data near the terminus for these three glaciers, where mélange was continually present over our study period at X and X, and seasonally present at X. Where mélange persisted, we calculated seasonally distinct freeboard heights....etc*”

On line 126, the results are described to support a hypothesis of thick mélange enabling winter terminus advance. Consider changing the language in the first paragraph of this section to better articulate this as a study hypothesis, which as written states a more general objective of identifying if there are ‘correlations between mélange thickness and calving dynamics.’

Section 1.6

Line 291 states that during winter advance, the buttressing force varies from 1.7 to 2.7×10^7 N/m (which exceeds the threshold suggested to impede calving at a floating terminus). However, later in the discussion, the average wintertime advance force is given of 6.5×10^6 N/m. What is the reason for this difference?

On filtering ArcticDEM strip data

The methods section of the manuscript describes the process of excluding strips when the terminus behavior is variable (both advance and retreat) within a 2-month window surrounding the strip acquisition date. This filtering step reduces the strip number from >300 to 108. I think

this is relevant information to describe earlier in the paper, in section 1.6. I understand why the authors used this approach given the study's main questions, but the fact that the majority of scenes featured episodically changing terminus behavior is important for evaluating when and under what conditions mélange thickness-to-buttressing force calculations are most applicable to understanding terminus change.

Minor comments/requests for clarification

-Heading of section 1.5, consider editing to “at *the* glacier terminus”.

-It may be helpful to specify the months corresponding to the seasonal descriptions (spring, winter, summer...). These may be the obvious 3-month definitions, but there are places in the text (for example, that 85% of advance occurred in winter) where it was unclear whether winter indicated DJF or more general “cold” vs “warm” season.

Line 339 through 342: consider adding glacier ID's in parentheses in the discussion text for easy comparison to figure 7.

Line 267 correct “We hypothesis” to “we hypothesize”

Figure 6: Consider again including a note that the blue line follows the highest density elevation in the figure caption.